# Scaling Laws for Precision in High-Dimensional Linear Regression

**Dechen Zhang** [1]  **Xuan Tang** [2]  **Yingyu Liang** [1,2]  **Difan Zou** [1,2]

## Abstract

Low-precision training is critical for optimizing the trade-off between model quality and training costs, necessitating the joint allocation of model size, dataset size, and numerical precision. While empirical scaling laws suggest that quantization impacts effective model and data capacities or acts as an additive error, the theoretical mechanisms governing these effects remain largely unexplored. In this work, we initiate a theoretical study of scaling laws for low-precision training within a high-dimensional sketched linear regression framework. By analyzing multiplicative (signal-dependent) and additive (signal-independent) quantization, we identify a critical dichotomy in their scaling behaviors. Our analysis reveals that in the worst case, while both schemes introduce an additive error and degrade the effective data size, they exhibit distinct effects on effective model size: multiplicative quantization maintains the full-precision model size, whereas additive quantization reduces the effective model size. Numerical experiments validate our theoretical findings. By rigorously characterizing the complex interplay among model scale, dataset size, and quantization error, our work provides a principled theoretical basis for optimizing training protocols under practical hardware constraints.

## 1. Introduction

The remarkable success of large language models (LLMs) has been largely driven by the scaling of model parameters and training datasets, governed by the now-canonical neural scaling laws (Kaplan et al., 2020; Hoffmann et al., 2022). However, the prohibitive computational and memory costs associated with such scaling have made low-precision train-

[1]Institute of Data Science, The University of Hong Kong [2]School of Computing & Data Science, The University of Hong Kong. Correspondence to: Difan Zou <dzou@hku.hk>.

*Proceedings of the 43$^{rd}$ International Conference on Machine Learning*, Seoul, South Korea. PMLR 306, 2026. Copyright 2026 by the author(s).

ing indispensable (Courbariaux et al., 2014; Wang et al., 2018; Sun et al., 2020; Hao et al., 2025). State-of-the-art frameworks now extensively leverage mixed- or low-precision formats for gradients, weights, and optimizer states (Peng et al., 2023; Wortsman et al., 2023; Xi et al., 2024; Fishman et al., 2024; Liu et al., 2024), showing that aggressively low-precision training can scale to trillion-token workloads without compromising accuracy. This shift fundamentally alters the scaling landscape, introducing a complex interplay between model size, dataset size, and numerical precision. Optimizing the performance of LLMs thus necessitates a rigorous understanding to guide the joint allocation of fixed compute or memory budgets across these three dimensions.

Despite the practical urgency, our understanding of low-precision scaling remains predominantly empirical. Recent studies have proposed different functional forms to describe how bit-width affects the scaling behavior in low-precision training (Kumar et al., 2024; Sun et al., 2025). One line of research posits that quantization effectively reduces the model's capacity: $L(M, N, Q) \approx AM_{\text{eff}}(M, Q)^{-\alpha} + BN^{-\beta} + E$, where $M_{\text{eff}}$ represents an effective model size reduced by quantization operations (Kumar et al., 2024). While others formulate quantization as an additive error term: $L(M, N, Q) \approx AM^{-\alpha} + BN^{-\beta} + E + \delta(M, N, Q)$, where $\delta$ acts as an explicit penalty term dependent on quantization (Sun et al., 2025). Crucially, these are purely empirical fits and there exists no unified theoretical framework to determine which formulation, effective size reduction or additive error, is physically correct, nor to mechanistically account for the intricate effects of specific training algorithms and mixed-precision strategies.

Recent studies on the theoretical understanding of scaling laws have focused on analyzing the exact training dynamics of SGD using linear models (Lin et al., 2024; 2025; Li et al., 2025; Yan et al., 2025). In particular, Lin et al. (2024) resolved the discrepancy between neural scaling laws and traditional statistical learning theory by adopting an infinite-dimensional sketched linear regression framework with power-law spectra. Building on this, Lin et al. (2025) and Yan et al. (2025) extended one-pass SGD to multi-pass SGD, showing the benefit of increasing the multi-epoch count $K$. In a parallel avenue of research, Li et al. (2025) characterized how the learning rate schedule shapes scaling

behaviors. These works have repeatedly demonstrated that such high-dimensional linear setups, despite their simplicity, can faithfully capture key phenomenological aspects of deep learning. Motivated by these successes, we initiate the theoretical study of scaling laws for low-precision training within a high-dimensional sketched linear regression setup.

**Our setting.** We assume access to $M$-dimensional sketched covariates and their responses, that is, $(\mathbf{Sx}, y)$, where $\mathbf{S} \in \mathbb{R}^M \times \mathbb{H}$ is a fixed sketch matrix, $\mathbf{x} \in \mathbb{H} \subset \mathbb{R}^p$ is the data vector and $\mathbb{H}$ is a Hilbert space that is either finite-dimensional or countably infinite-dimensional. We focus on the Gaussian sketch matrix (Lin et al., 2024; 2025; Chen et al., 2025b; Ding et al., 2025). That is, entries of $\mathbf{S}$ are independently sampled from $\mathcal{N}(0, 1/M)$. We then consider linear model with $M$ trainable parameters given by:

$$f_{\mathbf{v}} : \mathbb{H} \to \mathbb{R}, \quad \mathbf{x} \to \langle \mathbf{v}, \mathbf{Sx} \rangle,$$

where $\mathbf{v} \in \mathbb{R}^M$ are the trainable parameters. Our goal is to bound the population risk

$$\mathcal{R}_M(\mathbf{v}) := \frac{1}{2}\mathbb{E}[(\langle \mathbf{Sx}, \mathbf{v} \rangle - y)^2], \quad \mathbf{v} \in \mathbb{R}^M,$$

where the expectation is conditioned on the sketch matrix $\mathbf{S}$ [1]. We consider training $f_{\mathbf{v}}$ via constant-stepsize one-pass quantized stochastic gradient descent (SGD) (Zhang et al., 2025). The parameter $\mathbf{v}_t$ is updated as follows:

$$\mathbf{v}_t = \mathbf{v}_{t-1} + \gamma g^{(q)} \mathbf{f}^{(q)}, \quad t = 1, ..., N,$$
(quantized SGD)

$$\mathbf{f}^{(q)} = \mathcal{Q}_f\left(\mathcal{Q}_s(\mathbf{S})\mathcal{Q}_d(\mathbf{x}_t)\right),$$
$$g^{(q)} = \mathcal{Q}_o(\mathcal{Q}_l(y_t) - \mathcal{Q}_a(\mathcal{Q}_f\left(\mathcal{Q}_s(\mathbf{S})\mathcal{Q}_d(\mathbf{x}_t)\right)^\top \mathcal{Q}_p(\mathbf{v}_{t-1}))),$$

where $(\mathbf{x}_t, y_t)_{t=1}^N$ are independent samples and $\gamma$ is the stepsize, and $\mathcal{Q}_d, \mathcal{Q}_s, \mathcal{Q}_f, \mathcal{Q}_l, \mathcal{Q}_p, \mathcal{Q}_a, \mathcal{Q}_o$ are independent general quantization operations for data, sketch matrix, feature, labels, model parameters, activations and output gradients respectively, and $\mathbf{f}^{(q)}$ is the quantized feature and $g^{(q)}$ is the quantized output gradient. Without loss of generality, we assume the initial parameter is $\mathbf{v}_0 = 0$. The output of the SGD algorithm is the the iterate average $\overline{\mathbf{v}}_N := \frac{1}{N}\sum_{t=0}^{N-1} \mathbf{v}_t$.

**Notations.** For two positive-valued functions $f(x)$ and $g(x)$, we write $f(x) \lesssim g(x)$ or $f(x) \gtrsim g(x)$ if $f(x) \le cg(x)$ or $f(x) \ge cg(x)$ holds for some absolute (if not otherwise specified) constant $c > 0$ respectively. We write $f(x) \asymp g(x)$ if $f(x) \lesssim g(x) \lesssim f(x)$. For two vectors $\mathbf{u}$ and $\mathbf{v}$ in a Hilbert space, we denote their inner product by $\langle \mathbf{u}, \mathbf{v} \rangle$ or $\mathbf{u}^\top \mathbf{v}$. For two matrices $\mathbf{A}$ and $\mathbf{B}$ of appropriate dimensions, we define their inner product by $\langle \mathbf{A}, \mathbf{B} \rangle := \text{tr}\left(\mathbf{A}^\top \mathbf{B}\right)$. We use $\|\cdot\|$ to denote the operator norm for matrices. For a positive semi-definite (PSD) matrix $\mathbf{A}$ and a vector $\mathbf{v}$ of appropriate dimension, we write $\|\mathbf{v}\|_{\mathbf{A}}^2 = \mathbf{v}^\top \mathbf{A}\mathbf{v}$.

---

[1] In this paper, all expectations are conditioned on $\mathbf{S}$.

**Our main results.** Assuming that the spectrum of the data covariance matrix satisfies a power-law of degree $a > 1$, we analyze scaling laws under two standard quantization schemes: multiplicative quantization (where error variance scales with signal magnitude) and additive quantization (where error variance is independent of the signal). Informally, the population risk upper bound for both schemes can be unified as:

$$\mathcal{R}_M(\overline{\mathbf{v}}_N) \lesssim \mathcal{R}^* + \frac{1}{M_{\text{eff}}(M, \epsilon)^{a-1}} + \frac{1}{N_{\text{eff}}(N, \epsilon)^{\frac{a-1}{a}}} + \delta(\epsilon),$$

where $M$ is the model size, $N$ is the data size and $\mathcal{R}^*$ represents a positive irreducible risk, $\epsilon$ generically represents the quantization error (which vanishes in full-precision training) and $\delta(\epsilon)$ denotes an *additive error* induced by $\epsilon$. The key quantities $M_{\text{eff}}$ and $N_{\text{eff}}$ represent the *effective model size* and *effective data size*, respectively. We demonstrate a critical divergence in how the two quantization schemes affect these quantities.

- **Effective Data Size** ($N_{\text{eff}}$): Both schemes reduce the effective data size via noise-amplification quantization error $\epsilon_{\text{noise}}$ and spectral-distortion quantization error $\epsilon_{\text{spect}}$.
- **Effective Model Size** ($M_{\text{eff}}$): Multiplicative quantization (FP-like) preserves the full model capacity (i.e., $M_{\text{eff}} \approx M$), whereas additive quantization (INT-like) strictly contracts it driven by noise amplification and spectral distortion factors analogous to those reducing $N_{\text{eff}}$.

We refer to Theorem 4.1 and Theorem 4.2 for formal statements of upper bounds. This theoretical dichotomy provides a rigorous basis for recent empirical findings in low-precision training. Specifically, our additive quantization scaling law captures the effective model shrinkage observed in integer quantization (Kumar et al., 2024), while our multiplicative quantization scaling law corroborates the observation that floating-point quantization preserves effective model capacity (Sun et al., 2025). Complementing the upper bounds, we establish the first population risk lower bounds for low-precision training (see Theorem 4.3 and Theorem 4.4 for details). These lower bounds validate the existence of the additive error and the reduction of effective data size, confirming that these mechanisms are fundamental in low-precision training.

## 2. Related Work

**Empirical scaling laws for quantized training.** Recent research has focused on empirically characterizing the scaling behaviors of quantized training (Dettmers & Zettlemoyer, 2023; Ouyang et al., 2024; Kumar et al., 2024; Tao et al., 2024; Frantar et al., 2025; Chen et al., 2025a; Sun et al., 2025; Liu et al., 2025). One line of work conceptualizes quantization as a mechanism that effectively reduces model size (Kumar et al., 2024; Frantar et al., 2025). No-

tably, Kumar et al. (2024) proposed unified scaling laws under integer quantization covering low-precision training, quantization-aware training (QAT), and post-training quantization (PTQ). For low-precision training, they modeled the loss as $L(M, N, P) \approx AM_{\text{eff}}(M, P)^{-\alpha} + BN^{-\beta} + E$, where $M_{\text{eff}}(M, P) \approx M(1 - e^{-P/\gamma})$ represents the effective model capacity contracted by low precision. Another stream of research models quantization as an additive error (Chen et al., 2025a; Sun et al., 2025). Sun et al. (2025) established scaling laws for low-precision training under floating-point (FP) formats, formulating the loss with a precision-dependent error term: $L(M, N, P) \approx AM^{-\alpha} + BN^{-\beta} + E + \delta(M, N, P)$. They showed that quantization induces a predictable deviation from the standard power law. In a parallel effort targeting integer QAT, Chen et al. (2025a) extended this framework to account for quantization granularity ($G$), modeling the loss via a similar additive penalty: $L(M, N, P) \approx AM^{-\alpha} + BN^{-\beta} + E + \delta(M, N, P, G)$.

**High-dimensional linear regression via SGD.** Theoretical guarantees for generalization have garnered significant attention in machine learning. Seminal work by Bartlett et al. (2020); Tsigler & Bartlett (2023) established nearly tight upper and lower excess risk bounds for linear (ridge) regression under general regularization schemes. In the classical under-parameterized regime, extensive literature has explored the learnability of iterate-averaged SGD (Polyak & Juditsky, 1992; Bach & Moulines, 2013; Défossez & Bach, 2015; Dieuleveut et al., 2017; Jain et al., 2017; 2018). Conversely, in the modern overparameterized setting, one-pass SGD has been rigorously studied (Dieuleveut & Bach, 2015; Berthier et al., 2020; Varre et al., 2021; Zou et al., 2021; Wu et al., 2022a;b; Zhang et al., 2024), yielding frameworks to characterize how optimization dynamics influence generalization across various data distributions. Additionally, another line of work has analyzed multi-pass SGD for high-dimensional $\ell^2$-regularized least squares, detailing excess risk bounds (Lei et al., 2021; Zou et al., 2022) and exact risk dynamics (Paquette et al., 2024a). More recently, Zhang et al. (2025) established the first excess risk upper bounds for low-precision training, characterizing the impact of quantization on the learning dynamics of SGD in linear regression. Our work builds upon this foundation by extending their theoretical framework to sketched linear regression. Furthermore, we provide a critical missing piece by deriving the first excess risk lower bounds for low-precision training.

**Theoretical understandings of scaling laws.** Several recent studies have sought to formalize and explain empirical scaling laws using conceptually simplified linear models (Bahri et al., 2024; Atanasov et al., 2024; Paquette et al., 2024b; Bordelon et al., 2024; Lin et al., 2024; 2025; Yan et al., 2025; Li et al., 2025; Ding et al., 2025). Early theoretical attempts focused on asymptotic regimes: Bahri et al. (2024) analyzed a linear teacher-student model with power-

law spectra, showing that the test loss of the ordinary least squares (OLS) estimator decays as a power law in sample size $N$ (or model size $M$) when the other dimension approaches infinity. Similarly, Bordelon et al. (2024) studied gradient flow in linear random feature models, establishing power-law scaling with respect to one of $N$, $M$, or training time $T$, provided the other parameters remain effectively infinite. A pivotal step towards realistic finite-sample analysis is made by Lin et al. (2024). Building on analysis techniques from Zou et al. (2021) and Wu et al. (2022a), they analyzed the last iterate of one-pass SGD in a sketched linear model and presented the first systematic derivation of a finite-sample joint scaling law (in both $M$ and $N$) that aligns with empirical observations (Kaplan et al., 2020). Subsequent research expands this framework to more complex settings. Lin et al. (2025) extended the analysis to data reuse (multi-pass SGD), showing that for relatively small multi-epoch count $K$, every new epoch leads to a linear gain in effective sample size, i.e., $N_{\text{eff}} \approx NK$. Building on this, Yan et al. (2025) provided a finer-grained characterization for strongly convex or Zipf-distributed data. They demonstrated that for large multi-epoch count $K$, the effective reuse rate $N_{\text{eff}}/N$ plateaus at a problem-dependent value that grows with $N$. More recently, Li et al. (2025) established functional scaling laws and analyzed how learning rate schedules shape these scaling behaviors.

## 3. Theoretical Setup

### 3.1. Quantization Operation

For all quantization operations in (quantized SGD), we employ the stochastic quantization method (Markov et al., 2023; Modoranu et al., 2024; Ozkara et al., 2025), which unbiasedly rounds values using randomly adjusted probabilities. We summarize this in the following assumption.

**Assumption 3.1.** Let $\mathcal{Q}_i, i \in \{d, s, f, l, p, a, o\}$ be the coordinate-wise quantization operation for data, sketch matrix, feature, label, model parameters, activations, and output gradients, respectively. Then for any $\mathbf{u}$, the quantization operation is unbiased:

$$\mathbb{E}\left[\mathcal{Q}_i(\mathbf{u})|\mathbf{u}\right] = \mathbf{u}.$$

Furthermore, to better uncover the effect of quantization, we consider the following two types of quantization error: multiplicative quantization and additive quantization, which are motivated by abstracting the behavior of prevalent numerical formats used in practice (Zhang et al., 2025).

**Definition 3.2.** Let $\mathcal{Q}$ be an unbiased quantization operation. Denote $\mathbf{D} = \mathbb{E}[(\mathcal{Q}(\mathbf{x}) - \mathbf{x})(\mathcal{Q}(\mathbf{x}) - \mathbf{x})^\top |\mathbf{x}]$, and for any PSD matrix $\mathbf{A}$, denote $\mathbf{D}(\mathbf{A}) = \mathbb{E}[(\mathcal{Q}(\mathbf{X}) - \mathbf{X})\mathbf{A}(\mathcal{Q}(\mathbf{X}) - \mathbf{X})^\top |\mathbf{X}]$. We formalize two practical quantization schemes:

- **Multiplicative quantization.** We call the quantization to $\mathbf{x}$ is $(\underline{\epsilon}, \overline{\epsilon})$-multiplicative if the conditional second moment of quantization error is proportional to the outer product of raw data itself, i.e.,

$$\underline{\epsilon}\mathbf{x}\mathbf{x}^\top \preceq \mathbf{D} \preceq \overline{\epsilon}\mathbf{x}\mathbf{x}^\top.$$

For multiplicative quantization to matrix $\mathbf{X}$, we extend the definition to $\underline{\epsilon}\mathbf{X}\mathbf{A}\mathbf{X}^\top \preceq \mathbf{D}(\mathbf{A}) \preceq \overline{\epsilon}\mathbf{X}\mathbf{A}\mathbf{X}^\top$.
- **Additive quantization.** We call the quantization to $\mathbf{x}$ is $(\underline{\epsilon}, \overline{\epsilon})$-additive if the conditional second moment of quantization error is proportional to identity, i.e.,

$$\underline{\epsilon}\mathbf{I} \preceq \mathbf{D} \preceq \overline{\epsilon}\mathbf{I}.$$

For additive quantization to matrix $\mathbf{X}$, we extend the definition to $\underline{\epsilon}\mathrm{tr}(\mathbf{A})\mathbf{I} \preceq \mathbf{D}(\mathbf{A}) \preceq \overline{\epsilon}\mathrm{tr}(\mathbf{A})\mathbf{I}$.

This theoretical distinction is grounded in practical quantization schemes. For instance, integer quantization (e.g., INT8, INT16) uses a fixed bin length, resulting in an error that is largely independent of the value's magnitude (Wu et al., 2020). This characteristic aligns with our definition of additive quantization, where the error variance is uniform across coordinates. Conversely, floating-point quantization (e.g., FP8, FP32) employs a value-aware bin length via its exponent and mantissa bits (e.g., E4M3 format in FP8) (Kuzmin et al., 2022). This structure causes the quantization error to scale with the magnitude of the value itself, corresponding to multiplicative quantization.

### 3.2. Data Model

We then state the regularity assumptions on the data distribution, which align with those common in prior works (Zou et al., 2021; Wu et al., 2022a;b; 2023). As low-precision training is performed on quantized feature $\tilde{\mathbf{x}}^{(q)} = \mathcal{Q}_f\left(\mathcal{Q}_s(\mathbf{S})\mathcal{Q}_d(\mathbf{x})\right)$, we formulate these assumptions on the low-precision feature format following Zhang et al. (2025).

**Assumption 3.3** (Data covariance). Let $\mathbf{H} := \mathbb{E}[\mathbf{x}\mathbf{x}^\top]$ be the data covariance and $\mathbf{H}_f^{(q)} := \mathbb{E}[\tilde{\mathbf{x}}^{(q)}(\tilde{\mathbf{x}}^{(q)})^\top]$ be the quantized feature covariance. Assume that $\mathrm{tr}(\mathbf{H}), \mathrm{tr}(\mathbf{H}_f^q)$ and all entries of $\mathbf{H}, \mathbf{H}_f^{(q)}$ are finite. For convenience, we assume that $\mathbf{H}$ is strictly positive definite.

Let $\mathbf{H} = \sum_i \lambda_i \mathbf{v}_i \mathbf{v}_i^\top$ be the eigen-decomposition of $\mathbf{H}$, where $\{\lambda_i\}_{i=1}^\infty$ are the eigenvalues of $\mathbf{H}$ sorted in non-increasing order and $\mathbf{v}_i$ are the corresponding eigenvectors. We denote $\mathbf{H}_{0:k} := \sum_{i=1}^k \lambda_i \mathbf{v}_i \mathbf{v}_i^\top$, $\mathbf{H}_{k:\infty} := \sum_{i>k} \lambda_i \mathbf{v}_i \mathbf{v}_i^\top$, $\mathbf{I}_{0:k} := \sum_{i=1}^k \mathbf{v}_i \mathbf{v}_i^\top$, $\mathbf{I}_{k:\infty} := \sum_{i>k} \mathbf{v}_i \mathbf{v}_i^\top$. Similarly, we denote the eigen-decomposition of $\mathbf{H}_f^{(q)}$ as $\mathbf{H}_f^{(q)} = \sum_i \tilde{\lambda}_i^{(q)} \mathbf{v}_i^{(q)} \mathbf{v}_i^{(q)\top}$ and correspondingly obtain $\mathbf{H}_{f,0:k}^{(q)}, \mathbf{H}_{f,k:\infty}^{(q)}, \mathbf{I}_{f,0:k}^{(q)}, \mathbf{I}_{f,k:\infty}^{(q)}$, where $\{\tilde{\lambda}_i^{(q)}\}_{i=1}^\infty$ are the eigenvalues of $\mathbf{H}_f^{(q)}$. In line with Zhang et al. (2025),

we extend the fourth moment and noise assumptions (Zou et al., 2021; Wu et al., 2022b;a; 2023) to quantized features.

**Assumption 3.4** (Fourth-moment conditions). Denote $\mathbf{M} = \mathbb{E}\left[\tilde{\mathbf{x}}^{(q)}(\tilde{\mathbf{x}}^{(q)})^\top \mathbf{A}\tilde{\mathbf{x}}^{(q)}(\tilde{\mathbf{x}}^{(q)})^\top\right]$ for any PSD matrix $\mathbf{A}$. Assume that the fourth moment of $\tilde{\mathbf{x}}^{(q)}$ is finite and there exist constants $\alpha, \beta > 0$ such that

$$\mathbf{H}_f^{(q)}\mathbf{A}\mathbf{H}_f^{(q)} + \beta\,\mathrm{tr}(\mathbf{H}_f^{(q)}\mathbf{A})\mathbf{H}_f^{(q)} \preceq \mathbf{M} \preceq \alpha\,\mathrm{tr}(\mathbf{H}_f^{(q)}\mathbf{A})\mathbf{H}_f^{(q)}.$$

Regarding the noise assumptions, we first define the population risk and global optimum in quantized feature space:

$$\mathcal{R}_M^{(q)}(\mathbf{v}) := \frac{1}{2}\mathbb{E}[(\langle\tilde{\mathbf{x}}^{(q)}, \mathbf{v}\rangle - \mathcal{Q}_l(y))^2], \quad \mathbf{v} \in \mathbb{R}^M,$$

with global optimum $\mathbf{v}^{(q)*} := \mathrm{argmin}_\mathbf{v} \mathcal{R}_M^{(q)}(\mathbf{v})$.

**Assumption 3.5** (Noise conditions). Denote $\xi := \mathcal{Q}_l(y) - \langle\mathbf{v}^{(q)*}, \tilde{\mathbf{x}}^{(q)}\rangle$. Assume there exists a positive constants $\overline{\sigma}, \underline{\sigma} > 0$ such that

$$\underline{\sigma}^2\mathbf{H}_f^{(q)} \preceq \mathbb{E}[\xi^2\tilde{\mathbf{x}}^{(q)}(\tilde{\mathbf{x}}^{(q)})^\top] \preceq \overline{\sigma}^2\mathbf{H}_f^{(q)}.$$

A key distinction from the assumptions in Zhang et al. (2025) is that we require lower bounds on the noise and fourth moment to establish both upper and lower risk bounds. We would like to remark that under the fourth moment assumption and noise assumption on the full-precision data, Assumption 3.4 and 3.5 can be verified under specific multiplicative quantization and additive quantization schemes. We defer the verification in Section H.

To simplify the scaling-law behavior, we assume specific data distribution where the data spectrum satisfies a power law and the optimal parameter satisfies a prior (Lin et al., 2024). Specifically, we consider the population risk for $\mathbf{w} \in \mathbb{H}$:

$$\mathcal{R}(\mathbf{w}) := \frac{1}{2}\mathbb{E}[(\langle\mathbf{x}, \mathbf{w}\rangle - y)^2], \quad \mathbf{w} \in \mathbb{H},$$

with global optimum $\mathbf{w}^* := \mathrm{argmin}_\mathbf{w} \mathcal{R}(\mathbf{w})$.

**Assumption 3.6** (Distributional conditions). We assume the well-specified model, i.e., $\mathbb{E}[y|\mathbf{x}] = \mathbf{x}^\top\mathbf{w}^*$ and $\sigma^2 := \mathbb{E}\left[(y - \mathbf{x}^\top\mathbf{w}^*)^2\right]$, and the parameter prior, i.e., $\mathbb{E}[\mathbf{w}^*\mathbf{w}^{*\top}] = \mathbf{I}$. We also assume the data spectrum is polynomial, i.e., there exists $a > 1$ such that the eigenvalues of $\mathbf{H}$ satisfy $\lambda_i \approx i^{-a}$, $i > 0$.

## 4. Main Theory

In this section, we demonstrate low-precision training scaling laws when the data spectrum satisfies a power law. We state the scaling laws for multiplicative quantization and additive quantization respectively.

## 4.1. Multiplicative Quantization

In this section, we consider for any $i \in \{s, d, f, p, a, o\}$, there exist $\bar{\epsilon}_i$ such that quantization $\mathcal{Q}_i$ is $\bar{\epsilon}_i$-multiplicative [2]. Motivated by the insight from Zhang et al. (2025) that different quantization targets exert distinct influences on the risk, we first define a set of compound quantization coefficients to aggregate individual quantization errors based on their distinct physical effects on the learning dynamics. This formulation streamlines the presentation and elucidates the structural impact of quantization. Firstly, to capture the distortion to feature spectrum and the gap between quantized feature space and full-precision data space, we define

$$\bar{\epsilon}_3^{(M)} = 1 - \frac{1}{(1 + \bar{\epsilon}_d)(1 + \bar{\epsilon}_f)(1 + \bar{\epsilon}_s)},$$

which arises from feature, sketch and data quantization. Secondly, to characterize the noise amplification during training, we define

$$\bar{\epsilon}_2^{(M)} = (1 + \bar{\epsilon}_o)\left(1 + \bar{\epsilon}_p + (1 + \bar{\epsilon}_p)\bar{\epsilon}_a\right) - 1,$$

which arises from parameter, activation and output gradient quantization. Generally, the compound coefficients $\bar{\epsilon}_3^{(M)}$ and $\bar{\epsilon}_2^{(M)}$ scale monotonically with the underlying quantization severity. In standard training regimes where the individual quantization errors (e.g., $\bar{\epsilon}_o, \bar{\epsilon}_p, \bar{\epsilon}_d$) are small ($< 1$), these coefficients remain small quantities of comparable magnitude. In particular, $\bar{\epsilon}_3^{(M)}$ is strictly less than 1. Notably, they vanish strictly to zero in the full-precision limit. With these notations, we are now ready to state the main scaling laws under multiplicative quantization.

**Theorem 4.1** (Scaling law under multiplicative quantization, an upper bound). *Suppose* $\gamma < \frac{1}{(1 + \bar{\epsilon}_2^{(M)})\alpha \mathrm{tr}(\mathbf{H}_f^{(q)})}$. *For any* $i \in \{s, d, f, p, a, o\}$, *if there exist* $\bar{\epsilon}_i$ *such that quantization* $\mathcal{Q}_i$ *is* $\bar{\epsilon}_i$-*multiplicative, then under Assumption 3.1, 3.3, 3.4, 3.5 and 3.6, if* $\mathbf{H}_f^{(q)}$ *and* $\mathbf{SHS}^\top$ *commute, with probability at least* $1 - e^{-\Omega(M)}$ *over the randomness of* $\mathbf{S}$,

$$\mathbb{E}\mathcal{R}_M(\bar{\mathbf{v}}_N) \lesssim \frac{1}{M_{\mathrm{eff}}^{a-1}} + \frac{1}{N_{\mathrm{eff}}^{(a-1)/a}} + \sigma^2 + \bar{\epsilon}_3^{(M)}, \quad (1)$$

*where* $M_{\mathrm{eff}} = M$ *and* $N_{\mathrm{eff}} = N \left[ \frac{1 + \bar{\epsilon}_2^{(M)}}{(1 - \bar{\epsilon}_3^{(M)})^{\frac{1}{a}}} \right]^{-a/(a-1)}$.

Theorem 4.1 rigorously quantifies the dual impact of multiplicative quantization: the reduction of effective data size and the introduction of an additive error. Specifically, the reduction in effective data size $N_{\mathrm{eff}}$ stems from two mechanisms: the amplification of optimization noise due to quantized parameters, gradients and activations (captured by

---

$\bar{\epsilon}_2^{(M)}$), and the distortion of the feature spectrum (captured by $\bar{\epsilon}_3^{(M)}$). Meanwhile, the additive error term arises from the gap between the quantized feature space and the full-precision data space (captured by $\bar{\epsilon}_3^{(M)}$). These mechanisms align with the findings of how quantization affects learnability in Zhang et al. (2025). Notably, in the absence of quantization ($\bar{\epsilon}_i^{(M)} = 0$), Theorem 4.1 recovers the classical full-precision scaling law established in Lin et al. (2024).

A critical insight from Theorem 4.1 is that multiplicative quantization does not reduce the effective model size, which aligns with some empirical studies (Chen et al., 2025a; Sun et al., 2025). Intuitively, this invariance arises from the signal-dependent nature of multiplicative quantization, which preserves the spectral structure of the quantized feature covariance. Specifically, since the quantization error scales with the signal magnitude, it decays alongside the signal in the high-dimensional tail subspace. This ensures that the tail subspace of quantized feature spectrum decays as that of the full-precision spectrum (up to a constant scalar), thereby preserving the learnability of each parameter. Consequently, multiplicative quantization maintains $M_{\mathrm{eff}} = M$.

Our Theorem 4.1 assumes commutativity between the quantized feature covariance $\mathbf{H}_f^{(q)}$ and the sketched covariance $\mathbf{SHS}^\top$ to derive a sharper bound. A simple case which satisfies this assumption is when $\bar{\epsilon}_i = \underline{\epsilon}_i$ for any $i \in \{s, d, f, p, a, o\}$. We would also like to remark that, without this commutative condition, the quantization error may project non-trivially onto sensitive eigen-directions. While an upper bound can still be derived in general case (see Theorem C.29 for details), this misalignment introduces an additional penalty related to the condition number of $\mathbf{SHS}^\top$. To isolate the fundamental scaling behavior, we apply the commutativity assumption here.

## 4.2. Additive Quantization

In this section, we consider for any $i \in \{s, d, f, p, a, o\}$, there exist $\bar{\epsilon}_i$ such that quantization $\mathcal{Q}_i$ is $\bar{\epsilon}_i$-additive. Analogous to the multiplicative case, we define a set of compound quantization coefficients to streamline the presentation. Regarding the discrepancy between the quantized feature covariance and the original data covariance, we define:

$$\bar{\epsilon}_3^{(A)} = \frac{\bar{\epsilon}_f + \bar{\epsilon}_s(1 + \bar{\epsilon}_d p) + \bar{\epsilon}_d \frac{p}{M}}{M^{-a} + \left(\bar{\epsilon}_f + \bar{\epsilon}_s(1 + \bar{\epsilon}_d p) + \bar{\epsilon}_d \frac{p}{M}\right)}.$$

Regarding the noise amplification, we define

$$\bar{\epsilon}_2^{(A)} = \bar{\epsilon}_a + \bar{\epsilon}_o + \bar{\epsilon}_p \left[1 + p\bar{\epsilon}_d + M(\bar{\epsilon}_f + \bar{\epsilon}_s + \bar{\epsilon}_s \bar{\epsilon}_d p)\right].$$

Similar to the multiplicative case, these coefficients are small quantities that scale monotonically with the quantization severity and vanish strictly in the full-precision limit. However, we note that, unlike multiplicative coefficients

which are largely dimension-independent, $\bar{\epsilon}_2^{(A)}$ and $\bar{\epsilon}_3^{(A)}$ scale with the data dimension $p$ and model size $M$. This distinction arises because additive quantization introduces constant quantization variance that is independent across all coordinates. Moreover, since the additive quantization error constitutes a fixed floor rather than scaling with the signal, $\bar{\epsilon}_3^{(A)}$ must explicitly account for its magnitude relative to the minimum eigenvalues of the data spectrum ($M^{-a}$). With these notations, we now present the main scaling laws for low-precision training under additive quantization.

**Theorem 4.2** (Scaling law under additive quantization, an upper bound)**.** *Suppose* $\gamma < \frac{1}{\alpha \mathrm{tr}(\mathbf{H}_f^{(q)})}$. *For any* $i \in \{s, d, f, p, a, o\}$, *if there exist* $\bar{\epsilon}_i$ *such that quantization* $\mathcal{Q}_i$ *is* $\bar{\epsilon}_i$-*additive, then under Assumption 3.1, 3.3, 3.4, 3.5 and 3.6, if* $\mathbf{H}_f^{(q)}$ *and* $\mathbf{SHS}^\top$ *commute, with probability at least* $1 - e^{-\Omega(M)}$ *over the randomness of* $\mathbf{S}$,

$$\mathbb{E}\mathcal{R}_M(\bar{\mathbf{v}}_N) \lesssim \frac{1}{M_{\mathrm{eff}}^{a-1}} + \frac{1}{N_{\mathrm{eff}}^{(a-1)/a}} + \sigma^2 + \bar{\epsilon}_3^{(A)}, \qquad (2)$$

*where* $N_{\mathrm{eff}} = N \left[ \frac{1 + \bar{\epsilon}_2^{(A)}}{(1 - \bar{\epsilon}_3^{(A)})^{1/a}} \right]^{-\frac{a}{a-1}}$, *and crucially,*

$$M_{\mathrm{eff}} = M \left[ 1 + (1 + \bar{\epsilon}_2^{(A)}) \frac{(\bar{\epsilon}_3^{(A)})^2}{1 - \bar{\epsilon}_3^{(A)}} \right]^{-1/(a-1)}.$$

Theorem 4.2 characterizes a fundamental dichotomy between additive and multiplicative quantization. Unlike the multiplicative case, additive quantization not only introduces an additive error floor and reduces the effective data size, but also reduces the effective model size. The interpretation is that additive quantization injects an constant level quantization error across the entire spectrum of the quantized feature covariance $\mathbf{H}_f^{(q)}$. Consequently, this constant error overwhelms the intrinsic signal in the spectral tail and results in a flattened spectrum, rendering the tail dimensions useless for learning. Hence, the model cannot effectively leverage its full parameter count, leading to a reduction in $M_{\mathrm{eff}}$.

Our analysis further reveals that the degradation of effective model size ($M_{\mathrm{eff}}$) and effective data size ($N_{\mathrm{eff}}$) is governed by similar physical mechanisms. As derived in Theorem 4.2, both effective data size and effective model size are modulated by the same noise amplification factor $\bar{\epsilon}_2^{(A)}$ and spectral distortion factor $\bar{\epsilon}_3^{(A)}$. This mechanisms align with Theorem 4.1 under multiplicative quantization and prior work (Zhang et al., 2025). Similar to multiplicative case, under full precision, Theorem 4.2 recovers the result in Lin et al. (2024) and the commutativity condition is assumed here to isolate the fundamental scaling behavior. A general bound relaxing this assumption is in Theorem C.30.

**Connection with empirical scaling laws for low-precision training.** Our theoretical distinction between additive and multiplicative quantization provides a mechanistic explanation for the divergent empirical behaviors observed in integer versus floating-point training. Firstly, the empirical observation in Kumar et al. (2024) that integer quantization effectively reduces model capacity aligns with our additive quantization (INT-like) scaling law (Theorem 4.2). Our theory further reveals the mechanism: a constant level quantization error flattens the tail subspace, effectively rendering those dimensions uninformative and leading to the theoretically derived reduction in $M_{\mathrm{eff}}$. In contrast, Sun et al. (2025) found that floating-point quantization primarily introduces an additive loss term rather than shrinking the model size. This corroborates our multiplicative quantization (FP-like) scaling law (Theorem 4.1), which establishes that the effective model size remains invariant ($M_{\mathrm{eff}} = M$). The underlying mechanism is that multiplicative quantization preserves the relative spectral structure, ensuring the quantization error in the tail subspace scales down with the signal.

### 4.3. Lower Bound Analysis

To tighten our analysis, we establish scaling law lower bounds under multiplicative and additive quantization. In lower bound analysis, we consider low-precision well-specific model: $\mathbb{E}\left[\xi | \tilde{\mathbf{x}}^{(q)}\right] = 0$, which is extended from the standard full-precision well-specific model assumption (Zou et al., 2021; Wu et al., 2022a;b).

#### 4.3.1. MULTIPLICATIVE QUANTIZATION

We extend the compound coefficients defined in Section 4.1 to their lower-bound counterparts. The definitions utilize the minimum quantization errors $\underline{\epsilon}$. For simplicity, we provide explicit definitions for $\underline{\epsilon}_2^{(M)}, \underline{\epsilon}_3^{(M)}$ in Section E.1.

**Theorem 4.3** (Scaling law under multiplicative quantization, a lower bound)**.** *Suppose* $\gamma < 1/\tilde{\lambda}_1^{(q)}$. *For* $i \in \{d, f, s, p, a, o\}$, *if there exist constants* $(\bar{\epsilon}_i, \underline{\epsilon}_i)$ *such that* $\mathcal{Q}_i$ *is* $(\bar{\epsilon}_i, \underline{\epsilon}_i)$-*multiplicative, then under Assumption 3.1, 3.3, 3.4, 3.5 and 3.6, for sufficiently large* $N > 500$, *if* $\mathbf{H}_f^{(q)}$ *and* $\mathbf{SHS}^\top$ *are commutative, with probability at least* $1 - e^{-\Omega(M)}$ *over the randomness of* $\mathbf{S}$, *it holds*

$$\mathbb{E}\mathcal{R}_M(\bar{\mathbf{v}}_N) \gtrsim \frac{1}{M_{\mathrm{eff}}^{a-1}} + \frac{1}{N_{\mathrm{eff}}^{(a-1)/a}}$$
$$+ \sigma^2 + \left(\underline{\epsilon}_3^{(M)}\right)^2 + \frac{\underline{\epsilon}_3^{(M)}}{N}\left(1 - \bar{\epsilon}_3^{(M)}\right), \qquad (3)$$

*where* $M_{\mathrm{eff}} = M$ *and* $N_{\mathrm{eff}} = N \left[ \frac{(1 - \bar{\epsilon}_3^{(M)})(1 + \underline{\epsilon}_2^{(M)})}{(1 - \underline{\epsilon}_3^{(M)})^{\frac{1}{a}}} \right]^{-\frac{a}{a-1}}$.

Theorem 4.3 matches the form of scaling law derived in the upper bound: multiplicative quantization inherently reduces

the effective data size $N_{\text{eff}}$ via noise amplification ($\underline{\epsilon}_2^{(M)}$) and spectral distortion ($\underline{\epsilon}_3^{(M)}$), while introducing an unavoidable additive error via the gap between quantized feature space and full-precision data space ($\underline{\epsilon}_3^{(M)}$). Generally, the lower bound for the effective data size $N_{\text{eff}}$ in Theorem 4.3 does not strictly match the upper bound. This discrepancy stems from the gap between the worst-case ($\bar{\epsilon}$) and best-case ($\underline{\epsilon}$) quantization errors. Matching bounds are achieved in the sharp quantization limit where $\bar{\epsilon} \approx \underline{\epsilon}$ [3].

We note that this clean scaling law form holds in two asymptotic regimes where the interplay between $M$ and $N$ is well-separated, effectively rendering the ratio term $N/M$ of strict higher order. See Theorem E.2 for the explicit definition of these regimes. For completeness, a general population risk lower bound covering the full space of $(M, N)$ is provided in Theorem D.23 in the Appendix.

### 4.3.2. ADDITIVE QUANTIZATION

Analogous to the multiplicative case, we establish the lower bound for additive quantization by extending the compound coefficients to their lower-bound counterparts. For simplicity, we defer the definitions of $\underline{\epsilon}_2^{(A)}, \underline{\epsilon}_3^{(A)}$ to Section E.2.

**Theorem 4.4** (Scaling law under additive quantization, a lower bound). *Suppose $\gamma < \frac{1}{\bar{\lambda}_1^{(q)}}$. For $i \in \{d, f, s, p, a, o\}$, if there exist constants $(\bar{\epsilon}_i, \underline{\epsilon}_i)$ such that $\mathcal{Q}_i$ is $(\bar{\epsilon}_i, \underline{\epsilon}_i)$-additive, then under Assumption 3.1, 3.3, 3.4, 3.5 and 3.6, for sufficiently large $N > 500$, if $\mathbf{H}_f^{(q)}$ and $\mathbf{SHS}^\top$ are commutative, with probability at least $1 - e^{-\Omega(M)}$ over the randomness of $\mathbf{S}$, it holds*

$$\mathbb{E}\mathcal{R}_M(\bar{\mathbf{v}}_N) \gtrsim \frac{1}{M_{\text{eff}}^{a-1}} + \frac{1}{N_{\text{eff}}^{(a-1)/a}}$$
$$+ \sigma^2 + \left(\underline{\epsilon}_3^{(A)}\right)^2 + \frac{\epsilon_3^{(A)}}{N}\left(1 - \bar{\epsilon}_3^{(A)}\right), \quad (4)$$

*where $M_{\text{eff}} = M$, $N_{\text{eff}} = N\left[\frac{\left(1-\bar{\epsilon}_3^{(A)}\right)\left(1+\underline{\epsilon}_2^{(A)}\right)}{[1-N\gamma(\frac{1}{1-\underline{\epsilon}_3^{(A)}}-1)]^{\frac{1}{a}}}\right]^{-\frac{a}{a-1}}.$*

Theorem 4.4 rigorously validates the existence of the additive error floor (induced by the gap between quantized and low-precision space) and the reduction of effective data size (induced by noise amplification and spectral distortion), confirming the theoretical findings in upper bound analysis. Similar to Theorem 4.3, the clean scaling law in Theorem 4.4 holds in specific regimes under the condition that $\frac{1}{N\gamma} \geq \frac{1}{1-\underline{\epsilon}_3^{(A)}} - 1$. See Theorem E.4 for the explicit definition of these regimes. For completeness, a general pop-

---

³When $\bar{\epsilon} \approx \underline{\epsilon}$, our lower bound for $N_{\text{eff}}$ matches the refined upper bound established in Theorem E.1 (which incorporates the lower quantization limit $\underline{\epsilon}$ compared with Theorem 4.1).

ulation risk lower bound covering the full space of $(M, N)$ is established in Theorem D.24 in the Appendix.

We acknowledge that, unlike the upper bound, our lower bound does not explicitly exhibit the reduction in effective model size. This is a technical limitation rather than a physical one: while additive quantization error theoretically flattens the tail subspace, the induced error term becomes intricately coupled with $M$ and $N$ in the lower bound analysis (see the proof for Theorem E.4 for details). Decoupling this interaction to derive a clean scaling form that explicitly separates the shrinkage of $M_{\text{eff}}$ remains a non-trivial challenge, which we defer to future work.

**Experiments.** We generate data with polynomial spectral decay $\lambda_i \propto i^{-a}$ for $a \in \{1.5, 2.0\}$, with dimension $p = 10,000$ for $a = 1.5$ and $p = 1,000$ for $a = 2.0$. Models are trained via one-pass SGD with iterate averaging under multiplicative quantization ($\epsilon = 10^{-3}$) and additive quantization ($\epsilon = 10^{-8}$). We fit the excess risk $\mathbb{E}[\mathcal{R}_M] - \frac{1}{2}\sigma^2 = A \cdot M_{\text{eff}}^\alpha + B \cdot N_{\text{eff}}^\beta + C$. To isolate each scaling dimension, we conduct two sweeps: (i) fixing $M_{\text{eff}} = 2,000$ while varying $N_{\text{eff}} \in [10^2, 10^5]$ across 10 log-spaced values, and (ii) fixing $N_{\text{eff}} = 20,000$ while varying $M_{\text{eff}} \in [10, 200]$ across 10 log-spaced values. Each configuration is averaged over 20 seeds. Figures 1 and 2 show results. Across all configurations, the fitted exponents match theoretical predictions: for $a = 1.5$, we obtain $\alpha = -0.50$ (theory: $-\frac{1}{2}$) and $\beta = -0.34$ (theory: $-\frac{1}{3}$); for $a = 2.0$, we obtain $\alpha = -1.01$ (theory: $-1$) and $\beta = -0.50$ (theory: $-\frac{1}{2}$). All fits achieve $R^2 > 0.99$, confirming the scaling laws $\mathcal{R} \sim N_{\text{eff}}^{-(a-1)/a}$ and $\mathcal{R} \sim M_{\text{eff}}^{-(a-1)}$. These empirical results align with our theoretical scaling laws for low-precision training.

## 5. Proof Overview

In this section, we outline the proof strategy for the theoretical results established in Section 4. Moreover, we point out some key technical challenges and our strategy to address them.

**A proof roadmap.** Following Lin et al. (2024), we begin by decomposing the population risk into three components: irreducible risk, approximation error, and excess risk:

$$\mathcal{R}_M(\bar{\mathbf{v}}_N) = \underbrace{\min\mathcal{R}(\cdot)}_{\text{Irreducible}} + \underbrace{\min\mathcal{R}_M(\cdot) - \min\mathcal{R}(\cdot)}_{\text{Approx}}$$
$$+ \underbrace{\mathcal{R}_M(\bar{\mathbf{v}}_N) - \min\mathcal{R}_M(\cdot)}_{\text{Excess}}.$$

Since the quantized SGD algorithm (quantized SGD) operates within the quantized feature space rather than the exact sketch space, we further decompose the excess risk term into an *algorithm-dependent* excess risk and an *algorithm-independent* additive error, adopting the framework of

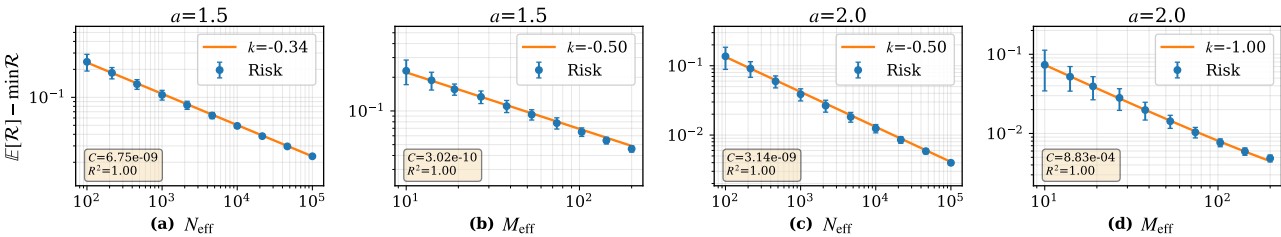

*Figure 1.* Scaling of excess risk $\mathbb{E}[\mathcal{R}] - \frac{1}{2}\sigma^2$ under multiplicative quantization with $\epsilon = 10^{-3}$, $\gamma = 0.1$, $\sigma = 1$. (a), (b): $a = 1.5$, $p = 10{,}000$; (c), (d): $a = 2.0$, $p = 1{,}000$. Panels (a), (c) fix $M_{\text{eff}}$ and vary $N_{\text{eff}}$; panels (b), (d) fix $N_{\text{eff}}$ and vary $M_{\text{eff}}$. Fitted exponents (orange curves) match theoretical predictions: $\alpha = -(a-1)$ and $\beta = -(a-1)/a$. All fits achieve $R^2 > 0.99$.

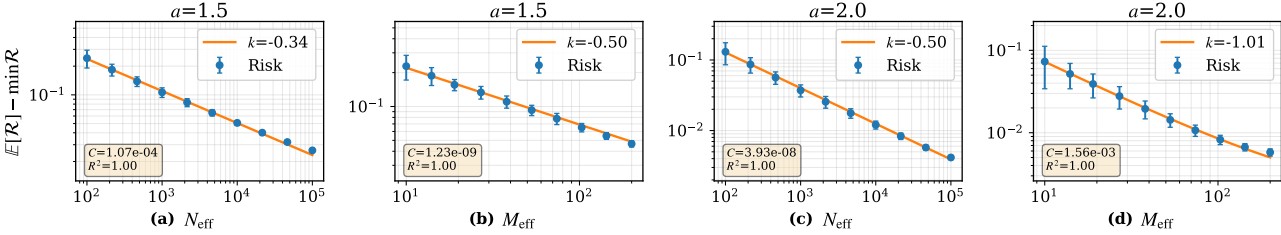

*Figure 2.* Scaling of excess risk $\mathbb{E}[\mathcal{R}] - \frac{1}{2}\sigma^2$ under additive quantization with $\epsilon = 10^{-8}$, $\gamma = 0.1$, $\sigma = 1$. (a), (b): $a = 1.5$, $p = 10{,}000$; (c), (d): $a = 2.0$, $p = 1{,}000$. Panels (a), (c) fix $M_{\text{eff}}$ and vary $N_{\text{eff}}$; panels (b), (d) fix $N_{\text{eff}}$ and vary $M_{\text{eff}}$. Fitted exponents (orange curves) match theoretical predictions: $\alpha = -(a-1)$ and $\beta = -(a-1)/a$. All fits achieve $R^2 > 0.99$.

Zhang et al. (2025):

$$\mathbb{E}\text{Excess} = \underbrace{\frac{1}{2}\langle \mathbf{SHS}^\top, \mathbb{E}[(\mathbf{v}^{(q)^*} - \overline{\mathbf{v}}_N)(\mathbf{v}^{(q)^*} - \overline{\mathbf{v}}_N)^\top]\rangle}_{R_N}$$

$$+ \text{AdditiveError}.$$

Consequently, our primary theoretical task reduces to deriving bounds for the algorithm-dependent risk $R_N$. The analysis proceeds in two logical stages: (1) we analyze the dynamics of the error covariance $\mathbb{E}[\boldsymbol{\eta}_t \boldsymbol{\eta}_t^\top]$ (where $\boldsymbol{\eta}_t = \mathbf{v}_t - \mathbf{v}^{(q)^*}$ denotes the centered SGD iterate) to establish risk bounds under general spectral conditions; and (2) we instantiate these general bounds under the polynomial spectrum assumption to explicitly derive the final scaling laws. We then point out some key technical challenges to analyze $R_N$ and present some high-level ideas to address these them.

**Challenge I: Lower bound analysis for multiplicative quantization.** Multiplicative quantization introduces noise variance proportional to the signal magnitude, creating a complex feedback loop where the noise geometry evolves with the iterate $\mathbf{v}_t$. To see this, we first rewrite (quantized SGD) using the quantization errors [4] (see Lemma C.1 for details):

$$\boldsymbol{\eta}_t = (\mathbf{I} - \gamma \tilde{\mathbf{x}}_t^{(q)} (\tilde{\mathbf{x}}_t^{(q)})^\top) \boldsymbol{\eta}_{t-1}$$
$$+ \gamma (\xi_t + \epsilon_t^{(o)} - \epsilon_t^{(a)} - (\tilde{\mathbf{x}}_t^{(q)})^\top \epsilon_{t-1}^{(p)}) \tilde{\mathbf{x}}_t^{(q)}.$$

---

[4]These quantization errors are defined as the difference of parameter, activation, output gradient and their quantized counterpart, respectively, e.g., $\epsilon_t^{(p)} = \mathcal{Q}_p(\mathbf{v}_t) - \mathbf{v}_t$.

Then in the subsequent analysis of $\mathbb{E}[\boldsymbol{\eta}_t \otimes \boldsymbol{\eta}_t]$, the second moment of parameter quantization error $\mathbb{E}[\epsilon_{t-1}^{(p)} \otimes \epsilon_{t-1}^{(p)}]$, activation quantization error $\mathbb{E}[\epsilon_t^{(a)} \otimes \epsilon_t^{(a)}]$ and output gradient quantization error $\mathbb{E}[\epsilon_t^{(o)} \otimes \epsilon_t^{(o)}]$ are all related to the magnitude of signal $\mathbb{E}[\mathbf{v}_{t-1} \otimes \mathbf{v}_{t-1}]$. While Zhang et al. (2025) successfully derived upper bounds by relaxing the quadratic forms (decoupling $\mathbb{E}[\epsilon_{t-1}^{(p)} \otimes \epsilon_{t-1}^{(p)}] \approx \epsilon_p \mathbb{E}[\mathbf{v}_{t-1} \otimes \mathbf{v}_{t-1}]$ into an iterate-dependent term $\mathbb{E}[\boldsymbol{\eta}_{t-1} \otimes \boldsymbol{\eta}_{t-1}]$ and a constant term $\mathbf{v}^{(q)^*} \otimes \mathbf{v}^{(q)^*}$), this approach is insufficient for lower bounds. The critical difficulty is the indefiniteness of the cross-term $\mathbb{E}[\boldsymbol{\eta}_{t-1}^\top \mathbf{v}^{(q)^*}]$. This negative component could theoretically cancel out the positive constant contribution $\mathbf{v}^{(q)^*} \otimes \mathbf{v}^{(q)^*}$, thereby precluding the derivation of a strictly positive noise using standard techniques.

**Our strategy.** Intuitively, the iterate's second moment $\mathbb{E}[\mathbf{v}_{t-1} \otimes \mathbf{v}_{t-1}]$ is always positive semi-definite and generally evolves from zero initialization towards the optimal covariance $\mathbf{v}^{(q)^*} \otimes \mathbf{v}^{(q)^*}$. Therefore, instead of roughly decoupling the second moment, we refine the analysis by establishing a crude lower bound for $\mathbb{E}[\mathbf{v}_{t-1} \otimes \mathbf{v}_{t-1}]$. Specifically, we achieve this by deriving a crude lower bound for $\mathbb{E}[\boldsymbol{\eta}_t \otimes \boldsymbol{\eta}_t]$ through the crude update rule: $\mathbb{E}[\boldsymbol{\eta}_t \boldsymbol{\eta}_t^\top] \succeq (\mathbf{I} - \gamma \tilde{\mathbf{x}}_t^{(q)} (\tilde{\mathbf{x}}_t^{(q)})^\top) \mathbb{E}[\boldsymbol{\eta}_{t-1} \boldsymbol{\eta}_{t-1}^\top] (\mathbf{I} - \gamma \tilde{\mathbf{x}}_t^{(q)} (\tilde{\mathbf{x}}_t^{(q)})^\top)$. We then show that (see Lemma D.2 for details)

$$\mathbb{E}[\mathbf{v}_t \mathbf{v}_t^\top] \succeq \frac{\gamma \beta}{2} (\mathbf{I} - \gamma \mathbf{H}_f^{(q)})^{2t} \mathbf{H}_f^{(q)} \|\mathbf{v}^{(q)^*}\|_{\mathbf{I}-(\mathbf{I}-\gamma\mathbf{H}_f^{(q)})^{2t}}^2$$

$$+ (\mathbf{I} - (\mathbf{I} - \gamma \mathbf{H}_f^{(q)})^t) \mathbf{v}^{(q)^*} \mathbf{v}^{(q)^{*\top}} (\mathbf{I} - (\mathbf{I} - \gamma \mathbf{H}_f^{(q)})^t).$$

With this lower bound, we can then apply standard techniques to derive risk lower bounds under general spectrum.

**Challenge II: Spectral distortion induced by quantized sketching.** Since the update rule (quantized SGD) operates strictly within the quantized feature space, our risk analysis hinges on the spectral properties of the quantized covariance $\mathbf{H}_f^{(q)}$. Unlike Lin et al. (2024) where the covariance $\mathbf{SHS}^{\top}$ preserves the polynomial decay of the data covariance $\mathbf{H}$, additive quantization fundamentally alters the polynomial spectral structure. This disruption necessitates a novel analysis to characterize the eigenvalues of $\mathbf{H}_f^{(q)}$ and derive risk bounds under this distorted spectrum.

**Our strategy.** We leverage the concentration properties of the random sketch matrix $\mathbf{S}$ to rigorously bound the eigenvalues of the quantized covariance under additive quantization, showing that the spectrum of $\mathbf{H}_f^{(q)}$ behaves as a superposition of the original power-law decay and a dimension-dependent quantization error: (see Lemma G.4 for upper bounds and Lemma G.5 for lower bounds):

$$\mu_j(\mathbf{H}_f^{(q)}) \lesssim j^{-a} + \overline{\epsilon}_f + (1 + \overline{\epsilon}_d p)\overline{\epsilon}_s + \overline{\epsilon}_d \frac{p}{M}.$$

Consequently, analyzing the variance error $\mathrm{Var} = \frac{k^*}{N} + N\gamma^2 \sum_{i>k^*} (\tilde{\lambda}_i^{(q)})^2$ necessitates a spectral decomposition that separates the constant quantization error from the decaying polynomial signal. This operation yields a penalty term scaling as $N\gamma^2\epsilon^2(M - k^*)$ [5] (see Lemma C.22 for upper bounds and Lemma D.19 for lower bounds). Physically, this term represents the cumulative noise injected into the tail subspace, providing a direct mechanism for the reduction in the effective model size $M_{\mathrm{eff}}$ characterized in Theorem 4.2.

## 6. Conclusion

We establish upper and lower bounds on the scaling laws for low-precision training under multiplicative and additive quantization within a high-dimensional sketched linear regression setting. Our theoretical analysis demonstrates that in the worst case, while both schemes reduce the effective data size and introduce an additive error, they fundamentally differ in their impact on model capacity: additive quantization reduces the effective model size, whereas multiplicative quantization preserves it. Our experiments validates our theory. These findings align with prior studies and offer actionable insights for designing low-precision training strategies.

**Limitations.** Future work may address three key limitations of this study: (1) establishing matching lower and upper bounds; (2) extending the theoretical framework to

---

[5] Here $\epsilon = \overline{\epsilon}_f + (1 + \overline{\epsilon}_d p)\overline{\epsilon}_s + \overline{\epsilon}_d \frac{p}{M}$.

non-linear models; and (3) analyzing other optimization methods.

## Acknowledgments

We would like to thank the anonymous reviewers and area chairs for their helpful comments. We acknowledge the support from NSFC 62306252, Hong Kong ECS award 27309624, Guangdong NSF 2024A1515012444, and the central fund from HKU.

## Impact Statement

This paper presents work whose goal is to advance the field of Machine Learning. There are many potential societal consequences of our work, none of which we feel must be specifically highlighted here.

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

# Appendix

# Contents

The following proof dependency graph visually encapsulates the main logical structure and organizational architecture of the theoretical results in our paper. In particular, the arrow from element $X$ to element $Y$ means the proof of $Y$ relies on $X$. To maintain visual clarity, we omit auxiliary and concentration lemmas from the graph. However, it is crucial to note that the concentration lemmas establish both upper and lower bounds for the eigen-spectra of $\mathbf{H}_f^{(q)}$ and $\mathbf{SHS}^\top$. These results facilitate the refinement of bounds from general spectra to power-law spectra and are essential for proving the upper bound lemmas (Lemmas C.20, C.22, C.28, C.26, C.21, and C.19) and the lower bound lemmas (Lemmas D.17, D.19, D.22, D.16, D.18, and D.21).

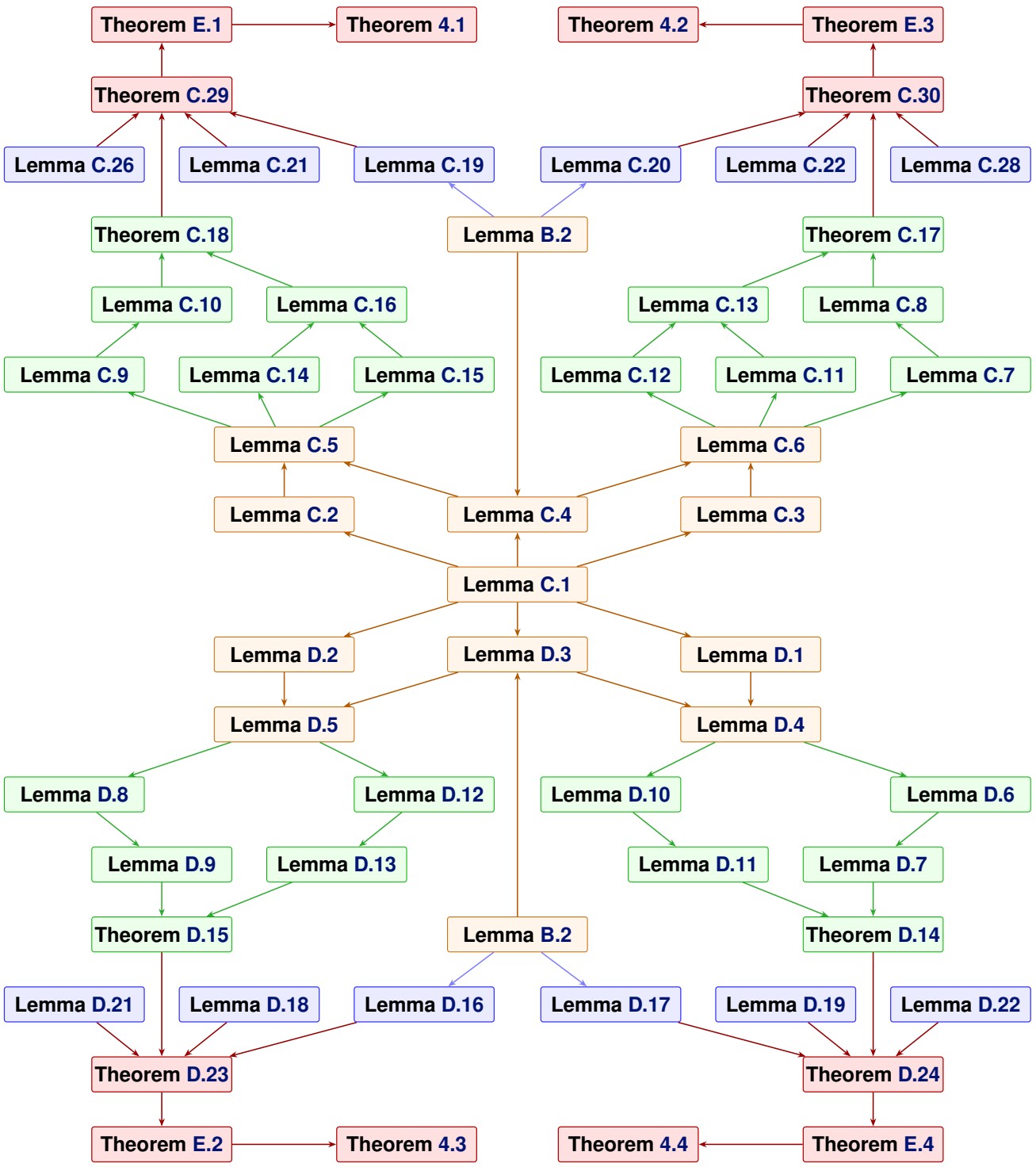

We provide detailed proofs in the Appendix. Recall the population risk

$$\mathcal{R}_M(\mathbf{v}) := \frac{1}{2}\mathbb{E}\left[(\langle \mathbf{Sx}, \mathbf{v}\rangle - y)^2\right], \quad \mathcal{R}(\mathbf{w}) := \frac{1}{2}\mathbb{E}\left[(\langle \mathbf{x}, \mathbf{w}^*\rangle - y)^2\right],$$

and the decomposition

$$\mathcal{R}_M(\overline{\mathbf{v}}_N) = \underbrace{\min \mathcal{R}(\cdot)}_{\text{Irreducible}} + \underbrace{\min \mathcal{R}_M(\cdot) - \min \mathcal{R}(\cdot)}_{\text{Approx}} + \underbrace{\mathcal{R}_M(\overline{\mathbf{v}}_N) - \min \mathcal{R}_M(\cdot)}_{\text{Excess}}.$$

We first provide bounds for the Irreducible. By the well-specified model Assumption 3.6,

$$\text{Irreducible} := \mathcal{R}(\mathbf{w}^*) = \frac{1}{2}\sigma^2. \tag{5}$$

We then provide matching bounds for Approx. As established in Lemma C.4 in (Lin et al., 2024), under Assumption 3.6, with probability at least $1 - e^{-\Omega(M)}$,

$$\mathbb{E}_{\mathbf{w}^*}\text{Approx} \asymp M^{1-a}. \tag{6}$$

In Section B-D, we will derive bounds for Excess. In Section E, we will derive scaling laws using risk bounds under general spectrum and Assumption 3.6. Unless otherwise specified, expectations are conditioned on $\mathbf{S}$ and $\mathbf{w}^*$.

## A. Omitted Proofs

### A.1. Proof for Theorem 4.1

*Proof.* The proof is completed by Theorem E.1 with $\underline{\epsilon}_i = 0$, $i = d, f, s, p, a, o$. $\qquad\square$

### A.2. Proof for Theorem 4.2

*Proof.* The proof is completed by Theorem E.3 with $\underline{\epsilon}_i = 0$, $i = d, f, s, p, a, o$. $\qquad\square$

### A.3. Proof for Theorem 4.3

*Proof.* The proof is completed by Theorem E.2. $\qquad\square$

### A.4. Proof for Theorem 4.4

*Proof.* The proof is completed by Theorem E.4. $\qquad\square$

## B. Initial Study

### B.1. Preliminary

Denote $\tilde{\mathbf{x}}^{(q)} = \mathcal{Q}_f\left(\mathcal{Q}_s(\mathbf{S})\mathcal{Q}_d(\mathbf{x}_t)\right)$, $\mathbf{H}_f^{(q)} := \mathbb{E}\left[\tilde{\mathbf{x}}^{(q)}(\tilde{\mathbf{x}}^{(q)})^\top\right]$. We first define the following linear operators as in Zou et al. (2021); Wu et al. (2022a); Zhang et al. (2025):

$$\mathcal{I} = \mathbf{I} \otimes \mathbf{I}, \quad \mathcal{M}^{(q)} = \mathbb{E}\left[\left(\tilde{\mathbf{x}}^{(q)}\right) \otimes \left(\tilde{\mathbf{x}}^{(q)}\right) \otimes \left(\tilde{\mathbf{x}}^{(q)}\right) \otimes \left(\tilde{\mathbf{x}}^{(q)}\right)\right],$$

$$\widetilde{\mathcal{M}}^{(q)} = \mathbf{H}_f^{(q)} \otimes \mathbf{H}_f^{(q)}, \quad \mathcal{T}^{(q)} = \mathbf{H}_f^{(q)} \otimes \mathbf{I} + \mathbf{I} \otimes \mathbf{H}_f^{(q)} - \gamma\mathcal{M}^{(q)},$$

$$\widetilde{\mathcal{T}}^{(q)} = \mathbf{H}_f^{(q)} \otimes \mathbf{I} + \mathbf{I} \otimes \mathbf{H}_f^{(q)} - \gamma\widetilde{\mathcal{M}}^{(q)}.$$

For a symmetric matrix $\mathbf{A}$, the above definitions result in:

$$\mathcal{I} \circ \mathbf{A} = \mathbf{A}, \quad \mathcal{M}^{(q)} \circ \mathbf{A} = \mathbb{E}\left[(\tilde{\mathbf{x}}^{(q)})^\top \mathbf{A}\tilde{\mathbf{x}}^{(q)}\tilde{\mathbf{x}}^{(q)}(\tilde{\mathbf{x}}^{(q)})^\top\right], \quad \widetilde{\mathcal{M}}^{(q)} \circ \mathbf{A} = \mathbf{H}_f^{(q)}\mathbf{A}\mathbf{H}_f^{(q)},$$

$$(\mathcal{I} - \gamma\mathcal{T}^{(q)}) \circ \mathbf{A} = \mathbb{E}\left[\left(\mathbf{I} - \gamma\tilde{\mathbf{x}}^{(q)}(\tilde{\mathbf{x}}^{(q)})^\top\right) \mathbf{A} \left(\mathbf{I} - \gamma\tilde{\mathbf{x}}^{(q)}(\tilde{\mathbf{x}}^{(q)})^\top\right)\right],$$

$$(\mathcal{I} - \gamma\widetilde{\mathcal{T}}^{(q)}) \circ \mathbf{A} = \left(\mathbf{I} - \gamma\mathbf{H}_f^{(q)}\right) \mathbf{A} \left(\mathbf{I} - \gamma\mathbf{H}_f^{(q)}\right).$$

## B.2. Excess Risk Decomposition

We first compute the global minimum of $\mathcal{R}_M(\mathbf{v})$:

$$\mathbf{v}^* := \operatorname*{argmin}_{\mathbf{v}} \mathcal{R}_M(\mathbf{v}) = \operatorname*{argmin}_{\mathbf{v}} \frac{1}{2} \mathbb{E}\left[ (\langle \mathbf{v}, \mathbf{Sx} \rangle - y)^2 \right].$$

Note that $\mathcal{R}_M(\mathbf{v})$ is a quadratic function, so its minimum is given by

$$\mathbf{v}^* = \left( \mathbf{SHS}^\top \right)^{-1} \mathbf{SHw}^*.$$

Further, we consider the global minimum of the risk on the quantized data space:

$$\mathbf{v}^{(q)^*} := \operatorname*{argmin}_{\mathbf{v}} \mathcal{R}_M^{(q)}(\mathbf{v}) = \operatorname*{argmin}_{\mathbf{v}} \frac{1}{2} \mathbb{E}\left[ \left( \langle \tilde{\mathbf{x}}^{(q)}, \mathbf{v} \rangle - \mathcal{Q}_l(y) \right)^2 \right].$$

Similarly,

$$\mathbf{v}^{(q)^*} = (\mathbf{H}_f^{(q)})^{-1} \mathbf{SHw}^*.$$

The optimality also implies the following first order optimality:

$$\mathbb{E}[(y - \langle \mathbf{v}^*, \mathbf{Sx} \rangle) \mathbf{Sx}] = 0, \quad \mathbb{E}[(\mathcal{Q}_l(y) - \langle \mathbf{v}^{(q)^*}, \tilde{\mathbf{x}}^{(q)} \rangle) \tilde{\mathbf{x}}^{(q)}] = 0. \tag{7}$$

**Lemma B.1** (Excess risk decomposition). *Under Assumption 3.1 and Assumption 3.3,*

$$\begin{aligned}
\mathbb{E}\left[ \mathcal{R}_M(\bar{\mathbf{v}}_N) - \mathcal{R}_M(\mathbf{v}^*) \right] =& \frac{1}{2} \left\langle \mathbf{H}_f^{(q)}, \mathbb{E}\left[ (\mathbf{v}^{(q)^*} - \bar{\mathbf{v}}_N) \otimes (\mathbf{v}^{(q)^*} - \bar{\mathbf{v}}_N) \right] \right\rangle \\
&+ \frac{1}{2} \left\langle \mathbf{SHS}^\top, (\mathbf{v}^{(q)^*} - \mathbf{v}^*) \otimes (\mathbf{v}^{(q)^*} - \mathbf{v}^*) \right\rangle \\
&+ \frac{1}{2} \mathbb{E}\left[ \langle \mathbf{v}^{(q)^*}, \mathbf{Sx} - \tilde{\mathbf{x}}^{(q)} \rangle^2 \right] \\
&- \frac{1}{2} \mathbb{E}\left[ \langle \bar{\mathbf{v}}_N, \mathbf{Sx} - \tilde{\mathbf{x}}^{(q)} \rangle^2 \right].
\end{aligned}$$

*Proof.* By definition,

$$\begin{aligned}
\mathbb{E}\left[ \mathcal{R}_M(\bar{\mathbf{v}}_N) - \mathcal{R}_M(\mathbf{v}^*) \right] =& \frac{1}{2} \mathbb{E}\left[ (\langle \mathbf{Sx}, \bar{\mathbf{v}}_N \rangle - y)^2 \right] - \frac{1}{2} \mathbb{E}\left[ (\langle \mathbf{Sx}, \mathbf{v}^* \rangle - y)^2 \right] \\
=& \underbrace{\frac{1}{2} \mathbb{E}\left[ (y - \langle \bar{\mathbf{v}}_N, \mathbf{Sx} \rangle)^2 \right] - \frac{1}{2} \mathbb{E}\left[ (\mathcal{Q}_l(y) - \langle \bar{\mathbf{v}}_N, \tilde{\mathbf{x}}^{(q)} \rangle)^2 \right]}_{R_1} \\
&+ \underbrace{\frac{1}{2} \mathbb{E}\left[ (\mathcal{Q}_l(y) - \langle \bar{\mathbf{v}}_N, \tilde{\mathbf{x}}^{(q)} \rangle)^2 \right] - \frac{1}{2} \mathbb{E}\left[ (\mathcal{Q}_l(y) - \langle \mathbf{v}^{(q)^*}, \tilde{\mathbf{x}}^{(q)} \rangle)^2 \right]}_{R_2} \\
&+ \underbrace{\frac{1}{2} \mathbb{E}\left[ (\mathcal{Q}_l(y) - \langle \mathbf{v}^{(q)^*}, \tilde{\mathbf{x}}^{(q)} \rangle)^2 \right] - \frac{1}{2} \mathbb{E}\left[ \left( y - \langle \mathbf{v}^{(q)^*}, \mathbf{Sx} \rangle \right)^2 \right]}_{R_3} \\
&+ \underbrace{\frac{1}{2} \mathbb{E}\left[ \left( y - \langle \mathbf{v}^{(q)^*}, \mathbf{Sx} \rangle \right)^2 \right] - \frac{1}{2} \mathbb{E}\left[ (y - \langle \mathbf{v}^*, \mathbf{Sx} \rangle)^2 \right]}_{R_4}.
\end{aligned}$$

We would like to remark that the quantization operations in $\mathcal{Q}_l(y)$ and $\tilde{\mathbf{x}}^{(q)}$ introduced in excess risk decomposition are independent of those quantization operators introduced in the training stage, i.e., $\bar{\mathbf{v}}_N$. We then deal with each term respectively. For $R_1$,

$$\begin{aligned}
&\frac{1}{2} \mathbb{E}\left[ (y - \langle \bar{\mathbf{v}}_N, \mathbf{Sx} \rangle)^2 \right] - \frac{1}{2} \mathbb{E}\left[ (\mathcal{Q}_l(y) - \langle \bar{\mathbf{v}}_N, \tilde{\mathbf{x}}^{(q)} \rangle)^2 \right] \\
=& \frac{1}{2} \mathbb{E}\left[ \left( y - \mathcal{Q}_l(y) - \langle \bar{\mathbf{v}}_N, \mathbf{Sx} - \tilde{\mathbf{x}}^{(q)} \rangle \right) \cdot \left( y + \mathcal{Q}_l(y) - \langle \bar{\mathbf{v}}_N, \mathbf{Sx} + \tilde{\mathbf{x}}^{(q)} \rangle \right) \right] \\
=& \frac{1}{2} \mathbb{E}\left[ (y - \mathcal{Q}_l(y))(y + \mathcal{Q}_l(y)) \right] - \frac{1}{2} \mathbb{E}\left[ \langle \bar{\mathbf{v}}_N, \mathbf{Sx} - \tilde{\mathbf{x}}^{(q)} \rangle^2 \right],
\end{aligned} \tag{8}$$

where the last equality uses the unbiased quantization Assumption 3.1. For $R_2$,

$$
\begin{aligned}
&\frac{1}{2}\mathbb{E}\left[(\mathcal{Q}_l(y) - \langle \overline{\mathbf{v}}_N, \tilde{\mathbf{x}}^{(q)}\rangle)^2\right] - \frac{1}{2}\mathbb{E}\left[(\mathcal{Q}_l(y) - \langle \mathbf{v}^{(q)^*}, \tilde{\mathbf{x}}^{(q)}\rangle)^2\right] \\
=&\frac{1}{2}\mathbb{E}\left[\langle \mathbf{v}^{(q)^*} - \overline{\mathbf{v}}_N, \tilde{\mathbf{x}}^{(q)}\rangle\left(2\mathcal{Q}_l(y) - \langle \mathbf{v}^{(q)^*} + \overline{\mathbf{v}}_N, \tilde{\mathbf{x}}^{(q)}\rangle\right)\right] \\
=&\frac{1}{2}\mathbb{E}\left[\langle \mathbf{v}^{(q)^*} - \overline{\mathbf{v}}_N, \tilde{\mathbf{x}}^{(q)}\rangle^2\right] \\
=&\frac{1}{2}\left\langle \mathbf{H}_f^{(q)}, \mathbb{E}\left[(\mathbf{v}^{(q)^*} - \overline{\mathbf{v}}_N) \otimes (\mathbf{v}^{(q)^*} - \overline{\mathbf{v}}_N)\right]\right\rangle,
\end{aligned}
\tag{9}
$$

where the second equality holds by the optimality (7). For $R_3$,

$$
\begin{aligned}
&\frac{1}{2}\mathbb{E}\left[(\mathcal{Q}_l(y) - \langle \mathbf{v}^{(q)^*}, \tilde{\mathbf{x}}^{(q)}\rangle)^2\right] - \frac{1}{2}\mathbb{E}\left[\left(y - \langle \mathbf{v}^{(q)^*}, \mathbf{Sx}\rangle\right)^2\right] \\
=&\frac{1}{2}\mathbb{E}\left[\left(\mathcal{Q}_l(y) - y - \langle \mathbf{v}^{(q)^*}, \tilde{\mathbf{x}}^{(q)} - \mathbf{Sx}\rangle\right)\left(\mathcal{Q}_l(y) + y - \langle \mathbf{v}^{(q)^*}, \tilde{\mathbf{x}}^{(q)} + \mathbf{Sx}\rangle\right)\right] \\
=&\frac{1}{2}\mathbb{E}\left[(\mathcal{Q}_l(y) - y)(y + \mathcal{Q}_l(y))\right] + \frac{1}{2}\mathbb{E}\left[\langle \mathbf{v}^{(q)^*}, \mathbf{Sx} - \tilde{\mathbf{x}}^{(q)}\rangle^2\right],
\end{aligned}
\tag{10}
$$

where the last equality holds by unbiased quantization Assumption 3.1. For $R_4$,

$$
\begin{aligned}
&\frac{1}{2}\mathbb{E}\left[\left(y - \langle \mathbf{v}^{(q)^*}, \mathbf{Sx}\rangle\right)^2\right] - \frac{1}{2}\mathbb{E}\left[(y - \langle \mathbf{v}^*, \mathbf{Sx}\rangle)^2\right] \\
=&\frac{1}{2}\mathbb{E}\left[\left(\langle \mathbf{v}^* - \mathbf{v}^{(q)^*}, \mathbf{Sx}\rangle\right)\left(2y - \langle \mathbf{v}^* + \mathbf{v}^{(q)^*}, \mathbf{Sx}\rangle\right)\right] \\
=&\frac{1}{2}\mathbb{E}\left[\langle \mathbf{v}^* - \mathbf{v}^{(q)^*}, \mathbf{Sx}\rangle^2\right] \\
=&\frac{1}{2}\left\langle \mathbf{SHS}^\top, (\mathbf{v}^{(q)^*} - \mathbf{v}^*) \otimes (\mathbf{v}^{(q)^*} - \mathbf{v}^*)\right\rangle,
\end{aligned}
\tag{11}
$$

where the second equality holds by the optimality (7).

Combining (8), (9), (10) and (11), it holds

$$
\begin{aligned}
\mathbb{E}\left[\mathcal{R}_M(\overline{\mathbf{v}}_N) - \mathcal{R}_M(\mathbf{v}^*)\right] =&\frac{1}{2}\left\langle \mathbf{H}_f^{(q)}, \mathbb{E}\left[(\mathbf{v}^{(q)^*} - \overline{\mathbf{v}}_N) \otimes (\mathbf{v}^{(q)^*} - \overline{\mathbf{v}}_N)\right]\right\rangle \\
&+\frac{1}{2}\left\langle \mathbf{SHS}^\top, (\mathbf{v}^{(q)^*} - \mathbf{v}^*) \otimes (\mathbf{v}^{(q)^*} - \mathbf{v}^*)\right\rangle \\
&+\frac{1}{2}\mathbb{E}\left[\langle \mathbf{v}^{(q)^*}, \mathbf{Sx} - \tilde{\mathbf{x}}^{(q)}\rangle^2\right] \\
&-\frac{1}{2}\mathbb{E}\left[\langle \overline{\mathbf{v}}_N, \mathbf{Sx} - \tilde{\mathbf{x}}^{(q)}\rangle^2\right].
\end{aligned}
$$

$\square$

**Lemma B.2** (Refined excess risk decomposition). *Under Assumption 3.1, Assumption 3.3, if the stepsize $\gamma < \frac{1}{\tilde{\lambda}_1^{(q)}}$, then*

$$
\begin{aligned}
\mathbb{E}\left[\mathcal{R}_M(\overline{\mathbf{v}}_N) - \mathcal{R}_M(\mathbf{v}^*)\right] =&\underbrace{\frac{1}{2}\left\langle \mathbf{SHS}^\top, \mathbb{E}\left[(\mathbf{v}^{(q)^*} - \overline{\mathbf{v}}_N) \otimes (\mathbf{v}^{(q)^*} - \overline{\mathbf{v}}_N)\right]\right\rangle}_{R_N} \\
&+\frac{1}{2}\left\langle \mathbf{SHS}^\top, (\mathbf{v}^{(q)^*} - \mathbf{v}^*) \otimes (\mathbf{v}^{(q)^*} - \mathbf{v}^*)\right\rangle \\
&+\left(\mathbf{v}^{(q)^*}\right)^\top \frac{1}{N\gamma}\left[\mathbf{I} - \left(\mathbf{I} - \gamma\mathbf{H}_f^{(q)}\right)^N\right]\left(\mathbf{H}_f^{(q)}\right)^{-1}\left(\mathbf{H}_f^{(q)} - \mathbf{SHS}^\top\right)\mathbf{v}^{(q)^*}.
\end{aligned}
$$

*Proof.* By Lemma B.1,

$$
\begin{aligned}
\mathbb{E}\left[\mathcal{R}_M(\overline{\mathbf{v}}_N) - \mathcal{R}_M(\mathbf{v}^*)\right] =& \frac{1}{2}\left\langle \mathbf{H}_f^{(q)}, \mathbb{E}\left[(\mathbf{v}^{(q)^*} - \overline{\mathbf{v}}_N) \otimes (\mathbf{v}^{(q)^*} - \overline{\mathbf{v}}_N)\right]\right\rangle \\
&+ \frac{1}{2}\left\langle \mathbf{SHS}^\top, (\mathbf{v}^{(q)^*} - \mathbf{v}^*) \otimes (\mathbf{v}^{(q)^*} - \mathbf{v}^*)\right\rangle \\
&+ \frac{1}{2}\mathbb{E}\left[\langle \mathbf{v}^{(q)^*}, \mathbf{Sx} - \tilde{\mathbf{x}}^{(q)}\rangle^2\right] \\
&- \frac{1}{2}\mathbb{E}\left[\langle \overline{\mathbf{v}}_N, \mathbf{Sx} - \tilde{\mathbf{x}}^{(q)}\rangle^2\right].
\end{aligned}
$$

Recall that $\overline{\mathbf{v}}_N = \overline{\mathbf{v}}_N - \mathbf{v}^{(q)^*} + \mathbf{v}^{(q)^*}$, it holds

$$
\begin{aligned}
\mathbb{E}\left[\langle \overline{\mathbf{v}}_N, \mathbf{Sx} - \tilde{\mathbf{x}}^{(q)}\rangle^2\right] =& \mathbb{E}\left[\overline{\mathbf{v}}_N^\top\left(\mathbf{H}_f^{(q)} - \mathbf{SHS}^\top\right)\overline{\mathbf{v}}_N\right] \\
=& \mathbb{E}\left[\left(\overline{\mathbf{v}}_N - \mathbf{v}^{(q)^*}\right)^\top\left(\mathbf{H}_f^{(q)} - \mathbf{SHS}^\top\right)\left(\overline{\mathbf{v}}_N - \mathbf{v}^{(q)^*}\right)\right] \\
&+ \mathbb{E}\left[\left(\mathbf{v}^{(q)^*}\right)^\top\left(\mathbf{H}_f^{(q)} - \mathbf{SHS}^\top\right)\mathbf{v}^{(q)^*}\right] \\
&+ 2\mathbb{E}\left[\left(\overline{\mathbf{v}}_N - \mathbf{v}^{(q)^*}\right)^\top\left(\mathbf{H}_f^{(q)} - \mathbf{SHS}^\top\right)\mathbf{v}^{(q)^*}\right].
\end{aligned}
$$

Hence,

$$
\begin{aligned}
\mathbb{E}\left[\mathcal{R}_M(\overline{\mathbf{v}}_N) - \mathcal{R}_M(\mathbf{v}^*)\right] =& \frac{1}{2}\left\langle \mathbf{SHS}^\top, \mathbb{E}\left[(\mathbf{v}^{(q)^*} - \overline{\mathbf{v}}_N) \otimes (\mathbf{v}^{(q)^*} - \overline{\mathbf{v}}_N)\right]\right\rangle \\
&+ \frac{1}{2}\left\langle \mathbf{SHS}^\top, (\mathbf{v}^{(q)^*} - \mathbf{v}^*) \otimes (\mathbf{v}^{(q)^*} - \mathbf{v}^*)\right\rangle \\
&- \mathbb{E}\left[\left(\overline{\mathbf{v}}_N - \mathbf{v}^{(q)^*}\right)^\top\left(\mathbf{H}_f^{(q)} - \mathbf{SHS}^\top\right)\mathbf{v}^{(q)^*}\right].
\end{aligned}
\tag{12}
$$

Denote $\boldsymbol{\eta}_t = \mathbf{v}_t - \mathbf{v}^{(q)^*}$, then by Lemma C.1,

$$
\boldsymbol{\eta}_t = \left(\mathbf{I} - \gamma\tilde{\mathbf{x}}_t^{(q)}(\tilde{\mathbf{x}}_t^{(q)})^\top\right)\boldsymbol{\eta}_{t-1} + \gamma\left(\xi_t + \epsilon_t^{(o)} - \epsilon_t^{(a)} - (\tilde{\mathbf{x}}_t^{(q)})^\top\epsilon_{t-1}^{(p)}\right)\tilde{\mathbf{x}}_t^{(q)},
$$

where

$$
\begin{aligned}
\epsilon_t^{(o)} :=& \mathcal{Q}_o\left(\mathcal{Q}_l(y_t) - \mathcal{Q}_a\left((\tilde{\mathbf{x}}_t^{(q)})^\top\mathcal{Q}_p(\mathbf{v}_{t-1})\right)\right) - \left[\mathcal{Q}_l(y_t) - \mathcal{Q}_a\left((\tilde{\mathbf{x}}_t^{(q)})^\top\mathcal{Q}_p(\mathbf{v}_{t-1})\right)\right], \\
\epsilon_t^{(a)} :=& \mathcal{Q}_a\left((\tilde{\mathbf{x}}_t^{(q)})^\top\mathcal{Q}_p(\mathbf{v}_{t-1})\right) - (\tilde{\mathbf{x}}_t^{(q)})^\top\mathcal{Q}_p(\mathbf{v}_{t-1}), \\
\epsilon_{t-1}^{(p)} :=& \mathcal{Q}_p(\mathbf{v}_{t-1}) - \mathbf{v}_{t-1}, \\
\xi_t :=& \mathcal{Q}_l(y_t) - (\tilde{\mathbf{x}}_t^{(q)})^\top\mathbf{v}^{(q)^*}.
\end{aligned}
$$

It follows by the unbiased quantization Assumption 3.1 and the optimality (7) that

$$
\mathbb{E}\left[\boldsymbol{\eta}_t\right] = \mathbb{E}\left[\mathbb{E}\left[\boldsymbol{\eta}_t|\boldsymbol{\eta}_{t-1}\right]\right] = \mathbb{E}\left[\left(\mathbf{I} - \gamma\mathbf{H}_f^{(q)}\right)\boldsymbol{\eta}_{t-1}\right] = \left(\mathbf{I} - \gamma\mathbf{H}_f^{(q)}\right)\mathbb{E}\left[\boldsymbol{\eta}_{t-1}\right] = \left(\mathbf{I} - \gamma\mathbf{H}_f^{(q)}\right)^t\boldsymbol{\eta}_0.
\tag{13}
$$

Hence,

$$
\mathbb{E}\left[\overline{\mathbf{v}}_N - \mathbf{v}^{(q)^*}\right]^\top \left(\mathbf{H}_f^{(q)} - \mathbf{SHS}^\top\right)\mathbf{v}^{(q)^*}
$$

$$
= \frac{1}{N}\sum_{t=0}^{N-1}\mathbb{E}[\boldsymbol{\eta}_t]^\top\left(\mathbf{H}_f^{(q)} - \mathbf{SHS}^\top\right)\mathbf{v}^{(q)^*}
$$

$$
= \left[\frac{1}{N}\sum_{t=0}^{N-1}\left(\mathbf{I} - \gamma\mathbf{H}_f^{(q)}\right)^t\boldsymbol{\eta}_0\right]^\top\left(\mathbf{H}_f^{(q)} - \mathbf{SHS}^\top\right)\mathbf{v}^{(q)^*} \tag{14}
$$

$$
= -\left(\mathbf{v}^{(q)^*}\right)^\top\frac{1}{N}\sum_{t=0}^{N-1}\left(\mathbf{I} - \gamma\mathbf{H}_f^{(q)}\right)^t\left(\mathbf{H}_f^{(q)} - \mathbf{SHS}^\top\right)\mathbf{v}^{(q)^*}
$$

$$
= -\left(\mathbf{v}^{(q)^*}\right)^\top\frac{1}{N\gamma}\left[\mathbf{I} - \left(\mathbf{I} - \gamma\mathbf{H}_f^{(q)}\right)^N\right]\left(\mathbf{H}_f^{(q)}\right)^{-1}\left(\mathbf{H}_f^{(q)} - \mathbf{SHS}^\top\right)\mathbf{v}^{(q)^*}.
$$

Together with (12) and (14), we have

$$
\mathbb{E}\left[\mathcal{R}_M(\overline{\mathbf{v}}_N) - \mathcal{R}_M(\mathbf{v}^*)\right] = \frac{1}{2}\left\langle\mathbf{SHS}^\top, \mathbb{E}\left[(\mathbf{v}^{(q)^*} - \overline{\mathbf{v}}_N)\otimes(\mathbf{v}^{(q)^*} - \overline{\mathbf{v}}_N)\right]\right\rangle
$$

$$
+ \frac{1}{2}\left\langle\mathbf{SHS}^\top, (\mathbf{v}^{(q)^*} - \mathbf{v}^*)\otimes(\mathbf{v}^{(q)^*} - \mathbf{v}^*)\right\rangle
$$

$$
+ \left(\mathbf{v}^{(q)^*}\right)^\top\frac{1}{N\gamma}\left[\mathbf{I} - \left(\mathbf{I} - \gamma\mathbf{H}_f^{(q)}\right)^N\right]\left(\mathbf{H}_f^{(q)}\right)^{-1}\left(\mathbf{H}_f^{(q)} - \mathbf{SHS}^\top\right)\mathbf{v}^{(q)^*}.
$$

$\square$

In the following part, we first establish upper bounds in Section C and then establish lower bounds in Section D. Specifically, we first analyze the algorithm-dependent excess risk

$$
R_N = \frac{1}{2}\left\langle\mathbf{SHS}^\top, \mathbb{E}\left[(\mathbf{v}^{(q)^*} - \overline{\mathbf{v}}_N)\otimes(\mathbf{v}^{(q)^*} - \overline{\mathbf{v}}_N)\right]\right\rangle,
$$

and then analyze the remaining algorithm-independent additive error

$$
\frac{1}{2}\left\langle\mathbf{SHS}^\top, (\mathbf{v}^{(q)^*} - \mathbf{v}^*)\otimes(\mathbf{v}^{(q)^*} - \mathbf{v}^*)\right\rangle + \left(\mathbf{v}^{(q)^*}\right)^\top\frac{1}{N\gamma}\left[\mathbf{I} - \left(\mathbf{I} - \gamma\mathbf{H}_f^{(q)}\right)^N\right]\left(\mathbf{H}_f^{(q)}\right)^{-1}\left(\mathbf{H}_f^{(q)} - \mathbf{SHS}^\top\right)\mathbf{v}^{(q)^*}.
$$

At last, we reorganize the population risk bounds to derive scaling laws in Section E.

## C. Upper Bound Analysis

We first derive the propagation of the deviation $\boldsymbol{\eta}_t = \mathbf{v}_t - \mathbf{v}^{(q)^*}$.

### C.1. Update Rule

**Lemma C.1.**

$$
\boldsymbol{\eta}_t = \left(\mathbf{I} - \gamma\tilde{\mathbf{x}}_t^{(q)}(\tilde{\mathbf{x}}_t^{(q)})^\top\right)\boldsymbol{\eta}_{t-1} + \gamma\left(\xi_t + \epsilon_t^{(o)} - \epsilon_t^{(a)} - (\tilde{\mathbf{x}}_t^{(q)})^\top\epsilon_{t-1}^{(p)}\right)\tilde{\mathbf{x}}_t^{(q)},
$$

*where*

$$
\epsilon_t^{(o)} := \mathcal{Q}_o\left(\mathcal{Q}_l(y_t) - \mathcal{Q}_a\left((\tilde{\mathbf{x}}_t^{(q)})^\top\mathcal{Q}_p(\mathbf{v}_{t-1})\right)\right) - \left[\mathcal{Q}_l(y_t) - \mathcal{Q}_a\left((\tilde{\mathbf{x}}_t^{(q)})^\top\mathcal{Q}_p(\mathbf{v}_{t-1})\right)\right],
$$

$$
\epsilon_t^{(a)} := \mathcal{Q}_a\left((\tilde{\mathbf{x}}_t^{(q)})^\top\mathcal{Q}_p(\mathbf{v}_{t-1})\right) - (\tilde{\mathbf{x}}_t^{(q)})^\top\mathcal{Q}_p(\mathbf{v}_{t-1}),
$$

$$
\epsilon_{t-1}^{(p)} := \mathcal{Q}_p(\mathbf{v}_{t-1}) - \mathbf{v}_{t-1},
$$

$$
\xi_t := \mathcal{Q}_l(y_t) - (\tilde{\mathbf{x}}_t^{(q)})^\top\mathbf{v}^{(q)^*}.
$$

*Proof.* By (quantized SGD),

$$\mathbf{v}_t = \mathbf{v}_{t-1} + \gamma \mathcal{Q}_o \left( \mathcal{Q}_l(y_t) - \mathcal{Q}_a \left( \mathcal{Q}_f \left( \mathcal{Q}_s(\mathbf{S}) \mathcal{Q}_d(\mathbf{x}_t) \right)^\top \mathcal{Q}_p(\mathbf{v}_{t-1}) \right) \right) \mathcal{Q}_f \left( \mathcal{Q}_s(\mathbf{S}) \mathcal{Q}_d(\mathbf{x}_t) \right).$$

Then we have

$$\boldsymbol{\eta}_t = \boldsymbol{\eta}_{t-1} + \gamma \mathcal{Q}_o \left( \mathcal{Q}_l(y_t) - \mathcal{Q}_a \left( \mathcal{Q}_f \left( \mathcal{Q}_s(\mathbf{S}) \mathcal{Q}_d(\mathbf{x}_t) \right)^\top \mathcal{Q}_p(\mathbf{v}_{t-1}) \right) \right) \mathcal{Q}_f \left( \mathcal{Q}_s(\mathbf{S}) \mathcal{Q}_d(\mathbf{x}_t) \right).$$

Denote $\tilde{\mathbf{x}}_t^{(q)} = \mathcal{Q}_f \left( \mathcal{Q}_s(\mathbf{S}) \mathcal{Q}_d(\mathbf{x}_t) \right)$. We then introduce quantization errors to better characterize each quantization operation $\mathcal{Q}(\cdot)$. In particular, define quantization errors:

$$\epsilon_t^{(o)} := \mathcal{Q}_o \left( \mathcal{Q}_l(y_t) - \mathcal{Q}_a \left( (\tilde{\mathbf{x}}_t^{(q)})^\top \mathcal{Q}_p(\mathbf{v}_{t-1}) \right) \right) - \left[ \mathcal{Q}_l(y_t) - \mathcal{Q}_a \left( (\tilde{\mathbf{x}}_t^{(q)})^\top \mathcal{Q}_p(\mathbf{v}_{t-1}) \right) \right],$$

$$\epsilon_t^{(a)} := \mathcal{Q}_a \left( (\tilde{\mathbf{x}}_t^{(q)})^\top \mathcal{Q}_p(\mathbf{v}_{t-1}) \right) - (\tilde{\mathbf{x}}_t^{(q)})^\top \mathcal{Q}_p(\mathbf{v}_{t-1}),$$

$$\boldsymbol{\epsilon}_{t-1}^{(p)} := \mathcal{Q}_p(\mathbf{v}_{t-1}) - \mathbf{v}_{t-1},$$

$$\xi_t := \mathcal{Q}_l(y_t) - (\tilde{\mathbf{x}}_t^{(q)})^\top \mathbf{v}^{(q)*}.$$

Then the update rule for the parameter deviation can be expressed as:

$$\begin{aligned}
\boldsymbol{\eta}_t &= \boldsymbol{\eta}_{t-1} + \gamma \mathcal{Q}_o \left( \mathcal{Q}_l(y_t) - \mathcal{Q}_a \left( (\tilde{\mathbf{x}}_t^{(q)})^\top \mathcal{Q}_p(\mathbf{v}_{t-1}) \right) \right) \tilde{\mathbf{x}}_t^{(q)} \\
&= \boldsymbol{\eta}_{t-1} + \gamma \left( \mathcal{Q}_l(y_t) - \mathcal{Q}_a \left( (\tilde{\mathbf{x}}_t^{(q)})^\top \mathcal{Q}_p(\mathbf{v}_{t-1}) \right) + \epsilon_t^{(o)} \right) \tilde{\mathbf{x}}_t^{(q)} \\
&= \boldsymbol{\eta}_{t-1} + \gamma \left( \mathcal{Q}_l(y_t) - \left( (\tilde{\mathbf{x}}_t^{(q)})^\top \mathcal{Q}_p(\mathbf{v}_{t-1}) + \epsilon_t^{(a)} \right) + \epsilon_t^{(o)} \right) \tilde{\mathbf{x}}_t^{(q)} \\
&= \boldsymbol{\eta}_{t-1} + \gamma \left( \mathcal{Q}_l(y_t) - \left( (\tilde{\mathbf{x}}_t^{(q)})^\top (\mathbf{v}_{t-1} + \boldsymbol{\epsilon}_{t-1}^{(p)} - \mathbf{v}^{(q)*} + \mathbf{v}^{(q)*}) \right) + \epsilon_t^{(o)} - \epsilon_t^{(a)} \right) \tilde{\mathbf{x}}_t^{(q)} \\
&= \boldsymbol{\eta}_{t-1} - \gamma \tilde{\mathbf{x}}_t^{(q)} (\tilde{\mathbf{x}}_t^{(q)})^\top \boldsymbol{\eta}_{t-1} + \gamma \left( \xi_t + \epsilon_t^{(o)} - \epsilon_t^{(a)} - (\tilde{\mathbf{x}}_t^{(q)})^\top \boldsymbol{\epsilon}_{t-1}^{(p)} \right) \tilde{\mathbf{x}}_t^{(q)}.
\end{aligned}$$

$\square$

We then derive the propagation of $\mathbb{E} \left[ \boldsymbol{\eta}_t \otimes \boldsymbol{\eta}_t \right]$. By Lemma C.1,

$$\boldsymbol{\eta}_t = \left( \mathbf{I} - \gamma \tilde{\mathbf{x}}_t^{(q)} (\tilde{\mathbf{x}}_t^{(q)})^\top \right) \boldsymbol{\eta}_{t-1} + \gamma \left( \xi_t + \epsilon_t^{(o)} - \epsilon_t^{(a)} - (\tilde{\mathbf{x}}_t^{(q)})^\top \boldsymbol{\epsilon}_{t-1}^{(p)} \right) \tilde{\mathbf{x}}_t^{(q)}.$$

Denote

$$\boldsymbol{\eta}_t^{\text{bias}} = \left( \mathbf{I} - \gamma \tilde{\mathbf{x}}_t^{(q)} (\tilde{\mathbf{x}}_t^{(q)})^\top \right) \boldsymbol{\eta}_{t-1}^{\text{bias}}, \quad \boldsymbol{\eta}_0^{\text{bias}} = \boldsymbol{\eta}_0,$$

$$\boldsymbol{\eta}_t^{\text{var}} = \left( \mathbf{I} - \gamma \tilde{\mathbf{x}}_t^{(q)} (\tilde{\mathbf{x}}_t^{(q)})^\top \right) \boldsymbol{\eta}_{t-1}^{\text{var}} + \gamma \left( \xi_t + \epsilon_t^{(o)} - \epsilon_t^{(a)} - (\tilde{\mathbf{x}}_t^{(q)})^\top \boldsymbol{\epsilon}_{t-1}^{(p)} \right) \tilde{\mathbf{x}}_t^{(q)}, \quad \boldsymbol{\eta}_0^{\text{var}} = \mathbf{0}.$$

Obviously, it holds

$$\boldsymbol{\eta}_t = \boldsymbol{\eta}_t^{\text{var}} + \boldsymbol{\eta}_t^{\text{bias}},$$

and

$$\mathbb{E} \left[ \boldsymbol{\eta}_t \otimes \boldsymbol{\eta}_t \right] \preceq 2 \left( \underbrace{\mathbb{E} \left[ \boldsymbol{\eta}_t^{\text{bias}} \otimes \boldsymbol{\eta}_t^{\text{bias}} \right]}_{\mathbf{B}_t} + \underbrace{\mathbb{E} \left[ \boldsymbol{\eta}_t^{\text{var}} \otimes \boldsymbol{\eta}_t^{\text{var}} \right]}_{\mathbf{C}_t} \right). \tag{15}$$

Regarding $\mathbf{B}_t$, we have

$$\mathbf{B}_t = \mathbb{E} \left[ \left( \mathbf{I} - \gamma \tilde{\mathbf{x}}_t^{(q)} (\tilde{\mathbf{x}}_t^{(q)})^\top \right) \mathbf{B}_{t-1} \left( \mathbf{I} - \gamma \tilde{\mathbf{x}}_t^{(q)} (\tilde{\mathbf{x}}_t^{(q)})^\top \right) \right]. \tag{16}$$

Regarding $\mathbf{C}_t$, by the unbiased quantization Assumption 3.1 and $\boldsymbol{\eta}_0^{\text{var}} = \mathbf{0}$, it holds

$$\mathbf{C}_t = \mathbb{E} \left[ \left( \mathbf{I} - \gamma \tilde{\mathbf{x}}_t^{(q)} (\tilde{\mathbf{x}}_t^{(q)})^\top \right) \mathbf{C}_{t-1} \left( \mathbf{I} - \gamma \tilde{\mathbf{x}}_t^{(q)} (\tilde{\mathbf{x}}_t^{(q)})^\top \right) \right] + \boldsymbol{\Sigma}_t, \tag{17}$$

where

$$\boldsymbol{\Sigma}_t = \gamma^2 \mathbb{E}\left[\left(\xi_t + \epsilon_t^{(o)} - \epsilon_t^{(a)} - (\tilde{\mathbf{x}}_t^{(q)})^\top \boldsymbol{\epsilon}_{t-1}^{(p)}\right)^2 \tilde{\mathbf{x}}_t^{(q)}(\tilde{\mathbf{x}}_t^{(q)})^\top\right]. \tag{18}$$

Further, it holds

$$\begin{aligned}
\boldsymbol{\Sigma}_t =& \gamma^2 \mathbb{E}\left[\left(\xi_t + \epsilon_t^{(o)} - \epsilon_t^{(a)} - (\tilde{\mathbf{x}}_t^{(q)})^\top \boldsymbol{\epsilon}_{t-1}^{(p)}\right)^2 \tilde{\mathbf{x}}_t^{(q)}(\tilde{\mathbf{x}}_t^{(q)})^\top\right] \\
=& \gamma^2 \mathbb{E}\left[\left(\xi_t^2 + \epsilon_t^{(o)2} + \epsilon_t^{(a)2} + (\tilde{\mathbf{x}}_t^{(q)})^\top \boldsymbol{\epsilon}_{t-1}^{(p)} \boldsymbol{\epsilon}_{t-1}^{(p)\top} \tilde{\mathbf{x}}_t^{(q)}\right) \tilde{\mathbf{x}}_t^{(q)}(\tilde{\mathbf{x}}_t^{(q)})^\top\right] \\
=& \gamma^2 \mathbb{E}\left[\xi_t^2 \tilde{\mathbf{x}}_t^{(q)}(\tilde{\mathbf{x}}_t^{(q)})^\top\right] + \gamma^2 \mathbb{E}\left[\epsilon_t^{(o)2} \tilde{\mathbf{x}}_t^{(q)}(\tilde{\mathbf{x}}_t^{(q)})^\top\right] \\
&+ \gamma^2 \mathbb{E}\left[\epsilon_t^{(a)2} \tilde{\mathbf{x}}_t^{(q)}(\tilde{\mathbf{x}}_t^{(q)})^\top\right] + \gamma^2 \mathbb{E}\left[(\tilde{\mathbf{x}}_t^{(q)})^\top \boldsymbol{\epsilon}_{t-1}^{(p)} \boldsymbol{\epsilon}_{t-1}^{(p)\top} \tilde{\mathbf{x}}_t^{(q)} \tilde{\mathbf{x}}_t^{(q)}(\tilde{\mathbf{x}}_t^{(q)})^\top\right],
\end{aligned} \tag{19}$$

where the second equality holds by the unbiased quantization Assumption 3.1. We then summarize the update rule for $\mathbb{E}\left[\boldsymbol{\eta}_t \otimes \boldsymbol{\eta}_t\right]$ as follows. Consider $\mathbf{B}_t$ and $\mathbf{C}_t$ defined in (15).

**Lemma C.2** (Update rule under multiplicative quantization, an upper bound). *If there exist $\bar{\epsilon}_p, \bar{\epsilon}_a$ and $\bar{\epsilon}_o$ such that for any $i \in \{p, a, o\}$, quantization $\mathcal{Q}_i$ is $\bar{\epsilon}_i$-multiplicative, then under Assumption 3.1, 3.3, 3.4 and 3.5,*

$$\begin{aligned}
\mathbf{C}_t \preceq& \mathbb{E}\left[\left(\mathbf{I} - \gamma \tilde{\mathbf{x}}_t^{(q)}(\tilde{\mathbf{x}}_t^{(q)})^\top\right) \mathbf{C}_{t-1} \left(\mathbf{I} - \gamma \tilde{\mathbf{x}}_t^{(q)}(\tilde{\mathbf{x}}_t^{(q)})^\top\right)\right] \\
&+ 2\gamma^2 \left(2\bar{\epsilon}_o + (2\bar{\epsilon}_o + 1)\left[2(1 + \bar{\epsilon}_p)\bar{\epsilon}_a + 2\bar{\epsilon}_p\right]\right) \mathbb{E}\left[\tilde{\mathbf{x}}_t^{(q)}(\tilde{\mathbf{x}}_t^{(q)})^\top \left(\mathbf{B}_{t-1} + \mathbf{C}_{t-1}\right) \tilde{\mathbf{x}}_t^{(q)}(\tilde{\mathbf{x}}_t^{(q)})^\top\right] \\
&+ \gamma^2(2\bar{\epsilon}_o + 1)\left[2\bar{\epsilon}_p + 2(1 + \bar{\epsilon}_p)\bar{\epsilon}_a\right] \alpha \operatorname{tr}\left(\mathbf{H}_f^{(q)} \mathbf{v}^{(q)*} \mathbf{v}^{(q)*\top}\right) \mathbf{H}_f^{(q)} + \gamma^2(2\bar{\epsilon}_o + 1)\bar{\sigma}^2 \mathbf{H}_f^{(q)}, \\
\mathbf{B}_t =& \mathbb{E}\left[\left(\mathbf{I} - \gamma \tilde{\mathbf{x}}_t^{(q)}(\tilde{\mathbf{x}}_t^{(q)})^\top\right) \mathbf{B}_{t-1} \left(\mathbf{I} - \gamma \tilde{\mathbf{x}}_t^{(q)}(\tilde{\mathbf{x}}_t^{(q)})^\top\right)\right].
\end{aligned}$$

*Proof.* By (16) and (17), the proof focuses on dealing with each term of $\boldsymbol{\Sigma}_t$ in (19). Firstly, by Assumption 3.5,

$$\mathbb{E}\left[\xi_t^2 \tilde{\mathbf{x}}_t^{(q)}(\tilde{\mathbf{x}}_t^{(q)})^\top\right] \preceq \bar{\sigma}^2 \mathbf{H}_f^{(q)}. \tag{20}$$

Secondly, by the definition of multiplicative quantization,

$$\begin{aligned}
&\mathbb{E}\left[(\tilde{\mathbf{x}}_t^{(q)})^\top \boldsymbol{\epsilon}_{t-1}^{(p)} \boldsymbol{\epsilon}_{t-1}^{(p)\top} \tilde{\mathbf{x}}_t^{(q)} \tilde{\mathbf{x}}_t^{(q)}(\tilde{\mathbf{x}}_t^{(q)})^\top\right] \\
\preceq& \bar{\epsilon}_p \mathbb{E}\left[(\tilde{\mathbf{x}}_t^{(q)})^\top \mathbf{v}_{t-1} \mathbf{v}_{t-1}^\top \tilde{\mathbf{x}}_t^{(q)} \tilde{\mathbf{x}}_t^{(q)}(\tilde{\mathbf{x}}_t^{(q)})^\top\right] \\
\preceq& 2\bar{\epsilon}_p \mathbb{E}\left[(\tilde{\mathbf{x}}_t^{(q)})^\top \left(\boldsymbol{\eta}_{t-1} \boldsymbol{\eta}_{t-1}^\top + \mathbf{v}^{(q)*} \mathbf{v}^{(q)*\top}\right) \tilde{\mathbf{x}}_t^{(q)} \tilde{\mathbf{x}}_t^{(q)}(\tilde{\mathbf{x}}_t^{(q)})^\top\right],
\end{aligned} \tag{21}$$

where the last inequality holds by the fact that: for two vectors $\mathbf{u}$ and $\mathbf{v}$, $(\mathbf{u} + \mathbf{v})(\mathbf{u} + \mathbf{v})^\top \preceq 2\left(\mathbf{u}\mathbf{u}^\top + \mathbf{v}\mathbf{v}^\top\right)$. Thirdly, by the definition of multiplicative quantization,

$$\begin{aligned}
&\mathbb{E}\left[\epsilon_t^{(a)2} \tilde{\mathbf{x}}_t^{(q)}(\tilde{\mathbf{x}}_t^{(q)})^\top\right] \\
\preceq& \bar{\epsilon}_a \mathbb{E}\left[(\tilde{\mathbf{x}}_t^{(q)})^\top \mathcal{Q}_p(\mathbf{v}_{t-1}) \mathcal{Q}_p(\mathbf{v}_{t-1})^\top \tilde{\mathbf{x}}_t^{(q)} \tilde{\mathbf{x}}_t^{(q)}(\tilde{\mathbf{x}}_t^{(q)})^\top\right] \\
=& \bar{\epsilon}_a \mathbb{E}\left[(\tilde{\mathbf{x}}_t^{(q)})^\top \boldsymbol{\epsilon}_{t-1}^{(p)} \boldsymbol{\epsilon}_{t-1}^{(p)\top} \tilde{\mathbf{x}}_t^{(q)} \tilde{\mathbf{x}}_t^{(q)}(\tilde{\mathbf{x}}_t^{(q)})^\top\right] \\
&+ \bar{\epsilon}_a \mathbb{E}\left[(\tilde{\mathbf{x}}_t^{(q)})^\top \mathbf{v}_{t-1} \mathbf{v}_{t-1}^\top \tilde{\mathbf{x}}_t^{(q)} \tilde{\mathbf{x}}_t^{(q)}(\tilde{\mathbf{x}}_t^{(q)})^\top\right] \\
\preceq& (1 + \bar{\epsilon}_p)\bar{\epsilon}_a \mathbb{E}\left[(\tilde{\mathbf{x}}_t^{(q)})^\top \mathbf{v}_{t-1} \mathbf{v}_{t-1}^\top \tilde{\mathbf{x}}_t^{(q)} \tilde{\mathbf{x}}_t^{(q)}(\tilde{\mathbf{x}}_t^{(q)})^\top\right] \\
\preceq& 2(1 + \bar{\epsilon}_p)\bar{\epsilon}_a \mathbb{E}\left[(\tilde{\mathbf{x}}_t^{(q)})^\top \left(\boldsymbol{\eta}_{t-1} \boldsymbol{\eta}_{t-1}^\top + \mathbf{v}^{(q)*} \mathbf{v}^{(q)*\top}\right) \tilde{\mathbf{x}}_t^{(q)} \tilde{\mathbf{x}}_t^{(q)}(\tilde{\mathbf{x}}_t^{(q)})^\top\right].
\end{aligned} \tag{22}$$

Fourthly, by the definition of multiplicative quantization,

$$
\mathbb{E}\left[\epsilon_t^{(o)^2}\tilde{\mathbf{x}}_t^{(q)}(\tilde{\mathbf{x}}_t^{(q)})^\top\right]
$$

$$
\preceq\bar{\epsilon}_o\mathbb{E}\left[\left[\mathcal{Q}_l(y_t) - \mathcal{Q}_a\left((\tilde{\mathbf{x}}_t^{(q)})^\top\mathcal{Q}_p(\mathbf{v}_{t-1})\right)\right]^2\tilde{\mathbf{x}}_t^{(q)}(\tilde{\mathbf{x}}_t^{(q)})^\top\right]
$$

$$
=\bar{\epsilon}_o\mathbb{E}\left[\left[\mathcal{Q}_l(y_t) - (\tilde{\mathbf{x}}_t^{(q)})^\top\mathcal{Q}_p(\mathbf{v}_{t-1}) - \epsilon_t^{(a)}\right]^2\tilde{\mathbf{x}}_t^{(q)}(\tilde{\mathbf{x}}_t^{(q)})^\top\right]
$$

$$
=\bar{\epsilon}_o\mathbb{E}\left[\left[\mathcal{Q}_l(y_t) - (\tilde{\mathbf{x}}_t^{(q)})^\top\mathbf{v}_{t-1} - (\tilde{\mathbf{x}}_t^{(q)})^\top\boldsymbol{\epsilon}_{t-1}^{(p)} - \epsilon_t^{(a)}\right]^2\tilde{\mathbf{x}}_t^{(q)}(\tilde{\mathbf{x}}_t^{(q)})^\top\right] \tag{23}
$$

$$
=\bar{\epsilon}_o\mathbb{E}\left[\left[\xi_t - (\tilde{\mathbf{x}}_t^{(q)})^\top\boldsymbol{\eta}_{t-1} - (\tilde{\mathbf{x}}_t^{(q)})^\top\boldsymbol{\epsilon}_{t-1}^{(p)} - \epsilon_t^{(a)}\right]^2\tilde{\mathbf{x}}_t^{(q)}(\tilde{\mathbf{x}}_t^{(q)})^\top\right]
$$

$$
\preceq 2\bar{\epsilon}_o\mathbb{E}\left[\left[\xi_t^2 + (\tilde{\mathbf{x}}_t^{(q)})^\top\boldsymbol{\epsilon}_{t-1}^{(p)}\boldsymbol{\epsilon}_{t-1}^{(p)^\top}\tilde{\mathbf{x}}_t^{(q)} + \epsilon_t^{(a)^2} + (\tilde{\mathbf{x}}_t^{(q)})^\top\boldsymbol{\eta}_{t-1}\boldsymbol{\eta}_{t-1}^\top\tilde{\mathbf{x}}_t^{(q)}\right]\tilde{\mathbf{x}}_t^{(q)}(\tilde{\mathbf{x}}_t^{(q)})^\top\right].
$$

Note that by Assumption 3.4,

$$
\mathbb{E}\left[(\tilde{\mathbf{x}}_t^{(q)})^\top\mathbf{v}^{(q)^*}\mathbf{v}^{(q)^{*\top}}\tilde{\mathbf{x}}_t^{(q)}\tilde{\mathbf{x}}_t^{(q)}(\tilde{\mathbf{x}}_t^{(q)})^\top\right] \preceq \alpha\mathrm{tr}\left(\mathbf{H}_f^{(q)}\mathbf{v}^{(q)^*}\mathbf{v}^{(q)^{*\top}}\right)\mathbf{H}_f^{(q)}. \tag{24}
$$

Therefore, together with (19), (20), (21), (22), (23) and (24), it holds

$$
\boldsymbol{\Sigma}_t/\gamma^2 \preceq (2\bar{\epsilon}_o + 1)\bar{\sigma}^2\mathbf{H}_f^{(q)}
$$

$$
+ (2\bar{\epsilon}_o + 1)\left[2\bar{\epsilon}_p + 2(1 + \bar{\epsilon}_p)\bar{\epsilon}_a\right]\alpha\mathrm{tr}\left(\mathbf{H}_f^{(q)}\mathbf{v}^{(q)^*}\mathbf{v}^{(q)^{*\top}}\right)\mathbf{H}_f^{(q)}
$$

$$
+ (2\bar{\epsilon}_o + (2\bar{\epsilon}_o + 1)\left[2(1 + \bar{\epsilon}_p)\bar{\epsilon}_a + 2\bar{\epsilon}_p\right])\mathbb{E}\left[\tilde{\mathbf{x}}_t^{(q)}(\tilde{\mathbf{x}}_t^{(q)})^\top\boldsymbol{\eta}_{t-1}\boldsymbol{\eta}_{t-1}^\top\tilde{\mathbf{x}}_t^{(q)}(\tilde{\mathbf{x}}_t^{(q)})^\top\right].
$$

The proof is completed by (15): $\mathbb{E}\left[\boldsymbol{\eta}_t \otimes \boldsymbol{\eta}_t\right] \preceq 2\left(\mathbf{B}_t + \mathbf{C}_t\right)$. $\qquad\square$

**Lemma C.3** (Update rule under general quantization, an upper bound)**.** *Under Assumption 3.1, 3.3, 3.4 and 3.5, it holds*

$$
\mathbf{B}_t = \mathbb{E}\left[\left(\mathbf{I} - \gamma\tilde{\mathbf{x}}_t^{(q)}(\tilde{\mathbf{x}}_t^{(q)})^\top\right)\mathbf{B}_{t-1}\left(\mathbf{I} - \gamma\tilde{\mathbf{x}}_t^{(q)}(\tilde{\mathbf{x}}_t^{(q)})^\top\right)\right],
$$

$$
\mathbf{C}_t = \mathbb{E}\left[\left(\mathbf{I} - \gamma\tilde{\mathbf{x}}_t^{(q)}(\tilde{\mathbf{x}}_t^{(q)})^\top\right)\mathbf{C}_{t-1}\left(\mathbf{I} - \gamma\tilde{\mathbf{x}}_t^{(q)}(\tilde{\mathbf{x}}_t^{(q)})^\top\right)\right]
$$

$$
+ \gamma^2\left[\bar{\sigma}^2 + \sup_t \alpha\mathrm{tr}\left(\mathbf{H}_f^{(q)}\mathbb{E}\left[\boldsymbol{\epsilon}_{t-1}^{(p)}\boldsymbol{\epsilon}_{t-1}^{(p)^\top}\right]\right) + \sup_t\left(\mathbb{E}\left[\epsilon_t^{(a)^2}\Big|a_t\right] + \mathbb{E}\left[\epsilon_t^{(o)^2}\Big|o_t\right]\right)\right]\mathbf{H}_f^{(q)}.
$$

*Proof.* By (16) and (17), the proof focuses on dealing with each term of $\boldsymbol{\Sigma}_t$ in (19). Firstly, by Assumption 3.4,

$$
\mathbb{E}\left[(\tilde{\mathbf{x}}_t^{(q)})^\top\boldsymbol{\epsilon}_{t-1}^{(p)}\boldsymbol{\epsilon}_{t-1}^{(p)^\top}\tilde{\mathbf{x}}_t^{(q)}\tilde{\mathbf{x}}_t^{(q)}(\tilde{\mathbf{x}}_t^{(q)})^\top\right] \preceq \alpha\mathrm{tr}\left(\mathbf{H}_f^{(q)}\mathbb{E}\left[\boldsymbol{\epsilon}_{t-1}^{(p)}\boldsymbol{\epsilon}_{t-1}^{(p)^\top}\right]\right)\mathbf{H}_f^{(q)}
$$

$$
\preceq \sup_t \alpha\mathrm{tr}\left(\mathbf{H}_f^{(q)}\mathbb{E}\left[\boldsymbol{\epsilon}_{t-1}^{(p)}\boldsymbol{\epsilon}_{t-1}^{(p)^\top}\right]\right)\mathbf{H}_f^{(q)}. \tag{25}
$$

Secondly, denote

$$
a_t = (\tilde{\mathbf{x}}_t^{(q)})^\top\mathcal{Q}_p(\mathbf{v}_{t-1}), \quad o_t = \mathcal{Q}_l(y_t) - \mathcal{Q}_a\left((\tilde{\mathbf{x}}_t^{(q)})^\top\mathcal{Q}_p(\mathbf{v}_{t-1})\right),
$$

then

$$
\mathbb{E}\left[\left(\epsilon_t^{(a)^2} + \epsilon_t^{(o)^2}\right)\tilde{\mathbf{x}}_t^{(q)}(\tilde{\mathbf{x}}_t^{(q)})^\top\right] \preceq \sup_t\left(\mathbb{E}\left[\epsilon_t^{(a)^2}\Big|a_t\right] + \mathbb{E}\left[\epsilon_t^{(o)^2}\Big|o_t\right]\right)\mathbf{H}_f^{(q)}. \tag{26}
$$

Therefore, together with (19), (20), (25) and (26), it holds,

$$
\boldsymbol{\Sigma}_t/\gamma^2 \preceq \left[\bar{\sigma}^2 + \sup_t \alpha\mathrm{tr}\left(\mathbf{H}_f^{(q)}\mathbb{E}\left[\boldsymbol{\epsilon}_{t-1}^{(p)}\boldsymbol{\epsilon}_{t-1}^{(p)^\top}\right]\right) + \sup_t\left(\mathbb{E}\left[\epsilon_t^{(a)^2}\Big|a_t\right] + \mathbb{E}\left[\epsilon_t^{(o)^2}\Big|o_t\right]\right)\right]\mathbf{H}_f^{(q)}.
$$

$\qquad\square$

## C.2. Bias-Variance Decomposition

Recall

$$
\begin{aligned}
R_N =& \frac{1}{2} \left\langle \mathbf{SHS}^\top, \mathbb{E}\left[\overline{\boldsymbol{\eta}}_N \otimes \overline{\boldsymbol{\eta}}_N\right] \right\rangle \\
\leq& \mu_{\max}\left((\mathbf{H}_f^{(q)})^{-1}\mathbf{SHS}^\top\right) \underbrace{\frac{1}{2}\left\langle \mathbf{H}_f^{(q)}, \mathbb{E}\left[\overline{\boldsymbol{\eta}}_N \otimes \overline{\boldsymbol{\eta}}_N\right]\right\rangle}_{R_N^{(0)}}.
\end{aligned}
\tag{27}
$$

We perform bias-variance decomposition for multiplicative and general cases respectively, to analyze $R_N^{(0)}$. Firstly, we express $\overline{\boldsymbol{\eta}}_N \otimes \overline{\boldsymbol{\eta}}_N$ into the sum of $\boldsymbol{\eta}_t$.

**Lemma C.4.** *Under Assumption 3.1 and Assumption 3.3, it holds*

$$
R_N^{(0)} \leq \frac{1}{N^2} \cdot \sum_{t=0}^{N-1}\sum_{k=t}^{N-1} \left\langle (\mathbf{I}-\gamma\mathbf{H}_f^{(q)})^{k-t}\mathbf{H}_f^{(q)}, \mathbb{E}[\boldsymbol{\eta}_t \otimes \boldsymbol{\eta}_t] \right\rangle.
$$

*Proof.* By definition $\overline{\boldsymbol{\eta}}_N = \frac{1}{N}\sum_{t=0}^{N-1}\boldsymbol{\eta}_t$, we have

$$
\begin{aligned}
\mathbb{E}[\bar{\boldsymbol{\eta}}_N \otimes \bar{\boldsymbol{\eta}}_N] =& \frac{1}{N^2} \cdot \left( \sum_{0 \leq k \leq t \leq N-1} \mathbb{E}[\boldsymbol{\eta}_t \otimes \boldsymbol{\eta}_k] + \sum_{0 \leq t < k \leq N-1} \mathbb{E}[\boldsymbol{\eta}_t \otimes \boldsymbol{\eta}_k] \right) \\
\preceq& \frac{1}{N^2} \cdot \left( \sum_{0 \leq k \leq t \leq N-1} \mathbb{E}\left[\mathbb{E}[\boldsymbol{\eta}_t \otimes \boldsymbol{\eta}_k | \boldsymbol{\eta}_k]\right] + \sum_{0 \leq t \leq k \leq N-1} \mathbb{E}\left[\mathbb{E}[\boldsymbol{\eta}_t \otimes \boldsymbol{\eta}_k | \boldsymbol{\eta}_t]\right] \right).
\end{aligned}
\tag{28}
$$

By the unbiased quantization Assumption 3.1 and the optimality (7), together with the update rule Lemma C.1, it holds

$$
\mathbb{E}\left[\boldsymbol{\eta}_t | \boldsymbol{\eta}_{t-1}\right] = \left(\mathbf{I} - \gamma\mathbf{H}_f^{(q)}\right)\boldsymbol{\eta}_{t-1}.
\tag{29}
$$

Therefore, by (28) and (29),

$$
\begin{aligned}
&\mathbb{E}[\bar{\boldsymbol{\eta}}_N \otimes \bar{\boldsymbol{\eta}}_N] \\
\preceq& \frac{1}{N^2} \cdot \left( \sum_{0 \leq k \leq t \leq N-1} \mathbb{E}\left[\mathbb{E}[\boldsymbol{\eta}_t \otimes \boldsymbol{\eta}_k | \boldsymbol{\eta}_k]\right] + \sum_{0 \leq t \leq k \leq N-1} \mathbb{E}\left[\mathbb{E}[\boldsymbol{\eta}_t \otimes \boldsymbol{\eta}_k | \boldsymbol{\eta}_t]\right] \right) \\
=& \frac{1}{N^2} \cdot \left( \sum_{0 \leq k \leq t \leq N-1} (\mathbf{I}-\gamma\mathbf{H}_f^{(q)})^{t-k}\mathbb{E}[\boldsymbol{\eta}_k \otimes \boldsymbol{\eta}_k] + \sum_{0 \leq t \leq k \leq N-1} \mathbb{E}[\boldsymbol{\eta}_t \otimes \boldsymbol{\eta}_t](\mathbf{I}-\gamma\mathbf{H}_f^{(q)})^{k-t} \right) \\
=& \frac{1}{N^2} \cdot \sum_{t=0}^{N-1}\sum_{k=t}^{N-1} \left((\mathbf{I}-\gamma\mathbf{H}_f^{(q)})^{k-t}\mathbb{E}[\boldsymbol{\eta}_t \otimes \boldsymbol{\eta}_t] + \mathbb{E}[\boldsymbol{\eta}_t \otimes \boldsymbol{\eta}_t](\mathbf{I}-\gamma\mathbf{H}_f^{(q)})^{k-t} \right).
\end{aligned}
\tag{30}
$$

Applying (30) into $R_N$, we have

$$
\begin{aligned}
R_N^{(0)} =& \frac{1}{2}\langle \mathbf{H}_f^{(q)}, \mathbb{E}[\bar{\boldsymbol{\eta}}_N \otimes \bar{\boldsymbol{\eta}}_N]\rangle \\
\leq& \frac{1}{2N^2} \cdot \sum_{t=0}^{N-1}\sum_{k=t}^{N-1} \left\langle \mathbf{H}_f^{(q)}, (\mathbf{I}-\gamma\mathbf{H}_f^{(q)})^{k-t}\mathbb{E}[\boldsymbol{\eta}_t \otimes \boldsymbol{\eta}_t] + \mathbb{E}[\boldsymbol{\eta}_t \otimes \boldsymbol{\eta}_t](\mathbf{I}-\gamma\mathbf{H}_f^{(q)})^{k-t} \right\rangle \\
=& \frac{1}{N^2} \cdot \sum_{t=0}^{N-1}\sum_{k=t}^{N-1} \left\langle (\mathbf{I}-\gamma\mathbf{H}_f^{(q)})^{k-t}\mathbf{H}_f^{(q)}, \mathbb{E}[\boldsymbol{\eta}_t \otimes \boldsymbol{\eta}_t] \right\rangle,
\end{aligned}
$$

where the last equality holds since $\mathbf{H}_f^{(q)}$ and $(\mathbf{I}-\gamma\mathbf{H}_f^{(q)})^{k-t}$ commute. This completes the proof. $\qquad \square$

**Lemma C.5** (Bias-variance decomposition under multiplicative quantization, an upper bound). *If there exist $\overline{\epsilon}_p, \overline{\epsilon}_a$ and $\overline{\epsilon}_o$ such that for any $i \in \{p, a, o\}$, quantization $\mathcal{Q}_i$ is $\overline{\epsilon}_i$-multiplicative, then under Assumption 3.1, 3.4, 3.3 and 3.5, if $\gamma < \frac{1}{(1+\widetilde{\epsilon})\alpha \mathrm{tr}(\mathbf{H}_f^{(q)})}$, it holds*

$$R_N^{(0)}/2 \le \frac{1}{N^2} \cdot \sum_{t=0}^{N-1} \sum_{k=t}^{N-1} \left\langle (\mathbf{I} - \gamma \mathbf{H}_f^{(q)})^{k-t} \mathbf{H}_f^{(q)}, \mathbf{B}_t^{(M)} + \mathbf{C}_t^{(M)} \right\rangle,$$

*where*

$$\mathbf{B}_t^{(M)} = (\mathcal{I} - \gamma \mathcal{T}^{(q)} + \widetilde{\epsilon}\gamma^2 \mathcal{M}^{(q)}) \circ \mathbf{B}_{t-1}^{(M)}, \quad \mathbf{B}_0^{(M)} = \mathbb{E}\left[\boldsymbol{\eta}_0 \otimes \boldsymbol{\eta}_0\right],$$
$$\mathbf{C}_t^{(M)} = (\mathcal{I} - \gamma \mathcal{T}^{(q)} + \widetilde{\epsilon}\gamma^2 \mathcal{M}^{(q)}) \circ \mathbf{C}_{t-1}^{(M)} + \gamma^2 \sigma_M^2 \mathbf{H}_f^{(q)}, \quad \mathbf{C}_0^{(M)} = \mathbf{0},$$

*with*

$$\widetilde{\epsilon} = 4\overline{\epsilon}_o + 2(2\overline{\epsilon}_o + 1)\left[2(1+\overline{\epsilon}_p)\overline{\epsilon}_a + 2\overline{\epsilon}_p\right],$$
$$\sigma_M^2 = (2\overline{\epsilon}_o + 1)\overline{\sigma}^2 + (2\overline{\epsilon}_o + 1)\left[2\overline{\epsilon}_p + 2(1+\overline{\epsilon}_p)\overline{\epsilon}_a\right]\alpha \mathrm{tr}\left(\mathbf{H}_f^{(q)} \mathbf{v}^{(q)^*} \mathbf{v}^{(q)^{*\top}}\right).$$

*Proof.* By (15), Lemma C.2 and Lemma C.4, this lemma can be proved by induction. $\square$

**Lemma C.6** (Bias-variance decomposition under general quantization, an upper bound). *Under Assumption 3.1, 3.3, 3.4 and 3.5, if $\gamma < \frac{1}{\alpha \mathrm{tr}(\mathbf{H}_f^{(q)})}$, it holds*

$$R_N^{(0)}/2 \le \frac{1}{N^2} \cdot \sum_{t=0}^{N-1} \sum_{k=t}^{N-1} \left\langle (\mathbf{I} - \gamma \mathbf{H}_f^{(q)})^{k-t} \mathbf{H}_f^{(q)}, \mathbf{B}_t + \mathbf{C}_t \right\rangle,$$

*where*

$$\mathbf{B}_t = (\mathcal{I} - \gamma \mathcal{T}^{(q)}) \circ \mathbf{B}_{t-1}, \quad \mathbf{B}_0 = \mathbb{E}\left[\boldsymbol{\eta}_0 \otimes \boldsymbol{\eta}_0\right],$$
$$\mathbf{C}_t = (\mathcal{I} - \gamma \mathcal{T}^{(q)}) \circ \mathbf{C}_{t-1} + \gamma^2 \sigma_G^2 \mathbf{H}_f^{(q)}, \quad \mathbf{C}_0 = \mathbf{0},$$

*with*

$$\sigma_G^2 = \overline{\sigma}^2 + \sup_t \alpha \mathrm{tr}\left(\mathbf{H}_f^{(q)} \mathbb{E}\left[\boldsymbol{\epsilon}_{t-1}^{(p)} \boldsymbol{\epsilon}_{t-1}^{(p)^{\top}}\right]\right) + \sup_t \left(\mathbb{E}\left[\epsilon_t^{(a)^2} \middle| a_t\right] + \mathbb{E}\left[\epsilon_t^{(o)^2} \middle| o_t\right]\right).$$

*Proof.* By (15), Lemma C.3 and Lemma C.4, this lemma can be proved by induction. $\square$

### C.3. Variance Upper Bounds

In this section, we derive upper bounds for $\frac{1}{N^2} \sum_{t=0}^{N-1} \sum_{k=t}^{N-1} \left\langle (\mathbf{I} - \gamma \mathbf{H}_f^{(q)})^{k-t} \mathbf{H}_f^{(q)}, \mathbf{C}_t^{(M)} \right\rangle$ and $\frac{1}{N^2} \sum_{t=0}^{N-1} \sum_{k=t}^{N-1} \left\langle (\mathbf{I} - \gamma \mathbf{H}_f^{(q)})^{k-t} \mathbf{H}_f^{(q)}, \mathbf{C}_t \right\rangle$.

#### C.3.1. GENERAL QUANTIZATION

**Lemma C.7** (A crude upper bound of variance under general quantization). *Under Assumption 3.3, Assumption 3.4, if $\gamma < \frac{1}{\alpha \mathrm{tr}(\mathbf{H}_f^{(q)})}$,*

$$\mathbf{C}_t \preceq \frac{\gamma \sigma_G^2}{1 - \gamma \alpha \mathrm{tr}(\mathbf{H}_f^{(q)})} \mathbf{I}.$$

*Proof.* We prove by induction. For $t = 0$, we have $\mathbf{C}_0 = \mathbf{0} \preceq \frac{\gamma\sigma_G^2}{1-\gamma\alpha\mathrm{tr}\left(\mathbf{H}_f^{(q)}\right)}\mathbf{I}$. We then assume that $\mathbf{C}_{t-1} \preceq \frac{\gamma\sigma_G^2}{1-\gamma\alpha\mathrm{tr}\left(\mathbf{H}_f^{(q)}\right)}\mathbf{I}$, and exam $\mathbf{C}_t$:

$$
\begin{aligned}
\mathbf{C}_t &= (\mathcal{I} - \gamma\mathcal{T}^{(q)}) \circ \mathbf{C}_{t-1} + \gamma^2\sigma_G^2\mathbf{H}_f^{(q)} \\
&= \left(\mathcal{I} - \gamma\mathbf{H}_f^{(q)} \otimes \mathbf{I} - \gamma\mathbf{I} \otimes \mathbf{H}_f^{(q)}\right) \circ \mathbf{C}_{t-1} + \gamma^2\mathcal{M}^{(q)} \circ \mathbf{C}_{t-1} + \gamma^2\sigma_G^2\mathbf{H}_f^{(q)} \\
&\preceq \frac{\gamma\sigma_G^2}{1-\gamma\alpha\mathrm{tr}\left(\mathbf{H}_f^{(q)}\right)} \cdot \left(\mathbf{I} - 2\gamma\mathbf{H}_f^{(q)}\right) + \frac{\gamma^2\gamma\sigma_G^2\alpha\mathrm{tr}\left(\mathbf{H}_f^{(q)}\right)}{1-\gamma\alpha\mathrm{tr}\left(\mathbf{H}_f^{(q)}\right)}\mathbf{H}_f^{(q)} + \gamma^2\sigma_G^2\mathbf{H}_f^{(q)} \\
&= \frac{\gamma\sigma_G^2}{1-\gamma\alpha\mathrm{tr}\left(\mathbf{H}_f^{(q)}\right)} \cdot \mathbf{I} - (2\gamma^2 - \gamma^2) \cdot \frac{\sigma_G^2}{1-\gamma\alpha\mathrm{tr}\left(\mathbf{H}_f^{(q)}\right)}\mathbf{H}_f^{(q)} \\
&\preceq \frac{\gamma\sigma_G^2}{1-\gamma\alpha\mathrm{tr}\left(\mathbf{H}_f^{(q)}\right)} \cdot \mathbf{I},
\end{aligned}
$$

where the first inequality holds by the induction assumption and $\mathcal{M}^{(q)} \circ \mathbf{I} \preceq \alpha\,\mathrm{tr}\left(\mathbf{H}_f^{(q)}\right)\mathbf{H}_f^{(q)}$. $\qquad\square$

**Lemma C.8** (A variance upper bound under general quantization). *Under Assumption 3.3, Assumption 3.4, if $\gamma < \frac{1}{\alpha\mathrm{tr}(\mathbf{H}_f^{(q)})}$,*

$$
\frac{1}{N^2} \cdot \sum_{t=0}^{N-1}\sum_{k=t}^{N-1} \left\langle (\mathbf{I} - \gamma\mathbf{H}_f^{(q)})^{k-t}\mathbf{H}_f^{(q)}, \mathbf{C}_t \right\rangle \leq \frac{\sigma_G^2}{1-\gamma\alpha\mathrm{tr}(\mathbf{H}_f^{(q)})}\left(\frac{k^*}{N} + N\gamma^2 \cdot \sum_{i>k^*}(\tilde{\lambda}_i^{(q)})^2\right).
$$

*where $(\tilde{\lambda}_i^{(q)})_{i=1}^M$ are eigenvalues of $\mathbf{H}_f^{(q)}$ and $k^* = \max\left\{k : \tilde{\lambda}_k^{(q)} \geq \frac{1}{N\gamma}\right\}$.*

*Proof.* We first provide a refined upper bound for $\mathbf{C}_t$. Note that by definition

$$
\begin{aligned}
\mathbf{C}_t &= (\mathcal{I} - \gamma\mathcal{T}^{(q)}) \circ \mathbf{C}_{t-1} + \gamma^2\sigma_G^2\mathbf{H}_f^{(q)} \\
&= (\mathcal{I} - \gamma\widetilde{\mathcal{T}}^{(q)}) \circ \mathbf{C}_{t-1} + \gamma(\widetilde{\mathcal{T}}^{(q)} - \mathcal{T}^{(q)}) \circ \mathbf{C}_{t-1} + \gamma^2\sigma_G^2\mathbf{H}_f^{(q)} \\
&= (\mathcal{I} - \gamma\widetilde{\mathcal{T}}^{(q)}) \circ \mathbf{C}_{t-1} + \gamma^2(\mathcal{M}^{(q)} - \widetilde{\mathcal{M}}^{(q)}) \circ \mathbf{C}_{t-1} + \gamma^2\sigma_G^2\mathbf{H}_f^{(q)} \\
&\preceq (\mathcal{I} - \gamma\widetilde{\mathcal{T}}^{(q)}) \circ \mathbf{C}_{t-1} + \gamma^2\mathcal{M}^{(q)} \circ \mathbf{C}_{t-1} + \gamma^2\sigma_G^2\mathbf{H}_f^{(q)},
\end{aligned}
\tag{31}
$$

together with Lemma C.7 and $\mathcal{M}^{(q)} \circ \mathbf{I} \preceq \alpha\,\mathrm{tr}(\mathbf{H}_f^{(q)})\mathbf{H}_f^{(q)}$, it holds

$$
\begin{aligned}
\mathbf{C}_t &\preceq (\mathcal{I} - \gamma\widetilde{\mathcal{T}}^{(q)}) \circ \mathbf{C}_{t-1} + \frac{\gamma^2\alpha\mathrm{tr}(\mathbf{H}_f^{(q)})\gamma\sigma_G^2}{1-\gamma\alpha\mathrm{tr}(\mathbf{H}_f^{(q)})}\mathbf{H}_f^{(q)} + \gamma^2\sigma_G^2\mathbf{H}_f^{(q)} \\
&= (\mathcal{I} - \gamma\widetilde{\mathcal{T}}^{(q)}) \circ \mathbf{C}_{t-1} + \frac{\gamma^2\sigma_G^2}{1-\gamma\alpha\mathrm{tr}(\mathbf{H}_f^{(q)})}\mathbf{H}_f^{(q)}.
\end{aligned}
$$

Solving recursion, it follows that

$$
\begin{aligned}
\mathbf{C}_t &\preceq \frac{\gamma^2\sigma_G^2}{1-\gamma\alpha\mathrm{tr}(\mathbf{H}_f^{(q)})} \cdot \sum_{k=0}^{t-1}(\mathcal{I} - \gamma\widetilde{\mathcal{T}}^{(q)})^k \circ \mathbf{H}_f^{(q)} \\
&= \frac{\gamma^2\sigma_G^2}{1-\gamma\alpha\mathrm{tr}(\mathbf{H}_f^{(q)})} \cdot \sum_{k=0}^{t-1}(\mathbf{I} - \gamma\mathbf{H}_f^{(q)})^k\mathbf{H}_f^{(q)}(\mathbf{I} - \gamma\mathbf{H}_f^{(q)})^k \\
&\preceq \frac{\gamma^2\sigma_G^2}{1-\gamma\alpha\mathrm{tr}(\mathbf{H}_f^{(q)})} \cdot \sum_{k=0}^{t-1}(\mathbf{I} - \gamma\mathbf{H}_f^{(q)})^k\mathbf{H}_f^{(q)} \\
&= \frac{\gamma\sigma_G^2}{1-\gamma\alpha\mathrm{tr}(\mathbf{H}_f^{(q)})} \cdot \left(\mathbf{I} - (\mathbf{I} - \gamma\mathbf{H}_f^{(q)})^t\right).
\end{aligned}
\tag{32}
$$

After providing a refined bound for $\mathbf{C}_t$, we are ready to bound the variance.

$$\frac{1}{N^2} \cdot \sum_{t=0}^{N-1} \sum_{k=t}^{N-1} \left\langle (\mathbf{I} - \gamma \mathbf{H}_f^{(q)})^{k-t} \mathbf{H}_f^{(q)}, \mathbf{C}_t \right\rangle$$

$$= \frac{1}{\gamma N^2} \sum_{t=0}^{N-1} \left\langle \mathbf{I} - (\mathbf{I} - \gamma \mathbf{H}_f^{(q)})^{N-t}, \mathbf{C}_t \right\rangle$$

$$\leq \frac{1}{\gamma^2 N^2} \frac{\gamma^2 \sigma_G^2}{1 - \gamma \alpha \mathrm{tr}(\mathbf{H}_f^{(q)})} \sum_{t=0}^{N-1} \left\langle \mathbf{I} - (\mathbf{I} - \gamma \mathbf{H}_f^{(q)})^{N-t}, \mathbf{I} - (\mathbf{I} - \gamma \mathbf{H}_f^{(q)})^t \right\rangle$$

$$= \frac{1}{\gamma^2 N^2} \frac{\gamma^2 \sigma_G^2}{1 - \gamma \alpha \mathrm{tr}(\mathbf{H}_f^{(q)})} \sum_{i} \sum_{t=0}^{N-1} \left[ 1 - (1 - \gamma \tilde{\lambda}_i^{(q)})^{N-t} \right] \left[ 1 - (1 - \gamma \tilde{\lambda}_i^{(q)})^t \right]$$

$$\leq \frac{1}{\gamma^2 N^2} \frac{\gamma^2 \sigma_G^2}{1 - \gamma \alpha \mathrm{tr}(\mathbf{H}_f^{(q)})} \sum_{i} \sum_{t=0}^{N-1} \left[ 1 - (1 - \gamma \tilde{\lambda}_i^{(q)})^{N} \right] \left[ 1 - (1 - \gamma \tilde{\lambda}_i^{(q)})^{N} \right]$$

$$\leq \frac{1}{\gamma^2 N} \frac{\gamma^2 \sigma_G^2}{1 - \gamma \alpha \mathrm{tr}(\mathbf{H}_f^{(q)})} \sum_{i} \min \left\{ 1, \gamma^2 N^2 (\tilde{\lambda}_i^{(q)})^2 \right\}$$

$$\leq \frac{\sigma_G^2}{1 - \gamma \alpha \mathrm{tr}(\mathbf{H}_f^{(q)})} \left( \frac{k^*}{N} + N \gamma^2 \cdot \sum_{i > k^*} (\tilde{\lambda}_i^{(q)})^2 \right),$$

where $(\tilde{\lambda}_i^{(q)})_{i=1}^{M}$ are eigenvalues of $\mathbf{H}_f^{(q)}$ and $k^* = \max \left\{ k : \tilde{\lambda}_k^{(q)} \geq \frac{1}{N\gamma} \right\}$. $\qquad\square$

### C.3.2. MULTIPLICATIVE QUANTIZATION

**Lemma C.9** (A crude upper bound of variance under multiplicative quantization). *Under Assumption 3.3, Assumption 3.4, if $\gamma < \frac{1}{(1+\tilde{\epsilon})\alpha \mathrm{tr}(\mathbf{H}_f^{(q)})}$,*

$$\mathbf{C}_t^{(M)} \preceq \frac{\gamma \sigma_M^2}{1 - \gamma(1 + \tilde{\epsilon})\alpha \mathrm{tr}\left(\mathbf{H}_f^{(q)}\right)} \mathbf{I}.$$

*Proof.* We prove by induction. For $t = 0$, we have $\mathbf{C}_0^{(M)} = \mathbf{0} \preceq \frac{\gamma \sigma_M^2}{1 - \gamma(1+\tilde{\epsilon})\alpha \mathrm{tr}\left(\mathbf{H}_f^{(q)}\right)} \mathbf{I}$. We then assume that $\mathbf{C}_{t-1}^{(M)} \preceq \frac{\gamma \sigma_M^2}{1 - \gamma(1+\tilde{\epsilon})\alpha \mathrm{tr}\left(\mathbf{H}_f^{(q)}\right)} \mathbf{I}$, and exam $\mathbf{C}_t^{(M)}$:

$$\mathbf{C}_t^{(M)} = (\mathcal{I} - \gamma \mathcal{T}^{(q)} + \tilde{\epsilon} \gamma^2 \mathcal{M}^{(q)}) \circ \mathbf{C}_{t-1}^{(M)} + \gamma^2 \sigma_M^2 \mathbf{H}_f^{(q)}$$

$$= \left( \mathcal{I} - \gamma \mathbf{H}_f^{(q)} \otimes \mathbf{I} - \gamma \mathbf{I} \otimes \mathbf{H}_f^{(q)} \right) \circ \mathbf{C}_{t-1}^{(M)} + (1 + \tilde{\epsilon}) \gamma^2 \mathcal{M}^{(q)} \circ \mathbf{C}_{t-1}^{(M)} + \gamma^2 \sigma_M^2 \mathbf{H}_f^{(q)}$$

$$\preceq \frac{\gamma \sigma_M^2}{1 - \gamma(1 + \tilde{\epsilon})\alpha \mathrm{tr}\left(\mathbf{H}_f^{(q)}\right)} \cdot \left( \mathbf{I} - 2\gamma \mathbf{H}_f^{(q)} \right) + \left( \frac{(1 + \tilde{\epsilon})\gamma^3 \sigma_M^2 \alpha \mathrm{tr}\left(\mathbf{H}_f^{(q)}\right)}{1 - \gamma(1 + \tilde{\epsilon})\alpha \mathrm{tr}\left(\mathbf{H}_f^{(q)}\right)} + \gamma^2 \sigma_M^2 \right) \mathbf{H}_f^{(q)}$$

$$= \frac{\gamma \sigma_M^2}{1 - \gamma(1 + \tilde{\epsilon})\alpha \mathrm{tr}\left(\mathbf{H}_f^{(q)}\right)} \cdot \mathbf{I} - (2\gamma^2 - \gamma^2) \cdot \frac{\sigma_M^2}{1 - \gamma(1 + \tilde{\epsilon})\alpha \mathrm{tr}\left(\mathbf{H}_f^{(q)}\right)} \mathbf{H}_f^{(q)}$$

$$\preceq \frac{\gamma \sigma_M^2}{1 - \gamma(1 + \tilde{\epsilon})\alpha \mathrm{tr}\left(\mathbf{H}_f^{(q)}\right)} \cdot \mathbf{I},$$

where the first inequality holds by the induction assumption and $\mathcal{M}^{(q)} \circ \mathbf{I} \preceq \alpha \mathrm{tr}\left(\mathbf{H}_f^{(q)}\right) \mathbf{H}_f^{(q)}$. $\qquad\square$

**Lemma C.10** (A variance upper bound under multiplicative quantization). *Under Assumption 3.3, Assumption 3.4, if $\gamma < \frac{1}{(1+\tilde{\epsilon})\alpha\mathrm{tr}(\mathbf{H}_f^{(q)})}$,*

$$\frac{1}{N^2} \cdot \sum_{t=0}^{N-1} \sum_{k=t}^{N-1} \left\langle (\mathbf{I} - \gamma\mathbf{H}_f^{(q)})^{k-t}\mathbf{H}_f^{(q)}, \mathbf{C}_t^{(M)} \right\rangle \leq \frac{\sigma_M^2}{1 - (1+\tilde{\epsilon})\gamma\alpha\mathrm{tr}(\mathbf{H}_f^{(q)})} \left( \frac{k^*}{N} + N\gamma^2 \cdot \sum_{i>k^*} (\tilde{\lambda}_i^{(q)})^2 \right),$$

*where $k^* = \max\left\{ k : \tilde{\lambda}_k^{(q)} \geq \frac{1}{N\gamma} \right\}$, and $(\tilde{\lambda}_i^{(q)})_{i=1}^M$ are eigenvalues of $\mathbf{H}_f^{(q)}$.*

*Proof.* We first provide a refined bound for $\mathbf{C}_t^{(M)}$. By the definition of $\mathbf{C}_t^{(M)}$,

$$\begin{aligned}
\mathbf{C}_t^{(M)} &= (\mathcal{I} - \gamma\mathcal{T}^{(q)} + \tilde{\epsilon}\gamma^2\mathcal{M}^{(q)}) \circ \mathbf{C}_{t-1}^{(M)} + \gamma^2\sigma_M^2\mathbf{H}_f^{(q)} \\
&\preceq (\mathcal{I} - \gamma\tilde{\mathcal{T}}^{(q)}) \circ \mathbf{C}_{t-1}^{(M)} + \gamma^2(1+\tilde{\epsilon})\mathcal{M}^{(q)} \circ \mathbf{C}_{t-1}^{(M)} + \gamma^2\sigma_M^2\mathbf{H}_f^{(q)} \\
&\preceq (\mathcal{I} - \gamma\tilde{\mathcal{T}}^{(q)}) \circ \mathbf{C}_{t-1}^{(M)} + \gamma^2(1+\tilde{\epsilon})\frac{\gamma\sigma_M^2\alpha\,\mathrm{tr}\left(\mathbf{H}_f^{(q)}\right)}{1 - \gamma(1+\tilde{\epsilon})\alpha\mathrm{tr}\left(\mathbf{H}_f^{(q)}\right)}\mathbf{H}_f^{(q)} + \gamma^2\sigma_M^2\mathbf{H}_f^{(q)} \\
&= (\mathcal{I} - \gamma\tilde{\mathcal{T}}^{(q)}) \circ \mathbf{C}_{t-1}^{(M)} + \frac{\gamma^2\sigma_M^2}{1 - \gamma(1+\tilde{\epsilon})\alpha\mathrm{tr}\left(\mathbf{H}_f^{(q)}\right)}\mathbf{H}_f^{(q)},
\end{aligned}$$

where the second inequality holds by Lemma C.9 and $\mathcal{M}^{(q)} \circ \mathbf{I} \preceq \alpha\,\mathrm{tr}\left(\mathbf{H}_f^{(q)}\right)\mathbf{H}_f^{(q)}$. Solving the recursion yields

$$\mathbf{C}_t^{(M)} \preceq \frac{\gamma\sigma_M^2}{1 - (1+\tilde{\epsilon})\gamma\alpha\mathrm{tr}(\mathbf{H}_f^{(q)})} \cdot \left( \mathbf{I} - (\mathbf{I} - \gamma\mathbf{H}_f^{(q)})^t \right).$$

After providing a refined bound for $\mathbf{C}_t^{(M)}$, we are ready to bound the variance.

$$\begin{aligned}
&\frac{1}{N^2} \cdot \sum_{t=0}^{N-1} \sum_{k=t}^{N-1} \left\langle (\mathbf{I} - \gamma\mathbf{H}_f^{(q)})^{k-t}\mathbf{H}_f^{(q)}, \mathbf{C}_t^{(M)} \right\rangle \\
&= \frac{1}{\gamma N^2} \sum_{t=0}^{N-1} \left\langle \mathbf{I} - (\mathbf{I} - \gamma\mathbf{H}_f^{(q)})^{N-t}, \mathbf{C}_t^{(M)} \right\rangle \\
&\leq \frac{1}{\gamma^2 N^2} \frac{\gamma^2\sigma_M^2}{1 - (1+\tilde{\epsilon})\gamma\alpha\mathrm{tr}(\mathbf{H}_f^{(q)})} \sum_{t=0}^{N-1} \left\langle \mathbf{I} - (\mathbf{I} - \gamma\mathbf{H}_f^{(q)})^{N-t}, \mathbf{I} - (\mathbf{I} - \gamma\mathbf{H}_f^{(q)})^t \right\rangle \\
&= \frac{1}{\gamma^2 N^2} \frac{\gamma^2\sigma_M^2}{1 - (1+\tilde{\epsilon})\gamma\alpha\mathrm{tr}(\mathbf{H}_f^{(q)})} \sum_i \sum_{t=0}^{N-1} \left[ 1 - (1 - \gamma\tilde{\lambda}_i^{(q)})^{N-t} \right] \left[ 1 - (1 - \gamma\tilde{\lambda}_i^{(q)})^t \right] \\
&\leq \frac{1}{\gamma^2 N^2} \frac{\gamma^2\sigma_M^2}{1 - (1+\tilde{\epsilon})\gamma\alpha\mathrm{tr}(\mathbf{H}_f^{(q)})} \sum_i \sum_{t=0}^{N-1} \left[ 1 - (1 - \gamma\tilde{\lambda}_i^{(q)})^N \right] \left[ 1 - (1 - \gamma\tilde{\lambda}_i^{(q)})^N \right] \\
&= \frac{1}{\gamma^2 N} \frac{\gamma^2\sigma_M^2}{1 - (1+\tilde{\epsilon})\gamma\alpha\mathrm{tr}(\mathbf{H}_f^{(q)})} \sum_i \left[ 1 - (1 - \gamma\tilde{\lambda}_i^{(q)})^N \right]^2 \\
&\leq \frac{1}{\gamma^2 N} \frac{\gamma^2\sigma_M^2}{1 - (1+\tilde{\epsilon})\gamma\alpha\mathrm{tr}(\mathbf{H}_f^{(q)})} \sum_i \min\left\{ 1, \gamma^2 N^2(\tilde{\lambda}_i^{(q)})^2 \right\} \\
&\leq \frac{1}{\gamma^2 N} \frac{\gamma^2\sigma_M^2}{1 - (1+\tilde{\epsilon})\gamma\alpha\mathrm{tr}(\mathbf{H}_f^{(q)})} \left( k^* + N^2\gamma^2 \cdot \sum_{i>k^*} (\tilde{\lambda}_i^{(q)})^2 \right) \\
&= \frac{\sigma_M^2}{1 - (1+\tilde{\epsilon})\gamma\alpha\mathrm{tr}(\mathbf{H}_f^{(q)})} \left( \frac{k^*}{N} + N\gamma^2 \cdot \sum_{i>k^*} (\tilde{\lambda}_i^{(q)})^2 \right),
\end{aligned}$$

where $(\tilde{\lambda}_i^{(q)})_{i=1}^M$ are eigenvalues of $\mathbf{H}_f^{(q)}$ and $k^* = \max\left\{k : \tilde{\lambda}_k^{(q)} \geq \frac{1}{N\gamma}\right\}$. $\qquad\square$

## C.4. Bias Upper Bounds

### C.4.1. GENERAL QUANTIZATION

Let $\mathbf{S}_n = \sum_{t=0}^{n-1} \mathbf{B}_t$.

**Lemma C.11** (Initial Study of $\mathbf{S}_t$). *For* $1 \leq t \leq N$,

$$\mathbf{S}_t \preceq (\mathcal{I} - \gamma\tilde{\mathcal{T}}^{(q)}) \circ \mathbf{S}_{t-1} + \gamma^2 \mathcal{M}^{(q)} \circ \mathbf{S}_N + \mathbf{B}_0.$$

*Proof.* By definition,

$$\begin{aligned}
\mathbf{S}_t &= \sum_{k=0}^{t-1} (\mathcal{I} - \gamma\mathcal{T}^{(q)})^k \circ \mathbf{B}_0 \\
&= (\mathcal{I} - \gamma\mathcal{T}^{(q)}) \circ \left(\sum_{k=1}^{t-1} (\mathcal{I} - \gamma\mathcal{T}^{(q)})^{k-1} \circ \mathbf{B}_0\right) + \mathbf{B}_0 \\
&= (\mathcal{I} - \gamma\mathcal{T}^{(q)}) \circ \mathbf{S}_{t-1} + \mathbf{B}_0.
\end{aligned}$$

(33)

Then we convert $\mathcal{T}^{(q)}$ to $\tilde{\mathcal{T}}^{(q)}$. By (33),

$$\begin{aligned}
\mathbf{S}_t &= (\mathcal{I} - \gamma\mathcal{T}^{(q)}) \circ \mathbf{S}_{t-1} + \mathbf{B}_0 \\
&= (\mathcal{I} - \gamma\tilde{\mathcal{T}}^{(q)}) \circ \mathbf{S}_{t-1} + \gamma(\tilde{\mathcal{T}}^{(q)} - \mathcal{T}^{(q)}) \circ \mathbf{S}_{t-1} + \mathbf{B}_0 \\
&= (\mathcal{I} - \gamma\tilde{\mathcal{T}}^{(q)}) \circ \mathbf{S}_{t-1} + \gamma^2(\mathcal{M}^{(q)} - \widetilde{\mathcal{M}}^{(q)}) \circ \mathbf{S}_{t-1} + \mathbf{B}_0 \\
&\preceq (\mathcal{I} - \gamma\tilde{\mathcal{T}}^{(q)}) \circ \mathbf{S}_{t-1} + \gamma^2 \mathcal{M}^{(q)} \circ \mathbf{S}_N + \mathbf{B}_0,
\end{aligned}$$

where the third equality holds by the definition of linear operators. $\qquad\square$

**Lemma C.12** (A Bound for $\mathcal{M}^{(q)} \circ \mathbf{S}_t$). *For* $1 \leq t \leq N$, *under Assumption 3.3, Assumption 3.4, if* $\gamma < \frac{1}{\alpha \mathrm{tr}(\mathbf{H}_f^{(q)})}$, *then*

$$\mathcal{M}^{(q)} \circ \mathbf{S}_t \preceq \frac{\alpha \cdot \mathrm{tr}\left(\left[\mathcal{I} - (\mathcal{I} - \gamma\tilde{\mathcal{T}}^{(q)})^t\right] \circ \mathbf{B}_0\right)}{\gamma(1 - \gamma\alpha\,\mathrm{tr}(\mathbf{H}_f^{(q)}))} \cdot \mathbf{H}_f^{(q)}.$$

*Proof.* The first step is to derive a crude bound for $\mathbf{S}_t$. Take summation via the update rule, we have

$$\mathbf{S}_t = \sum_{k=0}^{t-1} (\mathcal{I} - \gamma\mathcal{T}^{(q)})^k \circ \mathbf{B}_0 = \gamma^{-1}{\mathcal{T}^{(q)}}^{-1} \circ \left[\mathcal{I} - (\mathcal{I} - \gamma\mathcal{T}^{(q)})^t\right] \circ \mathbf{B}_0.$$

Note that

$$\mathcal{I} - \gamma\tilde{\mathcal{T}}^{(q)} \preceq \mathcal{I} - \gamma\mathcal{T}^{(q)}, \quad (\mathcal{I} - (\mathcal{I} - \gamma\mathcal{T}^{(q)})^t) \preceq (\mathcal{I} - (\mathcal{I} - \gamma\tilde{\mathcal{T}}^{(q)})^t),$$

and further note that ${\mathcal{T}^{(q)}}^{-1}$ is a PSD mapping [6], and $[\mathcal{I} - (\mathcal{I} - \gamma\tilde{\mathcal{T}}^{(q)})^t] \circ \mathbf{B}_0$ is a PSD matrix, we obtain

$$\mathbf{S}_t \preceq \gamma^{-1}{\mathcal{T}^{(q)}}^{-1} \circ (\mathcal{I} - (\mathcal{I} - \gamma\tilde{\mathcal{T}}^{(q)})^t) \circ \mathbf{B}_0.$$

For simplicity, we denote $\mathbf{A} := (\mathcal{I} - (\mathcal{I} - \gamma\tilde{\mathcal{T}}^{(q)})^t) \circ \mathbf{B}_0$. We then tackle ${\mathcal{T}^{(q)}}^{-1} \circ \mathbf{A}$. To be specific, we apply $\tilde{\mathcal{T}}^{(q)}$.

$$\begin{aligned}
\tilde{\mathcal{T}}^{(q)} \circ {\mathcal{T}^{(q)}}^{-1} \circ \mathbf{A} &= \gamma\mathcal{M}^{(q)} \circ {\mathcal{T}^{(q)}}^{-1} \circ \mathbf{A} + \mathbf{A} - \gamma\mathbf{H}_f^{(q)}({\mathcal{T}^{(q)}}^{-1} \circ \mathbf{A})\mathbf{H}_f^{(q)} \\
&\preceq \gamma\mathcal{M}^{(q)} \circ {\mathcal{T}^{(q)}}^{-1} \circ \mathbf{A} + \mathbf{A}.
\end{aligned}$$

---

[6] ${\mathcal{T}^{(q)}}^{-1}$ is a PSD mapping under the condition that $\gamma < \frac{1}{\alpha\mathrm{tr}(\mathbf{H}^{(q)})}$, which can be directly deduced by Lemma B.1 in Zou et al. (2021). We omit the proof here for simplicity.

Therefore,

$$\mathcal{T}^{(q)^{-1}} \circ \mathbf{A} \preceq \gamma(\widetilde{\mathcal{T}}^{(q)})^{-1} \circ \mathcal{M}^{(q)} \circ \mathcal{T}^{(q)^{-1}} \circ \mathbf{A} + (\widetilde{\mathcal{T}}^{(q)})^{-1} \circ \mathbf{A}.$$

Then we undertake the second step, applying $\mathcal{M}^{(q)}$ on both sides.

$$\begin{aligned}
\mathcal{M}^{(q)} \circ (\mathcal{T}^{(q)^{-1}} \circ \mathbf{A}) &\preceq \mathcal{M}^{(q)} \circ \gamma(\widetilde{\mathcal{T}}^{(q)})^{-1} \circ \mathcal{M}^{(q)} \circ \mathcal{T}^{(q)^{-1}} \circ \mathbf{A} + \mathcal{M}^{(q)} \circ (\widetilde{\mathcal{T}}^{(q)})^{-1} \circ \mathbf{A} \\
&\preceq \sum_{t=0}^{\infty} (\gamma \mathcal{M}^{(q)} \circ (\widetilde{\mathcal{T}}^{(q)})^{-1})^t \circ (\mathcal{M}^{(q)} \circ (\widetilde{\mathcal{T}}^{(q)})^{-1} \circ \mathbf{A}) \text{ (By recursion)}.
\end{aligned}$$
(34)

By Assumption 3.4,

$$\begin{aligned}
\mathcal{M}^{(q)} \circ (\widetilde{\mathcal{T}}^{(q)})^{-1} \circ \mathbf{A} &\preceq \alpha \operatorname{tr}(\mathbf{H}_f^{(q)}(\widetilde{\mathcal{T}}^{(q)})^{-1} \circ \mathbf{A})\mathbf{H}_f^{(q)} \\
&= \alpha\gamma \operatorname{tr}\left(\sum_{t=0}^{\infty} \mathbf{H}_f^{(q)}(\mathbf{I} - \gamma\mathbf{H}_f^{(q)})^t \mathbf{A}(\mathbf{I} - \gamma\mathbf{H}_f^{(q)})^t\right) \mathbf{H}_f^{(q)} \\
&= \alpha \operatorname{tr}\left(\mathbf{H}_f^{(q)}(2\mathbf{H}_f^{(q)} - \gamma(\mathbf{H}_f^{(q)})^2)^{-1}\mathbf{A}\right) \mathbf{H}_f^{(q)} \\
&\preceq \alpha \operatorname{tr}(\mathbf{A})\mathbf{H}_f^{(q)},
\end{aligned}$$

where the first equality holds by the definition of $\widetilde{\mathcal{T}}^{(q)}$ and the last inequality requires the condition that $\gamma < \frac{1}{\alpha \operatorname{tr}(\mathbf{H}_f^{(q)})}$. Hence, by (34), and further by $(\widetilde{\mathcal{T}}^{(q)})^{-1}\mathbf{H}_f^{(q)} \preceq \mathbf{I}$ and $\mathcal{M}^{(q)} \circ \mathbf{I} \preceq \alpha \operatorname{tr}(\mathbf{H}_f^{(q)})\mathbf{H}_f^{(q)}$, we obtain

$$\begin{aligned}
\mathcal{M}^{(q)} \circ (\mathcal{T}^{(q)^{-1}} \circ \mathbf{A}) &\preceq \sum_{t=0}^{\infty} (\gamma\mathcal{M}^{(q)} \circ (\widetilde{\mathcal{T}}^{(q)})^{-1})^t \circ (\mathcal{M}^{(q)} \circ (\widetilde{\mathcal{T}}^{(q)})^{-1} \circ \mathbf{A}) \\
&\preceq \alpha \operatorname{tr}(\mathbf{A})\sum_{t=0}^{\infty} (\gamma\alpha \operatorname{tr}(\mathbf{H}_f^{(q)}))^t \mathbf{H}_f^{(q)} \\
&\preceq \frac{\alpha \operatorname{tr}(\mathbf{A})}{1 - \gamma\alpha \operatorname{tr}(\mathbf{H}_f^{(q)})} \cdot \mathbf{H}_f^{(q)}.
\end{aligned}$$

Therefore,

$$\mathcal{M}^{(q)} \circ \mathbf{S}_t \preceq \gamma^{-1} \frac{\alpha \operatorname{tr}(\mathbf{A})}{1 - \gamma\alpha \operatorname{tr}(\mathbf{H}_f^{(q)})} \cdot \mathbf{H}_f^{(q)} = \frac{\alpha \cdot \operatorname{tr}\left(\left[\mathcal{I} - (\mathcal{I} - \gamma\widetilde{\mathcal{T}}^{(q)})^t\right] \circ \mathbf{B}_0\right)}{\gamma(1 - \gamma\alpha \operatorname{tr}(\mathbf{H}_f^{(q)}))} \cdot \mathbf{H}_f^{(q)}.$$

$\square$

**Lemma C.13** (A bias upper bound under general quantization). *Under Assumption 3.3, Assumption 3.4, if the stepsize satisfies $\gamma < \frac{1}{\alpha \operatorname{tr}(\mathbf{H}_f^{(q)})}$, then*

$$\begin{aligned}
&\frac{1}{N^2} \cdot \sum_{t=0}^{N-1}\sum_{k=t}^{N-1} \left\langle (\mathbf{I} - \gamma\mathbf{H}_f^{(q)})^{k-t}\mathbf{H}_f^{(q)}, \mathbf{B}_t \right\rangle \\
&\leq \frac{2\alpha\left(\|\mathbf{v}^{(q)*}\|_{\mathbf{I}_{f,0:k^*}^{(q)}}^2 + N\gamma\|\mathbf{v}^{(q)*}\|_{\mathbf{H}_{f,k^*:\infty}^{(q)}}^2\right)}{N\gamma(1 - \gamma\alpha \operatorname{tr}(\mathbf{H}_f^{(q)}))} \cdot \left(\frac{k^*}{N} + N\gamma^2\sum_{i>k^*}(\tilde{\lambda}_i^{(q)})^2\right) \\
&\quad + \frac{1}{\gamma^2 N^2} \cdot \|\mathbf{v}^{(q)*}\|_{(\mathbf{H}_{f,0:k^*}^{(q)})^{-1}}^2 + \|\mathbf{v}^{(q)*}\|_{\mathbf{H}_{f,k^*:\infty}^{(q)}}^2.
\end{aligned}$$

*Proof.* Recalling Lemma C.11, we can derive a refined upper bound for $\mathbf{S}_t$ by Lemma C.12:

$$
\begin{aligned}
\mathbf{S}_t \preceq & (\mathcal{I} - \gamma\tilde{\mathcal{T}}^{(q)}) \circ \mathbf{S}_{t-1} + \gamma^2\mathcal{M}^{(q)} \circ \mathbf{S}_N + \mathbf{B}_0 \\
\preceq & (\mathcal{I} - \gamma\tilde{\mathcal{T}}^{(q)}) \circ \mathbf{S}_{t-1} + \frac{\gamma\alpha \cdot \operatorname{tr}\left(\left[\mathcal{I} - (\mathcal{I} - \gamma\tilde{\mathcal{T}}^{(q)})^N\right] \circ \mathbf{B}_0\right)}{1 - \gamma\alpha\operatorname{tr}(\mathbf{H}_f^{(q)})} \cdot \mathbf{H}_f^{(q)} + \mathbf{B}_0 \\
= & \sum_{k=0}^{t-1} (\mathcal{I} - \gamma\tilde{\mathcal{T}}^{(q)})^k \left(\frac{\gamma\alpha \cdot \operatorname{tr}\left(\left[\mathcal{I} - (\mathcal{I} - \gamma\tilde{\mathcal{T}}^{(q)})^N\right] \circ \mathbf{B}_0\right)}{1 - \gamma\alpha\operatorname{tr}(\mathbf{H}_f^{(q)})} \cdot \mathbf{H}_f^{(q)} + \mathbf{B}_0\right) \\
= & \sum_{k=0}^{t-1} (\mathbf{I} - \gamma\mathbf{H}_f^{(q)})^k \left(\frac{\gamma\alpha \cdot \operatorname{tr}\left(\mathbf{B}_0 - (\mathbf{I} - \gamma\mathbf{H}_f^{(q)})^N\mathbf{B}_0(\mathbf{I} - \gamma\mathbf{H}_f^{(q)})^N\right)}{1 - \gamma\alpha\operatorname{tr}(\mathbf{H}_f^{(q)})} \cdot \mathbf{H}_f^{(q)} + \mathbf{B}_0\right) (\mathbf{I} - \gamma\mathbf{H}_f^{(q)})^k.
\end{aligned}
\tag{35}
$$

Before providing our upper bound for the bias error, we denote

$$
\mathbf{B}_{a,b} := \mathbf{B}_a - (\mathbf{I} - \gamma\mathbf{H}_f^{(q)})^{b-a}\mathbf{B}_a(\mathbf{I} - \gamma\mathbf{H}_f^{(q)})^{b-a}.
$$

Then by (35),

$$
\begin{aligned}
& \frac{1}{N^2} \cdot \sum_{t=0}^{N-1}\sum_{k=t}^{N-1} \left\langle (\mathbf{I} - \gamma\mathbf{H}_f^{(q)})^{k-t}\mathbf{H}_f^{(q)}, \mathbf{B}_t \right\rangle \\
= & \frac{1}{\gamma N^2} \sum_{t=0}^{N-1} \left\langle \mathbf{I} - (\mathbf{I} - \gamma\mathbf{H}_f^{(q)})^{N-t}, \mathbf{B}_t \right\rangle \\
\leq & \frac{1}{\gamma N^2} \langle \mathbf{I} - (\mathbf{I} - \gamma\mathbf{H}_f^{(q)})^N, \sum_{t=0}^{N-1}\mathbf{B}_t \rangle \\
\leq & \frac{1}{\gamma N^2} \sum_{k=0}^{N-1} \left\langle \mathbf{I} - (\mathbf{I} - \gamma\mathbf{H}_f^{(q)})^N, (\mathbf{I} - \gamma\mathbf{H}_f^{(q)})^k \left(\frac{\gamma\alpha \cdot \operatorname{tr}(\mathbf{B}_{0,N})}{1 - \gamma\alpha\operatorname{tr}(\mathbf{H}_f^{(q)})} \cdot \mathbf{H}_f^{(q)} + \mathbf{B}_0\right) (\mathbf{I} - \gamma\mathbf{H}_f^{(q)})^k \right\rangle \\
= & \frac{1}{\gamma N^2} \sum_{k=0}^{N-1} \left\langle (\mathbf{I} - \gamma\mathbf{H}_f^{(q)})^{2k} - (\mathbf{I} - \gamma\mathbf{H}_f^{(q)})^{N+2k}, \left(\frac{\gamma\alpha \cdot \operatorname{tr}(\mathbf{B}_{0,N})}{1 - \gamma\alpha\operatorname{tr}(\mathbf{H}_f^{(q)})} \cdot \mathbf{H}_f^{(q)} + \mathbf{B}_0\right) \right\rangle.
\end{aligned}
$$

Note that

$$
\begin{aligned}
(\mathbf{I} - \gamma\mathbf{H}_f^{(q)})^{2k} - (\mathbf{I} - \gamma\mathbf{H}_f^{(q)})^{N+2k} &= \left(\mathbf{I} - \gamma\mathbf{H}_f^{(q)}\right)^k \left(\left(\mathbf{I} - \gamma\mathbf{H}_f^{(q)}\right)^k - \left(\mathbf{I} - \gamma\mathbf{H}_f^{(q)}\right)^{N+k}\right) \\
&\preceq (\mathbf{I} - \gamma\mathbf{H}_f^{(q)})^k - (\mathbf{I} - \gamma\mathbf{H}_f^{(q)})^{N+k},
\end{aligned}
$$

we obtain

$$
\begin{aligned}
& \frac{1}{N^2} \cdot \sum_{t=0}^{N-1}\sum_{k=t}^{N-1} \left\langle (\mathbf{I} - \gamma\mathbf{H}_f^{(q)})^{k-t}\mathbf{H}_f^{(q)}, \mathbf{B}_t \right\rangle \\
\leq & \frac{1}{\gamma N^2} \sum_{k=0}^{N-1} \left\langle (\mathbf{I} - \gamma\mathbf{H}_f^{(q)})^k - (\mathbf{I} - \gamma\mathbf{H}_f^{(q)})^{N+k}, \frac{\gamma\alpha \cdot \operatorname{tr}(\mathbf{B}_{0,N})}{1 - \gamma\alpha\operatorname{tr}(\mathbf{H}_f^{(q)})} \cdot \mathbf{H}_f^{(q)} + \mathbf{B}_0 \right\rangle.
\end{aligned}
$$

Therefore, it suffices to upper bound the following two terms

$$
\begin{aligned}
I_1 &= \frac{\alpha\operatorname{tr}(\mathbf{B}_{0,N})}{N^2(1 - \gamma\alpha\operatorname{tr}(\mathbf{H}_f^{(q)}))} \sum_{k=0}^{N-1} \left\langle (\mathbf{I} - \gamma\mathbf{H}_f^{(q)})^k - (\mathbf{I} - \gamma\mathbf{H}_f^{(q)})^{N+k}, \mathbf{H}_f^{(q)} \right\rangle, \\
I_2 &= \frac{1}{\gamma N^2} \sum_{k=0}^{N-1} \left\langle (\mathbf{I} - \gamma\mathbf{H}_f^{(q)})^k - (\mathbf{I} - \gamma\mathbf{H}_f^{(q)})^{N+k}, \mathbf{B}_0 \right\rangle.
\end{aligned}
$$

Regarding $I_1$, since $\mathbf{H}_f^{(q)}$ and $\mathbf{I} - \gamma\mathbf{H}_f^{(q)}$ can be diagonalized simultaneously,

$$
\begin{aligned}
I_1 &= \frac{\alpha\operatorname{tr}(\mathbf{B}_{0,N})}{N^2(1 - \gamma\alpha\operatorname{tr}(\mathbf{H}_f^{(q)}))} \sum_{k=0}^{N-1} \sum_i \left[(1 - \gamma\tilde{\lambda}_i^{(q)})^k - (1 - \gamma\tilde{\lambda}_i^{(q)})^{N+k}\right] \tilde{\lambda}_i^{(q)} \\
&= \frac{\alpha\operatorname{tr}(\mathbf{B}_{0,N})}{\gamma N^2(1 - \gamma\alpha\operatorname{tr}(\mathbf{H}_f^{(q)}))} \sum_i \left[1 - (1 - \gamma\tilde{\lambda}_i^{(q)})^N\right]^2 \\
&\leq \frac{\alpha\operatorname{tr}(\mathbf{B}_{0,N})}{\gamma N^2(1 - \gamma\alpha\operatorname{tr}(\mathbf{H}_f^{(q)}))} \sum_i \min\left\{1, \gamma^2 N^2 (\tilde{\lambda}_i^{(q)})^2\right\} \\
&\leq \frac{\alpha\operatorname{tr}(\mathbf{B}_{0,N})}{\gamma(1 - \gamma\alpha\operatorname{tr}(\mathbf{H}_f^{(q)}))} \cdot \left(\frac{k^*}{N^2} + \gamma^2 \sum_{i>k^*} (\tilde{\lambda}_i^{(q)})^2\right),
\end{aligned}
$$

where $k^* = \max\{k : \tilde{\lambda}_k^{(q)} \geq \frac{1}{N\gamma}\}$ and $\{\tilde{\lambda}_i^{(q)}\}_{i=1}^M$ are eigenvalues of $\mathbf{H}_f^{(q)}$. Then we tackle $\operatorname{tr}(\mathbf{B}_{0,N})$.

$$
\begin{aligned}
\operatorname{tr}(\mathbf{B}_{0,N}) &= \operatorname{tr}\left(\mathbf{B}_0 - (\mathbf{I} - \gamma\mathbf{H}_f^{(q)})^N \mathbf{B}_0 (\mathbf{I} - \gamma\mathbf{H}_f^{(q)})^N\right) \\
&= \sum_i \left(1 - (1 - \gamma\tilde{\lambda}_i^{(q)})^{2N}\right) \cdot \left(\langle \mathbf{v}_0 - \mathbf{v}^{(q)*}, \mathbf{v}_i^{(q)}\rangle\right)^2 \\
&\leq 2\sum_i \min\{1, N\gamma\tilde{\lambda}_i^{(q)}\} \left(\langle \mathbf{v}_0 - \mathbf{v}^{(q)*}, \mathbf{v}_i^{(q)}\rangle\right)^2 \\
&\leq 2\left(\|\mathbf{v}_0 - \mathbf{v}^{(q)*}\|_{\mathbf{I}_{f,0:k^*}^{(q)}}^2 + N\gamma\|\mathbf{v}_0 - \mathbf{v}^{(q)*}\|_{\mathbf{H}_{f,k^*:\infty}^{(q)}}^2\right).
\end{aligned}
\tag{36}
$$

Hence,

$$
I_1 \leq \frac{2\alpha\left(\|\mathbf{v}_0 - \mathbf{v}^{(q)*}\|_{\mathbf{I}_{0:k^*}^{(q)}}^2 + N\gamma\|\mathbf{v}_0 - \mathbf{v}^{(q)*}\|_{\mathbf{H}_{k^*:\infty}^{(q)}}^2\right)}{N\gamma(1 - \gamma\alpha\operatorname{tr}(\mathbf{H}_f^{(q)}))} \cdot \left(\frac{k^*}{N} + N\gamma^2 \sum_{i>k^*} (\tilde{\lambda}_i^{(q)})^2\right).
$$

Regarding $I_2$, decompose $\mathbf{H}_f^{(q)} = \mathbf{V}^{(q)}\mathbf{\Lambda}^{(q)}\mathbf{V}^{(q)\top}$, then

$$
I_2 = \frac{1}{\gamma N^2} \sum_{k=0}^{N-1} \left\langle (\mathbf{I} - \gamma\mathbf{\Lambda}^{(q)})^k - (\mathbf{I} - \gamma\mathbf{\Lambda}^{(q)})^{N+k}, \mathbf{V}^{(q)\top}\mathbf{B}_0\mathbf{V}^{(q)}\right\rangle.
$$

Note that $\mathbf{B}_0 = \boldsymbol{\eta}_0\boldsymbol{\eta}_0^\top$, it can be shown that the diagonal entries of $\mathbf{V}^{(q)\top}\mathbf{B}_0\mathbf{V}^{(q)}$ are $\omega_1^2, \ldots$, where $\omega_i = \mathbf{v}_i^{(q)\top}\boldsymbol{\eta}_0 = \mathbf{v}_i^{(q)\top}(\mathbf{v}_0 - \mathbf{v}^{(q)*})$. Hence,

$$
\begin{aligned}
I_2 &= \frac{1}{\gamma N^2} \sum_{k=0}^{N-1} \sum_i \left[(1 - \gamma\tilde{\lambda}_i^{(q)})^k - (1 - \gamma\tilde{\lambda}_i^{(q)})^{N+k}\right] \omega_i^2 \\
&= \frac{1}{\gamma^2 N^2} \sum_i \frac{\omega_i^2}{\tilde{\lambda}_i^{(q)}} \left[1 - (1 - \gamma\tilde{\lambda}_i^{(q)})^N\right]^2 \\
&\leq \frac{1}{\gamma^2 N^2} \sum_i \frac{\omega_i^2}{\tilde{\lambda}_i^{(q)}} \min\left\{1, \gamma^2 N^2 (\tilde{\lambda}_i^{(q)})^2\right\} \\
&\leq \frac{1}{\gamma^2 N^2} \cdot \sum_{i\leq k^*} \frac{\omega_i^2}{\tilde{\lambda}_i^{(q)}} + \sum_{i>k^*} \tilde{\lambda}_i^{(q)}\omega_i^2 \\
&= \frac{1}{\gamma^2 N^2} \cdot \|\mathbf{v}_0 - \mathbf{v}^{(q)*}\|_{(\mathbf{H}_{f,0:k^*}^{(q)})^{-1}}^2 + \|\mathbf{v}_0 - \mathbf{v}^{(q)*}\|_{\mathbf{H}_{f,k^*:\infty}^{(q)}}^2.
\end{aligned}
$$

In conclusion, if the stepsize satisfies $\gamma < \frac{1}{\alpha \mathrm{tr}(\mathbf{H}_f^{(q)})}$,

$$
\frac{1}{N^2} \cdot \sum_{t=0}^{N-1} \sum_{k=t}^{N-1} \left\langle (\mathbf{I} - \gamma \mathbf{H}_f^{(q)})^{k-t} \mathbf{H}_f^{(q)}, \mathbf{B}_t \right\rangle
$$

$$
\leq \frac{2\alpha \left( \|\mathbf{v}_0 - \mathbf{v}^{(q)*}\|_{\mathbf{I}_{f,0:k^*}^{(q)}}^2 + N\gamma \|\mathbf{v}_0 - \mathbf{v}^{(q)*}\|_{\mathbf{H}_{f,k^*:\infty}^{(q)}}^2 \right)}{N\gamma(1 - \gamma\alpha\,\mathrm{tr}(\mathbf{H}_f^{(q)}))} \cdot \left( \frac{k^*}{N} + N\gamma^2 \sum_{i>k^*} (\tilde{\lambda}_i^{(q)})^2 \right)
$$

$$
+ \frac{1}{\gamma^2 N^2} \cdot \|\mathbf{v}_0 - \mathbf{v}^{(q)*}\|_{(\mathbf{H}_{f,0:k^*}^{(q)})^{-1}}^2 + \|\mathbf{v}_0 - \mathbf{v}^{(q)*}\|_{\mathbf{H}_{f,k^*:\infty}^{(q)}}^2.
$$

Applying $\mathbf{v}_0 = \mathbf{0}$ completes the proof. $\qquad\square$

### C.4.2. MULTIPLICATIVE QUANTIZATION

Let $\mathbf{S}_n^{(M)} = \sum_{t=0}^{n-1} \mathbf{B}_t^{(M)}$.

**Lemma C.14** (Initial Study of $\mathbf{S}_t^{(M)}$). *For* $1 \leq t \leq N$,

$$
\mathbf{S}_t^{(M)} \preceq (\mathcal{I} - \gamma\tilde{\mathcal{T}}^{(q)}) \circ \mathbf{S}_{t-1}^{(M)} + (1+\tilde{\epsilon})\gamma^2 \mathcal{M}^{(q)} \circ \mathbf{S}_N^{(M)} + \mathbf{B}_0.
$$

*Proof.* The proof is similar to the proof for Lemma C.11.

$$
\begin{aligned}
\mathbf{S}_t^{(M)} &= (\mathcal{I} - \gamma\mathcal{T}^{(q)} + \tilde{\epsilon}\gamma^2 \mathcal{M}^{(q)}) \circ \mathbf{S}_{t-1}^{(M)} + \mathbf{B}_0 \\
&= (\mathcal{I} - \gamma\tilde{\mathcal{T}}^{(q)}) \circ \mathbf{S}_{t-1} + \gamma(\tilde{\mathcal{T}}^{(q)} - \mathcal{T}^{(q)}) \circ \mathbf{S}_{t-1}^{(M)} + \tilde{\epsilon}\gamma^2 \mathcal{M}^{(q)} \circ \mathbf{S}_{t-1}^{(M)} + \mathbf{B}_0 \\
&= (\mathcal{I} - \gamma\tilde{\mathcal{T}}^{(q)}) \circ \mathbf{S}_{t-1}^{(M)} + \gamma^2((1+\tilde{\epsilon})\mathcal{M}^{(q)} - \widetilde{\mathcal{M}}^{(q)}) \circ \mathbf{S}_{t-1}^{(M)} + \mathbf{B}_0 \\
&\preceq (\mathcal{I} - \gamma\tilde{\mathcal{T}}^{(q)}) \circ \mathbf{S}_{t-1}^{(M)} + (1+\tilde{\epsilon})\gamma^2 \mathcal{M}^{(q)} \circ \mathbf{S}_N^{(M)} + \mathbf{B}_0.
\end{aligned}
$$

$\qquad\square$

**Lemma C.15** (A Bound for $\mathcal{M}^{(q)} \circ \mathbf{S}_t^{(M)}$). *For* $1 \leq t \leq N$, *under Assumption 3.3, Assumption 3.4, if* $\gamma < \frac{1}{(1+\tilde{\epsilon})\alpha \mathrm{tr}(\mathbf{H}_f^{(q)})}$,

$$
\mathcal{M}^{(q)} \circ \mathbf{S}_t^{(M)} \preceq \frac{\alpha \cdot \mathrm{tr}\left( \left[ \mathcal{I} - (\mathcal{I} - \gamma\tilde{\mathcal{T}}^{(q)})^t \right] \circ \mathbf{B}_0 \right)}{\gamma(1 - (1+\tilde{\epsilon})\gamma\alpha\,\mathrm{tr}(\mathbf{H}_f^{(q)}))} \cdot \mathbf{H}_f^{(q)}.
$$

*Proof.* The first step is to derive a crude bound for $\mathbf{S}_t^{(M)}$. Take summation via the update rule, we have [7]

$$
\mathbf{S}_t^{(M)} = \sum_{k=0}^{t-1} (\mathcal{I} - \gamma\mathcal{T}^{(q)} + \tilde{\epsilon}\gamma^2 \mathcal{M}^{(q)})^k \circ \mathbf{B}_0 = \gamma^{-1}(\mathcal{T}^{(q)} - \tilde{\epsilon}\gamma\mathcal{M}^{(q)})^{-1} \circ \left[ \mathcal{I} - (\mathcal{I} - \gamma\mathcal{T}^{(q)} + \tilde{\epsilon}\gamma^2 \mathcal{M}^{(q)})^t \right] \circ \mathbf{B}_0.
$$

Note that

$$
\mathcal{I} - \gamma\tilde{\mathcal{T}}^{(q)} \preceq \mathcal{I} - \gamma\mathcal{T}^{(q)}, \quad (\mathcal{I} - (\mathcal{I} - \gamma\mathcal{T}^{(q)} + \tilde{\epsilon}\gamma^2 \mathcal{M}^{(q)})^t) \preceq (\mathcal{I} - (\mathcal{I} - \gamma\tilde{\mathcal{T}}^{(q)} + \tilde{\epsilon}\gamma^2 \mathcal{M}^{(q)})^t),
$$

we obtain

$$
\mathbf{S}_t^{(M)} \preceq \gamma^{-1}(\mathcal{T}^{(q)} - \tilde{\epsilon}\gamma\mathcal{M}^{(q)})^{-1} \circ (\mathcal{I} - (\mathcal{I} - \gamma\tilde{\mathcal{T}}^{(q)} + \tilde{\epsilon}\gamma^2 \mathcal{M}^{(q)})^t) \circ \mathbf{B}_0.
$$

Denote $\mathbf{A} := (\mathcal{I} - (\mathcal{I} - \gamma\tilde{\mathcal{T}}^{(q)} + \tilde{\epsilon}\gamma^2 \mathcal{M}^{(q)})^t) \circ \mathbf{B}_0$, then

$$
\tilde{\mathcal{T}}^{(q)} \circ (\mathcal{T}^{(q)} - \tilde{\epsilon}\gamma\mathcal{M}^{(q)})^{-1} \circ \mathbf{A} \preceq (1+\tilde{\epsilon})\gamma\mathcal{M}^{(q)} \circ (\mathcal{T}^{(q)} - \tilde{\epsilon}\gamma\mathcal{M}^{(q)})^{-1} \circ \mathbf{A} + \mathbf{A}.
$$

---

[7] $(\mathcal{T}^{(q)} - \tilde{\epsilon}\gamma\mathcal{M}^{(q)})^{-1}$ is a PSD mapping under the condition that $\gamma < \frac{1}{(1+\tilde{\epsilon})\alpha \mathrm{tr}(\mathbf{H}_f^{(q)})}$, which can be directly deduced by Lemma B.1 in Zou et al. (2021). We omit the proof here for simplicity.

Therefore

$$(\mathcal{T}^{(q)} - \tilde{\epsilon}\gamma\mathcal{M}^{(q)})^{-1} \circ \mathbf{A} \preceq (1+\tilde{\epsilon})\gamma(\widetilde{\mathcal{T}}^{(q)})^{-1} \circ \mathcal{M}^{(q)} \circ (\mathcal{T}^{(q)} - \tilde{\epsilon}\gamma\mathcal{M}^{(q)})^{-1} \circ \mathbf{A} + (\widetilde{\mathcal{T}}^{(q)})^{-1} \circ \mathbf{A}.$$

Then we undertake the second step, applying $\mathcal{M}^{(q)}$ on both sides.

$$\mathcal{M}^{(q)} \circ (\mathcal{T}^{(q)} - \tilde{\epsilon}\gamma\mathcal{M}^{(q)})^{-1} \circ \mathbf{A} \preceq \sum_{t=0}^{\infty} ((1+\tilde{\epsilon})\gamma\mathcal{M}^{(q)} \circ (\widetilde{\mathcal{T}}^{(q)})^{-1})^t \circ (\mathcal{M}^{(q)} \circ (\widetilde{\mathcal{T}}^{(q)})^{-1} \circ \mathbf{A}). \tag{37}$$

By Assumption 3.4,

$$\begin{aligned}
\mathcal{M}^{(q)} \circ (\widetilde{\mathcal{T}}^{(q)})^{-1} \circ \mathbf{A} &\preceq \alpha\operatorname{tr}(\mathbf{H}_f^{(q)}(\widetilde{\mathcal{T}}^{(q)})^{-1} \circ \mathbf{A})\mathbf{H}_f^{(q)} \\
&= \alpha\gamma\operatorname{tr}\left(\sum_{t=0}^{\infty}\mathbf{H}_f^{(q)}(\mathbf{I} - \gamma\mathbf{H}_f^{(q)})^t \mathbf{A}(\mathbf{I} - \gamma\mathbf{H}_f^{(q)})^t\right)\mathbf{H}_f^{(q)} \\
&= \alpha\operatorname{tr}\left(\mathbf{H}_f^{(q)}(2\mathbf{H}_f^{(q)} - \gamma(\mathbf{H}_f^{(q)})^2)^{-1}\mathbf{A}\right)\mathbf{H}_f^{(q)} \\
&\preceq \alpha\operatorname{tr}(\mathbf{A})\mathbf{H}_f^{(q)},
\end{aligned} \tag{38}$$

where the last inequality requires the condition that $\gamma < \frac{1}{\alpha\operatorname{tr}(\mathbf{H}_f^{(q)})}$. Hence, by (37), (38), and further by $(\widetilde{\mathcal{T}}^{(q)})^{-1}\mathbf{H}_f^{(q)} \preceq \mathbf{I}$ and $\mathcal{M}^{(q)} \circ \mathbf{I} \preceq \alpha\operatorname{tr}(\mathbf{H}_f^{(q)})\mathbf{H}_f^{(q)}$, we obtain

$$\begin{aligned}
\mathcal{M}^{(q)} \circ ((\mathcal{T}^{(q)} - \tilde{\epsilon}\gamma\mathcal{M}^{(q)})^{-1} \circ \mathbf{A}) &\preceq \sum_{t=0}^{\infty}((1+\tilde{\epsilon})\gamma\mathcal{M}^{(q)} \circ (\widetilde{\mathcal{T}}^{(q)})^{-1})^t \circ (\mathcal{M}^{(q)} \circ (\widetilde{\mathcal{T}}^{(q)})^{-1} \circ \mathbf{A}) \\
&\preceq \alpha\operatorname{tr}(\mathbf{A})\sum_{t=0}^{\infty}((1+\tilde{\epsilon})\gamma\alpha\operatorname{tr}(\mathbf{H}_f^{(q)}))^t\mathbf{H}_f^{(q)} \\
&\preceq \frac{\alpha\operatorname{tr}(\mathbf{A})}{1 - (1+\tilde{\epsilon})\gamma\alpha\operatorname{tr}(\mathbf{H}_f^{(q)})} \cdot \mathbf{H}_f^{(q)}.
\end{aligned}$$

Therefore,

$$\mathcal{M}^{(q)} \circ \mathbf{S}_t^{(M)} \preceq \gamma^{-1}\frac{\alpha\operatorname{tr}(\mathbf{A})}{1 - (1+\tilde{\epsilon})\gamma\alpha\operatorname{tr}(\mathbf{H}_f^{(q)})} \cdot \mathbf{H}_f^{(q)} \preceq \frac{\alpha \cdot \operatorname{tr}\left(\left[\mathcal{I} - (\mathcal{I} - \gamma\widetilde{\mathcal{T}}^{(q)})^t\right] \circ \mathbf{B}_0\right)}{\gamma(1 - (1+\tilde{\epsilon})\gamma\alpha\operatorname{tr}(\mathbf{H}_f^{(q)}))} \cdot \mathbf{H}_f^{(q)}.$$

$\square$

**Lemma C.16** (A bias upper bound under multiplicative quantization). *Under Assumption 3.3, Assumption 3.4, if the stepsize satisfies $\gamma < \frac{1}{\alpha\operatorname{tr}(\mathbf{H}_f^{(q)})}$, then*

$$\begin{aligned}
&\frac{1}{N^2} \cdot \sum_{t=0}^{N-1}\sum_{k=t}^{N-1}\left\langle(\mathbf{I} - \gamma\mathbf{H}_f^{(q)})^{k-t}\mathbf{H}_f^{(q)}, \mathbf{B}_t^{(M)}\right\rangle \\
&\leq \frac{2(1+\tilde{\epsilon})\alpha\left(\|\mathbf{v}^{(q)*}\|_{\mathbf{I}_{f,0:k^*}^{(q)}}^2 + N\gamma\|\mathbf{v}^{(q)*}\|_{\mathbf{H}_{f,k^*:\infty}^{(q)}}^2\right)}{N\gamma(1 - (1+\tilde{\epsilon})\gamma\alpha\operatorname{tr}(\mathbf{H}_f^{(q)}))} \cdot \left(\frac{k^*}{N} + N\gamma^2\sum_{i>k^*}(\tilde{\lambda}_i^{(q)})^2\right) \\
&\quad + \frac{1}{\gamma^2 N^2} \cdot \|\mathbf{v}^{(q)*}\|_{(\mathbf{H}_{f,0:k^*}^{(q)})^{-1}}^2 + \|\mathbf{v}^{(q)*}\|_{\mathbf{H}_{f,k^*:\infty}^{(q)}}^2.
\end{aligned}$$

*Proof.* Recalling Lemma C.14, we can derive a refined upper bound for $\mathbf{S}_t$ by Lemma C.15:

$$
\begin{aligned}
\mathbf{S}_t^{(M)} \preceq & (\mathcal{I} - \gamma\tilde{\mathcal{T}}^{(q)}) \circ \mathbf{S}_{t-1}^{(M)} + (1+\tilde{\epsilon})\gamma^2\mathcal{M}^{(q)} \circ \mathbf{S}_N^{(M)} + \mathbf{B}_0 \\
\preceq & (\mathcal{I} - \gamma\tilde{\mathcal{T}}^{(q)}) \circ \mathbf{S}_{t-1}^{(M)} + \frac{(1+\tilde{\epsilon})\gamma\alpha \cdot \operatorname{tr}\left(\left[\mathcal{I} - (\mathcal{I} - \gamma\tilde{\mathcal{T}}^{(q)})^N\right] \circ \mathbf{B}_0\right)}{(1-(1+\tilde{\epsilon})\gamma\alpha\operatorname{tr}(\mathbf{H}_f^{(q)}))} \cdot \mathbf{H}_f^{(q)} + \mathbf{B}_0 \\
= & \sum_{k=0}^{t-1} (\mathcal{I} - \gamma\tilde{\mathcal{T}}^{(q)})^k \left(\frac{(1+\tilde{\epsilon})\gamma\alpha \cdot \operatorname{tr}\left(\left[\mathcal{I} - (\mathcal{I} - \gamma\tilde{\mathcal{T}}^{(q)})^N\right] \circ \mathbf{B}_0\right)}{(1-(1+\tilde{\epsilon})\gamma\alpha\operatorname{tr}(\mathbf{H}_f^{(q)}))} \cdot \mathbf{H}_f^{(q)} + \mathbf{B}_0\right) \\
= & \sum_{k=0}^{t-1} (\mathbf{I} - \gamma\mathbf{H}_f^{(q)})^k \left(\frac{(1+\tilde{\epsilon})\gamma\alpha \cdot \operatorname{tr}\left(\mathbf{B}_0 - (\mathbf{I} - \gamma\mathbf{H}_f^{(q)})^N\mathbf{B}_0(\mathbf{I} - \gamma\mathbf{H}_f^{(q)})^N\right)}{(1-(1+\tilde{\epsilon})\gamma\alpha\operatorname{tr}(\mathbf{H}_f^{(q)}))} \cdot \mathbf{H}_f^{(q)} + \mathbf{B}_0\right) (\mathbf{I} - \gamma\mathbf{H}_f^{(q)})^k.
\end{aligned}
\tag{39}
$$

Before providing our upper bound for the bias error, we denote

$$
\mathbf{B}_{a,b} := \mathbf{B}_a - (\mathbf{I} - \gamma\mathbf{H}_f^{(q)})^{b-a}\mathbf{B}_a(\mathbf{I} - \gamma\mathbf{H}_f^{(q)})^{b-a}.
$$

Then by (39),

$$
\begin{aligned}
& \frac{1}{N^2} \cdot \sum_{t=0}^{N-1}\sum_{k=t}^{N-1} \left\langle (\mathbf{I} - \gamma\mathbf{H}_f^{(q)})^{k-t}\mathbf{H}_f^{(q)}, \mathbf{B}_t^{(M)}\right\rangle \\
= & \frac{1}{\gamma N^2} \sum_{t=0}^{N-1} \left\langle \mathbf{I} - (\mathbf{I} - \gamma\mathbf{H}_f^{(q)})^{N-t}, \mathbf{B}_t^{(M)}\right\rangle \\
\leq & \frac{1}{\gamma N^2} \langle \mathbf{I} - (\mathbf{I} - \gamma\mathbf{H}_f^{(q)})^N, \sum_{t=0}^{N-1}\mathbf{B}_t^{(M)}\rangle \\
\leq & \frac{1}{\gamma N^2} \sum_{k=0}^{N-1} \left\langle \mathbf{I} - (\mathbf{I} - \gamma\mathbf{H}_f^{(q)})^N, (\mathbf{I} - \gamma\mathbf{H}_f^{(q)})^k \left(\frac{(1+\tilde{\epsilon})\gamma\alpha \cdot \operatorname{tr}(\mathbf{B}_{0,N})}{1-(1+\tilde{\epsilon})\gamma\alpha\operatorname{tr}(\mathbf{H}_f^{(q)})} \cdot \mathbf{H}_f^{(q)} + \mathbf{B}_0\right)(\mathbf{I} - \gamma\mathbf{H}_f^{(q)})^k\right\rangle \\
= & \frac{1}{\gamma N^2} \sum_{k=0}^{N-1} \left\langle (\mathbf{I} - \gamma\mathbf{H}_f^{(q)})^{2k} - (\mathbf{I} - \gamma\mathbf{H}_f^{(q)})^{N+2k}, \left(\frac{(1+\tilde{\epsilon})\gamma\alpha \cdot \operatorname{tr}(\mathbf{B}_{0,N})}{1-(1+\tilde{\epsilon})\gamma\alpha\operatorname{tr}(\mathbf{H}_f^{(q)})} \cdot \mathbf{H}_f^{(q)} + \mathbf{B}_0\right)\right\rangle.
\end{aligned}
$$

Note that

$$
\begin{aligned}
(\mathbf{I} - \gamma\mathbf{H}_f^{(q)})^{2k} - (\mathbf{I} - \gamma\mathbf{H}_f^{(q)})^{N+2k} &= \left(\mathbf{I} - \gamma\mathbf{H}_f^{(q)}\right)^k \left(\left(\mathbf{I} - \gamma\mathbf{H}_f^{(q)}\right)^k - \left(\mathbf{I} - \gamma\mathbf{H}_f^{(q)}\right)^{N+k}\right) \\
&\preceq (\mathbf{I} - \gamma\mathbf{H}_f^{(q)})^k - (\mathbf{I} - \gamma\mathbf{H}_f^{(q)})^{N+k},
\end{aligned}
$$

we obtain

$$
\begin{aligned}
& \frac{1}{N^2} \cdot \sum_{t=0}^{N-1}\sum_{k=t}^{N-1} \left\langle (\mathbf{I} - \gamma\mathbf{H}_f^{(q)})^{k-t}\mathbf{H}_f^{(q)}, \mathbf{B}_t^{(M)}\right\rangle \\
\leq & \frac{1}{\gamma N^2} \sum_{k=0}^{N-1} \left\langle (\mathbf{I} - \gamma\mathbf{H}_f^{(q)})^k - (\mathbf{I} - \gamma\mathbf{H}_f^{(q)})^{N+k}, \frac{(1+\tilde{\epsilon})\gamma\alpha \cdot \operatorname{tr}(\mathbf{B}_{0,N})}{1-(1+\tilde{\epsilon})\gamma\alpha\operatorname{tr}(\mathbf{H}_f^{(q)})} \cdot \mathbf{H}_f^{(q)} + \mathbf{B}_0\right\rangle.
\end{aligned}
$$

Therefore, it suffices to upper bound the following two terms

$$
\begin{aligned}
I_1 &= \frac{(1+\tilde{\epsilon})\alpha\operatorname{tr}(\mathbf{B}_{0,N})}{N^2(1-(1+\tilde{\epsilon})\gamma\alpha\operatorname{tr}(\mathbf{H}_f^{(q)}))} \sum_{k=0}^{N-1} \left\langle (\mathbf{I} - \gamma\mathbf{H}_f^{(q)})^k - (\mathbf{I} - \gamma\mathbf{H}_f^{(q)})^{N+k}, \mathbf{H}_f^{(q)}\right\rangle \\
I_2 &= \frac{1}{\gamma N^2} \sum_{k=0}^{N-1} \left\langle (\mathbf{I} - \gamma\mathbf{H}_f^{(q)})^k - (\mathbf{I} - \gamma\mathbf{H}_f^{(q)})^{N+k}, \mathbf{B}_0\right\rangle.
\end{aligned}
$$

Repeating the computation in the proof of Lemma C.13,

$$I_1 \leq \frac{2(1+\tilde{\epsilon})\alpha \left( \|\mathbf{v}_0 - \mathbf{v}^{(q)*}\|^2_{\mathbf{I}^{(q)}_{f,0:k^*}} + N\gamma \|\mathbf{v}_0 - \mathbf{v}^{(q)*}\|^2_{\mathbf{H}^{(q)}_{f,k^*:\infty}} \right)}{N\gamma(1 - (1+\tilde{\epsilon})\gamma\alpha \operatorname{tr}(\mathbf{H}^{(q)}_f))} \cdot \left( \frac{k^*}{N} + N\gamma^2 \sum_{i>k^*} (\tilde{\lambda}^{(q)}_i)^2 \right).$$

$$I_2 \leq \frac{1}{\gamma^2 N^2} \cdot \|\mathbf{v}_0 - \mathbf{v}^{(q)*}\|^2_{(\mathbf{H}^{(q)}_{f,0:k^*})^{-1}} + \|\mathbf{v}_0 - \mathbf{v}^{(q)*}\|^2_{\mathbf{H}^{(q)}_{f,k^*:\infty}}.$$

$\square$

## C.5. Final Upper Bounds

### C.5.1. GENERAL QUANTIZATION

**Theorem C.17.** *Suppose* $\gamma < 1/\left( \alpha\operatorname{tr}\left(\mathbf{H}^{(q)}_f\right) \right)$. *Under Assumption 3.1, 3.3, 3.4 and 3.5,*

$$R^{(0)}_N \leq 2\mathrm{BiasError} + 2\mathrm{VarianceError},$$

*where*

$$\mathrm{BiasError} \leq \frac{1}{\gamma^2 N^2} \cdot \left\| \mathbf{v}^{(q)*} \right\|^2_{(\mathbf{H}^{(q)}_{f,0:k^*})^{-1}} + \left\| \mathbf{v}^{(q)*} \right\|^2_{\mathbf{H}^{(q)}_{f,k^*:\infty}},$$

$$\mathrm{VarianceError} \leq \frac{\sigma^2_G + 2\alpha \left( \frac{\|\mathbf{v}^{(q)*}\|^2_{\mathbf{I}^{(q)}_{f,0:k^*}}}{N\gamma} + \left\| \mathbf{v}^{(q)*} \right\|^2_{\mathbf{H}^{(q)}_{f,k^*:\infty}} \right)}{1 - \gamma\alpha\operatorname{tr}(\mathbf{H}^{(q)}_f)} \left( \frac{k^*}{N} + N\gamma^2 \cdot \sum_{i>k^*} (\tilde{\lambda}^{(q)}_i)^2 \right).$$

*Here* $k^* = \max\{i : \tilde{\lambda}^{(q)}_i \geq 1/(\gamma N)\}$,

$$\sigma^2_G = \overline{\sigma}^2 + \sup_t \alpha\operatorname{tr}\left( \mathbf{H}^{(q)}_f \mathbb{E}\left[ \boldsymbol{\epsilon}^{(p)}_{t-1} \boldsymbol{\epsilon}^{(p)\top}_{t-1} \right] \right) + \sup_t \left( \mathbb{E}\left[ \epsilon^{(a)^2}_t \Big| a_t \right] + \mathbb{E}\left[ \epsilon^{(o)^2}_t \Big| o_t \right] \right).$$

*Proof.* The proof can be completed by Lemma C.6, Lemma C.8 and Lemma C.13. $\square$

### C.5.2. MULTIPLICATIVE QUANTIZATION

**Theorem C.18.** *Suppose* $\gamma < 1/\left( (1+\tilde{\epsilon})\alpha\operatorname{tr}\left(\mathbf{H}^{(q)}_f\right) \right)$. *If there exist* $\overline{\epsilon}_p, \overline{\epsilon}_a$ *and* $\overline{\epsilon}_o$ *such that for any* $i \in \{p, a, o\}$, *quantization* $\mathcal{Q}_i$ *is* $\overline{\epsilon}_i$-*multiplicative, then under Assumption 3.1, 3.3, 3.4 and 3.5,*

$$R^{(0)}_N \leq 2\mathrm{BiasError} + 2\mathrm{VarianceError},$$

*where*

$$\mathrm{BiasError} \leq \frac{1}{\gamma^2 N^2} \cdot \left\| \mathbf{v}^{(q)*} \right\|^2_{(\mathbf{H}^{(q)}_{f,0:k^*})^{-1}} + \left\| \mathbf{v}^{(q)*} \right\|^2_{\mathbf{H}^{(q)}_{f,k^*:\infty}},$$

$$\mathrm{VarianceError} \leq \frac{\sigma^2_M + 2(1+\tilde{\epsilon})\alpha \left( \frac{\|\mathbf{v}^{(q)*}\|^2_{\mathbf{I}^{(q)}_{f,0:k^*}}}{N\gamma} + \left\| \mathbf{v}^{(q)*} \right\|^2_{\mathbf{H}^{(q)}_{f,k^*:\infty}} \right)}{1 - (1+\tilde{\epsilon})\gamma\alpha\operatorname{tr}(\mathbf{H}^{(q)}_f)} \left( \frac{k^*}{N} + N\gamma^2 \cdot \sum_{i>k^*} (\tilde{\lambda}^{(q)}_i)^2 \right).$$

*Here* $k^* = \max\{i : \tilde{\lambda}^{(q)}_i \geq 1/(\gamma N)\}$ *and*

$$\tilde{\epsilon} = 4\overline{\epsilon}_o + 2(2\overline{\epsilon}_o + 1)\left[2(1+\overline{\epsilon}_p)\overline{\epsilon}_a + 2\overline{\epsilon}_p\right],$$

$$\sigma^2_M = (2\overline{\epsilon}_o + 1)\overline{\sigma}^2 + (2\overline{\epsilon}_o + 1)\left[2\overline{\epsilon}_p + 2(1+\overline{\epsilon}_p)\overline{\epsilon}_a\right]\alpha\operatorname{tr}\left( \mathbf{H}^{(q)}_f \mathbf{v}^{(q)*} \mathbf{v}^{(q)*\top} \right).$$

*Proof.* The proof can be completed by Lemma C.5, Lemma C.10 and Lemma C.16. $\square$

### C.6. Additive Error Upper Bounds under Power-law Spectrum

Here we analyze the additive error in Lemma B.2, and take expectation on $\mathbf{w}^*$. Denote

$$
\begin{aligned}
\text{AdditiveError} = &\frac{1}{2} \left\langle \mathbf{SHS}^\top, (\mathbf{v}^{(q)^*} - \mathbf{v}^*) \otimes (\mathbf{v}^{(q)^*} - \mathbf{v}^*) \right\rangle \\
&+ \left( \mathbf{v}^{(q)^*} \right)^\top \frac{1}{N\gamma} \left[ \mathbf{I} - \left( \mathbf{I} - \gamma \mathbf{H}_f^{(q)} \right)^N \right] \left( \mathbf{H}_f^{(q)} \right)^{-1} \left( \mathbf{H}_f^{(q)} - \mathbf{SHS}^\top \right) \mathbf{v}^{(q)^*}.
\end{aligned}
$$

Recall that

$$
\mathbf{v}^* = \left( \mathbf{SHS}^\top \right)^{-1} \mathbf{SHw}^*, \quad \mathbf{v}^{(q)^*} = (\mathbf{H}_f^{(q)})^{-1} \mathbf{SHw}^*.
$$

Denote $\mathbf{D} = \mathbf{H}_f^{(q)} - \mathbf{SHS}^\top$, then

$$
\mathbf{v}^{(q)^*} = \left( \mathbf{SHS}^\top + \mathbf{D} \right)^{-1} \mathbf{SHS}^\top \mathbf{v}^*.
$$

It follows that

$$
\mathbf{v}^* - \mathbf{v}^{(q)^*} = \left( \mathbf{SHS}^\top + \mathbf{D} \right)^{-1} \mathbf{Dv}^*.
$$

Hence,

$$
\frac{1}{2} \left\langle \mathbf{SHS}^\top, (\mathbf{v}^{(q)^*} - \mathbf{v}^*) \otimes (\mathbf{v}^{(q)^*} - \mathbf{v}^*) \right\rangle = \frac{1}{2} \| \mathbf{w}^* \|_{\mathbf{S}_1}^2, \tag{40}
$$

where

$$
\mathbf{S}_1 = \mathbf{HS}^\top \left( \mathbf{SHS}^\top \right)^{-1} \mathbf{D} \left( \mathbf{SHS}^\top + \mathbf{D} \right)^{-1} \mathbf{SHS}^\top \left( \mathbf{SHS}^\top + \mathbf{D} \right)^{-1} \mathbf{D} \left( \mathbf{SHS}^\top \right)^{-1} \mathbf{SH}.
$$

Next, we derive upper bounds for Additive via taking expectation on $\mathbf{w}^*$.

**Lemma C.19** (Additive Error under multiplicative quantization, an upper bound). *Under Assumption 3.1, 3.3 and 3.6, for any $i \in \{s, d, f\}$, if there exist $(\bar{\epsilon}_i, \underline{\epsilon}_i)$ such that quantization $\mathcal{Q}_i$ is $(\bar{\epsilon}_i, \underline{\epsilon}_i)$-multiplicative,*

$$
\mathbb{E}_{\mathbf{w}^*} \| \mathbf{w}^* \|_{\mathbf{S}_1}^2 \lesssim \frac{[(1 + \bar{\epsilon}_d)(1 + \bar{\epsilon}_f)(1 + \bar{\epsilon}_s) - 1]^2}{[(1 + \bar{\epsilon}_d)(1 + \bar{\epsilon}_f)(1 + \bar{\epsilon}_s)]^2},
$$

$$
\mathbb{E}_{\mathbf{w}^*} \left[ \left( \mathbf{v}^{(q)^*} \right)^\top \frac{1}{N\gamma} \left[ \mathbf{I} - \left( \mathbf{I} - \gamma \mathbf{H}_f^{(q)} \right)^N \right] \left( \mathbf{H}_f^{(q)} \right)^{-1} \left( \mathbf{H}_f^{(q)} - \mathbf{SHS}^\top \right) \mathbf{v}^{(q)^*} \right]
$$

$$
\lesssim \frac{(1 + \bar{\epsilon}_f)(1 + \bar{\epsilon}_d)(1 + \bar{\epsilon}_s) - 1}{(1 + \bar{\epsilon}_f)(1 + \bar{\epsilon}_d)(1 + \bar{\epsilon}_s)(1 + \underline{\epsilon}_f)(1 + \underline{\epsilon}_d)(1 + \underline{\epsilon}_s)}.
$$

*Proof.* Regarding the first inequality, noticing that under multiplicative quantization,

$$
\mathbf{H}_f^{(q)} \preceq (1 + \bar{\epsilon}_f)(1 + \bar{\epsilon}_d)(1 + \bar{\epsilon}_s) \mathbf{SHS}^\top,
$$

it follows that

$$
\mathbf{D} = \mathbf{H}_f^{(q)} - \mathbf{SHS}^\top \preceq [(1 + \bar{\epsilon}_f)(1 + \bar{\epsilon}_d)(1 + \bar{\epsilon}_s) - 1] \mathbf{SHS}^\top.
$$

Further by Assumption 3.6,

$$
\mathbb{E}_{\mathbf{w}^*} \| \mathbf{w}^* \|_{\mathbf{H}}^2 \approx 1,
$$

then we have

$$
\begin{aligned}
\mathbb{E}_{\mathbf{w}^*} \| \mathbf{w}^* \|_{\mathbf{S}_1}^2 &\lesssim \frac{[(1 + \bar{\epsilon}_d)(1 + \bar{\epsilon}_f)(1 + \bar{\epsilon}_s) - 1]^2}{[(1 + \bar{\epsilon}_d)(1 + \bar{\epsilon}_f)(1 + \bar{\epsilon}_s)]^2} \left\| \mathbf{H}^{1/2} \mathbf{S}^\top \left( \mathbf{SHS}^\top \right)^{-1} \mathbf{SH}^{1/2} \right\| \\
&\leq \frac{[(1 + \bar{\epsilon}_d)(1 + \bar{\epsilon}_f)(1 + \bar{\epsilon}_s) - 1]^2}{[(1 + \bar{\epsilon}_d)(1 + \bar{\epsilon}_f)(1 + \bar{\epsilon}_s)]^2},
\end{aligned}
$$

where the first inequality holds by Lemma F.3. Regarding the second inequality, by Assumption 3.6, it holds

$$
\mathbb{E}\left[\left(\mathbf{v}^{(q)*}\right)^{\top}\frac{1}{N\gamma}\left[\mathbf{I}-\left(\mathbf{I}-\gamma\mathbf{H}_f^{(q)}\right)^N\right]\left(\mathbf{H}_f^{(q)}\right)^{-1}\left(\mathbf{H}_f^{(q)}-\mathbf{SHS}^{\top}\right)\mathbf{v}^{(q)*}\right]
$$

$$
=\mathbb{E}\left[\mathbf{w}^{*\top}\mathbf{HS}^{\top}(\mathbf{H}_f^{(q)})^{-1}\frac{1}{N\gamma}\left[\mathbf{I}-\left(\mathbf{I}-\gamma\mathbf{H}_f^{(q)}\right)^N\right]\left(\mathbf{H}_f^{(q)}\right)^{-1}\left(\mathbf{H}_f^{(q)}-\mathbf{SHS}^{\top}\right)(\mathbf{H}_f^{(q)})^{-1}\mathbf{SHw}^*\right]
$$

$$
\lesssim\left\|\mathbf{H}^{1/2}\mathbf{S}^{\top}(\mathbf{H}_f^{(q)})^{-1}\frac{1}{N\gamma}\left[\mathbf{I}-\left(\mathbf{I}-\gamma\mathbf{H}_f^{(q)}\right)^N\right]\left(\mathbf{H}_f^{(q)}\right)^{-1}\left(\mathbf{H}_f^{(q)}-\mathbf{SHS}^{\top}\right)(\mathbf{H}_f^{(q)})^{-1}\mathbf{SH}^{1/2}\right\| \tag{41}
$$

$$
\leq\left\|(\mathbf{H}_f^{(q)})^{-\frac{1}{2}}\frac{1}{N\gamma}\left[\mathbf{I}-\left(\mathbf{I}-\gamma\mathbf{H}_f^{(q)}\right)^N\right](\mathbf{H}_f^{(q)})^{-\frac{1}{2}}\right\|\cdot\left\|(\mathbf{H}_f^{(q)})^{-\frac{1}{2}}\left(\mathbf{H}_f^{(q)}-\mathbf{SHS}^{\top}\right)(\mathbf{H}_f^{(q)})^{-\frac{1}{2}}\right\|
$$

$$
\cdot\left\|(\mathbf{H}_f^{(q)})^{-\frac{1}{2}}\mathbf{SHS}^{\top}(\mathbf{H}_f^{(q)})^{-\frac{1}{2}}\right\|.
$$

Noticing that

$$
[(1+\underline{\epsilon}_f)(1+\underline{\epsilon}_d)(1+\underline{\epsilon}_s)-1]\mathbf{SHS}^{\top}\preceq\mathbf{D}\preceq[(1+\overline{\epsilon}_f)(1+\overline{\epsilon}_d)(1+\overline{\epsilon}_s)-1]\mathbf{SHS}^{\top}.
$$

Firstly, by Lemma F.2,

$$
\left\|(\mathbf{H}_f^{(q)})^{-\frac{1}{2}}\left(\mathbf{H}_f^{(q)}-\mathbf{SHS}^{\top}\right)(\mathbf{H}_f^{(q)})^{-\frac{1}{2}}\right\|\leq\frac{(1+\overline{\epsilon}_f)(1+\overline{\epsilon}_d)(1+\overline{\epsilon}_s)-1}{(1+\overline{\epsilon}_f)(1+\overline{\epsilon}_d)(1+\overline{\epsilon}_s)}. \tag{42}
$$

Secondly, by Lemma F.1,

$$
\left\|(\mathbf{H}_f^{(q)})^{-\frac{1}{2}}\mathbf{SHS}^{\top}(\mathbf{H}_f^{(q)})^{-\frac{1}{2}}\right\|\leq\frac{1}{(1+\underline{\epsilon}_f)(1+\underline{\epsilon}_d)(1+\underline{\epsilon}_s)}. \tag{43}
$$

Thirdly,

$$
\left\|(\mathbf{H}_f^{(q)})^{-\frac{1}{2}}\frac{1}{N\gamma}\left[\mathbf{I}-\left(\mathbf{I}-\gamma\mathbf{H}_f^{(q)}\right)^N\right](\mathbf{H}_f^{(q)})^{-\frac{1}{2}}\right\|=\frac{1}{N\gamma}\max_i\frac{1-\left(1-\gamma\tilde{\lambda}_i^{(q)}\right)^N}{\tilde{\lambda}_i^{(q)}}
$$

$$
\leq\frac{1}{N\gamma}\max_i\frac{\min\left\{1,\gamma N\tilde{\lambda}_i^{(q)}\right\}}{\tilde{\lambda}_i^{(q)}} \tag{44}
$$

$$
=\frac{1}{N\gamma}\max_i\min\left\{\frac{1}{\tilde{\lambda}_i^{(q)}},\gamma N\right\}
$$

$$
\leq 1.
$$

Therefore, (41), (42), (43), and (44), we have

$$
\mathbb{E}_{\mathbf{w}^*}\left[\left(\mathbf{v}^{(q)*}\right)^{\top}\frac{1}{N\gamma}\left[\mathbf{I}-\left(\mathbf{I}-\gamma\mathbf{H}_f^{(q)}\right)^N\right]\left(\mathbf{H}_f^{(q)}\right)^{-1}\left(\mathbf{H}_f^{(q)}-\mathbf{SHS}^{\top}\right)\mathbf{v}^{(q)*}\right]
$$

$$
\lesssim\frac{(1+\overline{\epsilon}_f)(1+\overline{\epsilon}_d)(1+\overline{\epsilon}_s)-1}{(1+\overline{\epsilon}_f)(1+\overline{\epsilon}_d)(1+\overline{\epsilon}_s)(1+\underline{\epsilon}_f)(1+\underline{\epsilon}_d)(1+\underline{\epsilon}_s)}.
$$

$\square$

**Lemma C.20** (Additive Error under additive quantization, an upper bound). *Under Assumption 3.1, 3.3, 3.6, for any $i\in\{s,d,f\}$, if there exist $(\overline{\epsilon}_i,\underline{\epsilon}_i)$ such that quantization $\mathcal{Q}_i$ is $(\overline{\epsilon}_i,\underline{\epsilon}_i)$-additive, then with probability at least $1-e^{-\Omega(M)}$,*

$$
\mathbb{E}_{\mathbf{w}^*}\|\mathbf{w}^*\|_{\mathbf{S}_1}^2\lesssim\frac{\left(\overline{\epsilon}_s+\overline{\epsilon}_f+\overline{\epsilon}_s\overline{\epsilon}_d p+\overline{\epsilon}_d\frac{p}{M}\right)^2}{\left(M^{-a}+\overline{\epsilon}_s+\overline{\epsilon}_f+\overline{\epsilon}_s\overline{\epsilon}_d p+\overline{\epsilon}_d\frac{p}{M}\right)^2}.
$$

$$
\mathbb{E}_{\mathbf{w}^*}\left[\left(\mathbf{v}^{(q)*}\right)^{\top}\frac{1}{N\gamma}\left[\mathbf{I}-\left(\mathbf{I}-\gamma\mathbf{H}_f^{(q)}\right)^N\right]\left(\mathbf{H}_f^{(q)}\right)^{-1}\left(\mathbf{H}_f^{(q)}-\mathbf{SHS}^{\top}\right)\mathbf{v}^{(q)*}\right]
$$

$$
\lesssim\frac{\overline{\epsilon}_s+\overline{\epsilon}_s\overline{\epsilon}_d p+\overline{\epsilon}_f+\overline{\epsilon}_d\frac{p}{M}}{\overline{\epsilon}_s+\overline{\epsilon}_s\overline{\epsilon}_d p+\overline{\epsilon}_f+\overline{\epsilon}_d\frac{p}{M}+M^{-a}}\cdot\frac{1}{1+\underline{\epsilon}_s(1+\underline{\epsilon}_d p)+\underline{\epsilon}_f+\underline{\epsilon}_d\frac{p}{M}}.
$$

*Proof.* Regarding the first inequality, noticing that under additive quantization,

$$\mathbf{SHS}^\top + \underline{\epsilon}_s \text{tr}(\mathbf{H})\mathbf{I} + \underline{\epsilon}_d \mathbf{SS}^\top + (\underline{\epsilon}_s\underline{\epsilon}_d p + \underline{\epsilon}_f)\mathbf{I} \preceq \mathbf{H}_f^{(q)} \preceq \mathbf{SHS}^\top + \bar{\epsilon}_s \text{tr}(\mathbf{H})\mathbf{I} + \bar{\epsilon}_d \mathbf{SS}^\top + (\bar{\epsilon}_s\bar{\epsilon}_d p + \bar{\epsilon}_f)\mathbf{I}.$$

Then under the power-law Assumption 3.6, with probability at least $1 - e^{-\Omega(M)}$,

$$\mathbf{SHS}^\top + \left(\underline{\epsilon}_s + \underline{\epsilon}_s\underline{\epsilon}_d p + \underline{\epsilon}_f + \underline{\epsilon}_d \frac{p}{M}\right)\mathbf{I} \precsim \mathbf{H}_f^{(q)} \precsim \mathbf{SHS}^\top + \left(\bar{\epsilon}_s + \bar{\epsilon}_s\bar{\epsilon}_d p + \bar{\epsilon}_f + \bar{\epsilon}_d \frac{p}{M}\right)\mathbf{I}.$$

It follows that

$$\mathbf{D} = \mathbf{H}_f^{(q)} - \mathbf{SHS}^\top \precsim \left(\bar{\epsilon}_s + \bar{\epsilon}_s\bar{\epsilon}_d p + \bar{\epsilon}_f + \bar{\epsilon}_d \frac{p}{M}\right)\mathbf{I}.$$

Further by Assumption 3.6, we have with probability at least $1 - e^{-\Omega(M)}$,

$$\mathbb{E}_{\mathbf{w}^*} \|\mathbf{w}^*\|_{\mathbf{S}_1}^2 \precsim \frac{\left(\bar{\epsilon}_s + \bar{\epsilon}_s\bar{\epsilon}_d p + \bar{\epsilon}_f + \bar{\epsilon}_d \frac{p}{M}\right)^2}{\left(\mu_{\min}(\mathbf{SHS}^\top) + \bar{\epsilon}_s + \bar{\epsilon}_s\bar{\epsilon}_d p + \bar{\epsilon}_f + \bar{\epsilon}_d \frac{p}{M}\right)^2} \left\| \mathbf{H}^{1/2}\mathbf{S}^\top \left(\mathbf{SHS}^\top\right)^{-1}\mathbf{SH}^{1/2}\right\|$$

$$\approx \frac{\left(\bar{\epsilon}_s + \bar{\epsilon}_f + \bar{\epsilon}_s\bar{\epsilon}_d p + \bar{\epsilon}_d \frac{p}{M}\right)^2}{\left(M^{-a} + \bar{\epsilon}_s + \bar{\epsilon}_f + \bar{\epsilon}_s\bar{\epsilon}_d p + \bar{\epsilon}_d \frac{p}{M}\right)^2},$$

where the first inequality holds by Lemma F.4 and the last inequality holds by Lemma G.1. Regarding the second inequality, we prove by (41) and noticing that

$$\left(\underline{\epsilon}_s + \underline{\epsilon}_s\underline{\epsilon}_d p + \underline{\epsilon}_f + \underline{\epsilon}_d \frac{p}{M}\right)\mathbf{I} \precsim \mathbf{D} \precsim \left(\bar{\epsilon}_s + \bar{\epsilon}_s\bar{\epsilon}_d p + \bar{\epsilon}_f + \bar{\epsilon}_d \frac{p}{M}\right)\mathbf{I}.$$

Firstly, by Lemma F.2 and Lemma G.1, with probability at least $1 - e^{-\Omega(M)}$,

$$\left\|(\mathbf{H}_f^{(q)})^{-\frac{1}{2}}\left(\mathbf{H}_f^{(q)} - \mathbf{SHS}^\top\right)(\mathbf{H}_f^{(q)})^{-\frac{1}{2}}\right\| \lesssim \frac{\bar{\epsilon}_s + \bar{\epsilon}_s\bar{\epsilon}_d p + \bar{\epsilon}_f + \bar{\epsilon}_d \frac{p}{M}}{\bar{\epsilon}_s + \bar{\epsilon}_s\bar{\epsilon}_d p + \bar{\epsilon}_f + \bar{\epsilon}_d \frac{p}{M} + M^{-a}}. \tag{45}$$

By Lemma F.1 and Lemma G.1, with probability at least $1 - e^{-\Omega(M)}$,

$$\left\|(\mathbf{H}_f^{(q)})^{-\frac{1}{2}}\mathbf{SHS}^\top(\mathbf{H}_f^{(q)})^{-\frac{1}{2}}\right\| \lesssim \frac{1}{1 + \underline{\epsilon}_s(1 + \underline{\epsilon}_d p) + \underline{\epsilon}_f + \underline{\epsilon}_d \frac{p}{M}}. \tag{46}$$

Therefore, together with (41), (44), (45), and (46), we have, with probability at least $1 - e^{-\Omega(M)}$,

$$\mathbb{E}_{\mathbf{w}^*}\left[\left(\mathbf{v}^{(q)*}\right)^\top \frac{1}{N\gamma}\left[\mathbf{I} - \left(\mathbf{I} - \gamma\mathbf{H}_f^{(q)}\right)^N\right]\left(\mathbf{H}_f^{(q)}\right)^{-1}\left(\mathbf{H}_f^{(q)} - \mathbf{SHS}^\top\right)\mathbf{v}^{(q)*}\right]$$

$$\precsim \frac{\bar{\epsilon}_s + \bar{\epsilon}_s\bar{\epsilon}_d p + \bar{\epsilon}_f + \bar{\epsilon}_d \frac{p}{M}}{\bar{\epsilon}_s\bar{\epsilon}_d p + \bar{\epsilon}_f + \bar{\epsilon}_d \frac{p}{M} + M^{-a} + \bar{\epsilon}_s} \cdot \frac{1}{1 + \underline{\epsilon}_s(1 + \underline{\epsilon}_d p) + \underline{\epsilon}_f + \underline{\epsilon}_d \frac{p}{M}}.$$

□

## C.7. Variance Upper Bounds under Power-Law Spectrum

Denote

$$d_{\text{eff}} = k^* + \gamma^2 N^2 \sum_{i>k^*}(\tilde{\lambda}_i^{(q)})^2. \tag{47}$$

We then focus on bounding $d_{\text{eff}}/N$ with $k^* = \max\{k : \tilde{\lambda}_i^{(q)} \geq 1/(\gamma N)\}$ in this subsection.

### C.7.1. MULTIPLICATIVE QUANTIZATION

**Lemma C.21.** *If there exist constants $\bar{\epsilon}_s, \bar{\epsilon}_d, \bar{\epsilon}_f$ such that for $i \in \{s, d, f\}$, $\mathcal{Q}_i(\cdot)$ is $\bar{\epsilon}_i$-multiplicative, under Assumption 3.1, Assumption 3.3 and Assumption 3.6, with probability at least $1 - e^{-\Omega(M)}$ over the randomness of $\mathbf{S}$, with $d_{\text{eff}}$ defined in (47), it holds*

$$\frac{d_{\text{eff}}}{N} \precsim \frac{\min\left\{M, [N\gamma(1 + \bar{\epsilon}_f)(1 + \bar{\epsilon}_d)(1 + \bar{\epsilon}_s)]^{1/a}\right\}}{N}.$$

*Proof.* Define $k^\dagger := \max\{j : (1+\bar\epsilon_f)(1+\bar\epsilon_d)(1+\bar\epsilon_s)j^{-a} \geq 1/(\gamma N)\}$. Denote $N_{\text{eff}}^{(M)} = [N\gamma(1+\bar\epsilon_f)(1+\bar\epsilon_d)(1+\bar\epsilon_s)]^{1/a}$. By (47) and Lemma G.2, with probability at least $1 - e^{-\Omega(M)}$ over the randomness of $\mathbf{S}$,

$$
\begin{aligned}
\frac{d_{\text{eff}}}{N} &= \frac{k^* + \gamma^2 N^2 \sum_{i>k^*}(\tilde\lambda_i^{(q)})^2}{N} \\
&\leq \frac{k^\dagger + \gamma^2 N^2 \sum_{i>k^\dagger}(\tilde\lambda_i^{(q)})^2}{N} \\
&\lesssim \frac{k^\dagger + \gamma^2 N^2 \sum_{j>k^\dagger}\left[(1+\bar\epsilon_f)(1+\bar\epsilon_d)(1+\bar\epsilon_s)j^{-a}\right]^2}{N} \\
&\approx \frac{\min\left\{M, N_{\text{eff}}^{(M)} + (N_{\text{eff}}^{(M)})^{2a}(N_{\text{eff}}^{(M)})^{1-2a}\right\}}{N} \\
&\approx \frac{\min\left\{M, N_{\text{eff}}^{(M)}\right\}}{N} \\
&= \frac{\min\left\{M, [N\gamma(1+\bar\epsilon_f)(1+\bar\epsilon_d)(1+\bar\epsilon_s)]^{1/a}\right\}}{N},
\end{aligned}
$$

$\square$

### C.7.2. ADDITIVE QUANTIZATION

**Lemma C.22.** *If there exist constants $\bar\epsilon_s, \bar\epsilon_d, \bar\epsilon_f$ such that for $i \in \{s, d, f\}$, $\mathcal{Q}_i(\cdot)$ is $\bar\epsilon_i$-additive, under Assumption 3.1, Assumption 3.3 and Assumption 3.6, with probability at least $1 - e^{-\Omega(M)}$ over the randomness of $\mathbf{S}$, with $d_{\text{eff}}$ defined in (47), it holds*

$$
\frac{d_{\text{eff}}}{N} \lesssim \frac{k_{\text{eff}} + \gamma^2 N^2 \left(\bar\epsilon_f + (1+\bar\epsilon_d p)\bar\epsilon_s + \bar\epsilon_d \frac{p}{M}\right)^2 (M - k_{\text{eff}})}{N},
$$

*where*

$$
k_{\text{eff}} = \left[M^{-a} \vee \left(\frac{1}{N\gamma} - \bar\epsilon_f - (1+\bar\epsilon_d p)\bar\epsilon_s - \bar\epsilon_d \frac{p}{M}\right)\right]^{-\frac{1}{a}}.
$$

*Proof.* Define $k^\dagger := \max\{j : j^{-a} + \bar\epsilon_f + (1+\bar\epsilon_d p)\bar\epsilon_s + \bar\epsilon_d \frac{p}{M} \geq 1/(\gamma N)\}$. By (47) and Lemma G.4, with probability at least $1 - e^{-\Omega(M)}$ over the randomness of $\mathbf{S}$,

$$
\begin{aligned}
\frac{d_{\text{eff}}}{N} &= \frac{k^* + \gamma^2 N^2 \sum_{i>k^*}(\tilde\lambda_i^{(q)})^2}{N} \\
&\leq \frac{k^\dagger + \gamma^2 N^2 \sum_{i>k^\dagger}(\tilde\lambda_i^{(q)})^2}{N} \\
&\lesssim \frac{k^\dagger + \gamma^2 N^2 \sum_{j>k^\dagger}\left[j^{-a} + \bar\epsilon_f + (1+\bar\epsilon_d p)\bar\epsilon_s + \bar\epsilon_d \frac{p}{M}\right]^2}{N}.
\end{aligned}
\tag{48}
$$

We then consider two cases to complete the proof.

- Case one: $M^{-a} + \bar\epsilon_f + (1+\bar\epsilon_d p)\bar\epsilon_s + \bar\epsilon_d \frac{p}{M} < \frac{1}{N\gamma}$

  Denote

$$
N_{\text{eff}}^{(A)} = \left(\frac{1}{N\gamma} - \bar\epsilon_f - (1+\bar\epsilon_d p)\bar\epsilon_s - \bar\epsilon_d \frac{p}{M}\right)^{-\frac{1}{a}}.
$$

Then by (48), with probability at least $1 - e^{-\Omega(M)}$ over the randomness of $\mathbf{S}$,

$$
\begin{aligned}
\frac{d_{\text{eff}}}{N} &\lesssim \frac{k^{\dagger} + \gamma^2 N^2 \sum_{j>k^{\dagger}} \left[ j^{-a} + \bar{\epsilon}_f + (1 + \bar{\epsilon}_d p)\bar{\epsilon}_s + \bar{\epsilon}_d \frac{p}{M} \right]^2}{N} \\
&\approx \frac{N_{\text{eff}}^{(A)} + \gamma^2 N^2 \left[ (N_{\text{eff}}^{(A)})^{1-2a} + \left( \bar{\epsilon}_f + (1 + \bar{\epsilon}_d p)\bar{\epsilon}_s + \bar{\epsilon}_d \frac{p}{M} \right)^2 \left( M - N_{\text{eff}}^{(A)} \right) \right]}{N} \\
&\approx \frac{N_{\text{eff}}^{(A)} + \gamma^2 N^2 \left( \bar{\epsilon}_f + (1 + \bar{\epsilon}_d p)\bar{\epsilon}_s + \bar{\epsilon}_d \frac{p}{M} \right)^2 \left( M - N_{\text{eff}}^{(A)} \right)}{N}.
\end{aligned}
$$

- Case two: $M^{-a} + \bar{\epsilon}_f + (1 + \bar{\epsilon}_d p)\bar{\epsilon}_s + \bar{\epsilon}_d \frac{p}{M} \geq \frac{1}{N\gamma}$

  By (48),

  $$
  \frac{d_{\text{eff}}}{N} \lesssim \frac{M}{N}.
  $$

Denote

$$
k_{\text{eff}} = \left[ M^{-a} \vee N_{\text{eff}}^{(A)^{-a}} \right]^{-\frac{1}{a}} = \left[ M^{-a} \vee \left( \frac{1}{N\gamma} - \bar{\epsilon}_f - (1 + \bar{\epsilon}_d p)\bar{\epsilon}_s - \bar{\epsilon}_d \frac{p}{M} \right) \right]^{-\frac{1}{a}},
$$

then with probability at least $1 - e^{-\Omega(M)}$ over the randomness of $\mathbf{S}$,

$$
\frac{d_{\text{eff}}}{N} \lesssim \frac{k_{\text{eff}} + \gamma^2 N^2 \left( \bar{\epsilon}_f + (1 + \bar{\epsilon}_d p)\bar{\epsilon}_s + \bar{\epsilon}_d \frac{p}{M} \right)^2 (M - k_{\text{eff}})}{N}.
$$

$\qquad\qquad\qquad\qquad\qquad\qquad\qquad\qquad\qquad\qquad\qquad\qquad\qquad\qquad\qquad\qquad\square$

### C.8. Bias Upper Bounds under Power-Law Spectrum

Noticing that

$$
\frac{1}{\gamma^2 N^2} \cdot \left\| \mathbf{v}^{(q)*} \right\|_{(\mathbf{H}_{f,0:k^*}^{(q)})^{-1}}^2 + \left\| \mathbf{v}^{(q)*} \right\|_{\mathbf{H}_{f,k^*:\infty}^{(q)}}^2 \leq \frac{1}{\gamma N} \cdot \left\| \mathbf{v}^{(q)*} \right\|_{\mathbf{I}_{f,0:k^*}^{(q)}}^2 + \left\| \mathbf{v}^{(q)*} \right\|_{\mathbf{H}_{f,k^*:\infty}^{(q)}}^2,
$$

we aim to derive upper bounds for $\frac{1}{\gamma N} \cdot \left\| \mathbf{v}^{(q)*} \right\|_{\mathbf{I}_{f,0:k^*}^{(q)}}^2 + \left\| \mathbf{v}^{(q)*} \right\|_{\mathbf{H}_{f,k^*:\infty}^{(q)}}^2$ in this section.

**Lemma C.23.** *For any $k \geq 0$,*

$$
\frac{\left\| \mathbf{v}^{(q)*} \right\|_{\mathbf{I}_{f,0:k^*}^{(q)}}^2}{\gamma N} + \left\| \mathbf{v}^{(q)*} \right\|_{\mathbf{H}_{f,k^*:\infty}^{(q)}}^2 \lesssim \frac{\|\mathbf{w}^*\|_{\mathbf{I}_{0:k}}^2}{\gamma N} \left\| \left( \mathbf{H}_f^{(q)} \right)^{-1} \mathbf{S} \mathbf{H}_{0:k} \right\|^2 + \|\mathbf{w}^*\|_{\mathbf{H}_{k:\infty}}^2 \left\| \left( \mathbf{H}_f^{(q)} \right)^{-1/2} \mathbf{S} \mathbf{H}_{k:\infty}^{\frac{1}{2}} \right\|^2.
$$

*Proof.* By the definition of

$$
\mathbf{v}^{(q)*} = (\mathbf{H}_f^{(q)})^{-1} \mathbf{S} \mathbf{H} \mathbf{w}^*,
$$

we have

$$
\begin{aligned}
\frac{1}{\gamma N} \cdot \left\| \mathbf{v}^{(q)*} \right\|_{\mathbf{I}_{f,0:k^*}^{(q)}}^2 &= \frac{1}{\gamma N} \left\| \left( \mathbf{H}_{f,0:k^*}^{(q)} \right)^{-1} \mathbf{S} \mathbf{H} \mathbf{w}^* \right\|^2 \\
&\lesssim \frac{1}{\gamma N} \left\| \left( \mathbf{H}_{f,0:k^*}^{(q)} \right)^{-1} \mathbf{S} \mathbf{H}_{0:k} \mathbf{w}^* \right\|^2 + \frac{1}{\gamma N} \left\| \left( \mathbf{H}_{f,0:k^*}^{(q)} \right)^{-1} \mathbf{S} \mathbf{H}_{k:\infty} \mathbf{w}^* \right\|^2 \\
&\leq \frac{1}{\gamma N} \left\| \left( \mathbf{H}_{f,0:k^*}^{(q)} \right)^{-1} \mathbf{S} \mathbf{H}_{0:k} \mathbf{w}^* \right\|^2 + \left\| \left( \mathbf{H}_{f,0:k^*}^{(q)} \right)^{-1/2} \mathbf{S} \mathbf{H}_{k:\infty} \mathbf{w}^* \right\|^2.
\end{aligned}
$$

$$\left\| \mathbf{v}^{(q)*} \right\|_{\mathbf{H}_{f,k^*:\infty}^{(q)}}^2 = \left\| \left( \mathbf{H}_{f,k^*:\infty}^{(q)} \right)^{1/2} \left( \mathbf{H}_f^{(q)} \right)^{-1} \mathbf{S} \mathbf{H} \mathbf{w}^* \right\|^2$$

$$= \left\| \left( \mathbf{H}_{f,k^*:\infty}^{(q)} \right)^{-1/2} \mathbf{S} \mathbf{H} \mathbf{w}^* \right\|^2$$

$$\lesssim \left\| \left( \mathbf{H}_{f,k^*:\infty}^{(q)} \right)^{-1/2} \mathbf{S} \mathbf{H}_{0:k} \mathbf{w}^* \right\|^2 + \left\| \left( \mathbf{H}_{f,k^*:\infty}^{(q)} \right)^{-1/2} \mathbf{S} \mathbf{H}_{k:\infty} \mathbf{w}^* \right\|^2$$

$$\leq \frac{1}{\gamma N} \left\| \left( \mathbf{H}_{f,k^*:\infty}^{(q)} \right)^{-1} \mathbf{S} \mathbf{H}_{0:k} \mathbf{w}^* \right\|^2 + \left\| \left( \mathbf{H}_{f,k^*:\infty}^{(q)} \right)^{-1/2} \mathbf{S} \mathbf{H}_{k:\infty} \mathbf{w}^* \right\|^2 .$$

Hence,

$$\frac{1}{\gamma N} \cdot \left\| \mathbf{v}^{(q)*} \right\|_{\mathbf{I}_{f,0:k^*}^{(q)}}^2 + \left\| \mathbf{v}^{(q)*} \right\|_{\mathbf{H}_{f,k^*:\infty}^{(q)}}^2$$

$$\lesssim \frac{1}{\gamma N} \left[ \left\| \left( \mathbf{H}_{f,0:k^*}^{(q)} \right)^{-1} \mathbf{S} \mathbf{H}_{0:k} \mathbf{w}^* \right\|^2 + \left\| \left( \mathbf{H}_{f,k^*:\infty}^{(q)} \right)^{-1} \mathbf{S} \mathbf{H}_{0:k} \mathbf{w}^* \right\|^2 \right]$$

$$+ \left\| \left( \mathbf{H}_{f,0:k^*}^{(q)} \right)^{-1/2} \mathbf{S} \mathbf{H}_{k:\infty} \mathbf{w}^* \right\|^2 + \left\| \left( \mathbf{H}_{f,k^*:\infty}^{(q)} \right)^{-1/2} \mathbf{S} \mathbf{H}_{k:\infty} \mathbf{w}^* \right\|^2$$

$$= \frac{1}{\gamma N} \left\| \left( \mathbf{H}_f^{(q)} \right)^{-1} \mathbf{S} \mathbf{H}_{0:k} \mathbf{w}^* \right\|^2 + \left\| \left( \mathbf{H}_f^{(q)} \right)^{-1/2} \mathbf{S} \mathbf{H}_{k:\infty} \mathbf{w}^* \right\|^2$$

$$\leq \frac{\| \mathbf{w}^* \|_{\mathbf{I}_{0:k}}^2}{\gamma N} \left\| \left( \mathbf{H}_f^{(q)} \right)^{-1} \mathbf{S} \mathbf{H}_{0:k} \right\|^2 + \| \mathbf{w}^* \|_{\mathbf{H}_{k:\infty}}^2 \left\| \left( \mathbf{H}_f^{(q)} \right)^{-1/2} \mathbf{S} \mathbf{H}_{k:\infty}^{\frac{1}{2}} \right\|^2 .$$

$\square$

**Lemma C.24** (Lemma D.1 in Lin et al. (2024)). *Under Assumption 3.3 and Assumption 3.6, for $k \leq M/2$, with probability at least $1 - e^{-\Omega(M)}$, it holds*

$$\left\| (\mathbf{S} \mathbf{H} \mathbf{S}^\top)^{-1} \mathbf{S} \mathbf{H}_{0:k} \right\|^2 \lesssim 1.$$

*Proof.* For completeness, we provide the proof here. Separating

$$\mathbf{S} \mathbf{H} \mathbf{S}^\top = \mathbf{S} \mathbf{I}_{0:k} \mathbf{H}_{0:k} \mathbf{I}_{0:k} \mathbf{S}^\top + \underbrace{\mathbf{S} \mathbf{I}_{k:\infty} \mathbf{H}_{k:\infty} \mathbf{I}_{k:\infty} \mathbf{S}^\top}_{\mathbf{A}_k} .$$

Then by the Woodbury's identity,

$$(\mathbf{S} \mathbf{H} \mathbf{S}^\top)^{-1} \mathbf{S} \mathbf{H}_{0:k} = \left( \mathbf{A}_k^{-1} - \mathbf{A}_k^{-1} \mathbf{S} \mathbf{I}_{0:k} \left[ \mathbf{H}_{0:k}^{-1} + \mathbf{I}_{0:k} \mathbf{S}^\top \mathbf{A}_k^{-1} \mathbf{S} \mathbf{I}_{0:k} \right]^{-1} \mathbf{I}_{0:k} \mathbf{S}^\top \mathbf{A}_k^{-1} \right) \mathbf{S} \mathbf{I}_{0:k} \mathbf{H}_{0:k}$$

$$= \mathbf{A}_k^{-1} \mathbf{S} \mathbf{I}_{0:k} \mathbf{H}_{0:k} - \mathbf{A}_k^{-1} \mathbf{S} \mathbf{I}_{0:k} \left[ \mathbf{H}_{0:k}^{-1} + \mathbf{I}_{0:k} \mathbf{S}^\top \mathbf{A}_k^{-1} \mathbf{S} \mathbf{I}_{0:k} \right]^{-1} \mathbf{I}_{0:k} \mathbf{S}^\top \mathbf{A}_k^{-1} \mathbf{S} \mathbf{I}_{0:k} \mathbf{H}_{0:k}$$

$$= \mathbf{A}_k^{-1} \mathbf{S} \mathbf{I}_{0:k} \left[ \mathbf{H}_{0:k}^{-1} + \mathbf{I}_{0:k} \mathbf{S}^\top \mathbf{A}_k^{-1} \mathbf{S} \mathbf{I}_{0:k} \right]^{-1} \mathbf{H}_{0:k}^{-1} \mathbf{H}_{0:k} .$$

Therefore,

$$\left\| \left( \mathbf{S} \mathbf{H} \mathbf{S}^\top \right)^{-1} \mathbf{S} \mathbf{H}_{0:k} \right\| = \left\| \mathbf{A}_k^{-1} \mathbf{S} \mathbf{I}_{0:k} \left[ \mathbf{H}_{0:k}^{-1} + \mathbf{I}_{0:k} \mathbf{S}^\top \mathbf{A}_k^{-1} \mathbf{S} \mathbf{I}_{0:k} \right]^{-1} \mathbf{H}_{0:k}^{-1} \mathbf{H}_{0:k} \right\|$$

$$\leq \left\| \mathbf{A}_k^{-1} \right\| \left\| \mathbf{S} \mathbf{I}_{0:k} \right\| \left\| \left[ \mathbf{I}_{0:k} \mathbf{S}^\top \mathbf{A}_k^{-1} \mathbf{S} \mathbf{I}_{0:k} \right]^{-1} \right\| . \tag{49}$$

Note that

$$\mathbf{I}_{0:k} = \mathbf{V}_k \mathbf{V}_k^\top, \quad \mathbf{V}_k = [\mathbf{v}_1, ..., \mathbf{v}_k] \in \mathbb{R}^{p \times k},$$

it follows that the eigenvalues of $\mathbf{S} \mathbf{I}_{0:k}$ correspond to the eigenvalues of $\mathbf{S} \mathbf{V}_k$. As $\mathbf{S}_{ij} \sim \mathcal{N}(0, \frac{1}{M})$, for $k \leq \frac{M}{2}$, with probability at least $1 - e^{-\Omega(M)}$,

$$\| \mathbf{S} \mathbf{I}_{0:k} \| \leq c, \tag{50}$$

where $c$ is a constant. Denote $\{\hat{\lambda}_i\}_{i=1}^M$ be the eigenvalues of $\mathbf{A}_k = \mathbf{SI}_{k:\infty}\mathbf{H}_{k:\infty}\mathbf{I}_{k:\infty}\mathbf{S}^\top + \mathbf{D}$.

$$\left\|\mathbf{A}_k^{-1}\right\| \leq \frac{1}{\hat{\lambda}_M}, \tag{51}$$

We then deal with $\mathbf{I}_{0:k}\mathbf{S}^\top\mathbf{A}_k^{-1}\mathbf{SI}_{0:k}$. With probability at least $1 - e^{-\Omega(M)}$, for $k \leq M/2$, it holds

$$
\begin{aligned}
\mathbf{I}_{0:k}\mathbf{S}^\top\mathbf{A}_k^{-1}\mathbf{SI}_{0:k} &= \mathbf{V}_k \sum_{i=1}^M \frac{1}{\mu_i(\mathbf{A}_k)}\tilde{\mathbf{s}}_i\tilde{\mathbf{s}}_i^\top \mathbf{V}_k^\top \\
&\succeq \mathbf{V}_k \sum_{i=M/2}^M \frac{1}{\mu_i(\mathbf{A}_k)}\tilde{\mathbf{s}}_i\tilde{\mathbf{s}}_i^\top \mathbf{V}_k^\top \\
&\succeq \mathbf{V}_k \sum_{i=M/2}^M \frac{1}{\mu_{M/2}(\mathbf{A}_k)}\tilde{\mathbf{s}}_i\tilde{\mathbf{s}}_i^\top \mathbf{V}_k^\top \\
&\succsim \frac{1}{\mu_{M/2}\left(\mathbf{SI}_{k:\infty}\mathbf{H}_{k:\infty}\mathbf{I}_{k:\infty}\mathbf{S}^\top\right)}\mathbf{I}_{0:k}.
\end{aligned}
$$

Together with (49), (50), (51), for $k \leq M/2$, with probability at least $1 - e^{-\Omega(M)}$,

$$\left\|\left(\mathbf{SHS}^\top\right)^{-1}\mathbf{SH}_{0:k}\right\| \lesssim \frac{\mu_{M/2}\left(\mathbf{SI}_{k:\infty}\mathbf{H}_{k:\infty}\mathbf{I}_{k:\infty}\mathbf{S}^\top\right)}{\mu_M\left(\mathbf{SI}_{k:\infty}\mathbf{H}_{k:\infty}\mathbf{I}_{k:\infty}\mathbf{S}^\top\right)} \lesssim 1, \tag{52}$$

where the last inequality holds by Lemma G.6. $\qquad\square$

### C.8.1. MULTIPLICATIVE QUANTIZATION

**Lemma C.25.** *Under Assumption 3.3 and Assumption 3.6, for any $i \in \{s, d, f, p, a, o\}$, if there exist $(\bar{\epsilon}_i, \underline{\epsilon}_i)$ such that quantization $\mathcal{Q}_i$ is $(\bar{\epsilon}_i, \underline{\epsilon}_i)$-multiplicative, for $k \leq M/2$, with probability at least $1 - e^{-\Omega(M)}$, it holds*

$$
\begin{aligned}
&\frac{1}{\gamma N} \cdot \left\|\mathbf{v}^{(q)*}\right\|^2_{\mathbf{I}^{(q)}_{f,0:k^*}} + \left\|\mathbf{v}^{(q)*}\right\|^2_{\mathbf{H}^{(q)}_{f,k^*:\infty}} \\
&\lesssim \frac{\left[1 + \frac{(1+\bar{\epsilon}_d)(1+\bar{\epsilon}_f)(1+\bar{\epsilon}_s) - (1+\underline{\epsilon}_d)(1+\underline{\epsilon}_f)(1+\underline{\epsilon}_s)}{(1+\bar{\epsilon}_d)(1+\bar{\epsilon}_f)(1+\bar{\epsilon}_s)}M^{a/2}\right]^2}{[(1+\underline{\epsilon}_d)(1+\underline{\epsilon}_f)(1+\underline{\epsilon}_s)]^2}\frac{\|\mathbf{w}^*\|^2_{\mathbf{I}_{0:k}}}{\gamma N} + \frac{\|\mathbf{w}^*\|^2_{\mathbf{H}_{k:\infty}}}{(1+\underline{\epsilon}_d)(1+\underline{\epsilon}_f)(1+\underline{\epsilon}_s)}.
\end{aligned}
$$

*Further, if $\mathbf{H}^{(q)}_f$ and $\mathbf{SHS}^\top$ commute,*

$$\frac{1}{\gamma N} \cdot \left\|\mathbf{v}^{(q)*}\right\|^2_{\mathbf{I}^{(q)}_{f,0:k^*}} + \left\|\mathbf{v}^{(q)*}\right\|^2_{\mathbf{H}^{(q)}_{f,k^*:\infty}} \lesssim \frac{1}{[(1+\underline{\epsilon}_d)(1+\underline{\epsilon}_f)(1+\underline{\epsilon}_s)]^2}\frac{\|\mathbf{w}^*\|^2_{\mathbf{I}_{0:k}}}{\gamma N} + \frac{\|\mathbf{w}^*\|^2_{\mathbf{H}_{k:\infty}}}{(1+\underline{\epsilon}_d)(1+\underline{\epsilon}_f)(1+\underline{\epsilon}_s)}.$$

*Proof.* We prove by using Lemma C.23. The key is to derive bounds for $\left\|\left(\mathbf{H}^{(q)}_f\right)^{-1/2}\mathbf{SH}^{\frac{1}{2}}_{k:\infty}\right\|^2$ and $\left\|\left(\mathbf{H}^{(q)}_f\right)^{-1}\mathbf{SH}_{0:k}\right\|^2$. Noticing that

$$\mathbf{H}^{(q)}_f \succeq (1+\underline{\epsilon}_d)(1+\underline{\epsilon}_f)(1+\underline{\epsilon}_s)\mathbf{SHS}^\top,$$

we have

$$
\begin{aligned}
\left\|\left(\mathbf{H}_f^{(q)}\right)^{-1/2}\mathbf{S}\mathbf{H}_{k:\infty}^{\frac{1}{2}}\right\|^2 &\leq \left\|\left(\mathbf{H}_f^{(q)}\right)^{-1/2}\left(\mathbf{S}\mathbf{H}\mathbf{S}^\top\right)^{\frac{1}{2}}\right\|^2 \cdot \left\|\left(\mathbf{S}\mathbf{H}\mathbf{S}^\top\right)^{-\frac{1}{2}}\mathbf{S}\mathbf{H}_{k:\infty}^{\frac{1}{2}}\right\|^2 \\
&= \mu_{\max}\left(\left(\mathbf{H}_f^{(q)}\right)^{-1/2}\mathbf{S}\mathbf{H}\mathbf{S}^\top\left(\mathbf{H}_f^{(q)}\right)^{-1/2}\right) \cdot \left\|\left(\mathbf{S}\mathbf{H}\mathbf{S}^\top\right)^{-\frac{1}{2}}\mathbf{S}\mathbf{H}_{k:\infty}^{\frac{1}{2}}\right\|^2 \\
&\leq \frac{1}{(1+\underline{\epsilon}_d)(1+\underline{\epsilon}_f)(1+\underline{\epsilon}_s)}\left\|\left(\mathbf{S}\mathbf{H}\mathbf{S}^\top\right)^{-\frac{1}{2}}\mathbf{S}\mathbf{H}_{k:\infty}^{\frac{1}{2}}\right\|^2 \\
&\leq \frac{1}{(1+\underline{\epsilon}_d)(1+\underline{\epsilon}_f)(1+\underline{\epsilon}_s)}\left\|\left(\mathbf{S}\mathbf{H}\mathbf{S}^\top\right)^{-\frac{1}{2}}\mathbf{S}\mathbf{H}^{\frac{1}{2}}\right\|^2 \\
&\leq \frac{1}{(1+\underline{\epsilon}_d)(1+\underline{\epsilon}_f)(1+\underline{\epsilon}_s)},
\end{aligned}
\tag{53}
$$

where the second inequality holds by Lemma F.1. We then focus on $\left\|\left(\mathbf{H}_f^{(q)}\right)^{-1}\mathbf{S}\mathbf{H}_{0:k}\right\|^2$. Noting that

$$
\left\|\left(\mathbf{H}_f^{(q)}\right)^{-1}\mathbf{S}\mathbf{H}_{0:k}\right\|^2 \leq \left\|\left(\mathbf{H}_f^{(q)}\right)^{-1}\mathbf{S}\mathbf{H}\mathbf{S}^\top\right\|^2 \left\|(\mathbf{S}\mathbf{H}\mathbf{S}^\top)^{-1}\mathbf{S}\mathbf{H}_{0:k}\right\|^2,
$$

we handle $\left\|\left(\mathbf{H}_f^{(q)}\right)^{-1}\mathbf{S}\mathbf{H}\mathbf{S}^\top\right\|^2$ and $\left\|(\mathbf{S}\mathbf{H}\mathbf{S}^\top)^{-1}\mathbf{S}\mathbf{H}_{0:k}\right\|^2$ respectively.

Regarding $\left\|\left(\mathbf{H}_f^{(q)}\right)^{-1}\mathbf{S}\mathbf{H}\mathbf{S}^\top\right\|^2$, as $\mathbf{H}_f^{(q)}$ and $\mathbf{S}\mathbf{H}\mathbf{S}^\top$ might not commute, we can only derive an upper bound related to the condition number of $\mathbf{S}\mathbf{H}\mathbf{S}^\top$. Specifically, denote $\mathbf{X} = (\mathbf{S}\mathbf{H}\mathbf{S}^\top)^{-1/2}\mathbf{H}_f^{(q)}(\mathbf{S}\mathbf{H}\mathbf{S}^\top)^{-1/2}$, then we have

$$
\left(\mathbf{H}_f^{(q)}\right)^{-1}\mathbf{S}\mathbf{H}\mathbf{S}^\top = (\mathbf{S}\mathbf{H}\mathbf{S}^\top)^{-1/2}\mathbf{X}^{-1}(\mathbf{S}\mathbf{H}\mathbf{S}^\top)^{1/2}.
$$

Further, recall that $(1+\underline{\epsilon}_d)(1+\underline{\epsilon}_f)(1+\underline{\epsilon}_s)\mathbf{S}\mathbf{H}\mathbf{S}^\top \preceq \mathbf{H}_f^{(q)} \preceq (1+\bar{\epsilon}_d)(1+\bar{\epsilon}_f)(1+\bar{\epsilon}_s)\mathbf{S}\mathbf{H}\mathbf{S}^\top$, we have

$$
\frac{1}{(1+\bar{\epsilon}_d)(1+\bar{\epsilon}_f)(1+\bar{\epsilon}_s)}\mathbf{I} \preceq \mathbf{X}^{-1} = \mathbf{S}\mathbf{H}\mathbf{S}^\top(\mathbf{H}_f^{(q)})^{-1}\mathbf{S}\mathbf{H}\mathbf{S}^\top \preceq \frac{1}{(1+\underline{\epsilon}_d)(1+\underline{\epsilon}_f)(1+\underline{\epsilon}_s)}\mathbf{I}.
$$

Denote $\mathbf{0} \preceq \mathbf{\Delta} = \frac{1}{(1+\underline{\epsilon}_d)(1+\underline{\epsilon}_f)(1+\underline{\epsilon}_s)}\mathbf{I} - \mathbf{X}^{-1} \preceq \left[\frac{1}{(1+\underline{\epsilon}_d)(1+\underline{\epsilon}_f)(1+\underline{\epsilon}_s)} - \frac{1}{(1+\bar{\epsilon}_d)(1+\bar{\epsilon}_f)(1+\bar{\epsilon}_s)}\right]\mathbf{I}$, then

$$
\begin{aligned}
\left\|\left(\mathbf{H}_f^{(q)}\right)^{-1}\mathbf{S}\mathbf{H}\mathbf{S}^\top\right\| &= \left\|(\mathbf{S}\mathbf{H}\mathbf{S}^\top)^{-1/2}\mathbf{X}^{-1}(\mathbf{S}\mathbf{H}\mathbf{S}^\top)^{1/2}\right\| \\
&= \left\|\frac{1}{(1+\underline{\epsilon}_d)(1+\underline{\epsilon}_f)(1+\underline{\epsilon}_s)}\mathbf{I} - (\mathbf{S}\mathbf{H}\mathbf{S}^\top)^{-1/2}\mathbf{\Delta}(\mathbf{S}\mathbf{H}\mathbf{S}^\top)^{1/2}\right\| \\
&\leq \frac{1}{(1+\underline{\epsilon}_d)(1+\underline{\epsilon}_f)(1+\underline{\epsilon}_s)} + \left\|(\mathbf{S}\mathbf{H}\mathbf{S}^\top)^{-1/2}\mathbf{\Delta}(\mathbf{S}\mathbf{H}\mathbf{S}^\top)^{1/2}\right\| \\
&\leq \frac{1}{(1+\underline{\epsilon}_d)(1+\underline{\epsilon}_f)(1+\underline{\epsilon}_s)} + \left\|(\mathbf{S}\mathbf{H}\mathbf{S}^\top)^{-1/2}\right\|\|\mathbf{\Delta}\|\left\|(\mathbf{S}\mathbf{H}\mathbf{S}^\top)^{1/2}\right\| \\
&\lesssim \frac{1}{(1+\underline{\epsilon}_d)(1+\underline{\epsilon}_f)(1+\underline{\epsilon}_s)} + \left[\frac{1}{(1+\underline{\epsilon}_d)(1+\underline{\epsilon}_f)(1+\underline{\epsilon}_s)} - \frac{1}{(1+\bar{\epsilon}_d)(1+\bar{\epsilon}_f)(1+\bar{\epsilon}_s)}\right]M^{a/2},
\end{aligned}
$$

where the last inequality holds with probability at least $1 - e^{-\Omega(M)}$ by Lemma G.1. We would like to remark that, the term related to $M^{a/2}$ is from the misalignment between $\mathbf{H}_f^{(q)}$ and $\mathbf{S}\mathbf{H}\mathbf{S}^\top$. Specifically, if $\mathbf{H}_f^{(q)}$ and $\mathbf{S}\mathbf{H}\mathbf{S}^\top$ commute, then this term will be vanished.

$$
\left\|\left(\mathbf{H}_f^{(q)}\right)^{-1}\mathbf{S}\mathbf{H}\mathbf{S}^\top\right\| = \left\|(\mathbf{S}\mathbf{H}\mathbf{S}^\top)^{-1/2}\mathbf{X}^{-1}(\mathbf{S}\mathbf{H}\mathbf{S}^\top)^{1/2}\right\| = \|\mathbf{X}^{-1}\| \leq \frac{1}{(1+\underline{\epsilon}_d)(1+\underline{\epsilon}_f)(1+\underline{\epsilon}_s)}.
$$

Regarding $\left\|(\mathbf{S}\mathbf{H}\mathbf{S}^\top)^{-1}\mathbf{S}\mathbf{H}_{0:k}\right\|^2$, by Lemma C.24, for $k \le M/2$, with probability at least $1 - e^{-\Omega(M)}$,

$$\left\|(\mathbf{S}\mathbf{H}\mathbf{S}^\top)^{-1}\mathbf{S}\mathbf{H}_{0:k}\right\|^2 \lesssim 1.$$

Overall, together with Lemma C.23, for $k \le M/2$, with probability at least $1 - e^{-\Omega(M)}$, it holds

$$\frac{1}{\gamma N} \cdot \left\|\mathbf{v}^{(q)*}\right\|^2_{\mathbf{I}^{(q)}_{f,0:k^*}} + \left\|\mathbf{v}^{(q)*}\right\|^2_{\mathbf{H}^{(q)}_{f,k^*:\infty}}$$

$$\lesssim \frac{\left[1 + \frac{(1+\bar{\epsilon}_d)(1+\bar{\epsilon}_f)(1+\bar{\epsilon}_s) - (1+\underline{\epsilon}_d)(1+\underline{\epsilon}_f)(1+\underline{\epsilon}_s)}{(1+\bar{\epsilon}_d)(1+\bar{\epsilon}_f)(1+\bar{\epsilon}_s)} M^{a/2}\right]^2}{[(1+\underline{\epsilon}_d)(1+\underline{\epsilon}_f)(1+\underline{\epsilon}_s)]^2} \frac{\|\mathbf{w}^*\|^2_{\mathbf{I}_{0:k}}}{\gamma N} + \frac{\|\mathbf{w}^*\|^2_{\mathbf{H}_{k:\infty}}}{(1+\underline{\epsilon}_d)(1+\underline{\epsilon}_f)(1+\underline{\epsilon}_s)}.$$

Further, if $\mathbf{H}^{(q)}_f$ and $\mathbf{S}\mathbf{H}\mathbf{S}^\top$ commute,

$$\frac{1}{\gamma N} \cdot \left\|\mathbf{v}^{(q)*}\right\|^2_{\mathbf{I}^{(q)}_{f,0:k^*}} + \left\|\mathbf{v}^{(q)*}\right\|^2_{\mathbf{H}^{(q)}_{f,k^*:\infty}} \lesssim \frac{1}{[(1+\underline{\epsilon}_d)(1+\underline{\epsilon}_f)(1+\underline{\epsilon}_s)]^2} \frac{\|\mathbf{w}^*\|^2_{\mathbf{I}_{0:k}}}{\gamma N} + \frac{\|\mathbf{w}^*\|^2_{\mathbf{H}_{k:\infty}}}{(1+\underline{\epsilon}_d)(1+\underline{\epsilon}_f)(1+\underline{\epsilon}_s)}.$$

$\square$

**Lemma C.26.** *Under Assumption 3.3 and Assumption 3.6, for any $i \in \{s, d, f, p, a, o\}$, if there exist $(\bar{\epsilon}_i, \underline{\epsilon}_i)$ such that quantization $\mathcal{Q}_i$ is $(\bar{\epsilon}_i, \underline{\epsilon}_i)$-multiplicative, with probability at least $1 - e^{-\Omega(M)}$, it holds*

$$\mathbb{E}_{\mathbf{w}^*}\left[\frac{\left\|\mathbf{v}^{(q)*}\right\|^2_{\mathbf{I}^{(q)}_{f,0:k^*}}}{\gamma N} + \left\|\mathbf{v}^{(q)*}\right\|^2_{\mathbf{H}^{(q)}_{f,k^*:\infty}}\right] \lesssim \frac{\max\left\{\left[\frac{N\gamma(1+\underline{\epsilon}_d)(1+\underline{\epsilon}_f)(1+\underline{\epsilon}_s)}{1+\left[\frac{(1+\bar{\epsilon}_d)(1+\bar{\epsilon}_f)(1+\bar{\epsilon}_s) - (1+\underline{\epsilon}_d)(1+\underline{\epsilon}_f)(1+\underline{\epsilon}_s)}{(1+\bar{\epsilon}_d)(1+\bar{\epsilon}_f)(1+\bar{\epsilon}_s)}\right]^2 M^a}\right]^{\frac{1}{a}-1}, M^{1-a}\right\}}{(1+\underline{\epsilon}_d)(1+\underline{\epsilon}_f)(1+\underline{\epsilon}_s)}.$$

*Further if $\mathbf{H}^{(q)}_f$ and $\mathbf{S}\mathbf{H}\mathbf{S}^\top$ commute,*

$$\mathbb{E}_{\mathbf{w}^*}\left[\frac{\left\|\mathbf{v}^{(q)*}\right\|^2_{\mathbf{I}^{(q)}_{f,0:k^*}}}{\gamma N} + \left\|\mathbf{v}^{(q)*}\right\|^2_{\mathbf{H}^{(q)}_{f,k^*:\infty}}\right] \lesssim \frac{\max\left\{\left[N\gamma(1+\underline{\epsilon}_d)(1+\underline{\epsilon}_f)(1+\underline{\epsilon}_s)\right]^{\frac{1}{a}-1}, M^{1-a}\right\}}{(1+\underline{\epsilon}_d)(1+\underline{\epsilon}_f)(1+\underline{\epsilon}_s)}.$$

*Proof.* By Lemma C.25, for $k \le M/2$, with probability at least $1 - e^{-\Omega(M)}$,

$$\mathbb{E}_{\mathbf{w}^*}\left[\frac{\left\|\mathbf{v}^{(q)*}\right\|^2_{\mathbf{I}^{(q)}_{f,0:k^*}}}{\gamma N} + \left\|\mathbf{v}^{(q)*}\right\|^2_{\mathbf{H}^{(q)}_{f,k^*:\infty}}\right]$$

$$\lesssim \mathbb{E}_{\mathbf{w}^*}\left[\frac{\left[1 + \frac{(1+\bar{\epsilon}_d)(1+\bar{\epsilon}_f)(1+\bar{\epsilon}_s) - (1+\underline{\epsilon}_d)(1+\underline{\epsilon}_f)(1+\underline{\epsilon}_s)}{(1+\bar{\epsilon}_d)(1+\bar{\epsilon}_f)(1+\bar{\epsilon}_s)} M^{a/2}\right]^2}{[(1+\underline{\epsilon}_d)(1+\underline{\epsilon}_f)(1+\underline{\epsilon}_s)]^2} \frac{\|\mathbf{w}^*\|^2_{\mathbf{I}_{0:k}}}{\gamma N} + \frac{\|\mathbf{w}^*\|^2_{\mathbf{H}_{k:\infty}}}{(1+\underline{\epsilon}_d)(1+\underline{\epsilon}_f)(1+\underline{\epsilon}_s)}\right]$$

$$\approx \frac{\left[1 + \frac{(1+\bar{\epsilon}_d)(1+\bar{\epsilon}_f)(1+\bar{\epsilon}_s) - (1+\underline{\epsilon}_d)(1+\underline{\epsilon}_f)(1+\underline{\epsilon}_s)}{(1+\bar{\epsilon}_d)(1+\bar{\epsilon}_f)(1+\bar{\epsilon}_s)} M^{a/2}\right]^2}{[(1+\underline{\epsilon}_d)(1+\underline{\epsilon}_f)(1+\underline{\epsilon}_s)]^2} \frac{k}{N\gamma} + \frac{1}{(1+\underline{\epsilon}_d)(1+\underline{\epsilon}_f)(1+\underline{\epsilon}_s)} \sum_{i>k} i^{-a}$$

$$\approx \frac{1 + \left[\frac{(1+\bar{\epsilon}_d)(1+\bar{\epsilon}_f)(1+\bar{\epsilon}_s) - (1+\underline{\epsilon}_d)(1+\underline{\epsilon}_f)(1+\underline{\epsilon}_s)}{(1+\bar{\epsilon}_d)(1+\bar{\epsilon}_f)(1+\bar{\epsilon}_s)}\right]^2 M^a}{[(1+\underline{\epsilon}_d)(1+\underline{\epsilon}_f)(1+\underline{\epsilon}_s)]^2} \frac{k}{N\gamma} + \frac{k^{1-a}}{(1+\underline{\epsilon}_d)(1+\underline{\epsilon}_f)(1+\underline{\epsilon}_s)}$$

$$\lesssim \frac{\max\left\{\left[\frac{N\gamma(1+\underline{\epsilon}_d)(1+\underline{\epsilon}_f)(1+\underline{\epsilon}_s)}{1+\left[\frac{(1+\bar{\epsilon}_d)(1+\bar{\epsilon}_f)(1+\bar{\epsilon}_s) - (1+\underline{\epsilon}_d)(1+\underline{\epsilon}_f)(1+\underline{\epsilon}_s)}{(1+\bar{\epsilon}_d)(1+\bar{\epsilon}_f)(1+\bar{\epsilon}_s)}\right]^2 M^a}\right]^{\frac{1}{a}-1}, M^{1-a}\right\}}{(1+\underline{\epsilon}_d)(1+\underline{\epsilon}_f)(1+\underline{\epsilon}_s)},$$

where in the last inequality we choose $k = [M/2] \wedge \left[ \frac{N\gamma(1+\underline{\epsilon}_d)(1+\underline{\epsilon}_f)(1+\underline{\epsilon}_s)}{1 + \left[ \frac{(1+\overline{\epsilon}_d)(1+\overline{\epsilon}_f)(1+\overline{\epsilon}_s) - (1+\underline{\epsilon}_d)(1+\underline{\epsilon}_f)(1+\underline{\epsilon}_s)}{(1+\overline{\epsilon}_d)(1+\overline{\epsilon}_f)(1+\overline{\epsilon}_s)} \right]^2 M^a} \right]^{1/a}$. The statement when

$\mathbf{H}_f^{(q)}$ and $\mathbf{SHS}^\top$ commute can be deduced directly. We omit here for simplicity. $\qquad \square$

### C.8.2. ADDITIVE QUANTIZATION

**Lemma C.27.** *For any $i \in \{s, d, f, p, a, o\}$, if there exist $(\overline{\epsilon}_i, \underline{\epsilon}_i)$ such that quantization $\mathcal{Q}_i$ is $(\overline{\epsilon}_i, \underline{\epsilon}_i)$-additive, for $k \leq M/2$, with probability at least $1 - e^{-\Omega(M)}$, under Assumption 3.3 and Assumption 3.6, it holds*

$$\frac{1}{\gamma N} \cdot \left\| \mathbf{v}^{(q)*} \right\|_{\mathbf{I}_{f,0:k^*}^{(q)}}^2 + \left\| \mathbf{v}^{(q)*} \right\|_{\mathbf{H}_{f,k^*:\infty}^{(q)}}^2$$

$$\lesssim \frac{\left( 1 + M^{a/2} \frac{M^a\left( \overline{\epsilon}_f + \overline{\epsilon}_s(1+\overline{\epsilon}_d p) + \overline{\epsilon}_d \frac{p}{M} \right) - \left( \underline{\epsilon}_f + \underline{\epsilon}_s(1+\underline{\epsilon}_d p) + \underline{\epsilon}_d \frac{p}{M} \right)}{1 + M^a\left( \overline{\epsilon}_f + \overline{\epsilon}_s(1+\overline{\epsilon}_d p) + \overline{\epsilon}_d \frac{p}{M} \right)} \right)^2}{\left( 1 + \underline{\epsilon}_f + \underline{\epsilon}_s(1 + \underline{\epsilon}_d p) + \underline{\epsilon}_d \frac{p}{M} \right)^2} \frac{\|\mathbf{w}^*\|_{\mathbf{I}_{0:k}}^2}{\gamma N} + \frac{\|\mathbf{w}^*\|_{\mathbf{H}_{k:\infty}}^2}{1 + \underline{\epsilon}_f + \underline{\epsilon}_s(1 + \underline{\epsilon}_d p) + \underline{\epsilon}_d \frac{p}{M}}.$$

*Further, if $\mathbf{H}_f^{(q)}$ and $\mathbf{SHS}^\top$ commute,*

$$\frac{1}{\gamma N} \cdot \left\| \mathbf{v}^{(q)*} \right\|_{\mathbf{I}_{f,0:k^*}^{(q)}}^2 + \left\| \mathbf{v}^{(q)*} \right\|_{\mathbf{H}_{f,k^*:\infty}^{(q)}}^2$$

$$\lesssim \frac{1}{\left( 1 + \underline{\epsilon}_f + \underline{\epsilon}_s(1 + \underline{\epsilon}_d p) + \underline{\epsilon}_d \frac{p}{M} \right)^2} \frac{\|\mathbf{w}^*\|_{\mathbf{I}_{0:k}}^2}{\gamma N} + \frac{\|\mathbf{w}^*\|_{\mathbf{H}_{k:\infty}}^2}{1 + \underline{\epsilon}_f + \underline{\epsilon}_s(1 + \underline{\epsilon}_d p) + \underline{\epsilon}_d \frac{p}{M}}.$$

*Proof.* We prove by using Lemma C.23. The key is to derive bounds for $\left\| \left( \mathbf{H}_f^{(q)} \right)^{-1/2} \mathbf{SH}_{k:\infty}^{\frac{1}{2}} \right\|^2$ and $\left\| \left( \mathbf{H}_f^{(q)} \right)^{-1} \mathbf{SH}_{0:k} \right\|^2$. Noticing that with probability at least $1 - e^{-\Omega(M)}$,

$$\mathbf{H}_f^{(q)} \succsim \mathbf{SHS}^\top + \left( \underline{\epsilon}_f + \underline{\epsilon}_s(1 + \underline{\epsilon}_d p) + \underline{\epsilon}_d \frac{p}{M} \right) \mathbf{I},$$

it follows by Lemma F.1 that

$$\left\| \left( \mathbf{H}_f^{(q)} \right)^{-1/2} \mathbf{SH}_{k:\infty}^{\frac{1}{2}} \right\|^2 \leq \left\| \left( \mathbf{H}_f^{(q)} \right)^{-1/2} \left( \mathbf{SHS}^\top \right)^{\frac{1}{2}} \right\|^2 \left\| \left( \mathbf{SHS}^\top \right)^{-\frac{1}{2}} \mathbf{SH}_{k:\infty}^{\frac{1}{2}} \right\|^2$$

$$\leq \frac{\mu_{\max}(\mathbf{SHS}^\top)}{\mu_{\max}(\mathbf{SHS}^\top) + \underline{\epsilon}_f + \underline{\epsilon}_s(1 + \underline{\epsilon}_d p) + \underline{\epsilon}_d \frac{p}{M}} \left\| \left( \mathbf{SHS}^\top \right)^{-\frac{1}{2}} \mathbf{SH}^{\frac{1}{2}} \right\|^2 \qquad (54)$$

$$\lesssim \frac{1}{1 + \underline{\epsilon}_f + \underline{\epsilon}_s(1 + \underline{\epsilon}_d p) + \underline{\epsilon}_d \frac{p}{M}},$$

where the last inequality holds by Lemma G.1. We then focus on $\left\| \left( \mathbf{H}_f^{(q)} \right)^{-1} \mathbf{SH}_{0:k} \right\|^2$. Similarly, we consider

$$\left\| \left( \mathbf{H}_f^{(q)} \right)^{-1} \mathbf{SH}_{0:k} \right\|^2 \leq \left\| \left( \mathbf{H}_f^{(q)} \right)^{-1} \mathbf{SHS}^\top \right\|^2 \left\| (\mathbf{SHS}^\top)^{-1} \mathbf{SH}_{0:k} \right\|^2.$$

By Lemma C.24, for $k \leq M/2$, with probability at least $1 - e^{-\Omega(M)}$,

$$\left\| (\mathbf{SHS}^\top)^{-1} \mathbf{SH}_{0:k} \right\| \lesssim 1.$$

We merely need to drive upper bound for $\left\| \left( \mathbf{H}_f^{(q)} \right)^{-1} \mathbf{SHS}^\top \right\|$. Similar to the multiplicative quantization case, denote $\mathbf{X} = (\mathbf{SHS}^\top)^{-1/2} \mathbf{H}_f^{(q)} (\mathbf{SHS}^\top)^{-1/2}$. Recall that

$$\mathbf{SHS}^\top + \left( \underline{\epsilon}_f + \underline{\epsilon}_s(1 + \underline{\epsilon}_d p) + \underline{\epsilon}_d \frac{p}{M} \right) \mathbf{I} \preceq \mathbf{H}_f^{(q)} \preceq \mathbf{SHS}^\top + \left( \overline{\epsilon}_f + \overline{\epsilon}_s(1 + \overline{\epsilon}_d p) + \overline{\epsilon}_d \frac{p}{M} \right) \mathbf{I},$$

we have

$$\mathbf{I} + \left(\underline{\epsilon}_f + \underline{\epsilon}_s(1 + \underline{\epsilon}_d p) + \underline{\epsilon}_d \frac{p}{M}\right)(\mathbf{SHS}^\top)^{-1} \preceq \mathbf{X} \preceq \mathbf{I} + \left(\overline{\epsilon}_f + \overline{\epsilon}_s(1 + \overline{\epsilon}_d p) + \overline{\epsilon}_d \frac{p}{M}\right)(\mathbf{SHS}^\top)^{-1}.$$

Denote $\boldsymbol{\Delta} = \frac{1}{1+\mu_{\min}\left((\mathbf{SHS}^\top)^{-1}\right)\left[\underline{\epsilon}_f + \underline{\epsilon}_s(1 + \underline{\epsilon}_d p) + \underline{\epsilon}_d \frac{p}{M}\right]}\mathbf{I} - \mathbf{X}^{-1}$, then it holds

$$0 \preceq \boldsymbol{\Delta} \preceq \frac{1}{1 + \mu_{\min}\left((\mathbf{SHS}^\top)^{-1}\right)\left[\underline{\epsilon}_f + \underline{\epsilon}_s(1 + \underline{\epsilon}_d p) + \underline{\epsilon}_d \frac{p}{M}\right]}\mathbf{I} - \left[\mathbf{I} + \left(\overline{\epsilon}_f + \overline{\epsilon}_s(1 + \overline{\epsilon}_d p) + \overline{\epsilon}_d \frac{p}{M}\right)(\mathbf{SHS}^\top)^{-1}\right]^{-1}.$$

Hence, using Lemma G.1 we have

$$\left\|\left(\mathbf{H}_f^{(q)}\right)^{-1}\mathbf{SHS}^\top\right\|$$

$$= \left\|(\mathbf{SHS}^\top)^{-1/2}\mathbf{X}^{-1}(\mathbf{SHS}^\top)^{1/2}\right\|$$

$$= \left\|(\mathbf{SHS}^\top)^{-1/2}\left(\frac{1}{1 + \mu_{\min}\left((\mathbf{SHS}^\top)^{-1}\right)\left[\underline{\epsilon}_f + \underline{\epsilon}_s(1 + \underline{\epsilon}_d p) + \underline{\epsilon}_d \frac{p}{M}\right]}\mathbf{I} - \boldsymbol{\Delta}\right)(\mathbf{SHS}^\top)^{1/2}\right\|$$

$$\leq \frac{1}{1 + \mu_{\min}\left((\mathbf{SHS}^\top)^{-1}\right)\left[\underline{\epsilon}_f + \underline{\epsilon}_s(1 + \underline{\epsilon}_d p) + \underline{\epsilon}_d \frac{p}{M}\right]} + \left\|(\mathbf{SHS}^\top)^{-1/2}\boldsymbol{\Delta}(\mathbf{SHS}^\top)^{1/2}\right\|$$

$$\lesssim \frac{1}{1 + \underline{\epsilon}_f + \underline{\epsilon}_s(1 + \underline{\epsilon}_d p) + \underline{\epsilon}_d \frac{p}{M}} + M^{a/2}\|\boldsymbol{\Delta}\|$$

$$\lesssim \frac{1}{1 + \underline{\epsilon}_f + \underline{\epsilon}_s(1 + \underline{\epsilon}_d p) + \underline{\epsilon}_d \frac{p}{M}} + M^{a/2}\left(\frac{1}{1 + \underline{\epsilon}_f + \underline{\epsilon}_s(1 + \underline{\epsilon}_d p) + \underline{\epsilon}_d \frac{p}{M}} - \frac{1}{1 + M^a\left(\overline{\epsilon}_f + \overline{\epsilon}_s(1 + \overline{\epsilon}_d p) + \overline{\epsilon}_d \frac{p}{M}\right)}\right)$$

$$= \frac{1 + M^{a/2}\frac{M^a\left(\overline{\epsilon}_f + \overline{\epsilon}_s(1 + \overline{\epsilon}_d p) + \overline{\epsilon}_d \frac{p}{M}\right) - \left(\underline{\epsilon}_f + \underline{\epsilon}_s(1 + \underline{\epsilon}_d p) + \underline{\epsilon}_d \frac{p}{M}\right)}{1 + M^a\left(\overline{\epsilon}_f + \overline{\epsilon}_s(1 + \overline{\epsilon}_d p) + \overline{\epsilon}_d \frac{p}{M}\right)}}{1 + \underline{\epsilon}_f + \underline{\epsilon}_s(1 + \underline{\epsilon}_d p) + \underline{\epsilon}_d \frac{p}{M}}.$$

If $\mathbf{H}_f^{(q)}$ and $\mathbf{SHS}^\top$ commute,

$$\left\|\left(\mathbf{H}_f^{(q)}\right)^{-1}\mathbf{SHS}^\top\right\| = \left\|(\mathbf{SHS}^\top)^{-1/2}\mathbf{X}^{-1}(\mathbf{SHS}^\top)^{1/2}\right\| = \|\mathbf{X}^{-1}\| \lesssim \frac{1}{1 + \underline{\epsilon}_f + \underline{\epsilon}_s(1 + \underline{\epsilon}_d p) + \underline{\epsilon}_d \frac{p}{M}}.$$

$\square$

**Lemma C.28.** *Under Assumption 3.3 and Assumption 3.6, for any $i \in \{s, d, f, p, a, o\}$, if there exist $(\overline{\epsilon}_i, \underline{\epsilon}_i)$ such that quantization $\mathcal{Q}_i$ is $(\overline{\epsilon}_i, \underline{\epsilon}_i)$-additive, with probability at least $1 - e^{-\Omega(M)}$, it holds*

$$\mathbb{E}_{\mathbf{w}^*}\left[\frac{\left\|\mathbf{v}^{(q)*}\right\|_{\mathbf{I}_{f,0:k^*}^{(q)}}^2}{\gamma N} + \left\|\mathbf{v}^{(q)*}\right\|_{\mathbf{H}_{f,k^*:\infty}^{(q)}}^2\right] \lesssim \frac{\max\left\{\left[\frac{N\gamma\left(1 + \underline{\epsilon}_f + \underline{\epsilon}_s(1 + \underline{\epsilon}_d p) + \underline{\epsilon}_d \frac{p}{M}\right)}{1 + M^a\left[\frac{M^a\left(\overline{\epsilon}_f + \overline{\epsilon}_s(1 + \overline{\epsilon}_d p) + \overline{\epsilon}_d \frac{p}{M}\right) - \left(\underline{\epsilon}_f + \underline{\epsilon}_s(1 + \underline{\epsilon}_d p) + \underline{\epsilon}_d \frac{p}{M}\right)}{1 + M^a\left(\overline{\epsilon}_f + \overline{\epsilon}_s(1 + \overline{\epsilon}_d p) + \overline{\epsilon}_d \frac{p}{M}\right)}\right]^2}\right]^{\frac{1}{a}-1}, M^{1-a}\right\}}{1 + \underline{\epsilon}_f + \underline{\epsilon}_s(1 + \underline{\epsilon}_d p) + \underline{\epsilon}_d \frac{p}{M}}.$$

*Further if $\mathbf{H}_f^{(q)}$ and $\mathbf{SHS}^\top$ commute,*

$$\mathbb{E}_{\mathbf{w}^*}\left[\frac{\left\|\mathbf{v}^{(q)*}\right\|_{\mathbf{I}_{f,0:k^*}^{(q)}}^2}{\gamma N} + \left\|\mathbf{v}^{(q)*}\right\|_{\mathbf{H}_{f,k^*:\infty}^{(q)}}^2\right] \lesssim \frac{\max\left\{\left[N\gamma\left(1 + \underline{\epsilon}_f + \underline{\epsilon}_s(1 + \underline{\epsilon}_d p) + \underline{\epsilon}_d \frac{p}{M}\right)\right]^{\frac{1}{a}-1}, M^{1-a}\right\}}{1 + \underline{\epsilon}_f + \underline{\epsilon}_s(1 + \underline{\epsilon}_d p) + \underline{\epsilon}_d \frac{p}{M}}.$$

*Proof.* By Lemma C.27, for $k \leq M/2$, with probability at least $1 - e^{-\Omega(M)}$,

$$\mathbb{E}_{\mathbf{w}^*}\left[ \frac{\left\| \mathbf{v}^{(q)*} \right\|^2_{\mathbf{I}^{(q)}_{f,0:k^*}}}{\gamma N} + \left\| \mathbf{v}^{(q)*} \right\|^2_{\mathbf{H}^{(q)}_{f,k^*:\infty}} \right]$$

$$\lesssim \mathbb{E}_{\mathbf{w}^*}\left[ \frac{\left(1 + M^{a/2}\frac{M^a\left(\overline{\epsilon}_f + \overline{\epsilon}_s(1+\overline{\epsilon}_d p) + \overline{\epsilon}_d \frac{p}{M}\right) - \left(\underline{\epsilon}_f + \underline{\epsilon}_s(1+\underline{\epsilon}_d p) + \underline{\epsilon}_d \frac{p}{M}\right)}{1 + M^a\left(\overline{\epsilon}_f + \overline{\epsilon}_s(1+\overline{\epsilon}_d p) + \overline{\epsilon}_d \frac{p}{M}\right)}\right)^2}{\left(1 + \underline{\epsilon}_f + \underline{\epsilon}_s(1+\underline{\epsilon}_d p) + \underline{\epsilon}_d \frac{p}{M}\right)^2} \frac{\|\mathbf{w}^*\|^2_{\mathbf{I}_{0:k}}}{\gamma N} + \frac{\|\mathbf{w}^*\|^2_{\mathbf{H}_{k:\infty}}}{1 + \underline{\epsilon}_f + \underline{\epsilon}_s(1+\underline{\epsilon}_d p) + \underline{\epsilon}_d \frac{p}{M}} \right]$$

$$\approx \frac{\left(1 + M^{a/2}\frac{M^a\left(\overline{\epsilon}_f + \overline{\epsilon}_s(1+\overline{\epsilon}_d p) + \overline{\epsilon}_d \frac{p}{M}\right) - \left(\underline{\epsilon}_f + \underline{\epsilon}_s(1+\underline{\epsilon}_d p) + \underline{\epsilon}_d \frac{p}{M}\right)}{1 + M^a\left(\overline{\epsilon}_f + \overline{\epsilon}_s(1+\overline{\epsilon}_d p) + \overline{\epsilon}_d \frac{p}{M}\right)}\right)^2}{\left(1 + \underline{\epsilon}_f + \underline{\epsilon}_s(1+\underline{\epsilon}_d p) + \underline{\epsilon}_d \frac{p}{M}\right)^2} \frac{k}{N\gamma} + \frac{1}{1 + \underline{\epsilon}_f + \underline{\epsilon}_s(1+\underline{\epsilon}_d p) + \underline{\epsilon}_d \frac{p}{M}} \sum_{i>k} i^{-a}$$

$$\approx \frac{\left(1 + M^{a/2}\frac{M^a\left(\overline{\epsilon}_f + \overline{\epsilon}_s(1+\overline{\epsilon}_d p) + \overline{\epsilon}_d \frac{p}{M}\right) - \left(\underline{\epsilon}_f + \underline{\epsilon}_s(1+\underline{\epsilon}_d p) + \underline{\epsilon}_d \frac{p}{M}\right)}{1 + M^a\left(\overline{\epsilon}_f + \overline{\epsilon}_s(1+\overline{\epsilon}_d p) + \overline{\epsilon}_d \frac{p}{M}\right)}\right)^2}{\left(1 + \underline{\epsilon}_f + \underline{\epsilon}_s(1+\underline{\epsilon}_d p) + \underline{\epsilon}_d \frac{p}{M}\right)^2} \frac{k}{N\gamma} + \frac{k^{1-a}}{1 + \underline{\epsilon}_f + \underline{\epsilon}_s(1+\underline{\epsilon}_d p) + \underline{\epsilon}_d \frac{p}{M}}$$

$$\lesssim \frac{\max\left\{ \left[ \frac{N\gamma\left(1+\underline{\epsilon}_f + \underline{\epsilon}_s(1+\underline{\epsilon}_d p) + \underline{\epsilon}_d \frac{p}{M}\right)}{1 + M^a\left[\frac{M^a\left(\overline{\epsilon}_f + \overline{\epsilon}_s(1+\overline{\epsilon}_d p) + \overline{\epsilon}_d \frac{p}{M}\right) - \left(\underline{\epsilon}_f + \underline{\epsilon}_s(1+\underline{\epsilon}_d p) + \underline{\epsilon}_d \frac{p}{M}\right)}{1 + M^a\left(\overline{\epsilon}_f + \overline{\epsilon}_s(1+\overline{\epsilon}_d p) + \overline{\epsilon}_d \frac{p}{M}\right)}\right]^2} \right]^{\frac{1}{a}-1} , M^{1-a} \right\}}{1 + \underline{\epsilon}_f + \underline{\epsilon}_s(1+\underline{\epsilon}_d p) + \underline{\epsilon}_d \frac{p}{M}},$$

where in the last inequality we choose

$$k = [M/2] \wedge \left[ \frac{N\gamma\left(1 + \underline{\epsilon}_f + \underline{\epsilon}_s(1 + \underline{\epsilon}_d p) + \underline{\epsilon}_d \frac{p}{M}\right)}{\left(1 + M^{a/2}\frac{M^a\left(\overline{\epsilon}_f + \overline{\epsilon}_s(1+\overline{\epsilon}_d p) + \overline{\epsilon}_d \frac{p}{M}\right) - \left(\underline{\epsilon}_f + \underline{\epsilon}_s(1+\underline{\epsilon}_d p) + \underline{\epsilon}_d \frac{p}{M}\right)}{1 + M^a\left(\overline{\epsilon}_f + \overline{\epsilon}_s(1+\overline{\epsilon}_d p) + \overline{\epsilon}_d \frac{p}{M}\right)}\right)^2} \right]^{1/a}.$$

The statement when $\mathbf{H}^{(q)}_f$ and $\mathbf{SHS}^\top$ commute can be deduced directly. We omit here for simplicity. $\qquad\square$

## C.9. Population Risk Upper Bounds under Power-law Spectrum

### C.9.1. MULTIPLICATIVE QUANTIZATION

**Theorem C.29.** *Suppose* $\gamma < 1/\left((1+\tilde{\epsilon})\alpha \mathrm{tr}\left(\mathbf{H}^{(q)}_f\right)\right)$. *For any* $i \in \{s, d, f, p, a, o\}$, *if there exist* $(\overline{\epsilon}_i, \underline{\epsilon}_i)$ *such that quantization* $\mathcal{Q}_i$ *is* $(\overline{\epsilon}_i, \underline{\epsilon}_i)$-*multiplicative, then under Assumption 3.1, 3.3, 3.4, 3.5 and 3.6,*

- Irreducible := $\mathcal{R}(\mathbf{w}^*) = \frac{1}{2}\sigma^2$.

- *with probability at least* $1 - e^{-\Omega(M)}$ *over the randomness of* $\mathbf{S}$, $\mathbb{E}_{\mathbf{w}^*}\mathrm{Approx} \lesssim M^{1-a}$.

- *with probability at least* $1 - e^{-\Omega(M)}$ *over the randomness of* $\mathbf{S}$ [8],

$$\mathbb{E}\mathrm{Excess} \lesssim \mathrm{BiasError} + \mathrm{VarianceError} + \mathrm{AdditiveError},$$

---

[8] Here we take expectation on the prior $\mathbf{w}^*$.

*where*

$$\text{BiasError} \lesssim \frac{\max\left\{\left[\frac{N\gamma(1+\underline{\epsilon}_d)(1+\underline{\epsilon}_f)(1+\underline{\epsilon}_s)}{1+\left[\frac{(1+\overline{\epsilon}_d)(1+\overline{\epsilon}_f)(1+\overline{\epsilon}_s)-(1+\underline{\epsilon}_d)(1+\underline{\epsilon}_f)(1+\underline{\epsilon}_s)}{(1+\overline{\epsilon}_d)(1+\overline{\epsilon}_f)(1+\overline{\epsilon}_s)}\right]^2 M^a}\right]^{\frac{1}{a}-1}, M^{1-a}\right\}}{\left[(1+\underline{\epsilon}_d)(1+\underline{\epsilon}_f)(1+\underline{\epsilon}_s)\right]^2},$$

$$\text{VarianceError} \lesssim \frac{\sigma_M^2 + \frac{(1+\tilde{\epsilon})\alpha}{(1+\underline{\epsilon}_d)(1+\underline{\epsilon}_s)(1+\underline{\epsilon}_f)}}{(1+\underline{\epsilon}_d)(1+\underline{\epsilon}_f)(1+\underline{\epsilon}_s)} \frac{\min\left\{M, \left[N\gamma(1+\overline{\epsilon}_f)(1+\overline{\epsilon}_d)(1+\overline{\epsilon}_s)\right]^{1/a}\right\}}{N},$$

$$\text{AdditiveError} \lesssim \frac{\left[(1+\overline{\epsilon}_d)(1+\overline{\epsilon}_f)(1+\overline{\epsilon}_s)-1\right]^2}{\left[(1+\overline{\epsilon}_d)(1+\overline{\epsilon}_f)(1+\overline{\epsilon}_s)\right]^2} + \frac{(1+\overline{\epsilon}_f)(1+\overline{\epsilon}_d)(1+\overline{\epsilon}_s)-1}{(1+\overline{\epsilon}_f)(1+\overline{\epsilon}_d)(1+\overline{\epsilon}_s)(1+\underline{\epsilon}_f)(1+\underline{\epsilon}_d)(1+\underline{\epsilon}_s)},$$

*with*

$$\widetilde{\epsilon} = \overline{\epsilon}_o + (\overline{\epsilon}_o + 1)\left[(1+\overline{\epsilon}_p)\overline{\epsilon}_a + \overline{\epsilon}_p\right],$$

$$\sigma_M^2 = (\overline{\epsilon}_o + 1)\overline{\sigma}^2 + \frac{(\overline{\epsilon}_o + 1)\left[\overline{\epsilon}_p + (1+\overline{\epsilon}_p)\overline{\epsilon}_a\right]\alpha}{(1+\underline{\epsilon}_d)(1+\underline{\epsilon}_s)(1+\underline{\epsilon}_f)}.$$

*Further, if $\mathbf{H}_f^{(q)}$ and $\mathbf{SHS}^\top$ commute,*

$$\text{BiasError} \lesssim \frac{\max\left\{\left[N\gamma(1+\underline{\epsilon}_d)(1+\underline{\epsilon}_f)(1+\underline{\epsilon}_s)\right]^{\frac{1}{a}-1}, M^{1-a}\right\}}{\left[(1+\underline{\epsilon}_d)(1+\underline{\epsilon}_f)(1+\underline{\epsilon}_s)\right]^2}.$$

*Proof.* The proof can be completed by (5), (6), Lemma B.2, (27), Theorem C.18, Lemma C.19, Lemma C.21, Lemma C.26, and noticing the following facts. Firstly, under multiplicative quantization,

$$\mu_{\max}\left((\mathbf{H}_f^{(q)})^{-1}\mathbf{SHS}^\top\right) \leq \frac{1}{(1+\underline{\epsilon}_d)(1+\underline{\epsilon}_f)(1+\underline{\epsilon}_s)}.$$

Secondly,

$$
\begin{aligned}
\mathbb{E}_{\mathbf{w}^*}\left\|\mathbf{v}^{(q)*}\right\|_{\mathbf{H}_f^{(q)}}^2 &= \mathbb{E}_{\mathbf{w}^*}\left[\mathbf{w}^{*\top}\mathbf{HS}^\top(\mathbf{H}_f^{(q)})^{-1}\mathbf{SHw}^*\right] \\
&\leq \mathbb{E}_{\mathbf{w}^*}\|\mathbf{w}^*\|_{\mathbf{H}}^2 \left\|\mathbf{H}^{1/2}\mathbf{S}^\top(\mathbf{H}_f^{(q)})^{-1}\mathbf{SH}^{1/2}\right\| \\
&\asymp \left\|\mathbf{H}^{1/2}\mathbf{S}^\top(\mathbf{H}_f^{(q)})^{-1}\mathbf{SH}^{1/2}\right\| \\
&\leq \frac{1}{(1+\underline{\epsilon}_d)(1+\underline{\epsilon}_s)(1+\underline{\epsilon}_f)}\left\|\mathbf{H}^{1/2}\mathbf{S}^\top(\mathbf{SHS}^\top)^{-1}\mathbf{SH}^{1/2}\right\| \\
&\leq \frac{1}{(1+\underline{\epsilon}_d)(1+\underline{\epsilon}_s)(1+\underline{\epsilon}_f)}.
\end{aligned}
\tag{55}
$$

Thirdly, we derive crude upper bounds for $\frac{1}{\gamma N} \cdot \left\|\mathbf{v}^{(q)*}\right\|_{\mathbf{I}_{f,0:k^*}^{(q)}}^2 + \left\|\mathbf{v}^{(q)*}\right\|_{\mathbf{H}_{f,k^*:\infty}^{(q)}}^2$ in VarianceError.

$$\mathbb{E}_{\mathbf{w}^*}\left[\frac{1}{\gamma N}\cdot\left\|\mathbf{v}^{(q)*}\right\|_{\mathbf{I}_{f,0:k^*}^{(q)}}^2 + \left\|\mathbf{v}^{(q)*}\right\|_{\mathbf{H}_{f,k^*:\infty}^{(q)}}^2\right] \leq \mathbb{E}_{\mathbf{w}^*}\left\|\mathbf{v}^{(q)*}\right\|_{\mathbf{H}_f^{(q)}}^2 \lesssim \frac{1}{(1+\underline{\epsilon}_d)(1+\underline{\epsilon}_s)(1+\underline{\epsilon}_f)},$$

where the last inequality holds by (55). $\qquad\square$

### C.9.2. ADDITIVE QUANTIZATION

**Theorem C.30.** *Suppose $\gamma < 1/\left(\alpha\mathrm{tr}\left(\mathbf{H}_f^{(q)}\right)\right)$. For any $i \in \{s, d, f, p, a, o\}$, if there exist $(\overline{\epsilon}_i, \underline{\epsilon}_i)$ such that quantization $\mathcal{Q}_i$ is $(\overline{\epsilon}_i, \underline{\epsilon}_i)$-additive, then under Assumption 3.1, 3.3, 3.4, 3.5 and 3.6,*

- Irreducible $:= \mathcal{R}(\mathbf{w}^*) = \frac{1}{2}\sigma^2$.

- *with probability at least $1 - e^{-\Omega(M)}$ over the randomness of $\mathbf{S}$, $\mathbb{E}_{\mathbf{w}^*}\text{Approx} \lesssim M^{1-a}$.*

- *with probability at least $1 - e^{-\Omega(M)}$ over the randomness of $\mathbf{S}$ [9],*

$$\mathbb{E}\text{Excess} \lesssim \text{BiasError} + \text{VarianceError} + \text{AdditiveError},$$

*where*

$$\text{BiasError} \lesssim \frac{\max\left\{\left[\frac{N\gamma\left(1+\underline{\epsilon}_f+\underline{\epsilon}_s(1+\underline{\epsilon}_dp)+\underline{\epsilon}_d\frac{p}{M}\right)}{1+M^a\left[\frac{M^a\left(\overline{\epsilon}_f+\overline{\epsilon}_s(1+\overline{\epsilon}_dp)+\overline{\epsilon}_d\frac{p}{M}\right)-\left(\underline{\epsilon}_f+\underline{\epsilon}_s(1+\underline{\epsilon}_dp)+\underline{\epsilon}_d\frac{p}{M}\right)}{1+M^a\left(\overline{\epsilon}_f+\overline{\epsilon}_s(1+\overline{\epsilon}_dp)+\overline{\epsilon}_d\frac{p}{M}\right)}\right]^2}\right]^{\frac{1}{a}-1}, M^{1-a}\right\}}{\left[1+\underline{\epsilon}_f+\underline{\epsilon}_s(1+\underline{\epsilon}_dp)+\underline{\epsilon}_d\frac{p}{M}\right]^2},$$

$$\text{VarianceError} \lesssim \frac{\sigma_G^2 + \frac{\alpha}{1+\underline{\epsilon}_s+\underline{\epsilon}_f+\underline{\epsilon}_s\underline{\epsilon}_dp+\underline{\epsilon}_d\frac{p}{M}}}{1+\underline{\epsilon}_s+\underline{\epsilon}_f+\underline{\epsilon}_s\underline{\epsilon}_dp+\underline{\epsilon}_d\frac{p}{M}}k_{\text{eff}} + \frac{\gamma^2N^2\left(\overline{\epsilon}_f+(1+\overline{\epsilon}_dp)\overline{\epsilon}_s+\overline{\epsilon}_d\frac{p}{M}\right)^2(M-k_{\text{eff}})}{N},$$

$$\text{AdditiveError} \lesssim \frac{\left(\overline{\epsilon}_s+\overline{\epsilon}_f+\overline{\epsilon}_s\overline{\epsilon}_dp+\overline{\epsilon}_d\frac{p}{M}\right)^2}{\left(M^{-a}+\overline{\epsilon}_s+\overline{\epsilon}_f+\overline{\epsilon}_s\overline{\epsilon}_dp+\overline{\epsilon}_d\frac{p}{M}\right)^2}+$$

$$\frac{\overline{\epsilon}_s+\overline{\epsilon}_s\overline{\epsilon}_dp+\overline{\epsilon}_f+\overline{\epsilon}_d\frac{p}{M}}{\overline{\epsilon}_s+\overline{\epsilon}_s\overline{\epsilon}_dp+\overline{\epsilon}_f+\overline{\epsilon}_d\frac{p}{M}+M^{-a}} \cdot \frac{1}{1+\underline{\epsilon}_s(1+\underline{\epsilon}_dp)+\underline{\epsilon}_f+\underline{\epsilon}_d\frac{p}{M}},$$

*with*

$$\sigma_G^2 = \overline{\sigma}^2 + \overline{\epsilon}_a + \overline{\epsilon}_o + \alpha\overline{\epsilon}_p\left[1+p\overline{\epsilon}_d+M(\overline{\epsilon}_f+\overline{\epsilon}_s+\overline{\epsilon}_s\overline{\epsilon}_dp)\right],$$

$$k_{\text{eff}} = \left[M^{-a} \vee \left(\frac{1}{N\gamma}-\overline{\epsilon}_f-(1+\overline{\epsilon}_dp)\overline{\epsilon}_s-\overline{\epsilon}_d\frac{p}{M}\right)\right]^{-\frac{1}{a}}.$$

*Further, if $\mathbf{H}_f^{(q)}$ and $\mathbf{SHS}^\top$ commute,*

$$\text{BiasError} \lesssim \frac{\max\left\{\left[N\gamma\left(1+\underline{\epsilon}_f+\underline{\epsilon}_s(1+\underline{\epsilon}_dp)+\underline{\epsilon}_d\frac{p}{M}\right)\right]^{\frac{1}{a}-1}, M^{1-a}\right\}}{\left[1+\underline{\epsilon}_f+\underline{\epsilon}_s(1+\underline{\epsilon}_dp)+\underline{\epsilon}_d\frac{p}{M}\right]^2}.$$

*Proof.* The proof can be completed by (5), (6), Lemma B.2, (27), Theorem C.17, Lemma C.20, Lemma C.22, Lemma C.28, and noticing the following facts. Firstly, by Lemma F.1, with probability at least $1 - e^{-\Omega(M)}$,

$$\mu_{\max}\left((\mathbf{H}_f^{(q)})^{-1}\mathbf{SHS}^\top\right)$$

$$\leq \mu_{\max}\left(\left(\mathbf{SHS}^\top + (\underline{\epsilon}_f + \underline{\epsilon}_s + \underline{\epsilon}_s\underline{\epsilon}_dp) + \underline{\epsilon}_d\frac{p}{M})\mathbf{I}\right)^{-1}\mathbf{SHS}^\top\right)$$

$$\lesssim \frac{1}{1+\underline{\epsilon}_s+\underline{\epsilon}_f+\underline{\epsilon}_s\underline{\epsilon}_dp+\underline{\epsilon}_d\frac{p}{M}}.$$

Secondly, with probability at least $1 - e^{-\Omega(M)}$,

$$\text{tr}(\mathbf{H}_f^{(q)}) \lesssim \text{tr}\left(\mathbf{SHS}^\top + (\overline{\epsilon}_f + \overline{\epsilon}_s(1+\overline{\epsilon}_dp) + \overline{\epsilon}_d\frac{p}{M})\mathbf{I}\right) \asymp 1 + M(\overline{\epsilon}_f + \overline{\epsilon}_s(1+\overline{\epsilon}_dp)) + p\overline{\epsilon}_d.$$

---

[9]Here we take expectation on the prior $\mathbf{w}^*$.

Thirdly, we derive crude upper bounds for $\frac{1}{\gamma N} \cdot \left\| \mathbf{v}^{(q)*} \right\|^2_{\mathbf{I}^{(q)}_{f,0:k^*}} + \left\| \mathbf{v}^{(q)*} \right\|^2_{\mathbf{H}^{(q)}_{f,k^*:\infty}}$ in VarianceError. By Lemma G.1,

$$
\mathbb{E}_{\mathbf{w}^*} \left[ \frac{1}{\gamma N} \cdot \left\| \mathbf{v}^{(q)*} \right\|^2_{\mathbf{I}^{(q)}_{f,0:k^*}} + \left\| \mathbf{v}^{(q)*} \right\|^2_{\mathbf{H}^{(q)}_{f,k^*:\infty}} \right]
$$

$$
\leq \mathbb{E}_{\mathbf{w}^*} \left\| \mathbf{v}^{(q)*} \right\|^2_{\mathbf{H}^{(q)}_f}
$$

$$
= \mathbb{E}_{\mathbf{w}^*} \operatorname{tr} \left( \mathbf{w}^{*\top} \mathbf{H} \mathbf{S}^\top (\mathbf{H}^{(q)}_f)^{-1} \mathbf{S} \mathbf{H} \mathbf{w}^* \right)
$$

$$
\leq \mathbb{E}_{\mathbf{w}^*} \left\| \mathbf{w}^* \right\|^2_{\mathbf{H}} \left\| \mathbf{H}^{1/2} \mathbf{S}^\top (\mathbf{H}^{(q)}_f)^{-1} \mathbf{S} \mathbf{H}^{1/2} \right\|
$$

$$
\asymp \left\| \mathbf{H}^{1/2} \mathbf{S}^\top (\mathbf{H}^{(q)}_f)^{-1} \mathbf{S} \mathbf{H}^{1/2} \right\|
$$

$$
\lesssim \frac{1}{1 + \underline{\epsilon}_s + \underline{\epsilon}_f + \underline{\epsilon}_s \underline{\epsilon}_d p + \underline{\epsilon}_d \frac{p}{M}} .
$$

$\square$

# D. Lower Bound Analysis

## D.1. Update Rule

Recall Lemma B.2,

$$
\mathbb{E} \left[ \mathcal{R}_M(\overline{\mathbf{v}}_N) - \mathcal{R}_M(\mathbf{v}^*) \right] = \underbrace{\frac{1}{2} \left\langle \mathbf{S} \mathbf{H} \mathbf{S}^\top, \mathbb{E} \left[ (\mathbf{v}^{(q)*} - \overline{\mathbf{v}}_N) \otimes (\mathbf{v}^{(q)*} - \overline{\mathbf{v}}_N) \right] \right\rangle}_{R_N}
$$

$$
+ \frac{1}{2} \left\langle \mathbf{S} \mathbf{H} \mathbf{S}^\top, (\mathbf{v}^{(q)*} - \mathbf{v}^*) \otimes (\mathbf{v}^{(q)*} - \mathbf{v}^*) \right\rangle
$$

$$
+ \left( \mathbf{v}^{(q)*} \right)^\top \frac{1}{N\gamma} \left[ \mathbf{I} - \left( \mathbf{I} - \gamma \mathbf{H}^{(q)}_f \right)^N \right] \left( \mathbf{H}^{(q)}_f \right)^{-1} \left( \mathbf{H}^{(q)}_f - \mathbf{S} \mathbf{H} \mathbf{S}^\top \right) \mathbf{v}^{(q)*} .
$$

We first derive update rule for $\mathbb{E} \left[ \boldsymbol{\eta}_t \otimes \boldsymbol{\eta}_t \right]$. By Lemma C.1,

$$
\boldsymbol{\eta}_t = \left( \mathbf{I} - \gamma \tilde{\mathbf{x}}^{(q)}_t (\tilde{\mathbf{x}}^{(q)}_t)^\top \right) \boldsymbol{\eta}_{t-1} + \gamma \left( \xi_t + \epsilon^{(o)}_t - \epsilon^{(a)}_t - (\tilde{\mathbf{x}}^{(q)}_t)^\top \boldsymbol{\epsilon}^{(p)}_{t-1} \right) \tilde{\mathbf{x}}^{(q)}_t .
$$

Then by (7) and unbiased Assumption 3.1,

$$
\mathbb{E} \left[ \boldsymbol{\eta}_t \otimes \boldsymbol{\eta}_t \right] = \mathbb{E} \left[ \left( \mathbf{I} - \gamma \tilde{\mathbf{x}}^{(q)}_t (\tilde{\mathbf{x}}^{(q)}_t)^\top \right) \mathbb{E} \left[ \boldsymbol{\eta}_{t-1} \otimes \boldsymbol{\eta}_{t-1} \right] \left( \mathbf{I} - \gamma \tilde{\mathbf{x}}^{(q)}_t (\tilde{\mathbf{x}}^{(q)}_t)^\top \right) \right] + \boldsymbol{\Sigma}_t
$$

$$
- 2\gamma^2 \mathbb{E} \left[ \tilde{\mathbf{x}}^{(q)}_t (\tilde{\mathbf{x}}^{(q)}_t)^\top \boldsymbol{\eta}_{t-1} (\tilde{\mathbf{x}}^{(q)}_t)^\top \xi_t \right] ,
$$

where $\boldsymbol{\Sigma}_t$ is defined in (19). In lower bound analysis, we consider low-precision well-specific model:

$$
\mathbb{E} \left[ \xi_t | \tilde{\mathbf{x}}^{(q)}_t \right] = 0 .
$$

Therefore,

$$
\mathbb{E} \left[ \boldsymbol{\eta}_t \otimes \boldsymbol{\eta}_t \right] = \mathbb{E} \left[ \left( \mathbf{I} - \gamma \tilde{\mathbf{x}}^{(q)}_t (\tilde{\mathbf{x}}^{(q)}_t)^\top \right) \mathbb{E} \left[ \boldsymbol{\eta}_{t-1} \otimes \boldsymbol{\eta}_{t-1} \right] \left( \mathbf{I} - \gamma \tilde{\mathbf{x}}^{(q)}_t (\tilde{\mathbf{x}}^{(q)}_t)^\top \right) \right] + \boldsymbol{\Sigma}_t . \tag{56}
$$

We then summarize the update rule for $\mathbb{E} \left[ \boldsymbol{\eta}_t \otimes \boldsymbol{\eta}_t \right]$ as follows.

**Lemma D.1** (Update rule under general quantization, a lower bound). *Under Assumption 3.1, 3.3, 3.4 and 3.5, it holds*

$$
\mathbb{E} \left[ \boldsymbol{\eta}_t \otimes \boldsymbol{\eta}_t \right] \succeq \mathbb{E} \left[ \left( \mathbf{I} - \gamma \tilde{\mathbf{x}}^{(q)}_t (\tilde{\mathbf{x}}^{(q)}_t)^\top \right) \mathbb{E} \left[ \boldsymbol{\eta}_{t-1} \otimes \boldsymbol{\eta}_{t-1} \right] \left( \mathbf{I} - \gamma \tilde{\mathbf{x}}^{(q)}_t (\tilde{\mathbf{x}}^{(q)}_t)^\top \right) \right]
$$

$$
+ \gamma^2 \left[ \underline{\sigma}^2 + \inf_t \beta \operatorname{tr} \left( \mathbf{H}^{(q)}_f \mathbb{E} \left[ \boldsymbol{\epsilon}^{(p)}_{t-1} \boldsymbol{\epsilon}^{(p)\top}_{t-1} \right] \right) + \inf_t \left( \mathbb{E} \left[ \epsilon^{(a)2}_t \Big| a_t \right] + \mathbb{E} \left[ \epsilon^{(o)2}_t \Big| o_t \right] \right) \right] \mathbf{H}^{(q)}_f .
$$

*Proof.* The proof focuses on dealing with each term of $\mathbf{\Sigma}_t$ in (19). Firstly, by Assumption 3.4,

$$\mathbb{E}\left[(\tilde{\mathbf{x}}_t^{(q)})^\top \boldsymbol{\epsilon}_{t-1}^{(p)} \boldsymbol{\epsilon}_{t-1}^{(p)}{}^\top \tilde{\mathbf{x}}_t^{(q)} \tilde{\mathbf{x}}_t^{(q)} (\tilde{\mathbf{x}}_t^{(q)})^\top\right] \succeq \beta \mathrm{tr}\left(\mathbf{H}_f^{(q)} \mathbb{E}\left[\boldsymbol{\epsilon}_{t-1}^{(p)} \boldsymbol{\epsilon}_{t-1}^{(p)}{}^\top\right]\right) \mathbf{H}_f^{(q)}$$
$$\succeq \inf_t \beta \mathrm{tr}\left(\mathbf{H}_f^{(q)} \mathbb{E}\left[\boldsymbol{\epsilon}_{t-1}^{(p)} \boldsymbol{\epsilon}_{t-1}^{(p)}{}^\top\right]\right) \mathbf{H}_f^{(q)}. \tag{57}$$

Secondly, denote

$$a_t = (\tilde{\mathbf{x}}_t^{(q)})^\top \mathcal{Q}_p(\mathbf{v}_{t-1}), \quad o_t = \mathcal{Q}_l(y_t) - \mathcal{Q}_a\left((\tilde{\mathbf{x}}_t^{(q)})^\top \mathcal{Q}_p(\mathbf{v}_{t-1})\right),$$

then

$$\mathbb{E}\left[\left(\epsilon_t^{(a)^2} + \epsilon_t^{(o)^2}\right) \tilde{\mathbf{x}}_t^{(q)} (\tilde{\mathbf{x}}_t^{(q)})^\top\right] \succeq \inf_t \left\{\mathbb{E}\left[\epsilon_t^{(a)^2}\Big| a_t\right] + \mathbb{E}\left[\epsilon_t^{(o)^2}\Big| o_t\right]\right\} \mathbf{H}_f^{(q)}. \tag{58}$$

Thirdly, by Assumption 3.5,

$$\mathbb{E}\left[\xi_t^2 \tilde{\mathbf{x}}_t^{(q)} (\tilde{\mathbf{x}}_t^{(q)})^\top\right] \succeq \underline{\sigma}^2 \mathbf{H}_f^{(q)}. \tag{59}$$

Therefore, together with (19), (57), (58), and (59), it holds,

$$\mathbf{\Sigma}_t/\gamma^2 \succeq \left[\underline{\sigma}^2 + \inf_t \beta \mathrm{tr}\left(\mathbf{H}_f^{(q)} \mathbb{E}\left[\boldsymbol{\epsilon}_{t-1}^{(p)} \boldsymbol{\epsilon}_{t-1}^{(p)}{}^\top\right]\right) + \inf_t \left(\mathbb{E}\left[\epsilon_t^{(a)^2}\Big| a_t\right] + \mathbb{E}\left[\epsilon_t^{(o)^2}\Big| o_t\right]\right)\right] \mathbf{H}_f^{(q)}.$$

This immediately implies that

$$\mathbb{E}\left[\boldsymbol{\eta}_t \otimes \boldsymbol{\eta}_t\right]$$
$$\succeq \mathbb{E}\left[\left(\mathbf{I} - \gamma \tilde{\mathbf{x}}_t^{(q)} (\tilde{\mathbf{x}}_t^{(q)})^\top\right) \mathbb{E}\left[\boldsymbol{\eta}_{t-1} \otimes \boldsymbol{\eta}_{t-1}\right] \left(\mathbf{I} - \gamma \tilde{\mathbf{x}}_t^{(q)} (\tilde{\mathbf{x}}_t^{(q)})^\top\right)\right]$$
$$+ \gamma^2 \left[\underline{\sigma}^2 + \inf_t \beta \mathrm{tr}\left(\mathbf{H}_f^{(q)} \mathbb{E}\left[\boldsymbol{\epsilon}_{t-1}^{(p)} \boldsymbol{\epsilon}_{t-1}^{(p)}{}^\top\right]\right) + \inf_t \left(\mathbb{E}\left[\epsilon_t^{(a)^2}\Big| a_t\right] + \mathbb{E}\left[\epsilon_t^{(o)^2}\Big| o_t\right]\right)\right] \mathbf{H}_f^{(q)}.$$

$\square$

**Lemma D.2** (Update rule under multiplicative quantization, a lower bound). *If there exist $\underline{\epsilon}_p, \underline{\epsilon}_a$ and $\underline{\epsilon}_o$ such that for any $i \in \{p, a, o\}$, quantization $\mathcal{Q}_i$ is $\underline{\epsilon}_i$-multiplicative, then under Assumption 3.1, 3.3, 3.4, 3.5, if the stepsize $\gamma < \frac{1}{\bar{\lambda}_1^{(q)}}$,*

$$\mathbb{E}\left[\boldsymbol{\eta}_t \otimes \boldsymbol{\eta}_t\right]$$
$$\succeq \mathbb{E}\left[\left(\mathbf{I} - \gamma \tilde{\mathbf{x}}_t^{(q)} (\tilde{\mathbf{x}}_t^{(q)})^\top\right) \mathbb{E}\left[\boldsymbol{\eta}_{t-1} \otimes \boldsymbol{\eta}_{t-1}\right] \left(\mathbf{I} - \gamma \tilde{\mathbf{x}}_t^{(q)} (\tilde{\mathbf{x}}_t^{(q)})^\top\right)\right] + \gamma^2 (1 + \underline{\epsilon}_o) \underline{\sigma}^2 \mathbf{H}_f^{(q)}$$
$$+ \gamma^2 (1 + \underline{\epsilon}_o) \left[\underline{\epsilon}_p + (1 + \underline{\epsilon}_p)\underline{\epsilon}_a\right] \beta \mathrm{tr}\left(\mathbf{H}_f^{(q)} \left[\mathbf{I} - (\mathbf{I} - \gamma \mathbf{H}_f^{(q)})^{(t-1)}\right]^2 \mathbf{v}^{(q)*} \left(\mathbf{v}^{(q)*}\right)^\top\right) \mathbf{H}_f^{(q)}$$
$$+ \gamma^2 (1 + \underline{\epsilon}_o) \left[\underline{\epsilon}_p + (1 + \underline{\epsilon}_p)\underline{\epsilon}_a\right] \frac{\gamma \beta^2}{2} \mathrm{tr}\left(\mathbf{H}_f^{(q)} (\mathbf{I} - \gamma \mathbf{H}_f^{(q)})^{2(t-1)} \mathbf{H}_f^{(q)}\right) \left\|\mathbf{v}^{(q)*}\right\|_{\mathbf{I} - (\mathbf{I} - \gamma \mathbf{H}_f^{(q)})^{2(t-1)}}^2 \mathbf{H}_f^{(q)}.$$

*Proof.* The proof focuses on dealing with each term of $\mathbf{\Sigma}_t$ in (19). Firstly, by Assumption 3.5,

$$\mathbb{E}\left[\xi_t^2 \tilde{\mathbf{x}}_t^{(q)} (\tilde{\mathbf{x}}_t^{(q)})^\top\right] \succeq \underline{\sigma}^2 \mathbf{H}_f^{(q)}. \tag{60}$$

Secondly, by the definition of multiplicative quantization,

$$\mathbb{E}\left[(\tilde{\mathbf{x}}_t^{(q)})^\top \boldsymbol{\epsilon}_{t-1}^{(p)} \boldsymbol{\epsilon}_{t-1}^{(p)}{}^\top \tilde{\mathbf{x}}_t^{(q)} \tilde{\mathbf{x}}_t^{(q)} (\tilde{\mathbf{x}}_t^{(q)})^\top\right] \succeq \underline{\epsilon}_p \mathbb{E}\left[(\tilde{\mathbf{x}}_t^{(q)})^\top \mathbf{v}_{t-1} \mathbf{v}_{t-1}^\top \tilde{\mathbf{x}}_t^{(q)} \tilde{\mathbf{x}}_t^{(q)} (\tilde{\mathbf{x}}_t^{(q)})^\top\right], \tag{61}$$

Thirdly, by the definition of multiplicative quantization,

$$\mathbb{E}\left[\epsilon_t^{(a)^2} \tilde{\mathbf{x}}_t^{(q)} (\tilde{\mathbf{x}}_t^{(q)})^\top\right] \succeq \underline{\epsilon}_a \mathbb{E}\left[(\tilde{\mathbf{x}}_t^{(q)})^\top \mathcal{Q}_p(\mathbf{v}_{t-1}) \mathcal{Q}_p(\mathbf{v}_{t-1})^\top \tilde{\mathbf{x}}_t^{(q)} \tilde{\mathbf{x}}_t^{(q)} (\tilde{\mathbf{x}}_t^{(q)})^\top\right]$$
$$= \underline{\epsilon}_a \mathbb{E}\left[(\tilde{\mathbf{x}}_t^{(q)})^\top \boldsymbol{\epsilon}_{t-1}^{(p)} \boldsymbol{\epsilon}_{t-1}^{(p)}{}^\top \tilde{\mathbf{x}}_t^{(q)} \tilde{\mathbf{x}}_t^{(q)} (\tilde{\mathbf{x}}_t^{(q)})^\top\right] + \underline{\epsilon}_a \mathbb{E}\left[(\tilde{\mathbf{x}}_t^{(q)})^\top \mathbf{v}_{t-1} \mathbf{v}_{t-1}^\top \tilde{\mathbf{x}}_t^{(q)} \tilde{\mathbf{x}}_t^{(q)} (\tilde{\mathbf{x}}_t^{(q)})^\top\right] \tag{62}$$
$$\succeq (1 + \underline{\epsilon}_p) \underline{\epsilon}_a \mathbb{E}\left[(\tilde{\mathbf{x}}_t^{(q)})^\top \mathbf{v}_{t-1} \mathbf{v}_{t-1}^\top \tilde{\mathbf{x}}_t^{(q)} \tilde{\mathbf{x}}_t^{(q)} (\tilde{\mathbf{x}}_t^{(q)})^\top\right],$$

where the equality holds by the unbiased quantization Assumption 3.1. Fourthly, by the definition of multiplicative quantization,

$$
\begin{aligned}
&\mathbb{E}\left[\epsilon_t^{(o)^2}\tilde{\mathbf{x}}_t^{(q)}(\tilde{\mathbf{x}}_t^{(q)})^\top\right]\\
&\succeq_{\epsilon_o}\mathbb{E}\left[\left[\mathcal{Q}_l(y_t)-\mathcal{Q}_a\left((\tilde{\mathbf{x}}_t^{(q)})^\top\mathcal{Q}_p(\mathbf{v}_{t-1})\right)\right]^2\tilde{\mathbf{x}}_t^{(q)}(\tilde{\mathbf{x}}_t^{(q)})^\top\right]\\
&=_{\epsilon_o}\mathbb{E}\left[\left[\mathcal{Q}_l(y_t)-(\tilde{\mathbf{x}}_t^{(q)})^\top\mathcal{Q}_p(\mathbf{v}_{t-1})-\epsilon_t^{(a)}\right]^2\tilde{\mathbf{x}}_t^{(q)}(\tilde{\mathbf{x}}_t^{(q)})^\top\right]\\
&=_{\epsilon_o}\mathbb{E}\left[\left[\mathcal{Q}_l(y_t)-(\tilde{\mathbf{x}}_t^{(q)})^\top\mathbf{v}_{t-1}-(\tilde{\mathbf{x}}_t^{(q)})^\top\boldsymbol{\epsilon}_{t-1}^{(p)}-\epsilon_t^{(a)}\right]^2\tilde{\mathbf{x}}_t^{(q)}(\tilde{\mathbf{x}}_t^{(q)})^\top\right]\\
&=_{\epsilon_o}\mathbb{E}\left[\left[\xi_t-(\tilde{\mathbf{x}}_t^{(q)})^\top\boldsymbol{\eta}_{t-1}-(\tilde{\mathbf{x}}_t^{(q)})^\top\boldsymbol{\epsilon}_{t-1}^{(p)}-\epsilon_t^{(a)}\right]^2\tilde{\mathbf{x}}_t^{(q)}(\tilde{\mathbf{x}}_t^{(q)})^\top\right]\\
&=_{\epsilon_o}\mathbb{E}\left[\left[\xi_t^2+(\tilde{\mathbf{x}}_t^{(q)})^\top\boldsymbol{\epsilon}_{t-1}^{(p)}\boldsymbol{\epsilon}_{t-1}^{(p)^\top}\tilde{\mathbf{x}}_t^{(q)}+\epsilon_t^{(a)^2}+(\tilde{\mathbf{x}}_t^{(q)})^\top\boldsymbol{\eta}_{t-1}\boldsymbol{\eta}_{t-1}^\top\tilde{\mathbf{x}}_t^{(q)}\right]\tilde{\mathbf{x}}_t^{(q)}(\tilde{\mathbf{x}}_t^{(q)})^\top\right],
\end{aligned}
\tag{63}
$$

where the last inequality holds by the unbiased quantization Assumption 3.1 and the low-precision well-specified model assumption $\mathbb{E}\left[\xi_t|\tilde{\mathbf{x}}_t^{(q)}\right]=0$.

We then focus on deriving lower bounds for $\mathbb{E}\left[(\tilde{\mathbf{x}}_t^{(q)})^\top\mathbf{v}_{t-1}\mathbf{v}_{t-1}^\top\tilde{\mathbf{x}}_t^{(q)}\tilde{\mathbf{x}}_t^{(q)}(\tilde{\mathbf{x}}_t^{(q)})^\top\right]$. Noticing that

$$
\begin{aligned}
&\mathbb{E}\left[(\tilde{\mathbf{x}}_t^{(q)})^\top\mathbf{v}_{t-1}\mathbf{v}_{t-1}^\top\tilde{\mathbf{x}}_t^{(q)}\tilde{\mathbf{x}}_t^{(q)}(\tilde{\mathbf{x}}_t^{(q)})^\top\right]\\
&=\mathbb{E}\left[(\tilde{\mathbf{x}}_t^{(q)})^\top\left(\boldsymbol{\eta}_{t-1}+\mathbf{v}^{(q)^*}\right)\left(\boldsymbol{\eta}_{t-1}+\mathbf{v}^{(q)^*}\right)^\top\tilde{\mathbf{x}}_t^{(q)}\tilde{\mathbf{x}}_t^{(q)}(\tilde{\mathbf{x}}_t^{(q)})^\top\right]\\
&=\mathbb{E}\left[(\tilde{\mathbf{x}}_t^{(q)})^\top\mathbb{E}\left[\left(\boldsymbol{\eta}_{t-1}+\mathbf{v}^{(q)^*}\right)\left(\boldsymbol{\eta}_{t-1}+\mathbf{v}^{(q)^*}\right)^\top\right]\tilde{\mathbf{x}}_t^{(q)}\tilde{\mathbf{x}}_t^{(q)}(\tilde{\mathbf{x}}_t^{(q)})^\top\right].
\end{aligned}
$$

We first utilize the accurate expectation:

$$
\mathbb{E}\left[\boldsymbol{\eta}_t\right]=-\left(\mathbf{I}-\gamma\mathbf{H}_f^{(q)}\right)^t\mathbf{v}^{(q)^*}.
$$

Next, we utilize the crude bound of $\mathbb{E}\left[\boldsymbol{\eta}_t\boldsymbol{\eta}_t^\top\right]$. From the update rule of $\mathbb{E}\left[\boldsymbol{\eta}_t\boldsymbol{\eta}_t^\top\right]$ (56), we have

$$
\begin{aligned}
\mathbb{E}\left[\boldsymbol{\eta}_t\boldsymbol{\eta}_t^\top\right]&\succeq(\mathcal{I}-\gamma\mathcal{T}^{(q)})\circ\mathbb{E}\left[\boldsymbol{\eta}_{t-1}\boldsymbol{\eta}_{t-1}^\top\right]\\
&=(\mathcal{I}-\gamma\widetilde{\mathcal{T}}^{(q)})\circ\mathbb{E}\left[\boldsymbol{\eta}_{t-1}\boldsymbol{\eta}_{t-1}^\top\right]+\gamma^2(\mathcal{M}^{(q)}-\widetilde{\mathcal{M}}^{(q)})\circ\mathbb{E}\left[\boldsymbol{\eta}_{t-1}\boldsymbol{\eta}_{t-1}^\top\right].
\end{aligned}
$$

Noticing that

$$
\mathbb{E}\left[\boldsymbol{\eta}_{t-1}\boldsymbol{\eta}_{t-1}^\top\right]\succeq\mathbb{E}\left[\boldsymbol{\eta}_{t-1}\right]\mathbb{E}\left[\boldsymbol{\eta}_{t-1}\right]^\top=\left(\mathbf{I}-\gamma\mathbf{H}_f^{(q)}\right)^{t-1}\mathbf{v}^{(q)^*}\left(\mathbf{v}^{(q)^*}\right)^\top\left(\mathbf{I}-\gamma\mathbf{H}_f^{(q)}\right)^{t-1},
$$

by Assumption 3.4 we have

$$
(\mathcal{M}^{(q)}-\widetilde{\mathcal{M}}^{(q)})\circ\mathbb{E}\left[\boldsymbol{\eta}_{t-1}\boldsymbol{\eta}_{t-1}^\top\right]\succeq\beta\mathrm{tr}\left(\mathbf{H}_f^{(q)}\left(\mathbf{I}-\gamma\mathbf{H}_f^{(q)}\right)^{t-1}\mathbf{v}^{(q)^*}\left(\mathbf{v}^{(q)^*}\right)^\top\left(\mathbf{I}-\gamma\mathbf{H}_f^{(q)}\right)^{t-1}\right)\mathbf{H}_f^{(q)}.
$$

Hence,

$$
\mathbb{E}\left[\boldsymbol{\eta}_t\boldsymbol{\eta}_t^\top\right]\succeq(\mathcal{I}-\gamma\widetilde{\mathcal{T}}^{(q)})\circ\mathbb{E}\left[\boldsymbol{\eta}_{t-1}\boldsymbol{\eta}_{t-1}^\top\right]+\gamma^2\beta\mathrm{tr}\left(\mathbf{H}_f^{(q)}\left(\mathbf{I}-\gamma\mathbf{H}_f^{(q)}\right)^{t-1}\mathbf{v}^{(q)^*}\left(\mathbf{v}^{(q)^*}\right)^\top\left(\mathbf{I}-\gamma\mathbf{H}_f^{(q)}\right)^{t-1}\right)\mathbf{H}_f^{(q)}.
$$

By solving recursion,

$$
\begin{aligned}
\mathbb{E}\left[\boldsymbol{\eta}_t \boldsymbol{\eta}_t^{\top}\right] \succeq & \gamma^2 \beta \sum_{i=0}^{t-1}(\mathcal{I}-\gamma \widetilde{\mathcal{T}}^{(q)})^i \circ \operatorname{tr}\left(\mathbf{H}_f^{(q)}\left(\mathbf{I}-\gamma \mathbf{H}_f^{(q)}\right)^{t-1-i} \mathbf{v}^{(q)*}\left(\mathbf{v}^{(q)*}\right)^{\top}\left(\mathbf{I}-\gamma \mathbf{H}_f^{(q)}\right)^{t-1-i}\right) \mathbf{H}_f^{(q)} \\
& +(\mathcal{I}-\gamma \widetilde{\mathcal{T}}^{(q)})^t \circ \mathbb{E}\left[\boldsymbol{\eta}_0 \boldsymbol{\eta}_0^{\top}\right] \\
= & \gamma^2 \beta \sum_{i=0}^{t-1}(\mathbf{I}-\gamma \mathbf{H}_f^{(q)})^{2i} \operatorname{tr}\left(\mathbf{H}_f^{(q)}\left(\mathbf{I}-\gamma \mathbf{H}_f^{(q)}\right)^{t-1-i} \mathbf{v}^{(q)*}\left(\mathbf{v}^{(q)*}\right)^{\top}\left(\mathbf{I}-\gamma \mathbf{H}_f^{(q)}\right)^{t-1-i}\right) \mathbf{H}_f^{(q)} \\
& +\left(\mathbf{I}-\gamma \mathbf{H}_f^{(q)}\right)^t \mathbf{v}^{(q)*}\left(\mathbf{v}^{(q)*}\right)^{\top}\left(\mathbf{I}-\gamma \mathbf{H}_f^{(q)}\right)^t \\
\succeq & \gamma^2 \beta \sum_{i=0}^{t-1}(\mathbf{I}-\gamma \mathbf{H}_f^{(q)})^{2t} \operatorname{tr}\left(\mathbf{H}_f^{(q)}\left(\mathbf{I}-\gamma \mathbf{H}_f^{(q)}\right)^{2(t-1-i)} \mathbf{v}^{(q)*}\left(\mathbf{v}^{(q)*}\right)^{\top}\right) \mathbf{H}_f^{(q)} \\
& +\left(\mathbf{I}-\gamma \mathbf{H}_f^{(q)}\right)^t \mathbf{v}^{(q)*}\left(\mathbf{v}^{(q)*}\right)^{\top}\left(\mathbf{I}-\gamma \mathbf{H}_f^{(q)}\right)^t \\
\succeq & \frac{\gamma \beta}{2}(\mathbf{I}-\gamma \mathbf{H}_f^{(q)})^{2t} \mathbf{H}_f^{(q)}\left\|\mathbf{v}^{(q)*}\right\|_{\mathbf{I}-(\mathbf{I}-\gamma \mathbf{H}_f^{(q)})^{2t}}^2+\left(\mathbf{I}-\gamma \mathbf{H}_f^{(q)}\right)^t \mathbf{v}^{(q)*}\left(\mathbf{v}^{(q)*}\right)^{\top}\left(\mathbf{I}-\gamma \mathbf{H}_f^{(q)}\right)^t .
\end{aligned}
$$

Therefore, it holds

$$
\begin{aligned}
& \mathbb{E}\left[(\tilde{\mathbf{x}}_t^{(q)})^{\top} \mathbf{v}_{t-1} \mathbf{v}_{t-1}^{\top} \tilde{\mathbf{x}}_t^{(q)} \tilde{\mathbf{x}}_t^{(q)}(\tilde{\mathbf{x}}_t^{(q)})^{\top}\right] \\
\succeq & \mathbb{E}\left[(\tilde{\mathbf{x}}_t^{(q)})^{\top}\left(\mathbf{I}-(\mathbf{I}-\gamma \mathbf{H}_f^{(q)})^{t-1}\right) \mathbf{v}^{(q)*}\left(\mathbf{v}^{(q)*}\right)^{\top}\left(\mathbf{I}-(\mathbf{I}-\gamma \mathbf{H}_f^{(q)})^{t-1}\right) \tilde{\mathbf{x}}_t^{(q)} \tilde{\mathbf{x}}_t^{(q)}(\tilde{\mathbf{x}}_t^{(q)})^{\top}\right] \\
& +\frac{\gamma \beta}{2} \mathbb{E}\left[(\tilde{\mathbf{x}}_t^{(q)})^{\top}(\mathbf{I}-\gamma \mathbf{H}_f^{(q)})^{2(t-1)} \mathbf{H}_f^{(q)}\left\|\mathbf{v}^{(q)*}\right\|_{\mathbf{I}-(\mathbf{I}-\gamma \mathbf{H}_f^{(q)})^{2(t-1)}}^2 \tilde{\mathbf{x}}_t^{(q)} \tilde{\mathbf{x}}_t^{(q)}(\tilde{\mathbf{x}}_t^{(q)})^{\top}\right] \\
\succeq & \beta \operatorname{tr}\left(\mathbf{H}_f^{(q)}\left[\mathbf{I}-(\mathbf{I}-\gamma \mathbf{H}_f^{(q)})^{t-1}\right] \mathbf{v}^{(q)*}\left(\mathbf{v}^{(q)*}\right)^{\top}\left[\mathbf{I}-(\mathbf{I}-\gamma \mathbf{H}_f^{(q)})^{t-1}\right]\right) \mathbf{H}_f^{(q)} \\
& +\frac{\gamma \beta^2}{2} \operatorname{tr}\left(\mathbf{H}_f^{(q)}(\mathbf{I}-\gamma \mathbf{H}_f^{(q)})^{2(t-1)} \mathbf{H}_f^{(q)}\right)\left\|\mathbf{v}^{(q)*}\right\|_{\mathbf{I}-(\mathbf{I}-\gamma \mathbf{H}_f^{(q)})^{2(t-1)}}^2 \mathbf{H}_f^{(q)} .
\end{aligned}
$$

Therefore,

$$
\begin{aligned}
\frac{\boldsymbol{\Sigma}_t}{\gamma^2(1+\underline{\epsilon}_o)} \succeq & \underline{\sigma}^2 \mathbf{H}_f^{(q)}+\left[\underline{\epsilon}_p+(1+\underline{\epsilon}_p) \underline{\epsilon}_a\right] \beta \operatorname{tr}\left(\mathbf{H}_f^{(q)}\left[\mathbf{I}-(\mathbf{I}-\gamma \mathbf{H}_f^{(q)})^{(t-1)}\right]^2 \mathbf{v}^{(q)*}\left(\mathbf{v}^{(q)*}\right)^{\top}\right) \mathbf{H}_f^{(q)} \\
& +\left[\underline{\epsilon}_p+(1+\underline{\epsilon}_p) \underline{\epsilon}_a\right] \frac{\gamma \beta^2}{2} \operatorname{tr}\left(\mathbf{H}_f^{(q)}(\mathbf{I}-\gamma \mathbf{H}_f^{(q)})^{2(t-1)} \mathbf{H}_f^{(q)}\right)\left\|\mathbf{v}^{(q)*}\right\|_{\mathbf{I}-(\mathbf{I}-\gamma \mathbf{H}_f^{(q)})^{2(t-1)}}^2 \mathbf{H}_f^{(q)} .
\end{aligned}
$$

Recall (56),
$$
\mathbb{E}[\boldsymbol{\eta}_t \otimes \boldsymbol{\eta}_t]=\mathbb{E}\left[\left(\mathbf{I}-\gamma \tilde{\mathbf{x}}_t^{(q)}(\tilde{\mathbf{x}}_t^{(q)})^{\top}\right) \mathbb{E}[\boldsymbol{\eta}_{t-1} \otimes \boldsymbol{\eta}_{t-1}]\left(\mathbf{I}-\gamma \tilde{\mathbf{x}}_t^{(q)}(\tilde{\mathbf{x}}_t^{(q)})^{\top}\right)\right]+\boldsymbol{\Sigma}_t,
$$

we have

$$
\begin{aligned}
& \mathbb{E}[\boldsymbol{\eta}_t \otimes \boldsymbol{\eta}_t] \\
\succeq & \mathbb{E}\left[\left(\mathbf{I}-\gamma \tilde{\mathbf{x}}_t^{(q)}(\tilde{\mathbf{x}}_t^{(q)})^{\top}\right) \mathbb{E}[\boldsymbol{\eta}_{t-1} \otimes \boldsymbol{\eta}_{t-1}]\left(\mathbf{I}-\gamma \tilde{\mathbf{x}}_t^{(q)}(\tilde{\mathbf{x}}_t^{(q)})^{\top}\right)\right]+\gamma^2(1+\underline{\epsilon}_o) \underline{\sigma}^2 \mathbf{H}_f^{(q)} \\
& +\gamma^2(1+\underline{\epsilon}_o)\left[\underline{\epsilon}_p+(1+\underline{\epsilon}_p) \underline{\epsilon}_a\right] \beta \operatorname{tr}\left(\mathbf{H}_f^{(q)}\left[\mathbf{I}-(\mathbf{I}-\gamma \mathbf{H}_f^{(q)})^{(t-1)}\right]^2 \mathbf{v}^{(q)*}\left(\mathbf{v}^{(q)*}\right)^{\top}\right) \mathbf{H}_f^{(q)} \\
& +\gamma^2(1+\underline{\epsilon}_o)\left[\underline{\epsilon}_p+(1+\underline{\epsilon}_p) \underline{\epsilon}_a\right] \frac{\gamma \beta^2}{2} \operatorname{tr}\left(\mathbf{H}_f^{(q)}(\mathbf{I}-\gamma \mathbf{H}_f^{(q)})^{2(t-1)} \mathbf{H}_f^{(q)}\right)\left\|\mathbf{v}^{(q)*}\right\|_{\mathbf{I}-(\mathbf{I}-\gamma \mathbf{H}_f^{(q)})^{2(t-1)}}^2 \mathbf{H}_f^{(q)} .
\end{aligned}
$$

$\square$

### D.2. Bias-Variance Decomposition

Noticing that

$$
\begin{aligned}
R_N =& \frac{1}{2} \left\langle \mathbf{S}\mathbf{H}\mathbf{S}^\top, \mathbb{E}\left[\overline{\boldsymbol{\eta}}_N \otimes \overline{\boldsymbol{\eta}}_N\right]\right\rangle \\
\geq& \mu_{\min}\left((\mathbf{H}_f^{(q)})^{-1}\mathbf{S}\mathbf{H}\mathbf{S}^\top\right)\underbrace{\frac{1}{2}\left\langle \mathbf{H}_f^{(q)}, \mathbb{E}\left[\overline{\boldsymbol{\eta}}_N \otimes \overline{\boldsymbol{\eta}}_N\right]\right\rangle}_{R_N^{(0)}},
\end{aligned}
\tag{64}
$$

We then perform bias-variance decomposition for $R_N^{(0)}$.

**Lemma D.3.** *Under Assumption 3.1, Assumption 3.3, it holds*

$$
R_N^{(0)} \geq \frac{1}{2N^2} \cdot \sum_{t=0}^{N-1}\sum_{k=t}^{N-1} \left\langle (\mathbf{I} - \gamma\mathbf{H}_f^{(q)})^{k-t}\mathbf{H}_f^{(q)}, \mathbb{E}[\boldsymbol{\eta}_t \otimes \boldsymbol{\eta}_t]\right\rangle.
$$

*Proof.* By definition $\overline{\boldsymbol{\eta}}_N = \frac{1}{N}\sum_{t=0}^{N-1}\boldsymbol{\eta}_t$, we have

$$
\begin{aligned}
\mathbb{E}[\overline{\boldsymbol{\eta}}_N \otimes \overline{\boldsymbol{\eta}}_N] =& \frac{1}{N^2} \cdot \left( \sum_{0 \leq k \leq t \leq N-1} \mathbb{E}[\boldsymbol{\eta}_t \otimes \boldsymbol{\eta}_k] + \sum_{0 \leq t < k \leq N-1} \mathbb{E}[\boldsymbol{\eta}_t \otimes \boldsymbol{\eta}_k]\right) \\
=& \frac{1}{N^2} \cdot \left( \sum_{0 \leq k \leq t \leq N-1} \mathbb{E}\left[\mathbb{E}[\boldsymbol{\eta}_t \otimes \boldsymbol{\eta}_k|\boldsymbol{\eta}_k]\right] + \sum_{0 \leq t < k \leq N-1} \mathbb{E}\left[\mathbb{E}[\boldsymbol{\eta}_t \otimes \boldsymbol{\eta}_k|\boldsymbol{\eta}_t]\right]\right).
\end{aligned}
\tag{65}
$$

By

$$
\mathbb{E}\left[\boldsymbol{\eta}_t|\boldsymbol{\eta}_{t-1}\right] = \left(\mathbf{I} - \gamma\mathbf{H}_f^{(q)}\right)\boldsymbol{\eta}_{t-1},
$$

we have

$$
\begin{aligned}
&\mathbb{E}[\overline{\boldsymbol{\eta}}_N \otimes \overline{\boldsymbol{\eta}}_N] \\
=& \frac{1}{N^2} \cdot \left( \sum_{0 \leq k \leq t \leq N-1} \mathbb{E}\left[\mathbb{E}[\boldsymbol{\eta}_t \otimes \boldsymbol{\eta}_k|\boldsymbol{\eta}_k]\right] + \sum_{0 \leq t < k \leq N-1} \mathbb{E}\left[\mathbb{E}[\boldsymbol{\eta}_t \otimes \boldsymbol{\eta}_k|\boldsymbol{\eta}_t]\right]\right) \\
=& \frac{1}{N^2} \cdot \left( \sum_{0 \leq k \leq t \leq N-1} (\mathbf{I} - \gamma\mathbf{H}_f^{(q)})^{t-k}\mathbb{E}[\boldsymbol{\eta}_k \otimes \boldsymbol{\eta}_k] + \sum_{0 \leq t < k \leq N-1} \mathbb{E}[\boldsymbol{\eta}_t \otimes \boldsymbol{\eta}_t](\mathbf{I} - \gamma\mathbf{H}_f^{(q)})^{k-t}\right) \\
\succeq& \frac{1}{N^2} \cdot \sum_{t=0}^{N-1}\sum_{k=t}^{N-1}(\mathbf{I} - \gamma\mathbf{H}_f^{(q)})^{k-t}\mathbb{E}[\boldsymbol{\eta}_t \otimes \boldsymbol{\eta}_t].
\end{aligned}
\tag{66}
$$

Applying into $R_N^{(0)}$, we have

$$
\begin{aligned}
R_N^{(0)} =& \frac{1}{2}\langle\mathbf{H}_f^{(q)}, \mathbb{E}[\overline{\boldsymbol{\eta}}_N \otimes \overline{\boldsymbol{\eta}}_N]\rangle \\
\geq& \frac{1}{2N^2} \cdot \sum_{t=0}^{N-1}\sum_{k=t}^{N-1}\left\langle\mathbf{H}_f^{(q)}, (\mathbf{I} - \gamma\mathbf{H}_f^{(q)})^{k-t}\mathbb{E}[\boldsymbol{\eta}_t \otimes \boldsymbol{\eta}_t]\right\rangle \\
=& \frac{1}{2N^2} \cdot \sum_{t=0}^{N-1}\sum_{k=t}^{N-1}\left\langle(\mathbf{I} - \gamma\mathbf{H}_f^{(q)})^{k-t}\mathbf{H}_f^{(q)}, \mathbb{E}[\boldsymbol{\eta}_t \otimes \boldsymbol{\eta}_t]\right\rangle.
\end{aligned}
$$

$\square$

**Lemma D.4** (Bias-variance decomposition under general quantization, a lower bound). *Under Assumption 3.1, 3.3, 3.4 and 3.5, if the stepsize $\gamma < \frac{1}{\overline{\lambda}_1^{(q)}}$, it holds*

$$
R_N^{(0)} \geq \frac{1}{2N^2} \cdot \sum_{t=0}^{N-1}\sum_{k=t}^{N-1} \left\langle (\mathbf{I} - \gamma\mathbf{H}_f^{(q)})^{k-t}\mathbf{H}_f^{(q)}, \underline{\mathbf{B}}_t + \underline{\mathbf{C}}_t\right\rangle,
$$

*where*

$$\underline{\mathbf{B}}_t = (\mathcal{I} - \gamma\mathcal{T}^{(q)}) \circ \underline{\mathbf{B}}_{t-1}, \quad \underline{\mathbf{B}}_0 = \mathbb{E}\left[\boldsymbol{\eta}_0 \otimes \boldsymbol{\eta}_0\right],$$
$$\underline{\mathbf{C}}_t = (\mathcal{I} - \gamma\mathcal{T}^{(q)}) \circ \underline{\mathbf{C}}_{t-1} + \gamma^2 \underline{\sigma}_G^2 \mathbf{H}_f^{(q)}, \quad \underline{\mathbf{C}}_0 = \mathbf{0},$$

*with*

$$\underline{\sigma}_G^2 = \underline{\sigma}^2 + \inf_t \beta \mathrm{tr}\left(\mathbf{H}_f^{(q)} \mathbb{E}\left[\boldsymbol{\epsilon}_{t-1}^{(p)} \boldsymbol{\epsilon}_{t-1}^{(p)}{}^\top\right]\right) + \inf_t \left(\mathbb{E}\left[\epsilon_t^{(a)^2}\Big|a_t\right] + \mathbb{E}\left[\epsilon_t^{(o)^2}\Big|o_t\right]\right).$$

*Proof.* The proof is completed by Lemma D.1 and Lemma D.3. $\qquad\square$

**Lemma D.5** (Bias-variance decomposition under multiplicative quantization, a lower bound). *If there exist $\underline{\epsilon}_p, \underline{\epsilon}_a$ and $\underline{\epsilon}_o$ such that for any $i \in \{p, a, o\}$, quantization $\mathcal{Q}_i$ is $\underline{\epsilon}_i$-multiplicative, then under Assumption 3.1, 3.3, 3.4 and 3.5, if the stepsize $\gamma < \frac{1}{\widetilde{\lambda}_1^{(q)}}$, it holds*

$$R_N^{(0)} \geq \frac{1}{2N^2} \cdot \sum_{t=0}^{N-1}\sum_{k=t}^{N-1} \left\langle (\mathbf{I} - \gamma\mathbf{H}_f^{(q)})^{k-t}\mathbf{H}_f^{(q)}, \underline{\mathbf{B}}_t^{(M)} + \underline{\mathbf{C}}_t^{(M)}\right\rangle,$$

*where*

$$\underline{\mathbf{B}}_t^{(M)} = (\mathcal{I} - \gamma\mathcal{T}^{(q)}) \circ \underline{\mathbf{B}}_{t-1}^{(M)}, \quad \underline{\mathbf{B}}_0^{(M)} = \mathbb{E}\left[\boldsymbol{\eta}_0 \otimes \boldsymbol{\eta}_0\right],$$
$$\underline{\mathbf{C}}_t^{(M)} \succeq (\mathcal{I} - \gamma\mathcal{T}^{(q)}) \circ \underline{\mathbf{C}}_{t-1}^{(M)} + \gamma^2(1 + \underline{\epsilon}_o)\left[\underline{\sigma}^2 \mathbf{H}_f^{(q)} + \left(\underline{\epsilon}_p + (1 + \underline{\epsilon}_p)\underline{\epsilon}_a\right)\beta\left(\mathbf{R}_{t-1} + \mathbf{T}_{t-1}\right)\right], \quad \underline{\mathbf{C}}_0^{(M)} = \mathbf{0},$$

*with*

$$\mathbf{R}_t := \mathrm{tr}\left(\mathbf{H}_f^{(q)}\left[\mathbf{I} - (\mathbf{I} - \gamma\mathbf{H}_f^{(q)})^t\right]^2 \mathbf{v}^{(q)*}\left(\mathbf{v}^{(q)*}\right)^\top\right)\mathbf{H}_f^{(q)},$$
$$\mathbf{T}_t := \frac{\gamma\beta}{2}\mathrm{tr}\left(\mathbf{H}_f^{(q)}(\mathbf{I} - \gamma\mathbf{H}_f^{(q)})^{2t}\mathbf{H}_f^{(q)}\right)\left\|\mathbf{v}^{(q)*}\right\|_{\mathbf{I}-(\mathbf{I}-\gamma\mathbf{H}_f^{(q)})^{2t}}^2\mathbf{H}_f^{(q)}.$$

*Proof.* The proof is completed by Lemma D.2 and Lemma D.3. $\qquad\square$

## D.3. Variance Lower Bounds

In this section, we derive lower bounds for $\frac{1}{2N^2}\sum_{t=0}^{N-1}\sum_{k=t}^{N-1}\left\langle(\mathbf{I} - \gamma\mathbf{H}_f^{(q)})^{k-t}\mathbf{H}_f^{(q)}, \underline{\mathbf{C}}_t^{(M)}\right\rangle$ and $\frac{1}{2N^2}\sum_{t=0}^{N-1}\sum_{k=t}^{N-1}\left\langle(\mathbf{I} - \gamma\mathbf{H}_f^{(q)})^{k-t}\mathbf{H}_f^{(q)}, \underline{\mathbf{C}}_t\right\rangle$.

### D.3.1. GENERAL QUANTIZATION

**Lemma D.6** (A crude lower bound of variance under general quantization). *If the stepsize $\gamma < \frac{1}{\widetilde{\lambda}_1^{(q)}}$, under Assumption 3.3, it holds*

$$\underline{\mathbf{C}}_t \succeq \frac{\gamma\underline{\sigma}_G^2}{2}\left(\mathbf{I} - \left(\mathbf{I} - \gamma\mathbf{H}_f^{(q)}\right)^{2t}\right).$$

*Proof.* Note that $\mathcal{M}^{(q)} - \widetilde{\mathcal{M}}^{(q)}$ is a PSD mapping and $\underline{\mathbf{C}}_{t-1}$ is PSD, then by definition

$$\begin{aligned}\underline{\mathbf{C}}_t &= (\mathcal{I} - \gamma\mathcal{T}^{(q)}) \circ \underline{\mathbf{C}}_{t-1} + \gamma^2\underline{\sigma}_G^2\mathbf{H}_f^{(q)}\\&= (\mathcal{I} - \gamma\widetilde{\mathcal{T}}^{(q)}) \circ \underline{\mathbf{C}}_{t-1} + \gamma(\widetilde{\mathcal{T}}^{(q)} - \mathcal{T}^{(q)}) \circ \underline{\mathbf{C}}_{t-1} + \gamma^2\underline{\sigma}_G^2\mathbf{H}_f^{(q)}\\&= (\mathcal{I} - \gamma\widetilde{\mathcal{T}}^{(q)}) \circ \underline{\mathbf{C}}_{t-1} + \gamma^2(\mathcal{M}^{(q)} - \widetilde{\mathcal{M}}^{(q)}) \circ \underline{\mathbf{C}}_{t-1} + \gamma^2\underline{\sigma}_G^2\mathbf{H}_f^{(q)}\\&\succeq (\mathcal{I} - \gamma\widetilde{\mathcal{T}}^{(q)}) \circ \underline{\mathbf{C}}_{t-1} + \gamma^2\underline{\sigma}_G^2\mathbf{H}_f^{(q)}.\end{aligned}$$

By solving recursion, it holds

$$
\begin{aligned}
\underline{\mathbf{C}}_t &\succeq \gamma^2 \underline{\sigma}_G^2 \sum_{k=0}^{t-1} (\mathcal{I} - \gamma \widetilde{\mathcal{T}}^{(q)})^k \circ \mathbf{H}_f^{(q)} \\
&= \gamma^2 \underline{\sigma}_G^2 \sum_{k=0}^{t-1} (\mathbf{I} - \gamma \mathbf{H}_f^{(q)})^k \mathbf{H}_f^{(q)} (\mathbf{I} - \gamma \mathbf{H}_f^{(q)})^k \\
&= \gamma^2 \underline{\sigma}_G^2 \left( \mathbf{I} - \left( \mathbf{I} - \gamma \mathbf{H}_f^{(q)} \right)^{2t} \right) \left( 2\gamma \mathbf{I} - \gamma^2 \mathbf{H}_f^{(q)} \right)^{-1} \\
&\succeq \frac{\gamma \underline{\sigma}_G^2}{2} \left( \mathbf{I} - \left( \mathbf{I} - \gamma \mathbf{H}_f^{(q)} \right)^{2t} \right).
\end{aligned}
$$

$\square$

**Lemma D.7** (A variance lower bound under general quantization). *Suppose the stepsize $\gamma < \frac{1}{\tilde{\lambda}_1^{(q)}}$, then under Assumption 3.3, for sufficiently large $N > 500$,*

$$
\frac{1}{2N^2} \cdot \sum_{t=0}^{N-1} \sum_{k=t}^{N-1} \left\langle (\mathbf{I} - \gamma \mathbf{H}_f^{(q)})^{k-t} \mathbf{H}_f^{(q)}, \underline{\mathbf{C}}_t \right\rangle \geq \frac{\underline{\sigma}_G^2}{50} \left( \frac{k^*}{N} + N\gamma^2 \sum_{i > k^*} \left( \tilde{\lambda}_i^{(q)} \right)^2 \right),
$$

*where $k^* := \max\{k : \tilde{\lambda}_k^{(q)} \geq \frac{1}{\gamma N}\}$.*

*Proof.* By Lemma D.6,

$$
\begin{aligned}
&\frac{1}{2N^2} \sum_{t=0}^{N-1} \sum_{k=t}^{N-1} \left\langle (\mathbf{I} - \gamma \mathbf{H}_f^{(q)})^{k-t} \mathbf{H}_f^{(q)}, \underline{\mathbf{C}}_t \right\rangle \\
&\geq \frac{\gamma \underline{\sigma}_G^2}{4N^2} \sum_{t=0}^{N-1} \sum_{k=t}^{N-1} \left\langle (\mathbf{I} - \gamma \mathbf{H}_f^{(q)})^{k-t} \mathbf{H}_f^{(q)}, \mathbf{I} - (\mathbf{I} - \gamma \mathbf{H}_f^{(q)})^{2t} \right\rangle \\
&= \frac{\underline{\sigma}_G^2}{4N^2} \sum_{t=0}^{N-1} \left\langle (\mathbf{I} - (\mathbf{I} - \gamma \mathbf{H}_f^{(q)})^{N-t}), \mathbf{I} - (\mathbf{I} - \gamma \mathbf{H}_f^{(q)})^{2t} \right\rangle \\
&= \frac{\underline{\sigma}_G^2}{4N^2} \sum_{i} \sum_{t=0}^{N-1} (1 - (1 - \gamma \tilde{\lambda}_i^{(q)})^{N-t})(1 - (1 - \gamma \tilde{\lambda}_i^{(q)})^{2t}) \\
&\geq \frac{\underline{\sigma}_G^2}{4N^2} \sum_{i} \sum_{t=0}^{N-1} (1 - (1 - \gamma \tilde{\lambda}_i^{(q)})^{N-t-1})(1 - (1 - \gamma \tilde{\lambda}_i^{(q)})^{t}).
\end{aligned}
$$

Define

$$
f(x) = \sum_{t=0}^{N-1} (1 - (1 - x)^{N-t-1})(1 - (1 - x)^{t}), \quad 0 < x < 1.
$$

Note that if $N \geq 500$,

$$
f(x) \geq \begin{cases} \frac{N}{10}, & \frac{1}{N} \leq x < 1, \\ \frac{2N^3}{25} x^2, & 0 < x < \frac{1}{N}, \end{cases}
$$

it follows that

$$
\begin{aligned}
\frac{1}{2N^2} \sum_{t=0}^{N-1} \sum_{k=t}^{N-1} \left\langle (\mathbf{I} - \gamma \mathbf{H}_f^{(q)})^{k-t} \mathbf{H}_f^{(q)}, \underline{\mathbf{C}}_t \right\rangle &\geq \frac{\underline{\sigma}_G^2}{4N^2} \sum_{i} f(\gamma \tilde{\lambda}_i^{(q)}) \\
&\geq \frac{\underline{\sigma}_G^2}{50} \left( \frac{k^*}{N} + N\gamma^2 \sum_{i > k^*} \left( \tilde{\lambda}_i^{(q)} \right)^2 \right).
\end{aligned}
$$

$\square$

### D.3.2. MULTIPLICATIVE QUANTIZATION

**Lemma D.8** (A crude lower bound of variance under multiplicative quantization)**.** *If the stepsize* $\gamma < \frac{1}{\tilde{\lambda}_1^{(q)}}$*, then*

$$
\begin{aligned}
\underline{\mathbf{C}}_t^{(M)} \succeq &\frac{\gamma}{2}(1 + \underline{\epsilon}_o)\underline{\sigma}^2 \left(\mathbf{I} - (\mathbf{I} - \gamma\mathbf{H}_f^{(q)})^t\right) \\
&+ \frac{\gamma}{2}(1 + \underline{\epsilon}_o)\beta\left(\underline{\epsilon}_p + (1 + \underline{\epsilon}_p)\underline{\epsilon}_a\right) \operatorname{tr}\left(\mathbf{H}_f^{(q)}\left[\mathbf{I} - (\mathbf{I} - \gamma\mathbf{H}_f^{(q)})^{t/2}\right]^2 \mathbf{v}^{(q)*}\left(\mathbf{v}^{(q)*}\right)^{\top}\right)\left(\mathbf{I} - (\mathbf{I} - \gamma\mathbf{H}_f^{(q)})^t\right) \\
&+ \frac{\gamma^2\beta}{4}(1 + \underline{\epsilon}_o)\beta\left(\underline{\epsilon}_p + (1 + \underline{\epsilon}_p)\underline{\epsilon}_a\right)\operatorname{tr}\left(\mathbf{H}_f^{(q)}(\mathbf{I} - \gamma\mathbf{H}_f^{(q)})^{2t}\mathbf{H}_f^{(q)}\right)\left\|\mathbf{v}^{(q)*}\right\|_{\mathbf{I} - (\mathbf{I} - \gamma\mathbf{H}_f^{(q)})^t}^2\left(\mathbf{I} - (\mathbf{I} - \gamma\mathbf{H}_f^{(q)})^t\right).
\end{aligned}
$$

*Proof.* Similar to the general quantization case,

$$
\underline{\mathbf{C}}_t^{(M)} \succeq (\mathcal{I} - \gamma\widetilde{\mathcal{T}}^{(q)}) \circ \underline{\mathbf{C}}_{t-1}^{(M)} + \gamma^2(1 + \underline{\epsilon}_o)\left[\underline{\sigma}^2\mathbf{H}_f^{(q)} + \left(\underline{\epsilon}_p + (1 + \underline{\epsilon}_p)\underline{\epsilon}_a\right)\beta\left(\mathbf{R}_{t-1} + \mathbf{T}_{t-1}\right)\right].
$$

By solving recursion, it holds

$$
\begin{aligned}
\underline{\mathbf{C}}_t^{(M)} &\succeq \sum_{k=0}^{t-1}(\mathcal{I} - \gamma\widetilde{\mathcal{T}}^{(q)})^k \circ \gamma^2(1 + \underline{\epsilon}_o)\left[\underline{\sigma}^2\mathbf{H}_f^{(q)} + \left(\underline{\epsilon}_p + (1 + \underline{\epsilon}_p)\underline{\epsilon}_a\right)\beta\left(\mathbf{R}_{t-1-k} + \mathbf{T}_{t-1-k}\right)\right] \\
&= \gamma^2(1 + \underline{\epsilon}_o)\sum_{k=0}^{t-1}(\mathbf{I} - \gamma\mathbf{H}_f^{(q)})^{2k}\left[\underline{\sigma}^2\mathbf{H}_f^{(q)} + \left(\underline{\epsilon}_p + (1 + \underline{\epsilon}_p)\underline{\epsilon}_a\right)\beta\left(\mathbf{R}_{t-1-k} + \mathbf{T}_{t-1-k}\right)\right].
\end{aligned}
$$

Regarding the time-independent noise $\underline{\sigma}^2$,

$$
\gamma^2\underline{\sigma}^2(1 + \underline{\epsilon}_o)\sum_{k=0}^{t-1}(\mathbf{I} - \gamma\mathbf{H}_f^{(q)})^{2k}\mathbf{H}_f^{(q)} \succeq \frac{\gamma\underline{\sigma}^2(1 + \underline{\epsilon}_o)}{2}\left(\mathbf{I} - \left(\mathbf{I} - \gamma\mathbf{H}_f^{(q)}\right)^{2t}\right).
$$

Regarding the time-dependent term $\mathbf{R}_t$,

$$
\begin{aligned}
&\sum_{k=0}^{t-1}(\mathbf{I} - \gamma\mathbf{H}_f^{(q)})^{2k}\mathbf{R}_{t-1-k} \\
&= \sum_{k=0}^{t-1}(\mathbf{I} - \gamma\mathbf{H}_f^{(q)})^{2k}\mathbf{H}_f^{(q)}\operatorname{tr}\left(\mathbf{H}_f^{(q)}\left[\mathbf{I} - (\mathbf{I} - \gamma\mathbf{H}_f^{(q)})^{t-1-k}\right]^2\mathbf{v}^{(q)*}\left(\mathbf{v}^{(q)*}\right)^{\top}\right) \\
&\succeq \sum_{k=0}^{t/2-1}(\mathbf{I} - \gamma\mathbf{H}_f^{(q)})^{2k}\mathbf{H}_f^{(q)}\operatorname{tr}\left(\mathbf{H}_f^{(q)}\left[\mathbf{I} - (\mathbf{I} - \gamma\mathbf{H}_f^{(q)})^{t-1-k}\right]^2\mathbf{v}^{(q)*}\left(\mathbf{v}^{(q)*}\right)^{\top}\right) \\
&\succeq \sum_{k=0}^{t/2-1}(\mathbf{I} - \gamma\mathbf{H}_f^{(q)})^{2k}\mathbf{H}_f^{(q)}\operatorname{tr}\left(\mathbf{H}_f^{(q)}\left[\mathbf{I} - (\mathbf{I} - \gamma\mathbf{H}_f^{(q)})^{t/2}\right]^2\mathbf{v}^{(q)*}\left(\mathbf{v}^{(q)*}\right)^{\top}\right) \\
&\succeq \frac{1}{2\gamma}\left[\mathbf{I} - (\mathbf{I} - \gamma\mathbf{H}_f^{(q)})^t\right]\operatorname{tr}\left(\mathbf{H}_f^{(q)}\left[\mathbf{I} - (\mathbf{I} - \gamma\mathbf{H}_f^{(q)})^{t/2}\right]^2\mathbf{v}^{(q)*}\left(\mathbf{v}^{(q)*}\right)^{\top}\right).
\end{aligned}
$$

Regarding the time-dependent term $\mathbf{T}_t$,

$$\sum_{k=0}^{t-1}(\mathbf{I}-\gamma\mathbf{H}_f^{(q)})^{2k}\mathbf{T}_{t-1-k}$$

$$=\frac{\gamma\beta}{2}\sum_{k=0}^{t-1}(\mathbf{I}-\gamma\mathbf{H}_f^{(q)})^{2k}\mathrm{tr}\left(\mathbf{H}_f^{(q)}(\mathbf{I}-\gamma\mathbf{H}_f^{(q)})^{2(t-1-k)}\mathbf{H}_f^{(q)}\right)\left\|\mathbf{v}^{(q)*}\right\|_{\mathbf{I}-(\mathbf{I}-\gamma\mathbf{H}_f^{(q)})^{2(t-1-k)}}^2\mathbf{H}_f^{(q)}$$

$$\succeq\frac{\gamma\beta}{2}\sum_{k=0}^{t/2-1}(\mathbf{I}-\gamma\mathbf{H}_f^{(q)})^{2k}\mathrm{tr}\left(\mathbf{H}_f^{(q)}(\mathbf{I}-\gamma\mathbf{H}_f^{(q)})^{2(t-1-k)}\mathbf{H}_f^{(q)}\right)\left\|\mathbf{v}^{(q)*}\right\|_{\mathbf{I}-(\mathbf{I}-\gamma\mathbf{H}_f^{(q)})^{2(t-1-k)}}^2\mathbf{H}_f^{(q)}$$

$$\succeq\frac{\gamma\beta}{2}\sum_{k=0}^{t/2-1}(\mathbf{I}-\gamma\mathbf{H}_f^{(q)})^{2k}\mathrm{tr}\left(\mathbf{H}_f^{(q)}(\mathbf{I}-\gamma\mathbf{H}_f^{(q)})^{2t}\mathbf{H}_f^{(q)}\right)\left\|\mathbf{v}^{(q)*}\right\|_{\mathbf{I}-(\mathbf{I}-\gamma\mathbf{H}_f^{(q)})^t}^2\mathbf{H}_f^{(q)}$$

$$\succeq\frac{\beta}{4}\left(\mathbf{I}-(\mathbf{I}-\gamma\mathbf{H}_f^{(q)})^t\right)\mathrm{tr}\left(\mathbf{H}_f^{(q)}(\mathbf{I}-\gamma\mathbf{H}_f^{(q)})^{2t}\mathbf{H}_f^{(q)}\right)\left\|\mathbf{v}^{(q)*}\right\|_{\mathbf{I}-(\mathbf{I}-\gamma\mathbf{H}_f^{(q)})^t}^2.$$

Therefore,

$$\underline{\mathbf{C}}_t^{(M)}\succeq\frac{\gamma}{2}(1+\underline{\epsilon}_o)\underline{\sigma}^2\left(\mathbf{I}-(\mathbf{I}-\gamma\mathbf{H}_f^{(q)})^t\right)$$

$$+\frac{\gamma}{2}(1+\underline{\epsilon}_o)\beta\left(\underline{\epsilon}_p+(1+\underline{\epsilon}_p)\underline{\epsilon}_a\right)\mathrm{tr}\left(\mathbf{H}_f^{(q)}\left[\mathbf{I}-(\mathbf{I}-\gamma\mathbf{H}_f^{(q)})^{t/2}\right]^2\mathbf{v}^{(q)*}\left(\mathbf{v}^{(q)*}\right)^\top\right)\left(\mathbf{I}-(\mathbf{I}-\gamma\mathbf{H}_f^{(q)})^t\right)$$

$$+\frac{\gamma^2\beta}{4}(1+\underline{\epsilon}_o)\beta\left(\underline{\epsilon}_p+(1+\underline{\epsilon}_p)\underline{\epsilon}_a\right)\mathrm{tr}\left(\mathbf{H}_f^{(q)}(\mathbf{I}-\gamma\mathbf{H}_f^{(q)})^{2t}\mathbf{H}_f^{(q)}\right)\left\|\mathbf{v}^{(q)*}\right\|_{\mathbf{I}-(\mathbf{I}-\gamma\mathbf{H}_f^{(q)})^t}^2\left(\mathbf{I}-(\mathbf{I}-\gamma\mathbf{H}_f^{(q)})^t\right).$$

$$\square$$

**Lemma D.9** (A variance lower bound under multiplicative quantization). *For $i\in\{d,f,s,p,a,o\}$, if there exist constants $(\bar{\epsilon}_i,\underline{\epsilon}_i)$ such that $\mathcal{Q}_i$ is $(\bar{\epsilon}_i,\underline{\epsilon}_i)$-multiplicative, suppose the stepsize $\gamma<\frac{1}{\tilde{\lambda}_1^{(q)}}$, then under Assumption 3.3, for sufficiently large $N>500$,*

$$\frac{1}{2N^2}\sum_{t=0}^{N-1}\sum_{k=t}^{N-1}\left\langle(\mathbf{I}-\gamma\mathbf{H}_f^{(q)})^{k-t}\mathbf{H}_f^{(q)},\underline{\mathbf{C}}_t^{(M)}\right\rangle$$

$$\geq\left[\frac{(1+\underline{\epsilon}_o)\underline{\sigma}^2}{50}+\frac{\beta(1+\underline{\epsilon}_o)(\underline{\epsilon}_p+(1+\underline{\epsilon}_p)\underline{\epsilon}_a)}{2500}\left(\left\|\mathbf{v}^{(q)*}\right\|_{\mathbf{H}_{f,0:k^*}^{(q)}}^2+N^2\gamma^2\left\|\mathbf{v}^{(q)*}\right\|_{(\mathbf{H}_{f,k^*:\infty}^{(q)})^3}^2\right)\right]\frac{d_{\mathrm{eff}}}{N}$$

$$+\frac{\gamma(1+\underline{\epsilon}_o)\beta^2\left(\underline{\epsilon}_p+(1+\underline{\epsilon}_p)\underline{\epsilon}_a\right)}{600}\left(\left\|\mathbf{v}^{(q)*}\right\|_{\mathbf{I}_{f,0:k^*}^{(q)}}^2+N\gamma\left\|\mathbf{v}^{(q)*}\right\|_{\mathbf{H}_{f,k^*:\infty}^{(q)}}^2\right)\cdot\sum_i(\tilde{\lambda}_i^{(q)})^2(1-\gamma\tilde{\lambda}_i^{(q)})^{2N}\frac{d_{\mathrm{eff}}}{N},$$

*where $\frac{d_{\mathrm{eff}}}{N}=\frac{k^*}{N}+N\gamma^2\sum_{i>k^*}\left(\tilde{\lambda}_i^{(q)}\right)^2$ and $k^*:=\max\{k:\tilde{\lambda}_k^{(q)}\geq\frac{1}{\gamma N}\}$.*

*Proof.* Recall Lemma D.8,

$$\underline{\mathbf{C}}_t^{(M)}\succeq\underbrace{\frac{\gamma}{2}(1+\underline{\epsilon}_o)\underline{\sigma}^2\left(\mathbf{I}-(\mathbf{I}-\gamma\mathbf{H}_f^{(q)})^t\right)}_{\mathbf{r}_1}$$

$$+\underbrace{\frac{\gamma}{2}(1+\underline{\epsilon}_o)\beta\left(\underline{\epsilon}_p+(1+\underline{\epsilon}_p)\underline{\epsilon}_a\right)\mathrm{tr}\left(\mathbf{H}_f^{(q)}\left[\mathbf{I}-(\mathbf{I}-\gamma\mathbf{H}_f^{(q)})^{t/2}\right]^2\mathbf{v}^{(q)*}\left(\mathbf{v}^{(q)*}\right)^\top\right)\left(\mathbf{I}-(\mathbf{I}-\gamma\mathbf{H}_f^{(q)})^t\right)}_{\mathbf{r}_2}$$

$$+\underbrace{\frac{\gamma^2\beta}{4}(1+\underline{\epsilon}_o)\beta\left(\underline{\epsilon}_p+(1+\underline{\epsilon}_p)\underline{\epsilon}_a\right)\mathrm{tr}\left(\mathbf{H}_f^{(q)}(\mathbf{I}-\gamma\mathbf{H}_f^{(q)})^{2t}\mathbf{H}_f^{(q)}\right)\left\|\mathbf{v}^{(q)*}\right\|_{\mathbf{I}-(\mathbf{I}-\gamma\mathbf{H}_f^{(q)})^t}^2\left(\mathbf{I}-(\mathbf{I}-\gamma\mathbf{H}_f^{(q)})^t\right)}_{\mathbf{r}_3}.$$

Regarding the term $\mathbf{r}_1$,

$$\frac{1}{2N^2}\sum_{t=0}^{N-1}\sum_{k=t}^{N-1}\left\langle(\mathbf{I}-\gamma\mathbf{H}_f^{(q)})^{k-t}\mathbf{H}_f^{(q)},\mathbf{r}_1\right\rangle\geq\frac{\gamma(1+\underline{\epsilon}_o)\underline{\sigma}^2}{4N^2}\sum_{t=0}^{N-1}\sum_{k=t}^{N-1}\left\langle(\mathbf{I}-\gamma\mathbf{H}_f^{(q)})^{k-t}\mathbf{H}_f^{(q)},\mathbf{I}-(\mathbf{I}-\gamma\mathbf{H}_f^{(q)})^t\right\rangle$$

$$=\frac{(1+\underline{\epsilon}_o)\underline{\sigma}^2}{4N^2}\sum_{t=0}^{N-1}\left\langle\mathbf{I}-(\mathbf{I}-\gamma\mathbf{H}_f^{(q)})^{N-t},\mathbf{I}-(\mathbf{I}-\gamma\mathbf{H}_f^{(q)})^t\right\rangle$$

$$=\frac{(1+\underline{\epsilon}_o)\underline{\sigma}^2}{4N^2}\sum_i\sum_{t=0}^{N-1}(1-(1-\gamma\tilde{\lambda}_i^{(q)})^{N-t})(1-(1-\gamma\tilde{\lambda}_i^{(q)})^t)$$

$$\geq\frac{(1+\underline{\epsilon}_o)\underline{\sigma}^2}{4N^2}\sum_i\sum_{t=0}^{N-1}(1-(1-\gamma\tilde{\lambda}_i^{(q)})^{N-t-1})(1-(1-\gamma\tilde{\lambda}_i^{(q)})^t).$$

Define

$$f(x)=\sum_{t=0}^{N-1}(1-(1-x)^{N-t-1})(1-(1-x)^t),\quad 0<x<1.$$

Note that if $N\geq 500$,

$$f(x)\geq\begin{cases}\frac{N}{10}, & \frac{1}{N}\leq x<1,\\ \frac{2N^3}{25}x^2, & 0<x<\frac{1}{N},\end{cases}\tag{67}$$

it follows that

$$\frac{1}{2N^2}\sum_{t=0}^{N-1}\sum_{k=t}^{N-1}\left\langle(\mathbf{I}-\gamma\mathbf{H}_f^{(q)})^{k-t}\mathbf{H}_f^{(q)},\mathbf{r}_1\right\rangle\geq\frac{(1+\underline{\epsilon}_o)\underline{\sigma}^2}{4N^2}\sum_i f(\gamma\tilde{\lambda}_i^{(q)})$$

$$\geq\frac{(1+\underline{\epsilon}_o)\underline{\sigma}^2}{50}\left(\frac{k^*}{N}+N\gamma^2\sum_{i>k^*}\left(\tilde{\lambda}_i^{(q)}\right)^2\right).$$

Regarding the time-dependent term $\mathbf{r}_2$,

$$\frac{1}{2N^2}\sum_{t=0}^{N-1}\sum_{k=t}^{N-1}\left\langle(\mathbf{I}-\gamma\mathbf{H}_f^{(q)})^{k-t}\mathbf{H}_f^{(q)},\mathbf{r}_2\right\rangle$$

$$\geq\frac{\beta\gamma(1+\underline{\epsilon}_o)(\underline{\epsilon}_p+(1+\underline{\epsilon}_p)\underline{\epsilon}_a)}{4N^2}\sum_{t=0}^{N-1}\sum_{k=t}^{N-1}\left\langle(\mathbf{I}-\gamma\mathbf{H}_f^{(q)})^{k-t}\mathbf{H}_f^{(q)},\mathbf{I}-(\mathbf{I}-\gamma\mathbf{H}_f^{(q)})^t\right\rangle\left\|\mathbf{v}^{(q)*}\right\|_{\mathbf{H}_f^{(q)}\left[\mathbf{I}-(\mathbf{I}-\gamma\mathbf{H}_f^{(q)})^{t/2}\right]^2}^2$$

$$=\frac{\beta(1+\underline{\epsilon}_o)(\underline{\epsilon}_p+(1+\underline{\epsilon}_p)\underline{\epsilon}_a)}{4N^2}\sum_{t=0}^{N-1}\left\langle\mathbf{I}-(\mathbf{I}-\gamma\mathbf{H}_f^{(q)})^{N-t},\mathbf{I}-(\mathbf{I}-\gamma\mathbf{H}_f^{(q)})^t\right\rangle\left\|\mathbf{v}^{(q)*}\right\|_{\mathbf{H}_f^{(q)}\left[\mathbf{I}-(\mathbf{I}-\gamma\mathbf{H}_f^{(q)})^{t/2}\right]^2}^2$$

$$=\frac{\beta(1+\underline{\epsilon}_o)(\underline{\epsilon}_p+(1+\underline{\epsilon}_p)\underline{\epsilon}_a)}{4N^2}\sum_{t=0}^{N-1}\sum_i(1-(1-\gamma\tilde{\lambda}_i^{(q)})^{N-t})(1-(1-\gamma\tilde{\lambda}_i^{(q)})^t)\sum_j\tilde{\lambda}_j^{(q)}\left(1-(1-\gamma\tilde{\lambda}_j^{(q)})^{t/2}\right)^2\omega_j^2$$

$$\geq\frac{\beta(1+\underline{\epsilon}_o)(\underline{\epsilon}_p+(1+\underline{\epsilon}_p)\underline{\epsilon}_a)}{4N^2}\sum_{t=N/2}^{N-1}\sum_i(1-(1-\gamma\tilde{\lambda}_i^{(q)})^{N-t})(1-(1-\gamma\tilde{\lambda}_i^{(q)})^t)\sum_j\tilde{\lambda}_j^{(q)}\left(1-(1-\gamma\tilde{\lambda}_j^{(q)})^{t/2}\right)^2\omega_j^2$$

$$\geq\frac{\beta(1+\underline{\epsilon}_o)(\underline{\epsilon}_p+(1+\underline{\epsilon}_p)\underline{\epsilon}_a)}{4N^2}\sum_{t=N/2}^{N-1}\sum_i(1-(1-\gamma\tilde{\lambda}_i^{(q)})^{N-t})(1-(1-\gamma\tilde{\lambda}_i^{(q)})^t)\sum_j\tilde{\lambda}_j^{(q)}\left(1-(1-\gamma\tilde{\lambda}_j^{(q)})^{N/4}\right)^2\omega_j^2$$

$$\geq\frac{\beta(1+\underline{\epsilon}_o)(\underline{\epsilon}_p+(1+\underline{\epsilon}_p)\underline{\epsilon}_a)}{8N^2}\sum_{t=0}^{N-1}\sum_i(1-(1-\gamma\tilde{\lambda}_i^{(q)})^{N-t})(1-(1-\gamma\tilde{\lambda}_i^{(q)})^t)\sum_j\tilde{\lambda}_j^{(q)}\left(1-(1-\gamma\tilde{\lambda}_j^{(q)})^{N/4}\right)^2\omega_j^2$$

$$\geq\frac{\beta(1+\underline{\epsilon}_o)(\underline{\epsilon}_p+(1+\underline{\epsilon}_p)\underline{\epsilon}_a)}{100}\left(\frac{k^*}{N}+N\gamma^2\sum_{i>k^*}\left(\tilde{\lambda}_i^{(q)}\right)^2\right)\sum_j\tilde{\lambda}_j^{(q)}\left(1-(1-\gamma\tilde{\lambda}_j^{(q)})^{N/4}\right)^2\omega_j^2,$$

where $\omega_j = (\mathbf{v}^{(q)*})^\top \mathbf{v}_j^{(q)}$ with $\mathbf{v}_j^{(q)}$ being the eigenvectors of $\mathbf{H}_f^{(q)}$ and the last inequality reuses the property (67). Noticing that

$$1 - (1 - \gamma\tilde{\lambda}_i^{(q)})^{\frac{N}{4}} \geq \begin{cases} 1 - (1 - \frac{1}{N})^{\frac{N}{4}} \geq 1 - e^{-\frac{1}{4}} \geq \frac{1}{5}, & \tilde{\lambda}_i^{(q)} \geq \frac{1}{\gamma N}, \\ \frac{N}{4} \cdot \gamma\tilde{\lambda}_i^{(q)} - \frac{N(N-4)}{32} \cdot \gamma^2(\tilde{\lambda}_i^{(q)})^2 \geq \frac{N}{5} \cdot \gamma\tilde{\lambda}_i^{(q)}, & \tilde{\lambda}_i^{(q)} < \frac{1}{\gamma N}. \end{cases}$$

We have

$$\sum_j \tilde{\lambda}_j^{(q)} \left(1 - (1 - \gamma\tilde{\lambda}_j^{(q)})^{N/4}\right)^2 \omega_j^2 \geq \sum_{j \leq k^*} \frac{\tilde{\lambda}_j^{(q)}}{25} \omega_j^2 + \sum_{j > k^*} \left(\tilde{\lambda}_j^{(q)}\right)^3 \frac{N^2\gamma^2}{25} \omega_j^2$$

$$\geq \frac{1}{25}\left(\left\|\mathbf{v}^{(q)*}\right\|^2_{\mathbf{H}_{f,0:k^*}^{(q)}} + N^2\gamma^2 \left\|\mathbf{v}^{(q)*}\right\|^2_{(\mathbf{H}_{f,k^*:\infty}^{(q)})^3}\right).$$

Hence,

$$\frac{1}{2N^2} \sum_{t=0}^{N-1} \sum_{k=t}^{N-1} \left\langle (\mathbf{I} - \gamma\mathbf{H}_f^{(q)})^{k-t}\mathbf{H}_f^{(q)}, \mathbf{r}_2 \right\rangle$$

$$\geq \frac{\beta(1 + \underline{\epsilon}_o)(\underline{\epsilon}_p + (1 + \underline{\epsilon}_p)\underline{\epsilon}_a)}{2500}\left(\frac{k^*}{N} + N\gamma^2 \sum_{i > k^*} \left(\tilde{\lambda}_i^{(q)}\right)^2\right)\left(\left\|\mathbf{v}^{(q)*}\right\|^2_{\mathbf{H}_{f,0:k^*}^{(q)}} + N^2\gamma^2 \left\|\mathbf{v}^{(q)*}\right\|^2_{(\mathbf{H}_{f,k^*:\infty}^{(q)})^3}\right).$$

Regarding the term $\mathbf{r}_3$,

$$\frac{1}{2N^2} \sum_{t=0}^{N-1} \sum_{k=t}^{N-1} \left\langle (\mathbf{I} - \gamma\mathbf{H}_f^{(q)})^{k-t}\mathbf{H}_f^{(q)}, \mathbf{r}_3 \right\rangle$$

$$= \frac{\gamma^2(1 + \underline{\epsilon}_o)\beta^2(\underline{\epsilon}_p + (1 + \underline{\epsilon}_p)\underline{\epsilon}_a)}{8N^2} \sum_{t=0}^{N-1} \sum_{k=t}^{N-1} \left\langle (\mathbf{I} - \gamma\mathbf{H}_f^{(q)})^{k-t}\mathbf{H}_f^{(q)}, \mathbf{I} - (\mathbf{I} - \gamma\mathbf{H}_f^{(q)})^t \right\rangle$$

$$\cdot \left\|\mathbf{v}^{(q)*}\right\|^2_{\mathbf{I} - (\mathbf{I} - \gamma\mathbf{H}_f^{(q)})^t} \operatorname{tr}\left(\mathbf{H}_f^{(q)}(\mathbf{I} - \gamma\mathbf{H}_f^{(q)})^{2t}\mathbf{H}_f^{(q)}\right)$$

$$= \frac{\gamma(1 + \underline{\epsilon}_o)\beta^2(\underline{\epsilon}_p + (1 + \underline{\epsilon}_p)\underline{\epsilon}_a)}{8N^2} \sum_{t=0}^{N-1} \left\langle \mathbf{I} - (\mathbf{I} - \gamma\mathbf{H}_f^{(q)})^{N-t}, \mathbf{I} - (\mathbf{I} - \gamma\mathbf{H}_f^{(q)})^t \right\rangle$$

$$\cdot \left\|\mathbf{v}^{(q)*}\right\|^2_{\mathbf{I} - (\mathbf{I} - \gamma\mathbf{H}_f^{(q)})^t} \operatorname{tr}\left(\mathbf{H}_f^{(q)}(\mathbf{I} - \gamma\mathbf{H}_f^{(q)})^{2t}\mathbf{H}_f^{(q)}\right)$$

$$\geq \frac{\gamma(1 + \underline{\epsilon}_o)\beta^2(\underline{\epsilon}_p + (1 + \underline{\epsilon}_p)\underline{\epsilon}_a)}{8N^2} \sum_{t=N/2}^{N-1} \left\langle \mathbf{I} - (\mathbf{I} - \gamma\mathbf{H}_f^{(q)})^{N-t}, \mathbf{I} - (\mathbf{I} - \gamma\mathbf{H}_f^{(q)})^t \right\rangle$$

$$\cdot \left\|\mathbf{v}^{(q)*}\right\|^2_{\mathbf{I} - (\mathbf{I} - \gamma\mathbf{H}_f^{(q)})^t} \operatorname{tr}\left(\mathbf{H}_f^{(q)}(\mathbf{I} - \gamma\mathbf{H}_f^{(q)})^{2t}\mathbf{H}_f^{(q)}\right)$$

$$\geq \frac{\gamma(1 + \underline{\epsilon}_o)\beta^2(\underline{\epsilon}_p + (1 + \underline{\epsilon}_p)\underline{\epsilon}_a)}{8N^2} \sum_{t=N/2}^{N-1} \left\langle \mathbf{I} - (\mathbf{I} - \gamma\mathbf{H}_f^{(q)})^{N-t}, \mathbf{I} - (\mathbf{I} - \gamma\mathbf{H}_f^{(q)})^t \right\rangle$$

$$\cdot \left\|\mathbf{v}^{(q)*}\right\|^2_{\mathbf{I} - (\mathbf{I} - \gamma\mathbf{H}_f^{(q)})^{N/2}} \operatorname{tr}\left(\mathbf{H}_f^{(q)}(\mathbf{I} - \gamma\mathbf{H}_f^{(q)})^{2N}\mathbf{H}_f^{(q)}\right)$$

$$\geq \frac{\gamma(1 + \underline{\epsilon}_o)\beta^2(\underline{\epsilon}_p + (1 + \underline{\epsilon}_p)\underline{\epsilon}_a)}{16N^2} \sum_{t=0}^{N-1} \left\langle \mathbf{I} - (\mathbf{I} - \gamma\mathbf{H}_f^{(q)})^{N-t}, \mathbf{I} - (\mathbf{I} - \gamma\mathbf{H}_f^{(q)})^t \right\rangle$$

$$\cdot \left\|\mathbf{v}^{(q)*}\right\|^2_{\mathbf{I} - (\mathbf{I} - \gamma\mathbf{H}_f^{(q)})^{N/2}} \operatorname{tr}\left(\mathbf{H}_f^{(q)}(\mathbf{I} - \gamma\mathbf{H}_f^{(q)})^{2N}\mathbf{H}_f^{(q)}\right).$$

Noticing that

$$1 - (1 - \gamma\tilde{\lambda}_i^{(q)})^{\frac{N}{2}} \geq \begin{cases} 1 - (1 - \frac{1}{N})^{\frac{N}{2}} \geq 1 - e^{-\frac{1}{2}} \geq \frac{1}{3}, & \tilde{\lambda}_i^{(q)} \geq \frac{1}{\gamma N}, \\ \frac{N}{2} \cdot \gamma\tilde{\lambda}_i^{(q)} - \frac{N(N-2)}{8} \cdot \gamma^2(\tilde{\lambda}_i^{(q)})^2 \geq \frac{N}{3} \cdot \gamma\tilde{\lambda}_i^{(q)}, & \tilde{\lambda}_i^{(q)} < \frac{1}{\gamma N}, \end{cases}$$

it holds

$$\left\| \mathbf{v}^{(q)*} \right\|_{\mathbf{I}-(\mathbf{I}-\gamma\mathbf{H}_f^{(q)})^{N/2}}^2 = \sum_i (1-(1-\gamma\tilde{\lambda}_i^{(q)})^{N/2})\omega_i^2 \geq \frac{1}{3}\left( \left\| \mathbf{v}^{(q)*} \right\|_{\mathbf{I}_{f,0:k^*}^{(q)}}^2 + N\gamma \left\| \mathbf{v}^{(q)*} \right\|_{\mathbf{H}_{f,k^*:\infty}^{(q)}}^2 \right).$$

Recall that

$$\sum_{t=0}^{N-1} \left\langle \mathbf{I}-(\mathbf{I}-\gamma\mathbf{H}_f^{(q)})^{N-t}, \mathbf{I}-(\mathbf{I}-\gamma\mathbf{H}_f^{(q)})^t \right\rangle = \sum_i \sum_{t=0}^{N-1} \left(1-(1-\gamma\tilde{\lambda}_i^{(q)})^{N-t}\right)\left(1-(1-\gamma\tilde{\lambda}_i^{(q)})^t\right)$$

$$\geq \frac{2}{25}N^2\left(\frac{k^*}{N} + N\gamma^2 \sum_{i>k^*} \left(\tilde{\lambda}_i^{(q)}\right)^2\right),$$

we have

$$\frac{1}{2N^2}\sum_{t=0}^{N-1}\sum_{k=t}^{N-1} \left\langle (\mathbf{I}-\gamma\mathbf{H}_f^{(q)})^{k-t}\mathbf{H}_f^{(q)}, \mathbf{r}_3 \right\rangle$$

$$\geq \frac{\gamma(1+\underline{\epsilon}_o)\beta^2\left(\underline{\epsilon}_p+(1+\underline{\epsilon}_p)\underline{\epsilon}_a\right)}{600}\left(\frac{k^*}{N}+N\gamma^2\sum_{i>k^*}\left(\tilde{\lambda}_i^{(q)}\right)^2\right)\left(\left\|\mathbf{v}^{(q)*}\right\|_{\mathbf{I}_{f,0:k^*}^{(q)}}^2 + N\gamma\left\|\mathbf{v}^{(q)*}\right\|_{\mathbf{H}_{f,k^*:\infty}^{(q)}}^2\right)$$

$$\cdot\sum_i (\tilde{\lambda}_i^{(q)})^2(1-\gamma\tilde{\lambda}_i^{(q)})^{2N}.$$

Therefore,

$$\frac{1}{2N^2}\sum_{t=0}^{N-1}\sum_{k=t}^{N-1} \left\langle (\mathbf{I}-\gamma\mathbf{H}_f^{(q)})^{k-t}\mathbf{H}_f^{(q)}, \underline{\mathbf{C}}_t^{(M)} \right\rangle$$

$$\geq \left[\frac{(1+\underline{\epsilon}_o)\underline{\sigma}^2}{50} + \frac{\beta(1+\underline{\epsilon}_o)(\underline{\epsilon}_p+(1+\underline{\epsilon}_p)\underline{\epsilon}_a)}{2500}\left(\left\|\mathbf{v}^{(q)*}\right\|_{\mathbf{H}_{f,0:k^*}^{(q)}}^2 + N^2\gamma^2\left\|\mathbf{v}^{(q)*}\right\|_{(\mathbf{H}_{f,k^*:\infty}^{(q)})^3}^2\right)\right]\frac{d_{\text{eff}}}{N}$$

$$+\frac{\gamma(1+\underline{\epsilon}_o)\beta^2\left(\underline{\epsilon}_p+(1+\underline{\epsilon}_p)\underline{\epsilon}_a\right)}{600}\left(\left\|\mathbf{v}^{(q)*}\right\|_{\mathbf{I}_{f,0:k^*}^{(q)}}^2 + N\gamma\left\|\mathbf{v}^{(q)*}\right\|_{\mathbf{H}_{f,k^*:\infty}^{(q)}}^2\right)\cdot\sum_i(\tilde{\lambda}_i^{(q)})^2(1-\gamma\tilde{\lambda}_i^{(q)})^{2N}\frac{d_{\text{eff}}}{N},$$

where $\frac{d_{\text{eff}}}{N} = \frac{k^*}{N} + N\gamma^2\sum_{i>k^*}\left(\tilde{\lambda}_i^{(q)}\right)^2$. $\qquad\square$

## D.4. Bias Lower Bounds

Recall that we have the following lower bound on the bias error. For general quantization,

$$\frac{1}{2N^2}\cdot\sum_{t=0}^{N-1}\sum_{k=t}^{N-1}\left\langle(\mathbf{I}-\gamma\mathbf{H}_f^{(q)})^{k-t}\mathbf{H}_f^{(q)}, \underline{\mathbf{B}}_t\right\rangle = \frac{1}{2\gamma N^2}\cdot\sum_{t=0}^{N-1}\left\langle\left(\mathbf{I}-(\mathbf{I}-\gamma\mathbf{H}_f^{(q)})^{N-t}\right), \underline{\mathbf{B}}_t\right\rangle$$

$$\geq \frac{1}{2\gamma N^2}\cdot\sum_{t=0}^{N/2}\left\langle\left(\mathbf{I}-(\mathbf{I}-\gamma\mathbf{H}_f^{(q)})^{N-t}\right), \underline{\mathbf{B}}_t\right\rangle \qquad (68)$$

$$\geq \frac{1}{2\gamma N^2}\cdot\left\langle\left(\mathbf{I}-(\mathbf{I}-\gamma\mathbf{H}_f^{(q)})^{N/2}\right), \sum_{t=0}^{N/2}\underline{\mathbf{B}}_t\right\rangle.$$

Let $\underline{\mathbf{S}}_n := \sum_{t=0}^{n-1}\underline{\mathbf{B}}_t$ and $\underline{\mathbf{S}}_n^{(M)} := \sum_{t=0}^{n-1}\underline{\mathbf{B}}_t^{(M)}$. Then the remaining challenge is to lower bound $\underline{\mathbf{S}}_{N/2+1}$ and $\underline{\mathbf{S}}_{N/2+1}^{(M)}$.

### D.4.1. GENERAL QUANTIZATION

**Lemma D.10.** *Suppose Assumption 3.3 and 3.4 holds, if the stepsize $\gamma < \frac{1}{\tilde{\lambda}_1^{(q)}}$, then*

$$\underline{\mathbf{S}}_n \succeq \frac{\beta}{4}\text{tr}\left(\left[\mathbf{I}-(\mathbf{I}-\gamma\mathbf{H}_f^{(q)})^{n/2}\right]\underline{\mathbf{B}}_0\right)\left[\mathbf{I}-(\mathbf{I}-\gamma\mathbf{H}_f^{(q)})^{n/2}\right] + \sum_{t=0}^{n-1}(\mathbf{I}-\gamma\mathbf{H}_f^{(q)})^t\underline{\mathbf{B}}_0(\mathbf{I}-\gamma\mathbf{H}_f^{(q)})^t.$$

*Proof.* We first build a crude bound. By the definition of $\underline{\mathbf{B}}_t$,

$$\mathbf{S}_n \succeq \sum_{t=0}^{n-1}(\mathcal{I} - \gamma\mathcal{T}^{(q)})^t \circ \underline{\mathbf{B}}_0 \succeq \sum_{t=0}^{n-1}(\mathcal{I} - \gamma\widetilde{\mathcal{T}}^{(q)})^t \circ \underline{\mathbf{B}}_0 = \sum_{t=0}^{n-1}(\mathbf{I} - \gamma\mathbf{H}_f^{(q)})^t\underline{\mathbf{B}}_0(\mathbf{I} - \gamma\mathbf{H}_f^{(q)})^t.$$

Further by Assumption 3.4, we have

$$(\mathcal{M}^{(q)} - \widetilde{\mathcal{M}}^{(q)}) \circ \underline{\mathbf{S}}_n \succeq \beta\mathrm{tr}\left(\mathbf{H}_f^{(q)}\underline{\mathbf{S}}_n\right)\mathbf{H}_f^{(q)}$$

$$\succeq \beta\mathrm{tr}\left(\mathbf{H}_f^{(q)}\sum_{t=0}^{n-1}(\mathbf{I} - \gamma\mathbf{H}_f^{(q)})^t\underline{\mathbf{B}}_0(\mathbf{I} - \gamma\mathbf{H}_f^{(q)})^t\right)\mathbf{H}_f^{(q)}$$

$$\succeq \beta\mathrm{tr}\left(\mathbf{H}_f^{(q)}\sum_{t=0}^{n-1}(\mathbf{I} - 2\gamma\mathbf{H}_f^{(q)})^t\underline{\mathbf{B}}_0\right)\mathbf{H}_f^{(q)}$$

$$= \frac{\beta}{2\gamma}\mathrm{tr}\left(\left[\mathbf{I} - (\mathbf{I} - 2\gamma\mathbf{H}_f^{(q)})^n\right]\underline{\mathbf{B}}_0\right)\mathbf{H}_f^{(q)}$$

$$\succeq \frac{\beta}{2\gamma}\mathrm{tr}\left(\left[\mathbf{I} - (\mathbf{I} - \gamma\mathbf{H}_f^{(q)})^n\right]\underline{\mathbf{B}}_0\right)\mathbf{H}_f^{(q)}.$$

Next we use the above inequality to build a refined lower bound.

$$\underline{\mathbf{S}}_n = (\mathcal{I} - \gamma\widetilde{\mathcal{T}}^{(q)}) \circ \underline{\mathbf{S}}_{n-1} + \gamma^2(\mathcal{M}^{(q)} - \widetilde{\mathcal{M}}^{(q)}) \circ \underline{\mathbf{S}}_{n-1} + \underline{\mathbf{B}}_0$$

$$\succeq (\mathcal{I} - \gamma\widetilde{\mathcal{T}}^{(q)}) \circ \underline{\mathbf{S}}_{n-1} + \gamma^2\frac{\beta}{2\gamma}\mathrm{tr}\left(\left[\mathbf{I} - (\mathbf{I} - \gamma\mathbf{H}_f^{(q)})^{n-1}\right]\underline{\mathbf{B}}_0\right)\mathbf{H}_f^{(q)} + \underline{\mathbf{B}}_0.$$

Solving the recursion yields

$$\underline{\mathbf{S}}_n \succeq \sum_{t=0}^{n-1}(\mathcal{I} - \gamma\widetilde{\mathcal{T}}^{(q)})^t \circ \left\{\frac{\beta\gamma}{2}\mathrm{tr}\left(\left[\mathbf{I} - (\mathbf{I} - \gamma\mathbf{H}_f^{(q)})^{n-1-t}\right]\underline{\mathbf{B}}_0\right)\mathbf{H}_f^{(q)} + \underline{\mathbf{B}}_0\right\}$$

$$= \frac{\beta\gamma}{2}\sum_{t=0}^{n-1}\mathrm{tr}\left(\left[\mathbf{I} - (\mathbf{I} - \gamma\mathbf{H}_f^{(q)})^{n-1-t}\right]\underline{\mathbf{B}}_0\right)(\mathbf{I} - \gamma\mathbf{H}_f^{(q)})^{2t}\mathbf{H}_f^{(q)} + \sum_{t=0}^{n-1}(\mathbf{I} - \gamma\mathbf{H}_f^{(q)})^t\underline{\mathbf{B}}_0(\mathbf{I} - \gamma\mathbf{H}_f^{(q)})^t.$$

For the first term, noticing the following:

$$\sum_{t=0}^{n-1}\mathrm{tr}\left(\left[\mathbf{I} - (\mathbf{I} - \gamma\mathbf{H}_f^{(q)})^{n-1-t}\right]\underline{\mathbf{B}}_0\right)(\mathbf{I} - \gamma\mathbf{H}_f^{(q)})^{2t}\mathbf{H}_f^{(q)}$$

$$\succeq \sum_{t=0}^{n-1}\mathrm{tr}\left(\left[\mathbf{I} - (\mathbf{I} - \gamma\mathbf{H}_f^{(q)})^{n-1-t}\right]\underline{\mathbf{B}}_0\right)(\mathbf{I} - 2\gamma\mathbf{H}_f^{(q)})^t\mathbf{H}_f^{(q)}$$

$$\succeq \sum_{t=0}^{n/2-1}\mathrm{tr}\left(\left[\mathbf{I} - (\mathbf{I} - \gamma\mathbf{H}_f^{(q)})^{n-1-t}\right]\underline{\mathbf{B}}_0\right)(\mathbf{I} - 2\gamma\mathbf{H}_f^{(q)})^t\mathbf{H}_f^{(q)}$$

$$\succeq \mathrm{tr}\left(\left[\mathbf{I} - (\mathbf{I} - \gamma\mathbf{H}_f^{(q)})^{n/2}\right]\underline{\mathbf{B}}_0\right)\sum_{t=0}^{n/2-1}(\mathbf{I} - 2\gamma\mathbf{H}_f^{(q)})^t\mathbf{H}_f^{(q)}$$

$$= \frac{1}{2\gamma}\mathrm{tr}\left(\left[\mathbf{I} - (\mathbf{I} - \gamma\mathbf{H}_f^{(q)})^{n/2}\right]\underline{\mathbf{B}}_0\right)\left[\mathbf{I} - (\mathbf{I} - 2\gamma\mathbf{H}_f^{(q)})^{n/2}\right]$$

$$\succeq \frac{1}{2\gamma}\mathrm{tr}\left(\left[\mathbf{I} - (\mathbf{I} - \gamma\mathbf{H}_f^{(q)})^{n/2}\right]\underline{\mathbf{B}}_0\right)\left[\mathbf{I} - (\mathbf{I} - \gamma\mathbf{H}_f^{(q)})^{n/2}\right],$$

The proof is immediately completed. $\square$

**Lemma D.11** (A bias lower bound under general quantization). *Suppose Assumption 3.3, 3.4 holds, if the stepsize $\gamma < \frac{1}{\tilde{\lambda}_1^{(q)}}$, then*

$$\frac{1}{2N^2} \cdot \sum_{t=0}^{N-1} \sum_{k=t}^{N-1} \left\langle (\mathbf{I} - \gamma \mathbf{H}_f^{(q)})^{k-t} \mathbf{H}_f^{(q)}, \underline{\mathbf{B}}_t \right\rangle$$

$$\geq \frac{1}{100\gamma^2 N^2} \left( \|\mathbf{v}^{(q)^*}\|_{(\mathbf{H}_{f,0:k^*}^{(q)})^{-1}}^2 + N^2 \gamma^2 \|\mathbf{v}^{(q)^*}\|_{\mathbf{H}_{f,k^*:\infty}^{(q)}}^2 \right)$$

$$+ \frac{\beta}{1000\gamma N^2} \left( \|\mathbf{v}^{(q)^*}\|_{\mathbf{I}_{f,0:k^*}^{(q)}}^2 + \gamma N \|\mathbf{v}^{(q)^*}\|_{\mathbf{H}_{f,k^*:\infty}^{(q)}}^2 \right) \left( k^* + \gamma^2 N^2 \sum_{i>k^*} (\tilde{\lambda}_i^{(q)})^2 \right).$$

*Proof.* According to (68) and Lemma D.10,

$$\frac{1}{2N^2} \cdot \sum_{t=0}^{N-1} \sum_{k=t}^{N-1} \left\langle (\mathbf{I} - \gamma \mathbf{H}_f^{(q)})^{k-t} \mathbf{H}_f^{(q)}, \underline{\mathbf{B}}_t \right\rangle \geq \frac{1}{2\gamma N^2} \cdot \left\langle \mathbf{I} - (\mathbf{I} - \gamma \mathbf{H}_f^{(q)})^{N/2}, \sum_{t=0}^{N/2} \underline{\mathbf{B}}_t \right\rangle$$

$$\geq \underbrace{\frac{1}{2\gamma N^2} \cdot \frac{\beta}{4} \mathrm{tr}\left( \left[ \mathbf{I} - (\mathbf{I} - \gamma \mathbf{H}_f^{(q)})^{N/4} \right] \underline{\mathbf{B}}_0 \right) \left\langle \mathbf{I} - (\mathbf{I} - \gamma \mathbf{H}_f^{(q)})^{N/2}, \mathbf{I} - (\mathbf{I} - \gamma \mathbf{H}_f^{(q)})^{N/4} \right\rangle}_{I_1}$$

$$+ \underbrace{\frac{1}{2\gamma N^2} \left\langle \mathbf{I} - (\mathbf{I} - \gamma \mathbf{H}_f^{(q)})^{N/2}, \sum_{t=0}^{N/2-1} (\mathbf{I} - \gamma \mathbf{H}_f^{(q)})^t \underline{\mathbf{B}}_0 (\mathbf{I} - \gamma \mathbf{H}_f^{(q)})^t \right\rangle}_{I_2}.$$

The first term $I_1$ can be lower bounded by

$$I_1 \geq \frac{\beta}{8\gamma N^2} \mathrm{tr}\left( \left[ \mathbf{I} - (\mathbf{I} - \gamma \mathbf{H}_f^{(q)})^{N/4} \right] \underline{\mathbf{B}}_0 \right) \mathrm{tr}\left( \left[ \mathbf{I} - (\mathbf{I} - \gamma \mathbf{H}_f^{(q)})^{N/4} \right]^2 \right)$$

$$= \frac{\beta}{8\gamma N^2} \left( \sum_i \left[ 1 - (1 - \gamma \tilde{\lambda}_i^{(q)})^{N/4} \right] \omega_i^2 \right) \left( \sum_i \left[ 1 - (1 - \gamma \tilde{\lambda}_i^{(q)})^{N/4} \right]^2 \right),$$

where $\omega_i = (\mathbf{v}_0 - \mathbf{v}^{(q)^*})^\top \mathbf{v}_i^{(q)}$ with $\mathbf{v}_i^{(q)}$ being the eigenvectors of $\mathbf{H}_f^{(q)}$. The second term $I_2$ can be lower bounded by

$$I_2 = \frac{1}{2\gamma N^2} \left\langle \sum_{t=0}^{N/2-1} (\mathbf{I} - \gamma \mathbf{H}_f^{(q)})^{2t} \left[ \mathbf{I} - (\mathbf{I} - \gamma \mathbf{H}_f^{(q)})^{N/2} \right], \underline{\mathbf{B}}_0 \right\rangle$$

$$\geq \frac{1}{2\gamma N^2} \left\langle \sum_{t=0}^{N/2-1} (\mathbf{I} - 2\gamma \mathbf{H}_f^{(q)})^t \left[ \mathbf{I} - (\mathbf{I} - \gamma \mathbf{H}_f^{(q)})^{N/2} \right], \underline{\mathbf{B}}_0 \right\rangle$$

$$\geq \frac{1}{4\gamma^2 N^2} \left\langle \mathbf{H}_f^{(q)-1} \left[ \mathbf{I} - (\mathbf{I} - \gamma \mathbf{H}_f^{(q)})^{N/2} \right]^2, \underline{\mathbf{B}}_0 \right\rangle$$

$$\geq \frac{1}{4\gamma^2 N^2} \left\langle \mathbf{H}_f^{(q)-1} \left[ \mathbf{I} - (\mathbf{I} - \gamma \mathbf{H}_f^{(q)})^{N/4} \right]^2, \underline{\mathbf{B}}_0 \right\rangle$$

$$= \frac{1}{4\gamma^2 N^2} \sum_i (\tilde{\lambda}_i^{(q)})^{-1} \left( 1 - (1 - \gamma \tilde{\lambda}_i^{(q)})^{N/4} \right)^2 \omega_i^2.$$

Noticing that

$$1 - (1 - \gamma \tilde{\lambda}_i^{(q)})^{\frac{N}{4}} \geq \begin{cases} 1 - (1 - \frac{1}{N})^{\frac{N}{4}} \geq 1 - e^{-\frac{1}{4}} \geq \frac{1}{5}, & \tilde{\lambda}_i^{(q)} \geq \frac{1}{\gamma N}, \\ \frac{N}{4} \cdot \gamma \tilde{\lambda}_i^{(q)} - \frac{N(N-4)}{32} \cdot \gamma^2 (\tilde{\lambda}_i^{(q)})^2 \geq \frac{N}{5} \cdot \gamma \tilde{\lambda}_i^{(q)}, & \tilde{\lambda}_i^{(q)} < \frac{1}{\gamma N}. \end{cases}$$

Plugging this into $I_1$ and $I_2$ yields

$$I_1 \geq \frac{\beta}{8\gamma N^2} \left( \sum_{i \leq k^*} \frac{1}{5}\omega_i^2 + \sum_{i > k^*} \frac{\gamma N}{5}\tilde{\lambda}_i^{(q)}\omega_i^2 \right) \left( \frac{k^*}{25} + \frac{\gamma^2 N^2}{25} \sum_{i > k^*} (\tilde{\lambda}_i^{(q)})^2 \right)$$

$$= \frac{\beta}{1000\gamma N^2} \left( \|\mathbf{v}^{(q)^*}\|^2_{\mathbf{I}^{(q)}_{f,0:k^*}} + \gamma N\|\mathbf{v}^{(q)^*}\|^2_{\mathbf{H}^{(q)}_{f,k^*:\infty}} \right) \left( k^* + \gamma^2 N^2 \sum_{i > k^*} (\tilde{\lambda}_i^{(q)})^2 \right),$$

and

$$I_2 \geq \frac{1}{4\gamma^2 N^2} \left( \sum_{i \leq k^*} (\tilde{\lambda}_i^{(q)})^{-1} \frac{1}{25}\omega_i^2 + \sum_{i > k^*} \tilde{\lambda}_i^{(q)} \frac{N^2\gamma^2}{25}\omega_i^2 \right)$$

$$= \frac{1}{100\gamma^2 N^2} \left( \|\mathbf{v}^{(q)^*}\|^2_{(\mathbf{H}^{(q)}_{f,0:k^*})^{-1}} + N^2\gamma^2\|\mathbf{v}^{(q)^*}\|^2_{\mathbf{H}^{(q)}_{f,k^*:\infty}} \right).$$

$\square$

### D.4.2. MULTIPLICATIVE QUANTIZATION

**Lemma D.12.** *Suppose Assumption 3.3 and 3.4 holds, if the stepsize $\gamma < \frac{1}{\tilde{\lambda}_1^{(q)}}$, then*

$$\underline{\mathbf{S}}_n^{(M)} \succeq \frac{\beta}{4}\mathrm{tr}\left( \left[ \mathbf{I} - (\mathbf{I} - \gamma\mathbf{H}_f^{(q)})^{n/2} \right] \underline{\mathbf{B}}_0^{(M)} \right) \left[ \mathbf{I} - (\mathbf{I} - \gamma\mathbf{H}_f^{(q)})^{n/2} \right] + \sum_{t=0}^{n-1} (\mathbf{I} - \gamma\mathbf{H}_f^{(q)})^t \underline{\mathbf{B}}_0^{(M)} (\mathbf{I} - \gamma\mathbf{H}_f^{(q)})^t.$$

*Proof.* As $\underline{\mathbf{S}}_n^{(M)}$ and $\underline{\mathbf{S}}_n$ have the same update rule, they own the same lower bound. $\square$

**Lemma D.13** (A bias lower bound under multiplicative quantization)**.** *For $i \in \{d, f, s, p, a, o\}$, if there exist constants $(\bar{\epsilon}_i, \underline{\epsilon}_i)$ such that $\mathcal{Q}_i$ is $(\bar{\epsilon}_i, \underline{\epsilon}_i)$-multiplicative, suppose the stepsize $\gamma < \frac{1}{\tilde{\lambda}_1^{(q)}}$, then under Assumption 3.3, 3.4,*

$$\frac{1}{2N^2} \cdot \sum_{t=0}^{N-1} \sum_{k=t}^{N-1} \left\langle (\mathbf{I} - \gamma\mathbf{H}_f^{(q)})^{k-t}\mathbf{H}_f^{(q)}, \underline{\mathbf{B}}_t^{(M)} \right\rangle$$

$$\geq \frac{1}{100\gamma^2 N^2} \left( \|\mathbf{v}^{(q)^*}\|^2_{(\mathbf{H}^{(q)}_{f,0:k^*})^{-1}} + N^2\gamma^2\|\mathbf{v}^{(q)^*}\|^2_{\mathbf{H}^{(q)}_{f,k^*:\infty}} \right)$$

$$+ \frac{\beta}{1000\gamma N^2} \left( \|\mathbf{v}^{(q)^*}\|^2_{\mathbf{I}^{(q)}_{f,0:k^*}} + \gamma N\|\mathbf{v}^{(q)^*}\|^2_{\mathbf{H}^{(q)}_{f,k^*:\infty}} \right) \left( k^* + \gamma^2 N^2 \sum_{i > k^*} (\tilde{\lambda}_i^{(q)})^2 \right).$$

*Proof.* The proof is the same as the proof for Lemma D.11. $\square$

## D.5. Final Lower Bounds

### D.5.1. GENERAL QUANTIZATION

**Theorem D.14.** *Suppose the stepsize $\gamma < \frac{1}{\tilde{\lambda}_1^{(q)}}$, then under Assumption 3.1, 3.3, 3.4 and 3.5, for sufficiently large $N > 500$,*

$$R_N^{(0)} \geq \mathrm{BiasError} + \mathrm{VarianceError},$$

*where*

$$\mathrm{BiasError} \geq \frac{1}{100\gamma^2 N^2} \left( \|\mathbf{v}^{(q)^*}\|^2_{(\mathbf{H}^{(q)}_{f,0:k^*})^{-1}} + N^2\gamma^2\|\mathbf{v}^{(q)^*}\|^2_{\mathbf{H}^{(q)}_{f,k^*:\infty}} \right),$$

$$\mathrm{VarianceError} \geq \frac{\beta}{1000\gamma N^2} \left( \|\mathbf{v}^{(q)^*}\|^2_{\mathbf{I}^{(q)}_{f,0:k^*}} + \gamma N\|\mathbf{v}^{(q)^*}\|^2_{\mathbf{H}^{(q)}_{f,k^*:\infty}} \right) \left( k^* + \gamma^2 N^2 \sum_{i > k^*} (\tilde{\lambda}_i^{(q)})^2 \right)$$

$$+ \frac{\sigma_G^2}{50} \left( \frac{k^*}{N} + N\gamma^2 \sum_{i > k^*} \left( \tilde{\lambda}_i^{(q)} \right)^2 \right),$$

*where $k^* = \max\{i : \tilde{\lambda}_i^{(q)} \geq 1/(\gamma N)\}$, and*

$$\underline{\sigma}_G^2 = \underline{\sigma}^2 + \inf_t \beta \mathrm{tr}\left(\mathbf{H}_f^{(q)} \mathbb{E}\left[\boldsymbol{\epsilon}_{t-1}^{(p)} \boldsymbol{\epsilon}_{t-1}^{(p)}{}^\top\right]\right) + \inf_t \left\{\mathbb{E}\left[\epsilon_t^{(a)^2}\Big|a_t\right] + \mathbb{E}\left[\epsilon_t^{(o)^2}\Big|o_t\right]\right\}.$$

*Proof.* The proof can be completed by Lemma D.4, Lemma D.7 and Lemma D.11. □

### D.5.2. MULTIPLICATIVE QUANTIZATION

**Theorem D.15.** *For $i \in \{d, f, s, p, a, o\}$, if there exist constants $(\bar{\epsilon}_i, \underline{\epsilon}_i)$ such that $\mathcal{Q}_i$ is $(\bar{\epsilon}_i, \underline{\epsilon}_i)$-multiplicative, suppose the stepsize $\gamma < \frac{1}{\tilde{\lambda}_1^{(q)}}$, then under Assumption 3.1, 3.3, 3.4 and 3.5, for sufficiently large $N > 500$,*

$$R_N^{(0)} \geq \mathrm{BiasError} + \mathrm{VarianceError},$$

*where*

$$\mathrm{BiasError} \geq \frac{1}{100\gamma^2 N^2}\left(\|\mathbf{v}^{(q)^*}\|_{(\mathbf{H}_{f,0:k^*}^{(q)})^{-1}}^2 + N^2\gamma^2\|\mathbf{v}^{(q)^*}\|_{\mathbf{H}_{f,k^*:\infty}^{(q)}}^2\right),$$

$$\mathrm{VarianceError} \geq \frac{(1+\underline{\epsilon}_o)\underline{\sigma}^2}{50}\left(\frac{k^*}{N} + N\gamma^2 \sum_{i>k^*}\left(\tilde{\lambda}_i^{(q)}\right)^2\right)$$

$$+ \beta(1+\underline{\epsilon}_o)(\underline{\epsilon}_p + (1+\underline{\epsilon}_p)\underline{\epsilon}_a)\left(\frac{k^*}{N} + N\gamma^2 \sum_{i>k^*}\left(\tilde{\lambda}_i^{(q)}\right)^2\right)P_{\mathrm{eff}}$$

$$+ \frac{\beta}{1000}\left(\frac{1}{N\gamma}\left\|\mathbf{v}^{(q)^*}\right\|_{\mathbf{I}_{f,0:k^*}^{(q)}}^2 + \left\|\mathbf{v}^{(q)^*}\right\|_{\mathbf{H}_{f,k^*:\infty}^{(q)}}^2\right)\left(\frac{k^*}{N} + N\gamma^2 \sum_{i>k^*}\left(\tilde{\lambda}_i^{(q)}\right)^2\right),$$

*where $k^* = \max\{i : \tilde{\lambda}_i^{(q)} \geq 1/(\gamma N)\}$ and*

$$P_{\mathrm{eff}} = \frac{1}{2500}\left(\left\|\mathbf{v}^{(q)^*}\right\|_{\mathbf{H}_{f,0:k^*}^{(q)}}^2 + N^2\gamma^2\left\|\mathbf{v}^{(q)^*}\right\|_{(\mathbf{H}_{f,k^*:\infty}^{(q)})^3}^2\right)$$

$$+ \frac{\gamma\beta}{600}\left(\left\|\mathbf{v}^{(q)^*}\right\|_{\mathbf{I}_{f,0:k^*}^{(q)}}^2 + N\gamma\left\|\mathbf{v}^{(q)^*}\right\|_{\mathbf{H}_{f,k^*:\infty}^{(q)}}^2\right)\sum_i(\tilde{\lambda}_i^{(q)})^2(1-\gamma\tilde{\lambda}_i^{(q)})^{2N}.$$

*Proof.* The proof can be completed by Lemma D.5, Lemma D.9 and Lemma D.13. □

### D.6. Additive Error Lower Bounds under Power-law Spectrum

We first analyze the additive error in Lemma B.2 and take expectation on $\mathbf{w}^*$:

$$\mathrm{AdditiveError} = \frac{1}{2}\left\langle \mathbf{SHS}^\top, (\mathbf{v}^{(q)^*} - \mathbf{v}^*) \otimes (\mathbf{v}^{(q)^*} - \mathbf{v}^*)\right\rangle$$

$$+ \left[\left(\mathbf{v}^{(q)^*}\right)^\top \frac{1}{N\gamma}\left[\mathbf{I} - \left(\mathbf{I} - \gamma\mathbf{H}_f^{(q)}\right)^N\right]\left(\mathbf{H}_f^{(q)}\right)^{-1}\left(\mathbf{H}_f^{(q)} - \mathbf{SHS}^\top\right)\mathbf{v}^{(q)^*}\right].$$

Recall that

$$\mathbf{v}^* = \left(\mathbf{SHS}^\top\right)^{-1}\mathbf{SHw}^*, \quad \mathbf{v}^{(q)^*} = (\mathbf{H}_f^{(q)})^{-1}\mathbf{SHw}^*.$$

Denote $\mathbf{D} = \mathbf{H}_f^{(q)} - \mathbf{SHS}^\top$, then

$$\mathbf{v}^{(q)^*} = \left(\mathbf{SHS}^\top + \mathbf{D}\right)^{-1}\mathbf{SHS}^\top\mathbf{v}^*.$$

It follows that

$$\mathbf{v}^* - \mathbf{v}^{(q)^*} = \left(\mathbf{SHS}^\top + \mathbf{D}\right)^{-1}\mathbf{Dv}^*.$$

Hence,

$$\frac{1}{2}\left\langle \mathbf{SHS}^\top, (\mathbf{v}^{(q)^*} - \mathbf{v}^*) \otimes (\mathbf{v}^{(q)^*} - \mathbf{v}^*)\right\rangle = \frac{1}{2}\|\mathbf{w}^*\|_{\mathbf{S}_1}^2, \tag{69}$$

where

$$\mathbf{S}_1 = \mathbf{HS}^\top\left(\mathbf{SHS}^\top\right)^{-1}\mathbf{D}\left(\mathbf{SHS}^\top + \mathbf{D}\right)^{-1}\mathbf{SHS}^\top\left(\mathbf{SHS}^\top + \mathbf{D}\right)^{-1}\mathbf{D}\left(\mathbf{SHS}^\top\right)^{-1}\mathbf{SH}.$$

**Lemma D.16** (Additive Error under multiplicative quantization, a lower bound). *Under Assumption 3.1, 3.3 and 3.6, for any $i \in \{s, d, f\}$, if there exist $(\bar{\epsilon}_i, \underline{\epsilon}_i)$ such that quantization $\mathcal{Q}_i$ is $(\bar{\epsilon}_i, \underline{\epsilon}_i)$-multiplicative, then*

$$\mathbb{E}_{\mathbf{w}^*} \|\mathbf{w}^*\|_{\mathbf{S}_1}^2 \gtrsim \frac{\left[(1 + \underline{\epsilon}_d)(1 + \underline{\epsilon}_f)(1 + \underline{\epsilon}_s) - 1\right]^2}{\left[(1 + \underline{\epsilon}_d)(1 + \underline{\epsilon}_f)(1 + \underline{\epsilon}_s)\right]^2}.$$

*Further if $\mathbf{H}_f^{(q)}$ and $\mathbf{SHS}^\top$ are commutative,*

$$\mathbb{E}\left[\left(\mathbf{v}^{(q)*}\right)^\top \frac{1}{N\gamma}\left[\mathbf{I} - \left(\mathbf{I} - \gamma\mathbf{H}_f^{(q)}\right)^N\right]\left(\mathbf{H}_f^{(q)}\right)^{-1}\left(\mathbf{H}_f^{(q)} - \mathbf{SHS}^\top\right)\mathbf{v}^{(q)*}\right]$$

$$\gtrsim \frac{1}{N(1 + \bar{\epsilon}_d)(1 + \bar{\epsilon}_f)(1 + \bar{\epsilon}_s)} \frac{(1 + \underline{\epsilon}_f)(1 + \underline{\epsilon}_d)(1 + \underline{\epsilon}_s) - 1}{(1 + \underline{\epsilon}_f)(1 + \underline{\epsilon}_d)(1 + \underline{\epsilon}_s)}.$$

*Proof.* Regarding the first inequality, noticing that

$$\left[(1 + \underline{\epsilon}_d)(1 + \underline{\epsilon}_f)(1 + \underline{\epsilon}_s) - 1\right]\mathbf{SHS}^\top \preceq \mathbf{D},$$

by Assumption 3.6,

$$\begin{aligned}
\mathbb{E}_{\mathbf{w}^*} \|\mathbf{w}^*\|_{\mathbf{S}_1}^2 &= \mathrm{tr}(\mathbf{S}_1) \\
&\geq \frac{\left[(1 + \underline{\epsilon}_d)(1 + \underline{\epsilon}_f)(1 + \underline{\epsilon}_s) - 1\right]^2}{\left[(1 + \underline{\epsilon}_d)(1 + \underline{\epsilon}_f)(1 + \underline{\epsilon}_s)\right]^2}\mathrm{tr}\left(\mathbf{SH}^2\mathbf{S}^\top\left(\mathbf{SHS}^\top\right)^{-1}\right) \\
&\gtrsim \frac{\left[(1 + \underline{\epsilon}_d)(1 + \underline{\epsilon}_f)(1 + \underline{\epsilon}_s) - 1\right]^2}{\left[(1 + \underline{\epsilon}_d)(1 + \underline{\epsilon}_f)(1 + \underline{\epsilon}_s)\right]^2},
\end{aligned}$$

where the first inequality holds by Lemma F.5, and the last inequality holds by Lemma G.1 and Von Neumann's trace inequality:

$$\mathrm{tr}\left(\mathbf{SH}^2\mathbf{S}^\top(\mathbf{SHS}^\top)^{-1}\right) \gtrsim \sum_i i^{-a} \approx 1.$$

Regarding the second inequality, by Assumption 3.6,

$$\begin{aligned}
&\mathbb{E}\left[\left(\mathbf{v}^{(q)*}\right)^\top \frac{1}{N\gamma}\left[\mathbf{I} - \left(\mathbf{I} - \gamma\mathbf{H}_f^{(q)}\right)^N\right]\left(\mathbf{H}_f^{(q)}\right)^{-1}\left(\mathbf{H}_f^{(q)} - \mathbf{SHS}^\top\right)\mathbf{v}^{(q)*}\right] \\
&= \mathbb{E}\left[\mathbf{w}^{*\top}\mathbf{HS}^\top(\mathbf{H}_f^{(q)})^{-1}\frac{1}{N\gamma}\left[\mathbf{I} - \left(\mathbf{I} - \gamma\mathbf{H}_f^{(q)}\right)^N\right]\left(\mathbf{H}_f^{(q)}\right)^{-1}\left(\mathbf{H}_f^{(q)} - \mathbf{SHS}^\top\right)(\mathbf{H}_f^{(q)})^{-1}\mathbf{SHw}^*\right] \quad (70) \\
&= \mathrm{tr}\left(\mathbf{HS}^\top(\mathbf{H}_f^{(q)})^{-1}\frac{1}{N\gamma}\left[\mathbf{I} - \left(\mathbf{I} - \gamma\mathbf{H}_f^{(q)}\right)^N\right]\left(\mathbf{H}_f^{(q)}\right)^{-1}\left(\mathbf{H}_f^{(q)} - \mathbf{SHS}^\top\right)(\mathbf{H}_f^{(q)})^{-1}\mathbf{SH}\right).
\end{aligned}$$

Noticing that

$$(1 + \underline{\epsilon}_f)(1 + \underline{\epsilon}_d)(1 + \underline{\epsilon}_s)\mathbf{SHS}^\top \preceq \mathbf{H}_f^{(q)} \preceq (1 + \bar{\epsilon}_f)(1 + \bar{\epsilon}_d)(1 + \bar{\epsilon}_s)\mathbf{SHS}^\top,$$

and $\mathbf{H}_f^{(q)}$ and $\mathbf{SHS}^\top$ are commutative, it holds

$$\begin{aligned}
&\mathrm{tr}\left(\mathbf{HS}^\top(\mathbf{H}_f^{(q)})^{-1}\frac{1}{N\gamma}\left[\mathbf{I} - \left(\mathbf{I} - \gamma\mathbf{H}_f^{(q)}\right)^N\right]\left(\mathbf{H}_f^{(q)}\right)^{-1}\left(\mathbf{H}_f^{(q)} - \mathbf{SHS}^\top\right)(\mathbf{H}_f^{(q)})^{-1}\mathbf{SH}\right) \\
&\geq \frac{(1 + \underline{\epsilon}_f)(1 + \underline{\epsilon}_d)(1 + \underline{\epsilon}_s) - 1}{(1 + \underline{\epsilon}_f)(1 + \underline{\epsilon}_d)(1 + \underline{\epsilon}_s)}\mathrm{tr}\left(\mathbf{HS}^\top(\mathbf{H}_f^{(q)})^{-1}\frac{1}{N\gamma}\left[\mathbf{I} - \left(\mathbf{I} - \gamma\mathbf{H}_f^{(q)}\right)^N\right](\mathbf{H}_f^{(q)})^{-1}\mathbf{SH}\right) \\
&\geq \frac{(1 + \underline{\epsilon}_f)(1 + \underline{\epsilon}_d)(1 + \underline{\epsilon}_s) - 1}{(1 + \underline{\epsilon}_f)(1 + \underline{\epsilon}_d)(1 + \underline{\epsilon}_s)}\mathrm{tr}\left((\mathbf{H}_f^{(q)})^{-\frac{1}{2}}\mathbf{SH}^2\mathbf{S}^\top(\mathbf{H}_f^{(q)})^{-\frac{1}{2}}\right) \\
&\quad \cdot\mu_{\min}\left((\mathbf{H}_f^{(q)})^{-\frac{1}{2}}\frac{1}{N\gamma}\left[\mathbf{I} - \left(\mathbf{I} - \gamma\mathbf{H}_f^{(q)}\right)^N\right](\mathbf{H}_f^{(q)})^{-\frac{1}{2}}\right).
\end{aligned}$$
(71)

Firstly, regarding

$$\mu_{\min}\left((\mathbf{H}_f^{(q)})^{-\frac{1}{2}}\frac{1}{N\gamma}\left[\mathbf{I}-\left(\mathbf{I}-\gamma\mathbf{H}_f^{(q)}\right)^N\right](\mathbf{H}_f^{(q)})^{-\frac{1}{2}}\right)=\frac{1}{N}\min_i\frac{1-\left(1-\gamma\tilde{\lambda}_i^{(q)}\right)^N}{\gamma\tilde{\lambda}_i^{(q)}},$$

note that $f(x)=\frac{1-(1-x)^N}{x}$ is decreasing in $(0,1)$, it holds

$$\mu_{\min}\left((\mathbf{H}_f^{(q)})^{-\frac{1}{2}}\frac{1}{N\gamma}\left[\mathbf{I}-\left(\mathbf{I}-\gamma\mathbf{H}_f^{(q)}\right)^N\right](\mathbf{H}_f^{(q)})^{-\frac{1}{2}}\right)\geq\frac{1}{N}. \tag{72}$$

Secondly, by Von Neumann's trace inequality, Lemma G.1 and Lemma G.2,

$$\operatorname{tr}\left((\mathbf{H}_f^{(q)})^{-\frac{1}{2}}\mathbf{S}\mathbf{H}^2\mathbf{S}^\top(\mathbf{H}_f^{(q)})^{-\frac{1}{2}}\right)\geq\sum_i\frac{\mu_i(\mathbf{S}\mathbf{H}^2\mathbf{S}^\top)}{\tilde{\lambda}_i^{(q)}}$$

$$\gtrsim\sum_i\frac{i^{-2a}}{(1+\bar{\epsilon}_d)(1+\bar{\epsilon}_f)(1+\bar{\epsilon}_s)i^{-a}} \tag{73}$$

$$\approx\frac{1}{(1+\bar{\epsilon}_d)(1+\bar{\epsilon}_f)(1+\bar{\epsilon}_s)}.$$

Hence, together with (70), (71), (72) and (73),

$$\mathbb{E}\left[\left(\mathbf{v}^{(q)*}\right)^\top\frac{1}{N\gamma}\left[\mathbf{I}-\left(\mathbf{I}-\gamma\mathbf{H}_f^{(q)}\right)^N\right]\left(\mathbf{H}_f^{(q)}\right)^{-1}\left(\mathbf{H}_f^{(q)}-\mathbf{S}\mathbf{H}\mathbf{S}^\top\right)\mathbf{v}^{(q)*}\right]$$

$$\gtrsim\frac{1}{N(1+\bar{\epsilon}_d)(1+\bar{\epsilon}_f)(1+\bar{\epsilon}_s)}\frac{(1+\underline{\epsilon}_f)(1+\underline{\epsilon}_d)(1+\underline{\epsilon}_s)-1}{(1+\underline{\epsilon}_f)(1+\underline{\epsilon}_d)(1+\underline{\epsilon}_s)}.$$

$\square$

**Lemma D.17** (Additive Error under additive quantization, a lower bound). *Under Assumption 3.1, 3.3, 3.6, for any $i\in\{s,d,f\}$, if there exist $(\bar{\epsilon}_i,\underline{\epsilon}_i)$ such that quantization $\mathcal{Q}_i$ is $(\bar{\epsilon}_i,\underline{\epsilon}_i)$-additive, then with probability at least $1-e^{-\Omega(M)}$,*

$$\mathbb{E}_{\mathbf{w}^*}\|\mathbf{w}^*\|_{\mathbf{S}_1}^2\gtrsim\frac{\left(\underline{\epsilon}_s+\underline{\epsilon}_f+\underline{\epsilon}_s\underline{\epsilon}_d p+\underline{\epsilon}_d\frac{p}{M}\right)^2}{\left(1+\underline{\epsilon}_s+\underline{\epsilon}_f+\underline{\epsilon}_s\underline{\epsilon}_d p+\underline{\epsilon}_d\frac{p}{M}\right)^2}.$$

*Further if $\mathbf{H}_f^{(q)}$ and $\mathbf{S}\mathbf{H}\mathbf{S}^\top$ are commutative,*

$$\mathbb{E}\left[\left(\mathbf{v}^{(q)*}\right)^\top\frac{1}{N\gamma}\left[\mathbf{I}-\left(\mathbf{I}-\gamma\mathbf{H}_f^{(q)}\right)^N\right]\left(\mathbf{H}_f^{(q)}\right)^{-1}\left(\mathbf{H}_f^{(q)}-\mathbf{S}\mathbf{H}\mathbf{S}^\top\right)\mathbf{v}^{(q)*}\right]$$

$$\gtrsim\frac{1}{N}\frac{M^{-a}}{M^{-a}+\bar{\epsilon}_f+\bar{\epsilon}_s(\bar{\epsilon}_d p+1)+\bar{\epsilon}_d\frac{p}{M}}\frac{\underline{\epsilon}_s+\underline{\epsilon}_s\underline{\epsilon}_d p+\underline{\epsilon}_f+\underline{\epsilon}_d\frac{p}{M}}{\underline{\epsilon}_s\underline{\epsilon}_d p+\underline{\epsilon}_f+\underline{\epsilon}_d\frac{p}{M}+1+\underline{\epsilon}_s}.$$

*Proof.* Regarding the first inequality, noticing that with probability at least $1-e^{-\Omega(M)}$,

$$\mathbf{D}\succsim\underbrace{\left(\underline{\epsilon}_s+\underline{\epsilon}_f+\underline{\epsilon}_s\underline{\epsilon}_d p+\underline{\epsilon}_d\frac{p}{M}\right)}_{\delta_{\min}}\mathbf{I},$$

by Assumption 3.6 we have

$$\mathbb{E}_{\mathbf{w}^*}\|\mathbf{w}^*\|_{\mathbf{S}_1}^2=\operatorname{tr}(\mathbf{S}_1)$$

$$\geq\frac{\delta_{\min}^2}{(\mu_{\max}(\mathbf{S}\mathbf{H}\mathbf{S}^\top)+\delta_{\min})^2}\operatorname{tr}\left(\mathbf{H}\mathbf{S}^\top\left(\mathbf{S}\mathbf{H}\mathbf{S}^\top\right)^{-1}\mathbf{S}\mathbf{H}\right)$$

$$\approx\frac{\delta_{\min}^2}{(1+\delta_{\min})^2}\operatorname{tr}\left(\mathbf{H}\mathbf{S}^\top\left(\mathbf{S}\mathbf{H}\mathbf{S}^\top\right)^{-1}\mathbf{S}\mathbf{H}\right)$$

$$\gtrsim\frac{\left(\underline{\epsilon}_s+\underline{\epsilon}_f+\underline{\epsilon}_s\underline{\epsilon}_d p+\underline{\epsilon}_d\frac{p}{M}\right)^2}{\left(1+\underline{\epsilon}_s+\underline{\epsilon}_f+\underline{\epsilon}_s\underline{\epsilon}_d p+\underline{\epsilon}_d\frac{p}{M}\right)^2},$$

where the first inequality holds by Lemma F.6, the last equality holds by Lemma G.1 and the last inequality holds by Lemma G.1 and the Von Neumann's trace inequality:

$$\text{tr}\left(\mathbf{S}\mathbf{H}^2\mathbf{S}^\top(\mathbf{S}\mathbf{H}\mathbf{S}^\top)^{-1}\right) \gtrsim \sum_i i^{-a} \asymp 1.$$

Regarding the second inequality, noticing that

$$\mathbf{S}\mathbf{H}\mathbf{S}^\top + \left(\underline{\epsilon}_s + \underline{\epsilon}_f + \underline{\epsilon}_s\underline{\epsilon}_d p + \underline{\epsilon}_d\frac{p}{M}\right)\mathbf{I} \precsim \mathbf{H}_f^{(q)} \precsim \mathbf{S}\mathbf{H}\mathbf{S}^\top + \left(\overline{\epsilon}_s + \overline{\epsilon}_f + \overline{\epsilon}_s\overline{\epsilon}_d p + \overline{\epsilon}_d\frac{p}{M}\right)\mathbf{I},$$

and $\mathbf{H}_f^{(q)}$ and $\mathbf{S}\mathbf{H}\mathbf{S}^\top$ are commutative, it holds

$$\text{tr}\left(\mathbf{H}\mathbf{S}^\top(\mathbf{H}_f^{(q)})^{-1}\frac{1}{N\gamma}\left[\mathbf{I}-\left(\mathbf{I}-\gamma\mathbf{H}_f^{(q)}\right)^N\right]\left(\mathbf{H}_f^{(q)}\right)^{-1}\left(\mathbf{H}_f^{(q)}-\mathbf{S}\mathbf{H}\mathbf{S}^\top\right)(\mathbf{H}_f^{(q)})^{-1}\mathbf{S}\mathbf{H}\right)$$

$$\geq \text{tr}\left((\mathbf{H}_f^{(q)})^{-\frac{1}{2}}\mathbf{S}\mathbf{H}^2\mathbf{S}^\top(\mathbf{H}_f^{(q)})^{-\frac{1}{2}}\right)\cdot\mu_{\min}\left((\mathbf{H}_f^{(q)})^{-\frac{1}{2}}\left(\mathbf{H}_f^{(q)}-\mathbf{S}\mathbf{H}\mathbf{S}^\top\right)(\mathbf{H}_f^{(q)})^{-\frac{1}{2}}\right) \quad (74)$$

$$\cdot\mu_{\min}\left((\mathbf{H}_f^{(q)})^{-\frac{1}{2}}\frac{1}{N\gamma}\left[\mathbf{I}-\left(\mathbf{I}-\gamma\mathbf{H}_f^{(q)}\right)^N\right](\mathbf{H}_f^{(q)})^{-\frac{1}{2}}\right).$$

Firstly, by Lemma F.2 and Lemma G.1,

$$\mu_{\min}\left((\mathbf{H}_f^{(q)})^{-\frac{1}{2}}\left(\mathbf{H}_f^{(q)}-\mathbf{S}\mathbf{H}\mathbf{S}^\top\right)(\mathbf{H}_f^{(q)})^{-\frac{1}{2}}\right) \gtrsim \frac{\underline{\epsilon}_s + \underline{\epsilon}_s\underline{\epsilon}_d p + \underline{\epsilon}_f + \underline{\epsilon}_d\frac{p}{M}}{\underline{\epsilon}_s\underline{\epsilon}_d p + \underline{\epsilon}_f + \underline{\epsilon}_d\frac{p}{M} + 1 + \underline{\epsilon}_s}. \quad (75)$$

Secondly, by Von Neumann's trace inequality, Lemma G.1 and Lemma G.4,

$$\text{tr}\left((\mathbf{H}_f^{(q)})^{-\frac{1}{2}}\mathbf{S}\mathbf{H}^2\mathbf{S}^\top(\mathbf{H}_f^{(q)})^{-\frac{1}{2}}\right) \geq \sum_i \frac{\mu_i(\mathbf{S}\mathbf{H}^2\mathbf{S}^\top)}{\tilde{\lambda}_i^{(q)}}$$

$$\gtrsim \sum_i \frac{i^{-2a}}{i^{-a}+\overline{\epsilon}_f+\overline{\epsilon}_s(\overline{\epsilon}_d p+1)+\overline{\epsilon}_d\frac{p}{M}}$$

$$\gtrsim \min \frac{i^{-a}}{i^{-a}+\overline{\epsilon}_f+\overline{\epsilon}_s(\overline{\epsilon}_d p+1)+\overline{\epsilon}_d\frac{p}{M}} \quad (76)$$

$$= \frac{M^{-a}}{M^{-a}+\overline{\epsilon}_f+\overline{\epsilon}_s(\overline{\epsilon}_d p+1)+\overline{\epsilon}_d\frac{p}{M}}.$$

Hence, together with (70), (72), (74), (75) and (76),

$$\mathbb{E}\left[\left(\mathbf{v}^{(q)*}\right)^\top\frac{1}{N\gamma}\left[\mathbf{I}-\left(\mathbf{I}-\gamma\mathbf{H}_f^{(q)}\right)^N\right]\left(\mathbf{H}_f^{(q)}\right)^{-1}\left(\mathbf{H}_f^{(q)}-\mathbf{S}\mathbf{H}\mathbf{S}^\top\right)\mathbf{v}^{(q)*}\right]$$

$$\gtrsim \frac{1}{N}\frac{M^{-a}}{M^{-a}+\overline{\epsilon}_f+\overline{\epsilon}_s(\overline{\epsilon}_d p+1)+\overline{\epsilon}_d\frac{p}{M}}\frac{\underline{\epsilon}_s+\underline{\epsilon}_s\underline{\epsilon}_d p+\underline{\epsilon}_f+\underline{\epsilon}_d\frac{p}{M}}{\underline{\epsilon}_s\underline{\epsilon}_d p+\underline{\epsilon}_f+\underline{\epsilon}_d\frac{p}{M}+1+\underline{\epsilon}_s}.$$

$\square$

## D.7. Variance Lower Bounds under Power-Law Spectrum

### D.7.1. MULTIPLICATIVE QUANTIZATION

**Lemma D.18** (A variance lower bound under multiplicative quantization, power-law spectrum)**.** *For $i \in \{d, f, s, p, a, o\}$, if there exist constants $(\overline{\epsilon}_i, \underline{\epsilon}_i)$ such that $\mathcal{Q}_i$ is $(\overline{\epsilon}_i, \underline{\epsilon}_i)$-multiplicative, then under Assumption 3.1, Assumption 3.3 and Assumption 3.6, for sufficiently large $N > 500$, with probability at least $1 - e^{-\Omega(M)}$,*

$$\frac{k^*}{N} + N\gamma^2\sum_{i>k^*}(\tilde{\lambda}_i^{(q)})^2 \gtrsim \frac{\min\left\{M, \left[N\gamma(1+\underline{\epsilon}_f)(1+\underline{\epsilon}_d)(1+\underline{\epsilon}_s)\right]^{\frac{1}{a}}\right\}}{N}.$$

*Proof.* By Lemma G.3, with probability at least $1 - e^{-\Omega(M)}$, for $j \in [M]$,

$$\mu_j(\mathbf{H}_f^{(q)}) \gtrsim (1 + \underline{\epsilon}_f)(1 + \underline{\epsilon}_d)(1 + \underline{\epsilon}_s)j^{-a}.$$

Hence, denote $k_0^* = \max\{k : (1 + \underline{\epsilon}_f)(1 + \underline{\epsilon}_d)(1 + \underline{\epsilon}_s)k^{-a} \geq \frac{1}{N\gamma}\}$, then with probability at least $1 - e^{-\Omega(M)}$, it holds

$$
\begin{aligned}
\frac{k^*}{N} + N\gamma^2 \sum_{i > k^*} (\tilde{\lambda}_i^{(q)})^2 &\gtrsim \frac{k^*}{N} + N\gamma^2 [(1 + \underline{\epsilon}_f)(1 + \underline{\epsilon}_d)(1 + \underline{\epsilon}_s)]^2 \sum_{i > k^*} i^{-2a} \\
&\gtrsim \frac{k_0^*}{N} + N\gamma^2 [(1 + \underline{\epsilon}_f)(1 + \underline{\epsilon}_d)(1 + \underline{\epsilon}_s)]^2 \sum_{i > k_0^*} i^{-2a} \\
&\approx \frac{\min\left\{ M, \left[ N\gamma(1 + \underline{\epsilon}_f)(1 + \underline{\epsilon}_d)(1 + \underline{\epsilon}_s) \right]^{\frac{1}{a}} \right\}}{N}.
\end{aligned}
\tag{77}
$$

$\square$

### D.7.2. ADDITIVE QUANTIZATION

**Lemma D.19** (A variance lower bound under additive quantization, power-law spectrum). *For $i \in \{d, f, s, p, a, o\}$, if there exist constants $(\bar{\epsilon}_i, \underline{\epsilon}_i)$ such that $\mathcal{Q}_i$ is $(\bar{\epsilon}_i, \underline{\epsilon}_i)$-additive, then under Assumption 3.1, Assumption 3.3 and Assumption 3.6, for sufficiently large $N > 500$, with probability at least $1 - e^{-\Omega(M)}$,*

$$\frac{k^*}{N} + N\gamma^2 \sum_{i > k^*} \left( \tilde{\lambda}_i^{(q)} \right)^2 \gtrsim \frac{\underline{k}_{\mathrm{eff}} + \gamma^2 N^2 \left( \underline{\epsilon}_f + \underline{\epsilon}_s(1 + \underline{\epsilon}_d p) + \underline{\epsilon}_d \frac{p}{M} \right)^2 (M - \underline{k}_{\mathrm{eff}})}{N},$$

*where*

$$\underline{k}_{\mathrm{eff}} = \left[ M^{-a} \vee \left( \frac{1}{N\gamma} - \underline{\epsilon}_f - (1 + \underline{\epsilon}_d p)\underline{\epsilon}_s - \underline{\epsilon}_d \frac{p}{M} \right) \right]^{-\frac{1}{a}}.$$

*Proof.* By Lemma G.5, with probability at least $1 - e^{-\Omega(M)}$,

$$\mu_j(\mathbf{H}_f^{(q)}) \gtrsim j^{-a} + \underline{\epsilon}_f + \underline{\epsilon}_s(1 + \underline{\epsilon}_d p) + \underline{\epsilon}_d \frac{p}{M}.$$

Denote $k_0^* = \max\{j : j^{-a} + \underline{\epsilon}_f + \underline{\epsilon}_s(1 + \underline{\epsilon}_d p) + \underline{\epsilon}_d \frac{p}{M} \geq \frac{1}{N\gamma}\}$. Then

$$
\begin{aligned}
\frac{k^*}{N} + N\gamma^2 \sum_{i > k^*} \left( \tilde{\lambda}_i^{(q)} \right)^2 &\gtrsim \frac{k^*}{N} + N\gamma^2 \sum_{i > k^*} \left( i^{-a} + \underline{\epsilon}_f + \underline{\epsilon}_s(1 + \underline{\epsilon}_d p) + \underline{\epsilon}_d \frac{p}{M} \right)^2 \\
&\gtrsim \frac{k_0^*}{N} + N\gamma^2 \sum_{i > k_0^*} \left( i^{-a} + \underline{\epsilon}_f + \underline{\epsilon}_s(1 + \underline{\epsilon}_d p) + \underline{\epsilon}_d \frac{p}{M} \right)^2.
\end{aligned}
\tag{78}
$$

We then consider two cases to complete the proof.

- $M^{-a} + \underline{\epsilon}_f + \underline{\epsilon}_s(1 + \underline{\epsilon}_d p) + \underline{\epsilon}_d \frac{p}{M} < \frac{1}{N\gamma}$

  Denote

  $$N_{\mathrm{eff}}^{(A)} = \left( \frac{1}{N\gamma} - \underline{\epsilon}_f - (1 + \underline{\epsilon}_d p)\underline{\epsilon}_s - \underline{\epsilon}_d \frac{p}{M} \right)^{-\frac{1}{a}}.$$

  Then with probability at least $1 - e^{-\Omega(M)}$,

$$
\begin{aligned}
\frac{k_0^*}{N} + N\gamma^2 \sum_{i > k_0^*} \left( i^{-a} + \underline{\epsilon}_f + \underline{\epsilon}_s(1 + \underline{\epsilon}_d p) + \underline{\epsilon}_d \frac{p}{M} \right)^2 &\gtrsim \frac{k_0^*}{N} + N\gamma^2 \sum_{i > k_0^*} \left[ i^{-2a} + \left( \underline{\epsilon}_f + \underline{\epsilon}_s(1 + \underline{\epsilon}_d p) + \underline{\epsilon}_d \frac{p}{M} \right)^2 \right] \\
&\gtrsim \frac{N_{\mathrm{eff}}^{(A)} + N^2\gamma^2 \left( \underline{\epsilon}_f + \underline{\epsilon}_s(1 + \underline{\epsilon}_d p) + \underline{\epsilon}_d \frac{p}{M} \right)^2 \left( M - N_{\mathrm{eff}}^{(A)} \right)}{N}.
\end{aligned}
\tag{79}
$$

- $M^{-a} + \underline{\epsilon}_f + \underline{\epsilon}_s(1 + \underline{\epsilon}_d p) + \underline{\epsilon}_d \frac{p}{M} \geq \frac{1}{N\gamma}$

$$\frac{k_0^*}{N} + N\gamma^2 \sum_{i>k_0^*} \left(i^{-a} + \underline{\epsilon}_f + \underline{\epsilon}_s(1 + \underline{\epsilon}_d p) + \underline{\epsilon}_d \frac{p}{M}\right)^2 = \frac{M}{N}. \tag{80}$$

Hence, together with (78), (79) and (80), with probability at least $1 - e^{-\Omega(M)}$,

$$\frac{k^*}{N} + N\gamma^2 \sum_{i>k^*} \left(\tilde{\lambda}_i^{(q)}\right)^2 \gtrsim \frac{\underline{k}_{\text{eff}} + \gamma^2 N^2 \left(\underline{\epsilon}_f + \underline{\epsilon}_s(1 + \underline{\epsilon}_d p) + \underline{\epsilon}_d \frac{p}{M}\right)^2 (M - \underline{k}_{\text{eff}})}{N},$$

where

$$\underline{k}_{\text{eff}} = \left[M^{-a} \vee \left(\frac{1}{N\gamma} - \underline{\epsilon}_f - (1 + \underline{\epsilon}_d p)\underline{\epsilon}_s - \underline{\epsilon}_d \frac{p}{M}\right)\right]^{-\frac{1}{a}}.$$

$\square$

### D.8. Bias Lower Bounds under Power-Law Spectrum

**Lemma D.20.** *Under Assumption 3.3 and Assumption 3.6, with probability at least $1 - e^{-\Omega(M)}$,*

$$\mathbb{E}_{\mathbf{w}^*}\left[\|\mathbf{v}^{(q)*}\|^2_{\mathbf{H}_{f,k^*:\infty}^{(q)}}\right] \geq \sum_{i>k^*} (\tilde{\lambda}_i^{(q)})^{-1} \mu_i(\mathbf{SH}^2\mathbf{S}^\top).$$

*Proof.* Recall that

$$\mathbf{v}^{(q)*} = (\mathbf{H}_f^{(q)})^{-1}\mathbf{SHw}^*,$$

it follows that

$$\begin{aligned}
\mathbb{E}_{\mathbf{w}^*}\|\mathbf{v}^{(q)*}\|^2_{\mathbf{H}_{f,k^*:\infty}^{(q)}} &= \mathbb{E}_{\mathbf{w}^*}\text{tr}\left(\mathbf{H}_{f,k^*:\infty}^{(q)}(\mathbf{H}_f^{(q)})^{-1}\mathbf{SHw}^*\mathbf{w}^{*\top}\mathbf{HS}^\top(\mathbf{H}_f^{(q)})^{-1}\right) \\
&= \mathbb{E}_{\mathbf{w}^*}\text{tr}\left((\mathbf{H}_{f,k^*:\infty}^{(q)})^{-1}\mathbf{SHw}^*\mathbf{w}^{*\top}\mathbf{HS}^\top\right) \\
&= \text{tr}\left((\mathbf{H}_{f,k^*:\infty}^{(q)})^{-1}\mathbf{SH}^2\mathbf{S}^\top\right) \\
&\geq \sum_{i>k^*}(\tilde{\lambda}_i^{(q)})^{-1}\mu_i(\mathbf{SH}^2\mathbf{S}^\top),
\end{aligned} \tag{81}$$

where the last inequality holds by Von Neumann's trace inequality. $\square$

#### D.8.1. Multiplicative Quantization

**Lemma D.21** (A bias lower bound under multiplicative quantization, power-law spectrum). *For $i \in \{d, f, s, p, a, o\}$, if there exist constants $(\bar{\epsilon}_i, \underline{\epsilon}_i)$ such that $\mathcal{Q}_i$ is $(\bar{\epsilon}_i, \underline{\epsilon}_i)$-multiplicative, then under Assumption 3.1, 3.3 and 3.6, with probability at least $1 - e^{-\Omega(M)}$, if $[N\gamma(1 + \bar{\epsilon}_f)(1 + \bar{\epsilon}_d)(1 + \bar{\epsilon}_s)]^{1/a} \leq M/C$ for some constant $C > 0$, then*

$$\mathbb{E}_{\mathbf{w}^*}\left[\|\mathbf{v}^{(q)*}\|^2_{\mathbf{H}_{f,k^*:\infty}^{(q)}}\right] \gtrsim \frac{[N\gamma(1 + \bar{\epsilon}_f)(1 + \bar{\epsilon}_d)(1 + \bar{\epsilon}_s)]^{1/a-1}}{(1 + \bar{\epsilon}_f)(1 + \bar{\epsilon}_d)(1 + \bar{\epsilon}_s)}.$$

*Proof.* By Lemma D.20,

$$\mathbb{E}_{\mathbf{w}^*}\left[\|\mathbf{v}^{(q)*}\|^2_{\mathbf{H}_{f,k^*:\infty}^{(q)}}\right] \geq \sum_{i>k^*}(\tilde{\lambda}_i^{(q)})^{-1}\mu_i(\mathbf{SH}^2\mathbf{S}^\top). \tag{82}$$

By Lemma G.1, with probability at least $1 - e^{-\Omega(M)}$,

$$\mu_i(\mathbf{SH}^2\mathbf{S}^\top) \asymp i^{-2a}. \tag{83}$$

By Lemma G.2, with probability at least $1 - e^{-\Omega(M)}$,

$$\tilde{\lambda}_i^{(q)} \lesssim (1 + \bar{\epsilon}_f)(1 + \bar{\epsilon}_d)(1 + \bar{\epsilon}_s)i^{-a}. \tag{84}$$

Therefore, if $[N\gamma(1+\bar{\epsilon}_f)(1+\bar{\epsilon}_d)(1+\bar{\epsilon}_s)]^{1/a} \le M/C$, then

$$
\begin{aligned}
&\mathbb{E}_{\mathbf{w}^*}\left[\|\mathbf{v}^{(q)^*}\|^2_{\mathbf{H}^{(q)}_{f,k^*:\infty}}\right] \\
&\gtrsim \sum_{i>k^*} \frac{i^{-2a}}{(1+\bar{\epsilon}_f)(1+\bar{\epsilon}_d)(1+\bar{\epsilon}_s)i^{-a}} \\
&= \frac{1}{(1+\bar{\epsilon}_f)(1+\bar{\epsilon}_d)(1+\bar{\epsilon}_s)} \sum_{i>k^*} i^{-a} \\
&\approx \frac{1}{(1+\bar{\epsilon}_f)(1+\bar{\epsilon}_d)(1+\bar{\epsilon}_s)} (k^*)^{1-a} \\
&\gtrsim \frac{1}{(1+\bar{\epsilon}_f)(1+\bar{\epsilon}_d)(1+\bar{\epsilon}_s)} [N\gamma(1+\bar{\epsilon}_f)(1+\bar{\epsilon}_d)(1+\bar{\epsilon}_s)]^{1/a-1}.
\end{aligned}
$$

$\square$

### D.8.2. ADDITIVE QUANTIZATION

**Lemma D.22** (A bias lower bound under additive quantization, power-law spectrum). *For $i \in \{d, f, s, p, a, o\}$, if there exist constants $(\bar{\epsilon}_i, \underline{\epsilon}_i)$ such that $\mathcal{Q}_i$ is $(\bar{\epsilon}_i, \underline{\epsilon}_i)$-additive, then under Assumption 3.1, 3.3 and 3.6, with probability at least $1 - e^{-\Omega(M)}$, if $M^{-a} + \bar{\epsilon}_f + (1+\bar{\epsilon}_d p)\bar{\epsilon}_s + \bar{\epsilon}_d \frac{p}{M} \le \frac{C}{N\gamma}$ for some constant $C > 0$, then*

$$
\mathbb{E}_{\mathbf{w}^*}\left[\|\mathbf{v}^{(q)^*}\|^2_{\mathbf{H}^{(q)}_{f,k^*:\infty}}\right] \gtrsim \frac{M^{-a}}{M^{-a} + \bar{\epsilon}_f + (1+\bar{\epsilon}_d p)\bar{\epsilon}_s + \bar{\epsilon}_d \frac{p}{M}} \left(\frac{1}{N\gamma} - \left[\bar{\epsilon}_f + (1+\bar{\epsilon}_d p)\bar{\epsilon}_s + \bar{\epsilon}_d \frac{p}{M}\right]\right)^{1-1/a}.
$$

*Proof.* By Lemma D.20,

$$
\mathbb{E}_{\mathbf{w}^*}\left[\|\mathbf{v}^{(q)^*}\|^2_{\mathbf{H}^{(q)}_{f,k^*:\infty}}\right] \ge \sum_{i>k^*} (\tilde{\lambda}^{(q)}_i)^{-1} \mu_i(\mathbf{S}\mathbf{H}^2\mathbf{S}^\top). \tag{85}
$$

By Lemma G.1, with probability at least $1 - e^{-\Omega(M)}$,

$$
\mu_i(\mathbf{S}\mathbf{H}^2\mathbf{S}^\top) \approx i^{-2a}. \tag{86}
$$

By Lemma G.4, with probability at least $1 - e^{-\Omega(M)}$,

$$
\tilde{\lambda}^{(q)}_i \lesssim i^{-a} + \bar{\epsilon}_f + (1+p\bar{\epsilon}_d)\bar{\epsilon}_s + \bar{\epsilon}_d \frac{p}{M}. \tag{87}
$$

Therefore, if $M^{-a} + \bar{\epsilon}_f + (1+p\bar{\epsilon}_d)\bar{\epsilon}_s + \bar{\epsilon}_d \frac{p}{M} \le \frac{C}{N\gamma}$, then by denoting $\Delta = \bar{\epsilon}_f + (1+\bar{\epsilon}_d p)\bar{\epsilon}_s + \bar{\epsilon}_d \frac{p}{M}$, we have

$$
\begin{aligned}
\mathbb{E}_{\mathbf{w}^*}\left[\|\mathbf{v}^{(q)^*}\|^2_{\mathbf{H}^{(q)}_{f,k^*:\infty}}\right] &\gtrsim \sum_{i>k^*} \frac{i^{-2a}}{i^{-a} + \Delta} \\
&\ge \min \frac{i^{-a}}{i^{-a} + \Delta} \sum_{i>k^*} i^{-a} \\
&\approx \frac{M^{-a}}{M^{-a} + \bar{\epsilon}_f + (1+\bar{\epsilon}_d p)\bar{\epsilon}_s + \bar{\epsilon}_d \frac{p}{M}} \cdot (k^*)^{1-a} \\
&\ge \frac{M^{-a}}{M^{-a} + \bar{\epsilon}_f + (1+\bar{\epsilon}_d p)\bar{\epsilon}_s + \bar{\epsilon}_d \frac{p}{M}} \left(\frac{1}{N\gamma} - \Delta\right)^{1-1/a}.
\end{aligned}
$$

$\square$

### D.9. Population Risk Lower Bounds under Power-law Spectrum

#### D.9.1. MULTIPLICATIVE QUANTIZATION

**Theorem D.23.** *Suppose $\gamma < 1/\tilde{\lambda}_1^{(q)}$. For $i \in \{d, f, s, p, a, o\}$, if there exist constants $(\bar{\epsilon}_i, \underline{\epsilon}_i)$ such that $\mathcal{Q}_i$ is $(\bar{\epsilon}_i, \underline{\epsilon}_i)$-multiplicative, then under Assumption 3.1, 3.3, 3.4, 3.5 and 3.6,*

- Irreducible $:= \mathcal{R}(\mathbf{w}^*) = \frac{1}{2}\sigma^2$.

- *with probability at least $1 - e^{-\Omega(M)}$ over the randomness of $\mathbf{S}$, $\mathbb{E}_{\mathbf{w}^*}\mathrm{Approx} \gtrsim M^{1-a}$.*

- *for sufficiently large $N > 500$, with probability at least $1 - e^{-\Omega(M)}$ over the randomness of $\mathbf{S}$ [10],*

$$\mathbb{E}\mathrm{Excess} \gtrsim \mathrm{BiasError} + \mathrm{VarianceError} + \mathrm{AdditiveError},$$

*where*

$$\mathrm{VarianceError} \gtrsim \frac{\underline{\sigma}_{M,1}^2 + \underline{\sigma}_{M,2}^2}{(1 + \bar{\epsilon}_f)(1 + \bar{\epsilon}_d)(1 + \bar{\epsilon}_s)} \frac{\min\left\{M, \left[N\gamma(1 + \underline{\epsilon}_f)(1 + \underline{\epsilon}_d)(1 + \underline{\epsilon}_s)\right]^{\frac{1}{a}}\right\}}{N},$$

*and if $[N\gamma(1 + \bar{\epsilon}_f)(1 + \bar{\epsilon}_d)(1 + \bar{\epsilon}_s)]^{1/a} \le M/C$ for some constant $C > 0$,*

$$\mathrm{BiasError} \gtrsim \frac{[N\gamma(1 + \bar{\epsilon}_f)(1 + \bar{\epsilon}_d)(1 + \bar{\epsilon}_s)]^{1/a-1}}{\left[(1 + \bar{\epsilon}_f)(1 + \bar{\epsilon}_d)(1 + \bar{\epsilon}_s)\right]^2},$$

*and if $\mathbf{H}_f^{(q)}$ and $\mathbf{SHS}^\top$ are commutative,*

$$\mathrm{AdditiveError} \gtrsim \frac{\left[(1 + \underline{\epsilon}_d)(1 + \underline{\epsilon}_f)(1 + \underline{\epsilon}_s) - 1\right]^2}{\left[(1 + \underline{\epsilon}_d)(1 + \underline{\epsilon}_f)(1 + \underline{\epsilon}_s)\right]^2} + \frac{(1 + \underline{\epsilon}_f)(1 + \underline{\epsilon}_d)(1 + \underline{\epsilon}_s) - 1}{(1 + \underline{\epsilon}_f)(1 + \underline{\epsilon}_d)(1 + \underline{\epsilon}_s)} \frac{1}{N(1 + \bar{\epsilon}_d)(1 + \bar{\epsilon}_f)(1 + \bar{\epsilon}_s)},$$

*with*

$$\underline{\sigma}_{M,1}^2 = (1 + \underline{\epsilon}_o)\underline{\sigma}^2, \quad \underline{\sigma}_{M,2}^2 = \frac{\beta(1 + \underline{\epsilon}_o)\left(\underline{\epsilon}_p + (1 + \underline{\epsilon}_p)\underline{\epsilon}_a\right)}{(1 + \bar{\epsilon}_f)(1 + \bar{\epsilon}_d)(1 + \bar{\epsilon}_s)}.$$

*Proof.* The proof can be completed by (5), (6), Lemma B.2, (64), Theorem D.15, Lemma D.16, Lemma D.18, Lemma D.21 and noticing the following facts. Firstly,

$$\mu_{\min}\left((\mathbf{H}_f^{(q)})^{-1}\mathbf{SHS}^\top\right) \ge \frac{1}{(1 + \bar{\epsilon}_d)(1 + \bar{\epsilon}_f)(1 + \bar{\epsilon}_s)}.$$

Secondly, recall that $\mathbf{v}^{(q)*} = (\mathbf{H}_f^{(q)})^{-1}\mathbf{SHw}^*$, we separate two cases to lower bound

$$\mathbb{E}_{\mathbf{w}^*}P_{\mathrm{eff}} \gtrsim \mathbb{E}_{\mathbf{w}^*}\left[\left\|\mathbf{v}^{(q)*}\right\|_{\mathbf{H}_{f,0:k^*}^{(q)}}^2 + N\gamma\left\|\mathbf{v}^{(q)*}\right\|_{\mathbf{H}_{f,k^*:\infty}^{(q)}}^2 \sum_i (\tilde{\lambda}_i^{(q)})^2(1 - \gamma\tilde{\lambda}_i^{(q)})^{2N}\right].$$

- $k^* = M$

  In this case,

$$\mathbb{E}_{\mathbf{w}^*}P_{\mathrm{eff}} \gtrsim \mathbb{E}_{\mathbf{w}^*}\left\|\mathbf{v}^{(q)*}\right\|_{\mathbf{H}_f^{(q)}}^2 = \mathrm{tr}\left(\mathbf{HS}^\top(\mathbf{H}_f^{(q)})^{-1}\mathbf{SH}\right) \ge \frac{1}{(1 + \bar{\epsilon}_d)(1 + \bar{\epsilon}_f)(1 + \bar{\epsilon}_s)}\mathrm{tr}\left(\mathbf{SH}^2\mathbf{S}^\top(\mathbf{SHS}^\top)^{-1}\right).$$

---

[10] Here we take expectation on the prior $\mathbf{w}^*$.

By the Von Neumann's trace inequality and Lemma G.1, with probability at least $1 - e^{-\Omega(M)}$,

$$\text{tr}\left(\mathbf{S}\mathbf{H}^2\mathbf{S}^\top(\mathbf{S}\mathbf{H}\mathbf{S}^\top)^{-1}\right) \geq \sum_i i^{-a} \approx 1,$$

then with probability at least $1 - e^{-\Omega(M)}$,

$$\mathbb{E}_{\mathbf{w}^*}P_{\text{eff}} \gtrsim \frac{1}{(1+\bar{\epsilon}_d)(1+\bar{\epsilon}_f)(1+\bar{\epsilon}_s)}.$$

- $k^* < M$

In this case, by Lemma G.1, with probability at least $1 - e^{-\Omega(M)}$,

$$
\begin{aligned}
\mathbb{E}_{\mathbf{w}^*}P_{\text{eff}} &\gtrsim \mathbb{E}_{\mathbf{w}^*}\left[\left\|\mathbf{v}^{(q)*}\right\|^2_{\mathbf{H}^{(q)}_{f,0:k^*}} + N\gamma\left\|\mathbf{v}^{(q)*}\right\|^2_{\mathbf{H}^{(q)}_{f,k^*:\infty}}\sum_i(\tilde{\lambda}^{(q)}_i)^2(1-\gamma\tilde{\lambda}^{(q)}_i)^{2N}\right] \\
&\gtrsim \mathbb{E}_{\mathbf{w}^*}\left[\left\|\mathbf{v}^{(q)*}\right\|^2_{\mathbf{H}^{(q)}_{f,0:k^*}} + \frac{1}{N\gamma}\left\|\mathbf{v}^{(q)*}\right\|^2_{\mathbf{H}^{(q)}_{f,k^*:\infty}}\right] \\
&= \text{tr}\left(\left[(\mathbf{H}^{(q)}_{f,0:k^*})^{-1} + \frac{1}{N\gamma}(\mathbf{H}^{(q)}_{f,k^*:\infty})^{-1}\right]\mathbf{S}\mathbf{H}^2\mathbf{S}^\top\right) \\
&\geq \mu_{\min}\left((\mathbf{H}^{(q)}_{f,0:k^*})^{-1} + \frac{1}{N\gamma}(\mathbf{H}^{(q)}_{f,k^*:\infty})^{-1}\right)\text{tr}\left(\mathbf{S}\mathbf{H}^2\mathbf{S}^\top\right) \\
&= \min\left\{\frac{1}{\tilde{\lambda}^{(q)}_1}, \frac{1}{N\gamma}\frac{1}{\tilde{\lambda}^{(q)}_{k^*+1}}\right\}\text{tr}\left(\mathbf{S}\mathbf{H}^2\mathbf{S}^\top\right) \\
&\geq \min\left\{\frac{1}{\tilde{\lambda}^{(q)}_1}, 1\right\}\text{tr}\left(\mathbf{S}\mathbf{H}^2\mathbf{S}^\top\right) \\
&\geq \frac{1}{(1+\bar{\epsilon}_d)(1+\bar{\epsilon}_f)(1+\bar{\epsilon}_s)}\text{tr}\left(\mathbf{S}\mathbf{H}^2\mathbf{S}^\top\right) \\
&\approx \frac{1}{(1+\bar{\epsilon}_d)(1+\bar{\epsilon}_f)(1+\bar{\epsilon}_s)}.
\end{aligned}
$$

$\square$

### D.9.2. ADDITIVE QUANTIZATION

**Theorem D.24.** *Suppose $\gamma < 1/\tilde{\lambda}^{(q)}_1$. For $i \in \{d, f, s, p, a, o\}$, if there exist constants $(\bar{\epsilon}_i, \underline{\epsilon}_i)$ such that $\mathcal{Q}_i$ is $(\bar{\epsilon}_i, \underline{\epsilon}_i)$-additive, then under Assumption 3.1, 3.3, 3.4, 3.5 and 3.6,*

- Irreducible $:= \mathcal{R}(\mathbf{w}^*) = \frac{1}{2}\sigma^2$.

- *with probability at least $1 - e^{-\Omega(M)}$ over the randomness of $\mathbf{S}$, $\mathbb{E}_{\mathbf{w}^*}\text{Approx} \gtrsim M^{1-a}$.*

- *for sufficiently large $N > 500$, with probability at least $1 - e^{-\Omega(M)}$ over the randomness of $\mathbf{S}$ [11],*

$$\mathbb{E}\text{Excess} \gtrsim \text{BiasError} + \text{VarianceError} + \text{AdditiveError},$$

*where*

$$\text{VarianceError} \gtrsim \frac{M^{-a}}{M^{-a} + \bar{\epsilon}_f + \bar{\epsilon}_s(1+\bar{\epsilon}_d p) + \bar{\epsilon}_d\frac{p}{M}}\sigma^2_G\frac{k_{\text{eff}} + \gamma^2N^2\left(\underline{\epsilon}_f + \underline{\epsilon}_s(1+\underline{\epsilon}_d p) + \underline{\epsilon}_d\frac{p}{M}\right)^2(M-\underline{k}_{\text{eff}})}{N},$$

---

[11] Here we take expectation on the prior $\mathbf{w}^*$.

and if $M^{-a} + \bar{\epsilon}_f + (1 + \bar{\epsilon}_d p)\bar{\epsilon}_s + \bar{\epsilon}_d \frac{p}{M} \le \frac{C}{N\gamma}$ for some constant $C > 0$, then

$$\text{BiasError} \gtrsim \left( \frac{M^{-a}}{M^{-a} + \bar{\epsilon}_f + \bar{\epsilon}_s(1 + \bar{\epsilon}_d p) + \bar{\epsilon}_d \frac{p}{M}} \right)^2 \cdot \left( \frac{1}{N\gamma} - \left[ \bar{\epsilon}_f + (1 + \bar{\epsilon}_d p)\bar{\epsilon}_s + \bar{\epsilon}_d \frac{p}{M} \right] \right)^{1-1/a},$$

and if $\mathbf{H}_f^{(q)}$ and $\mathbf{SHS}^\top$ are commutative,

$$\text{AdditiveError} \gtrsim \frac{\left( \underline{\epsilon}_s + \underline{\epsilon}_f + \underline{\epsilon}_s\underline{\epsilon}_d p + \underline{\epsilon}_d \frac{p}{M} \right)^2}{\left( 1 + \underline{\epsilon}_s + \underline{\epsilon}_f + \underline{\epsilon}_s\underline{\epsilon}_d p + \underline{\epsilon}_d \frac{p}{M} \right)^2}$$
$$+ \frac{1}{N} \frac{M^{-a}}{M^{-a} + \bar{\epsilon}_f + \bar{\epsilon}_s(\bar{\epsilon}_d p + 1) + \bar{\epsilon}_d \frac{p}{M}} \frac{\underline{\epsilon}_s + \underline{\epsilon}_s\underline{\epsilon}_d p + \underline{\epsilon}_f + \underline{\epsilon}_d \frac{p}{M}}{\underline{\epsilon}_s\underline{\epsilon}_d p + \underline{\epsilon}_f + \underline{\epsilon}_d \frac{p}{M} + 1 + \underline{\epsilon}_s},$$

with

$$\underline{\sigma}_G^2 = \underline{\sigma}^2 + \underline{\epsilon}_a + \underline{\epsilon}_o + \beta\underline{\epsilon}_p \left[ 1 + p\underline{\epsilon}_d + M(\underline{\epsilon}_f + \underline{\epsilon}_s(\underline{\epsilon}_d p + 1)) \right],$$
$$\underline{k}_{\text{eff}} = \left[ M^{-a} \vee \left( \frac{1}{N\gamma} - \underline{\epsilon}_f - (1 + \underline{\epsilon}_d p)\underline{\epsilon}_s - \underline{\epsilon}_d \frac{p}{M} \right) \right]^{-\frac{1}{a}}.$$

*Proof.* The proof can be completed by (5), (6), Lemma B.2, (64), Theorem D.14, Lemma D.17, Lemma D.19, Lemma D.22 and noticing the following facts. Firstly, by Lemma G.1, with probability at least $1 - e^{-\Omega(M)}$,

$$\mu_{\min}\left( (\mathbf{H}_f^{(q)})^{-1}\mathbf{SHS}^\top \right) \gtrsim \frac{M^{-a}}{\bar{\epsilon}_f + \bar{\epsilon}_s\bar{\epsilon}_d p + \bar{\epsilon}_d \frac{p}{M} + M^{-a} + \bar{\epsilon}_s}.$$

Secondly, by Lemma G.1, with probability at least $1 - e^{-\Omega(M)}$,

$$\text{tr}\left( \mathbf{H}_f^{(q)} \right) \ge \text{tr}\left( \mathbf{SHS}^\top + (\underline{\epsilon}_f + \underline{\epsilon}_s(1 + \underline{\epsilon}_d p) + \underline{\epsilon}_d \frac{p}{M})\mathbf{I} \right) \approx 1 + M(\underline{\epsilon}_f + \underline{\epsilon}_s(1 + \underline{\epsilon}_d p)) + p\underline{\epsilon}_d.$$

$\square$

# E. Scaling Laws

## E.1. Multiplicative Quantization

Denote

$$\bar{\epsilon}_2^{(M)} = (1 + \bar{\epsilon}_o)\left( 1 + \frac{\bar{\epsilon}_p + (1 + \bar{\epsilon}_p)\bar{\epsilon}_a}{(1 + \underline{\epsilon}_d)(1 + \underline{\epsilon}_f)(1 + \underline{\epsilon}_s)} \right) - 1,$$

$$\underline{\epsilon}_2^{(M)} = (1 + \underline{\epsilon}_o)\left( 1 + \frac{\underline{\epsilon}_p + (1 + \underline{\epsilon}_p)\underline{\epsilon}_a}{(1 + \bar{\epsilon}_d)(1 + \bar{\epsilon}_f)(1 + \bar{\epsilon}_s)} \right) - 1,$$

$$\bar{\epsilon}_3^{(M)} = 1 - \frac{1}{(1 + \bar{\epsilon}_d)(1 + \bar{\epsilon}_f)(1 + \bar{\epsilon}_s)}, \quad \underline{\epsilon}_3^{(M)} = 1 - \frac{1}{(1 + \underline{\epsilon}_d)(1 + \underline{\epsilon}_f)(1 + \underline{\epsilon}_s)}.$$

**Theorem E.1.** *Suppose $\gamma < 1/\left( (1 + \tilde{\epsilon})\alpha\text{tr}\left( \mathbf{H}_f^{(q)} \right) \right)$. For any $i \in \{s, d, f, p, a, o\}$, if there exist $(\bar{\epsilon}_i, \underline{\epsilon}_i)$ such that quantization $\mathcal{Q}_i$ is $(\bar{\epsilon}_i, \underline{\epsilon}_i)$-multiplicative, then under Assumption 3.1, 3.3, 3.4, 3.5 and 3.6, if $\mathbf{H}_f^{(q)}$ and $\mathbf{SHS}^\top$ commute, with probability at least $1 - e^{-\Omega(M)}$,*

$$\mathbb{E}\mathcal{R}_M(\bar{\mathbf{v}}_N) \lesssim \frac{1}{M_{\text{eff}}^{a-1}} + \frac{1}{N_{\text{eff}}^{(a-1)/a}} + \sigma^2 + \left( \bar{\epsilon}_3^{(M)} \right)^2 + \left( 1 - \underline{\epsilon}_3^{(M)} \right)\bar{\epsilon}_3^{(M)}, \tag{88}$$

*where*

$$M_{\text{eff}} = M, \quad N_{\text{eff}} = N \left[ \frac{\left( 1 - \underline{\epsilon}_3^{(M)} \right)\left( 1 + \bar{\epsilon}_2^{(M)} \right)}{\left( 1 - \bar{\epsilon}_3^{(M)} \right)^{1/a}} \right]^{-\frac{a}{a-1}}.$$

*Proof.* By Theorem C.29, with probability at least $1 - e^{-\Omega(M)}$ over the randomness of $\mathbf{S}$,

$$\mathbb{E}\mathcal{R}_M(\bar{\mathbf{v}}_N) \lesssim \sigma^2 + M^{1-a} + \text{BiasError} + \text{VarianceError} + \text{AdditiveError},$$

where

$$\text{BiasError} \lesssim (1 - \underline{\epsilon}_3^{(M)})^2 \max\left\{ \left(\frac{N\gamma}{1 - \underline{\epsilon}_3^{(M)}}\right)^{\frac{1}{a}-1}, M^{1-a} \right\},$$

$$\text{VarianceError} \lesssim (1 - \underline{\epsilon}_3^{(M)})\left(1 + \bar{\epsilon}_2^{(M)}\right) \frac{\min\left\{ M, \left(\frac{N\gamma}{1-\bar{\epsilon}_3^{(M)}}\right)^{1/a} \right\}}{N},$$

$$\text{AdditiveError} \lesssim \left(\bar{\epsilon}_3^{(M)}\right)^2 + \left(1 - \underline{\epsilon}_3^{(M)}\right)\bar{\epsilon}_3^{(M)}.$$

Denote

$$\mathcal{R}_0 = \mathbb{E}\mathcal{R}_M(\bar{\mathbf{v}}_N) - \left[\sigma^2 + \left(\bar{\epsilon}_3^{(M)}\right)^2 + \left(1 - \underline{\epsilon}_3^{(M)}\right)\bar{\epsilon}_3^{(M)}\right].$$

- $M^a \geq \frac{N\gamma}{1-\bar{\epsilon}_3^{(M)}}$

  In this case,

  $$\mathcal{R}_0 \lesssim M^{1-a} + \left(1 - \underline{\epsilon}_3^{(M)}\right)\frac{1 + \bar{\epsilon}_2^{(M)}}{\left(1 - \bar{\epsilon}_3^{(M)}\right)^{1/a}}N^{1/a-1}.$$

- $\frac{N\gamma}{1-\underline{\epsilon}_3^{(M)}} \leq M^a < \frac{N\gamma}{1-\bar{\epsilon}_3^{(M)}}$

  In this case,

  $$\mathcal{R}_0 \lesssim M^{1-a} + \left(1 - \underline{\epsilon}_3^{(M)}\right)\left(1 + \bar{\epsilon}_2^{(M)}\right)\frac{M}{N} + \left(1 - \underline{\epsilon}_3^{(M)}\right)^{3-1/a}N^{1/a-1}$$

  $$\lesssim M^{1-a} + \left(1 - \underline{\epsilon}_3^{(M)}\right)\frac{1 + \bar{\epsilon}_2^{(M)}}{\left(1 - \bar{\epsilon}_3^{(M)}\right)^{1/a}}N^{1/a-1} + \left(1 - \underline{\epsilon}_3^{(M)}\right)^{3-1/a}N^{1/a-1}$$

  $$\lesssim M^{1-a} + \left(1 - \underline{\epsilon}_3^{(M)}\right)\frac{1 + \bar{\epsilon}_2^{(M)}}{\left(1 - \bar{\epsilon}_3^{(M)}\right)^{1/a}}N^{1/a-1}.$$

- $M^a < \frac{N\gamma}{1-\underline{\epsilon}_3^{(M)}}$

  In this case,

  $$\mathcal{R}_0 \lesssim M^{1-a} + \left(1 - \underline{\epsilon}_3^{(M)}\right)\left(1 + \bar{\epsilon}_2^{(M)}\right)\frac{M}{N} \lesssim M^{1-a} + N^{1/a-1}\left(1 - \underline{\epsilon}_3^{(M)}\right)\left(1 + \bar{\epsilon}_2^{(M)}\right)\left(1 - \underline{\epsilon}_3^{(M)}\right)^{-1/a}.$$

Summarizing,

$$\mathcal{R}_0 \lesssim M^{1-a} + N^{1/a-1}\left[\frac{\left(1 - \underline{\epsilon}_3^{(M)}\right)\left(1 + \bar{\epsilon}_2^{(M)}\right)}{\left(1 - \bar{\epsilon}_3^{(M)}\right)^{1/a}}\right].$$

$\square$

**Theorem E.2.** *Suppose $\gamma < 1/\tilde{\lambda}_1^{(q)}$. For $i \in \{d, f, s, p, a, o\}$, if there exist constants $(\bar{\epsilon}_i, \underline{\epsilon}_i)$ such that $\mathcal{Q}_i$ is $(\bar{\epsilon}_i, \underline{\epsilon}_i)$-multiplicative, then under Assumption 3.1, 3.3, 3.4, 3.5 and 3.6, for sufficiently large $N > 500$, if $\mathbf{H}_f^{(q)}$ and $\mathbf{SHS}^\top$ are*

*commutative, with probability at least $1 - e^{-\Omega(M)}$ over the randomness of $\mathbf{S}$, it holds* [12]

$$\mathbb{ER}_M(\bar{\mathbf{v}}_N) \gtrsim \frac{1}{M_{\text{eff}}^{a-1}} + \frac{1}{N_{\text{eff}}^{(a-1)/a}} + \sigma^2 + \left(\underline{\epsilon}_3^{(M)}\right)^2 + \frac{\epsilon_3^{(M)}}{N}\left(1 - \bar{\epsilon}_3^{(M)}\right), \tag{89}$$

*where*

$$M_{\text{eff}} = M, \quad N_{\text{eff}} = N\left[\left(1 - \bar{\epsilon}_3^{(M)}\right)\left(1 + \underline{\epsilon}_2^{(M)}\right)\left(1 - \underline{\epsilon}_3^{(M)}\right)^{-1/a}\right]^{-\frac{a}{a-1}}.$$

*Proof.* By Theorem D.23, with probability at least $1 - e^{-\Omega(M)}$,

$$\mathbb{ER}_M(\bar{\mathbf{v}}_N) \gtrsim \sigma^2 + M^{1-a} + \text{BiasError} + \text{VarianceError} + \text{AdditiveError},$$

where

$$\text{VarianceError} \gtrsim \left(1 - \bar{\epsilon}_3^{(M)}\right)\left(1 + \underline{\epsilon}_2^{(M)}\right)\frac{\min\left\{M, \left(\frac{N\gamma}{1-\underline{\epsilon}_3^{(M)}}\right)^{\frac{1}{a}}\right\}}{N},$$

and if $\left(\frac{N\gamma}{1-\bar{\epsilon}_3^{(M)}}\right)^{1/a} \leq M/C$ for some constant $C > 0$,

$$\text{BiasError} \gtrsim \left(1 - \bar{\epsilon}_3^{(M)}\right)^2\left(\frac{N\gamma}{1 - \bar{\epsilon}_3^{(M)}}\right)^{1/a-1},$$

and if $\mathbf{H}_f^{(q)}$ and $\mathbf{SHS}^\top$ are commutative,

$$\text{AdditiveError} \gtrsim \left(\underline{\epsilon}_3^{(M)}\right)^2 + \frac{\epsilon_3^{(M)}}{N}\left(1 - \bar{\epsilon}_3^{(M)}\right).$$

Denote

$$\mathcal{R}_0 = \mathbb{ER}_M(\bar{\mathbf{v}}_N) - \sigma^2 - \left[\left(\underline{\epsilon}_3^{(M)}\right)^2 + \frac{\epsilon_3^{(M)}}{N}\left(1 - \bar{\epsilon}_3^{(M)}\right)\right].$$

It holds

$$\mathcal{R}_0 \gtrsim M^{1-a} + \left(1 - \bar{\epsilon}_3^{(M)}\right)\left(1 + \underline{\epsilon}_2^{(M)}\right)\frac{\min\left\{M, \left(\frac{N\gamma}{1-\underline{\epsilon}_3^{(M)}}\right)^{\frac{1}{a}}\right\}}{N}.$$

Let

$$\frac{1}{M^{a-1}} \geq \left(1 - \bar{\epsilon}_3^{(M)}\right)\left(1 + \underline{\epsilon}_2^{(M)}\right)\left(1 - \underline{\epsilon}_3^{(M)}\right)^{-1/a}(N\gamma)^{\frac{1}{a}-1},$$

solving that

$$M \leq (N\gamma)^{\frac{1}{a}}\left[\left(1 - \bar{\epsilon}_3^{(M)}\right)\left(1 + \underline{\epsilon}_2^{(M)}\right)\left(1 - \underline{\epsilon}_3^{(M)}\right)^{-1/a}\right]^{\frac{1}{1-a}}.$$

---

[12]This scaling law holds in the positive regime

$$\Omega = \left\{(M,N) : \left(M \leq (N\gamma)^{\frac{1}{a}}\left[\left(1 - \bar{\epsilon}_3^{(M)}\right)\left(1 + \underline{\epsilon}_2^{(M)}\right)\left(1 - \underline{\epsilon}_3^{(M)}\right)^{-1/a}\right]^{\frac{1}{1-a}}\right) \vee \left(M \geq \left(\frac{N\gamma}{1 - \underline{\epsilon}_3^{(M)}}\right)^{\frac{1}{a}}\right)\right\}.$$

- $M \geq \left(\frac{N\gamma}{1-\underline{\epsilon}_3^{(M)}}\right)^{\frac{1}{a}}$

  In this case,

  $$\mathcal{R}_0 \gtrsim M^{1-a} + \left(1 - \bar{\epsilon}_3^{(M)}\right)\left(1 + \underline{\epsilon}_2^{(M)}\right)\left(1 - \underline{\epsilon}_3^{(M)}\right)^{-1/a} N^{1/a-1}.$$

- $M \leq (N\gamma)^{\frac{1}{a}}\left[\left(1 - \bar{\epsilon}_3^{(M)}\right)\left(1 + \underline{\epsilon}_2^{(M)}\right)\left(1 - \underline{\epsilon}_3^{(M)}\right)^{-1/a}\right]^{\frac{1}{1-a}}$

  In this case,

  $$\mathcal{R}_0 \gtrsim M^{1-a} \asymp M^{1-a} + \left(1 - \bar{\epsilon}_3^{(M)}\right)\left(1 + \underline{\epsilon}_2^{(M)}\right)\left(1 - \underline{\epsilon}_3^{(M)}\right)^{-1/a} N^{1/a-1}.$$

$\square$

### E.2. Additive Quantization

Denote

$$\bar{\epsilon}_2^{(A)} = \bar{\epsilon}_a + \bar{\epsilon}_o + \bar{\epsilon}_p\left[1 + p\bar{\epsilon}_d + M(\bar{\epsilon}_f + \bar{\epsilon}_s + \bar{\epsilon}_s\bar{\epsilon}_d p)\right],$$

$$\underline{\epsilon}_2^{(A)} = \underline{\epsilon}_a + \underline{\epsilon}_o + \underline{\epsilon}_p\left[1 + p\underline{\epsilon}_d + M(\underline{\epsilon}_f + \underline{\epsilon}_s\underline{\epsilon}_d p + \underline{\epsilon}_s)\right],$$

$$\bar{\epsilon}_3^{(A)} = \frac{M^a\left(\bar{\epsilon}_f + \bar{\epsilon}_s(1 + \bar{\epsilon}_d p) + \bar{\epsilon}_d \frac{p}{M}\right)}{1 + M^a\left(\bar{\epsilon}_f + \bar{\epsilon}_s(1 + \bar{\epsilon}_d p) + \bar{\epsilon}_d \frac{p}{M}\right)},$$

$$\underline{\epsilon}_3^{(A)} = \frac{\underline{\epsilon}_f + (1 + \underline{\epsilon}_d p)\underline{\epsilon}_s + \underline{\epsilon}_d \frac{p}{M}}{1 + \underline{\epsilon}_f + (1 + \underline{\epsilon}_d p)\underline{\epsilon}_s + \underline{\epsilon}_d \frac{p}{M}}.$$

**Theorem E.3.** *Suppose $\gamma < 1/\left(\alpha\mathrm{tr}\left(\mathbf{H}_f^{(q)}\right)\right)$. For any $i \in \{s, d, f, p, a, o\}$, if there exist $(\bar{\epsilon}_i, \underline{\epsilon}_i)$ such that quantization $\mathcal{Q}_i$ is $(\bar{\epsilon}_i, \underline{\epsilon}_i)$-additive, then under Assumption 3.1, 3.3, 3.4, 3.5 and 3.6, if $\mathbf{H}_f^{(q)}$ and $\mathbf{SHS}^\top$ commute, with probability at least $1 - e^{-\Omega(M)}$,*

$$\mathbb{E}\mathcal{R}_M(\bar{\mathbf{v}}_N) \lesssim \frac{1}{M_{\mathrm{eff}}^{a-1}} + \frac{1}{N_{\mathrm{eff}}^{(a-1)/a}} + \sigma^2 + \left(\bar{\epsilon}_3^{(A)}\right)^2 + \left(1 - \underline{\epsilon}_3^{(A)}\right)\bar{\epsilon}_3^{(A)}, \tag{90}$$

*where*

$$M_{\mathrm{eff}} = M, \quad N_{\mathrm{eff}} = N\left[\left(1 - \underline{\epsilon}_3^{(A)}\right)\frac{1 + \bar{\epsilon}_2^{(A)}}{\left(1 - \bar{\epsilon}_3^{(A)}\right)^{1/a}}\left(1 + \left(\bar{\epsilon}_3^{(A)}\right)^2\right)\right]^{-\frac{a}{a-1}}.$$

*The scaling law can be also expressed as:*

$$M_{\mathrm{eff}} = M\left[1 + \left(1 - \underline{\epsilon}_3^{(A)}\right)\left(1 + \bar{\epsilon}_2^{(A)}\right)\frac{\left(\bar{\epsilon}_3^{(A)}\right)^2}{1 - \bar{\epsilon}_3^{(A)}}\right]^{-\frac{1}{a-1}}, \quad N_{\mathrm{eff}} = N\left[\left(1 - \underline{\epsilon}_3^{(A)}\right)\frac{1 + \bar{\epsilon}_2^{(A)}}{\left(1 - \bar{\epsilon}_3^{(A)}\right)^{1/a}}\right]^{-\frac{a}{a-1}}. \tag{91}$$

*Proof.* By Theorem C.30, with probability at least $1 - e^{-\Omega(M)}$,

$$\mathbb{E}\mathcal{R}_M(\bar{\mathbf{v}}_N) \lesssim \sigma^2 + M^{1-a} + \mathrm{BiasError} + \mathrm{VarianceError} + \mathrm{AdditiveError},$$

where

$$\text{BiasError} \lesssim \left(1 - \underline{\epsilon}_3^{(A)}\right)^2 \max\left\{\left[\frac{N\gamma}{1 - \underline{\epsilon}_3^{(A)}}\right]^{\frac{1}{a}-1}, M^{1-a}\right\},$$

$$\text{VarianceError} \lesssim \left(1 - \underline{\epsilon}_3^{(A)}\right)\left(1 + \bar{\epsilon}_2^{(A)}\right)\frac{k_{\text{eff}} + \gamma^2 N^2 M^{-2a}\left(\frac{1}{1-\bar{\epsilon}_3^{(A)}} - 1\right)^2 (M - k_{\text{eff}})}{N},$$

$$\text{AdditiveError} \lesssim \left(\bar{\epsilon}_3^{(A)}\right)^2 + \left(1 - \underline{\epsilon}_3^{(A)}\right)\bar{\epsilon}_3^{(A)},$$

with $k_{\text{eff}} = \left[M^{-a} \vee \left(\frac{1}{N\gamma} - M^{-a}\left(\frac{1}{1-\bar{\epsilon}_3^{(A)}} - 1\right)\right)\right]^{-\frac{1}{a}}$. Denote

$$\mathcal{R}_0 = \mathbb{E}\mathcal{R}_M(\bar{\mathbf{v}}_N) - \left[\sigma^2 + \left(\bar{\epsilon}_3^{(A)}\right)^2 + \left(1 - \underline{\epsilon}_3^{(A)}\right)\bar{\epsilon}_3^{(A)}\right].$$

- $\frac{1}{N\gamma} > \frac{M^{-a}}{1 - \bar{\epsilon}_3^{(A)}}$

  In this case, $\frac{1}{N\gamma} - M^{-a}\left(\frac{1}{1-\bar{\epsilon}_3^{(A)}} - 1\right) > M^{-a}$ and $\frac{1}{N\gamma} > \frac{M^{-a}}{1-\bar{\epsilon}_3^{(A)}} > \frac{M^{-a}}{1-\underline{\epsilon}_3^{(A)}}$. It follows that $\frac{N\gamma}{1-\underline{\epsilon}_3^{(A)}} < M^a$. Hence,

$$\mathcal{R}_0 \lesssim M^{1-a} + \left(1 - \underline{\epsilon}_3^{(A)}\right)^2 \left[\frac{N\gamma}{1 - \underline{\epsilon}_3^{(A)}}\right]^{1/a-1}$$

$$+ \left(1 - \underline{\epsilon}_3^{(A)}\right)\left(1 + \bar{\epsilon}_2^{(A)}\right)\frac{k_{\text{eff}}\left(1 - \gamma^2 N^2 M^{-2a}\left(\frac{1}{1-\bar{\epsilon}_3^{(A)}} - 1\right)^2\right) + \gamma^2 N^2 M^{-2a}\left(\frac{1}{1-\bar{\epsilon}_3^{(A)}} - 1\right)^2 M}{N},$$

with $k_{\text{eff}} = \left[\frac{1}{N\gamma} - M^{-a}\left(\frac{1}{1-\bar{\epsilon}_3^{(A)}} - 1\right)\right]^{-1/a}$. Noticing that

$$M^{1-2a} \leq \left(\frac{N\gamma}{1 - \bar{\epsilon}_3^{(A)}}\right)^{1/a-2},$$

and

$$k_{\text{eff}} = \left[\frac{1}{N\gamma} - M^{-a}\left(\frac{1}{1-\bar{\epsilon}_3^{(A)}} - 1\right)\right]^{-1/a}$$

$$\approx N^{1/a}\left[1 - N\gamma M^{-a}\left(\frac{1}{1-\bar{\epsilon}_3^{(A)}} - 1\right)\right]^{-1/a}$$

$$\leq N^{1/a}\left[1 - \left(1 - \bar{\epsilon}_3^{(A)}\right)\left(\frac{1}{1-\bar{\epsilon}_3^{(A)}} - 1\right)\right]^{-1/a}$$

$$= N^{1/a}\left(1 - \bar{\epsilon}_3^{(A)}\right)^{-1/a},$$

we have

$$\mathcal{R}_0 \lesssim M^{1-a} + \left(1 - \underline{\epsilon}_3^{(A)}\right)^{3-1/a}[N\gamma]^{1/a-1}$$

$$+ \left(1 - \underline{\epsilon}_3^{(A)}\right)\left(1 + \bar{\epsilon}_2^{(A)}\right)\frac{N^{1/a}\left(1 - \bar{\epsilon}_3^{(A)}\right)^{-1/a} + \gamma^2 N^2\left(\frac{1}{1-\bar{\epsilon}_3^{(A)}} - 1\right)^2\left(\frac{N\gamma}{1-\bar{\epsilon}_3^{(A)}}\right)^{1/a-2}}{N}$$

$$\lesssim M^{1-a} + N^{1/a-1}\left[\left(1 - \underline{\epsilon}_3^{(A)}\right)\left(1 + \bar{\epsilon}_2^{(A)}\right)\left(1 - \bar{\epsilon}_3^{(A)}\right)^{-1/a}\left(1 + \left(\bar{\epsilon}_3^{(A)}\right)^2\right)\right].$$

We would like to remark that, the term $NM^{1-2a}$ can also expressed by $M$, leading a decrease in effective $M$. Specifically, noticing that

$$\gamma^2 N M^{1-2a} \left(\frac{1}{1-\bar{\epsilon}_3^{(A)}} - 1\right)^2 \lesssim M^{1-a}\left(1 - \underline{\epsilon}_3^{(A)}\right)\left(\frac{1}{1-\bar{\epsilon}_3^{(A)}} - 1\right)^2,$$

we have

$$\mathcal{R}_0 \lesssim M^{1-a}\left[1 + \left(1 - \underline{\epsilon}_3^{(A)}\right)\left(1 + \bar{\epsilon}_2^{(A)}\right)\frac{\left(\bar{\epsilon}_3^{(A)}\right)^2}{1-\bar{\epsilon}_3^{(A)}}\right] + N^{1/a-1}\left[\left(1 - \underline{\epsilon}_3^{(A)}\right)\left(1 + \bar{\epsilon}_2^{(A)}\right)\left(1 - \bar{\epsilon}_3^{(A)}\right)^{-1/a}\right].$$

- $\frac{M^{-a}}{1-\underline{\epsilon}_3^{(A)}} < \frac{1}{N\gamma} \leq \frac{M^{-a}}{1-\bar{\epsilon}_3^{(A)}}$

  In this case,

$$\mathcal{R}_0 \lesssim M^{1-a} + \left(1 - \underline{\epsilon}_3^{(A)}\right)\left(1 + \bar{\epsilon}_2^{(A)}\right)\frac{M}{N} + \left(1 - \underline{\epsilon}_3^{(A)}\right)^{3-1/a}[N\gamma]^{1/a-1}$$

$$\lesssim M^{1-a} + N^{1/a-1}\left[\left(1 - \underline{\epsilon}_3^{(A)}\right)\left(1 + \bar{\epsilon}_2^{(A)}\right)\left(1 - \bar{\epsilon}_3^{(A)}\right)^{-1/a}\right].$$

- $\frac{1}{N\gamma} \leq \frac{M^{-a}}{1-\underline{\epsilon}_3^{(A)}}$

  In this case,

$$\mathcal{R}_0 \lesssim M^{1-a} + \left(1 - \underline{\epsilon}_3^{(A)}\right)\left(1 + \bar{\epsilon}_2^{(A)}\right)\frac{M}{N} \lesssim M^{1-a} + \left(1 - \underline{\epsilon}_3^{(A)}\right)^{1-1/a}\left(1 + \bar{\epsilon}_2^{(A)}\right)N^{1/a-1}.$$

Overall,

$$\mathcal{R}_0 \lesssim M^{1-a} + N^{1/a-1}\left(1 - \underline{\epsilon}_3^{(A)}\right)\left(1 + \bar{\epsilon}_2^{(A)}\right)\left(1 - \bar{\epsilon}_3^{(A)}\right)^{-1/a}\left[1 + \left(\bar{\epsilon}_3^{(A)}\right)^2\right].$$

Regarding analyzing $M_{\text{eff}}$, it holds

$$\mathcal{R}_0 \lesssim M^{1-a}\left[1 + \left(1 - \underline{\epsilon}_3^{(A)}\right)\left(1 + \bar{\epsilon}_2^{(A)}\right)\frac{\left(\bar{\epsilon}_3^{(A)}\right)^2}{1-\bar{\epsilon}_3^{(A)}}\right] + N^{1/a-1}\left[\left(1 - \underline{\epsilon}_3^{(A)}\right)\left(1 + \bar{\epsilon}_2^{(A)}\right)\left(1 - \bar{\epsilon}_3^{(A)}\right)^{-1/a}\right].$$

$\square$

**Theorem E.4.** *Suppose $\gamma < 1/\tilde{\lambda}_1^{(q)}$. For $i \in \{d, f, s, p, a, o\}$, if there exist constants $(\bar{\epsilon}_i, \underline{\epsilon}_i)$ such that $\mathcal{Q}_i$ is $(\bar{\epsilon}_i, \underline{\epsilon}_i)$-additive, then under Assumption 3.1, 3.3, 3.4, 3.5 and 3.6, for sufficiently large $N > 500$, if $\mathbf{H}_f^{(q)}$ and $\mathbf{SHS}^\top$ are commutative, with probability at least $1 - e^{-\Omega(M)}$ over the randomness of $\mathbf{S}$, it holds* [13]

$$\mathbb{E}\mathcal{R}_M(\bar{\mathbf{v}}_N) \gtrsim \frac{1}{M_{\text{eff}}^{a-1}} + \frac{1}{N_{\text{eff}}^{(a-1)/a}} + \sigma^2 + \left(\underline{\epsilon}_3^{(A)}\right)^2 + \frac{1-\bar{\epsilon}_3^{(A)}}{N}\underline{\epsilon}_3^{(A)}, \tag{92}$$

*where*

$$M_{\text{eff}} = M, \quad N_{\text{eff}} = N\left[\left(1 - \bar{\epsilon}_3^{(A)}\right)\left(1 + \underline{\epsilon}_2^{(A)}\right)\left[1 - N\gamma\left(\frac{1}{1-\underline{\epsilon}_3^{(A)}} - 1\right)\right]^{-1/a}\right]^{-\frac{a}{a-1}}.$$

---

[13] This scaling law holds in the positive regime

$$\Omega = \left\{(M, N) : \left(\frac{1}{N\gamma} - \left(\frac{1}{1-\underline{\epsilon}_3^{(A)}} - 1\right)\right) \geq M^{-a}\right) \vee \left(M^{1-a} \geq \left(1 - \bar{\epsilon}_3^{(A)}\right)\left(1 + \underline{\epsilon}_2^{(A)}\right)\frac{\left[\frac{1}{N\gamma} - \left(\frac{1}{1-\underline{\epsilon}_3^{(A)}} - 1\right)\right]^{-\frac{1}{a}}}{N\gamma}\right)\right\}$$

under the condition that $\frac{1}{N\gamma} - \left(\frac{1}{1-\underline{\epsilon}_3^{(A)}} - 1\right) \geq 0$.

*Proof.* By Lemma D.24,

$$\mathbb{E}\mathcal{R}_M(\bar{\mathbf{v}}_N) \gtrsim \sigma^2 + M^{1-a} + \text{BiasError} + \text{VarianceError} + \text{AdditiveError},$$

where

$$\text{VarianceError} \gtrsim \left(1 - \bar{\epsilon}_3^{(A)}\right)\left(1 + \underline{\epsilon}_2^{(A)}\right) \frac{\underline{k}_{\text{eff}} + \gamma^2 N^2 \left(\frac{1}{1-\underline{\epsilon}_3^{(A)}} - 1\right)^2 (M - \underline{k}_{\text{eff}})}{N},$$

and if $M^{-a} + M^{-a}\left(\frac{1}{1-\bar{\epsilon}_3^{(A)}} - 1\right) \leq \frac{C}{N\gamma}$ for some constant $C > 0$, then

$$\text{BiasError} \gtrsim \left(1 - \bar{\epsilon}_3^{(A)}\right)^2 \cdot \left(\frac{1}{N\gamma} - M^{-a}\left(\frac{1}{1-\bar{\epsilon}_3^{(A)}} - 1\right)\right)^{1-1/a},$$

$$\text{AdditiveError} \gtrsim \left(\underline{\epsilon}_3^{(A)}\right)^2 + \frac{1 - \bar{\epsilon}_3^{(A)}}{N}\underline{\epsilon}_3^{(A)},$$

with

$$\underline{k}_{\text{eff}} = \left[M^{-a} \vee \left(\frac{1}{N\gamma} - \left(\frac{1}{1-\underline{\epsilon}_3^{(A)}} - 1\right)\right)\right]^{-\frac{1}{a}}$$

Denote

$$\mathcal{R}_0 = \mathbb{E}\mathcal{R}_M(\bar{\mathbf{v}}_N) - \left[\sigma^2 + \left(\underline{\epsilon}_3^{(A)}\right)^2 + \frac{1 - \bar{\epsilon}_3^{(A)}}{N}\underline{\epsilon}_3^{(A)}\right].$$

- $\frac{1}{N\gamma} - \left(\frac{1}{1-\underline{\epsilon}_3^{(A)}} - 1\right) \geq M^{-a}$

   In this case,

$$\mathcal{R}_0 \gtrsim M^{1-a} + \left(1 - \bar{\epsilon}_3^{(A)}\right)\left(1 + \underline{\epsilon}_2^{(A)}\right) \frac{\underline{k}_{\text{eff}} + \gamma^2 N^2 \left(\frac{1}{1-\underline{\epsilon}_3^{(A)}} - 1\right)^2 (M - \underline{k}_{\text{eff}})}{N}$$

$$\geq M^{1-a} + \left(1 - \bar{\epsilon}_3^{(A)}\right)\left(1 + \underline{\epsilon}_2^{(A)}\right) \frac{\left[\frac{1}{N\gamma} - \left(\frac{1}{1-\underline{\epsilon}_3^{(A)}} - 1\right)\right]^{-\frac{1}{a}}}{N}.$$

- $\frac{1}{N\gamma} - \left(\frac{1}{1-\underline{\epsilon}_3^{(A)}} - 1\right) < M^{-a}$

   In this case,

$$\mathcal{R}_0 \gtrsim M^{1-a} + \left(1 - \bar{\epsilon}_3^{(A)}\right)\left(1 + \underline{\epsilon}_2^{(A)}\right)\frac{M}{N}.$$

Suppose $\frac{1}{N\gamma} - \left(\frac{1}{1-\underline{\epsilon}_3^{(A)}} - 1\right) \geq 0$. Further if

$$M^{1-a} \geq \left(1 - \bar{\epsilon}_3^{(A)}\right)\left(1 + \underline{\epsilon}_2^{(A)}\right) \frac{\left[\frac{1}{N\gamma} - \left(\frac{1}{1-\underline{\epsilon}_3^{(A)}} - 1\right)\right]^{-\frac{1}{a}}}{N\gamma},$$

then

$$\mathcal{R}_0 \gtrsim M^{1-a} + \left(1 - \bar{\epsilon}_3^{(A)}\right)\left(1 + \underline{\epsilon}_2^{(A)}\right)\frac{\left[\frac{1}{N\gamma} - \left(\frac{1}{1-\underline{\epsilon}_3^{(A)}} - 1\right)\right]^{-\frac{1}{a}}}{N}$$

$$\approx M^{1-a} + N^{1/a-1}\left(1 - \bar{\epsilon}_3^{(A)}\right)\left(1 + \underline{\epsilon}_2^{(A)}\right)\left[1 - N\gamma\left(\frac{1}{1-\underline{\epsilon}_3^{(A)}} - 1\right)\right]^{-1/a}.$$

$\square$

## F. Auxiliary Lemmas

**Lemma F.1.** *For PSD matrices $\mathbf{X} \preceq \mathbf{Y}$ and any PSD matrix $\mathbf{A}$,*

$$\mu_{\min}\left((\mathbf{A} + \mathbf{X})^{-1}\mathbf{A}\right) \geq \mu_{\min}\left((\mathbf{A} + \mathbf{Y})^{-1}\mathbf{A}\right), \quad \mu_{\max}\left((\mathbf{A} + \mathbf{X})^{-1}\mathbf{A}\right) \geq \mu_{\max}\left((\mathbf{A} + \mathbf{Y})^{-1}\mathbf{A}\right).$$

*Proof.* Define the generalized Rayleigh quotient for a matrix $\mathbf{M}$ as $R_{\mathbf{M}}(\mathbf{v}) = \frac{\mathbf{v}^\top \mathbf{A} \mathbf{v}}{\mathbf{v}^\top (\mathbf{M}+\mathbf{A})\mathbf{v}}$. Note that the eigenvalues of $(\mathbf{M} + \mathbf{A})^{-1}\mathbf{A}$ are given by the critical values of $R_{\mathbf{M}}(\mathbf{v})$. For any non-zero vector $\mathbf{v}$, since $\mathbf{X} \preceq \mathbf{Y}$ and $\mathbf{A} \succeq 0$, we have

$$R_{\mathbf{X}}(\mathbf{v}) = \frac{1}{1 + \frac{\mathbf{v}^\top \mathbf{X} \mathbf{v}}{\mathbf{v}^\top \mathbf{A} \mathbf{v}}} \geq \frac{1}{1 + \frac{\mathbf{v}^\top \mathbf{Y} \mathbf{v}}{\mathbf{v}^\top \mathbf{A} \mathbf{v}}} = R_{\mathbf{Y}}(\mathbf{v}).$$

By the Courant-Fischer Min-Max theorem, we have

$$\mu_{\min}\left((\mathbf{A} + \mathbf{X})^{-1}\mathbf{A}\right) = \min_{\mathbf{v} \neq 0} R_{\mathbf{X}}(\mathbf{v}) \geq \min_{\mathbf{v} \neq 0} R_{\mathbf{Y}}(\mathbf{v}) = \mu_{\min}\left((\mathbf{A} + \mathbf{Y})^{-1}\mathbf{A}\right).$$

Similarly

$$\mu_{\max}\left((\mathbf{A} + \mathbf{X})^{-1}\mathbf{A}\right) = \max_{\mathbf{v} \neq 0} R_{\mathbf{X}}(\mathbf{v}) \geq \max_{\mathbf{v} \neq 0} R_{\mathbf{Y}}(\mathbf{v}) = \mu_{\max}\left((\mathbf{A} + \mathbf{Y})^{-1}\mathbf{A}\right).$$

$\square$

**Lemma F.2.** *For PSD matrices $\mathbf{A} \preceq \mathbf{B}$ and any PSD matrix $\mathbf{X}$,*

$$\mu_{\min}\left((\mathbf{A} + \mathbf{X})^{-1}\mathbf{A}\right) \leq \mu_{\min}\left((\mathbf{B} + \mathbf{X})^{-1}\mathbf{B}\right), \quad \mu_{\max}\left((\mathbf{A} + \mathbf{X})^{-1}\mathbf{A}\right) \leq \mu_{\max}\left((\mathbf{B} + \mathbf{X})^{-1}\mathbf{B}\right).$$

*Proof.* Define the generalized Rayleigh quotient for a matrix $\mathbf{M}$ as $R_{\mathbf{M}}(\mathbf{v}) = \frac{\mathbf{v}^\top \mathbf{M} \mathbf{v}}{\mathbf{v}^\top (\mathbf{M}+\mathbf{X})\mathbf{v}}$. Note that the eigenvalues of $(\mathbf{M} + \mathbf{X})^{-1}\mathbf{M}$ are given by the critical values of $R_{\mathbf{M}}(\mathbf{v})$. For any non-zero vector $\mathbf{v}$, since $\mathbf{A} \preceq \mathbf{B}$ and $\mathbf{X} \succeq 0$, we have

$$R_{\mathbf{A}}(\mathbf{v}) = \frac{1}{1 + \frac{\mathbf{v}^\top \mathbf{X} \mathbf{v}}{\mathbf{v}^\top \mathbf{A} \mathbf{v}}} \leq \frac{1}{1 + \frac{\mathbf{v}^\top \mathbf{X} \mathbf{v}}{\mathbf{v}^\top \mathbf{B} \mathbf{v}}} = R_{\mathbf{B}}(\mathbf{v}).$$

By the Courant-Fischer Min-Max theorem, we have

$$\mu_{\min}\left((\mathbf{A} + \mathbf{X})^{-1}\mathbf{A}\right) = \min_{\mathbf{v} \neq 0} R_{\mathbf{A}}(\mathbf{v}) \leq \min_{\mathbf{v} \neq 0} R_{\mathbf{B}}(\mathbf{v}) = \mu_{\min}\left((\mathbf{B} + \mathbf{X})^{-1}\mathbf{B}\right).$$

Similarly,

$$\mu_{\max}\left((\mathbf{A} + \mathbf{X})^{-1}\mathbf{A}\right) = \max_{\mathbf{v} \neq 0} R_{\mathbf{A}}(\mathbf{v}) \leq \max_{\mathbf{v} \neq 0} R_{\mathbf{B}}(\mathbf{v}) = \mu_{\max}\left((\mathbf{B} + \mathbf{X})^{-1}\mathbf{B}\right).$$

$\square$

**Lemma F.3.** *For $\mathbf{A} \succ 0$, $\mathbf{D} \succeq 0$, if there exists $\epsilon > 0$ such that $\mathbf{D} \preceq \epsilon\mathbf{A}$, then*

$$\mathbf{A}^{-1}\mathbf{D}(\mathbf{A} + \mathbf{D})^{-1}\mathbf{A}(\mathbf{A} + \mathbf{D})^{-1}\mathbf{D}\mathbf{A}^{-1} \preceq \frac{\epsilon^2}{(\epsilon + 1)^2}\mathbf{A}^{-1}.$$

*Proof.* Denote $\mathbf{Q} = \mathbf{A}^{-1/2}\mathbf{D}\mathbf{A}^{-1/2}$. As $\mathbf{D} \succeq 0$, we have

$$\mathbf{Q} \preceq \epsilon\mathbf{I}.$$

Rewriting

$$
\begin{aligned}
&\mathbf{A}^{-1}\mathbf{D}(\mathbf{A}+\mathbf{D})^{-1}\mathbf{A}(\mathbf{A}+\mathbf{D})^{-1}\mathbf{D}\mathbf{A}^{-1}\\
=&\mathbf{A}^{-1}\left[\mathbf{A}^{1/2}\mathbf{Q}\mathbf{A}^{1/2}\right]\cdot\left[\mathbf{A}^{-1/2}(\mathbf{I}+\mathbf{Q})^{-1}\mathbf{A}^{-1/2}\right]\cdot\mathbf{A}\cdot\left[\mathbf{A}^{-1/2}(\mathbf{I}+\mathbf{Q})^{-1}\mathbf{A}^{-1/2}\right]\cdot\left[\mathbf{A}^{1/2}\mathbf{Q}\mathbf{A}^{1/2}\right]\mathbf{A}^{-1}\\
=&\left(\mathbf{A}^{-1/2}\mathbf{Q}\mathbf{A}^{1/2}\right)\left(\mathbf{A}^{-1/2}(\mathbf{I}+\mathbf{Q})^{-1}\mathbf{A}^{-1/2}\right)\mathbf{A}\left(\mathbf{A}^{-1/2}(\mathbf{I}+\mathbf{Q})^{-1}\mathbf{A}^{-1/2}\right)\left(\mathbf{A}^{1/2}\mathbf{Q}\mathbf{A}^{-1/2}\right)\\
=&\mathbf{A}^{-1/2}\mathbf{Q}\mathbf{A}^{1/2}\cdot\left[\mathbf{A}^{-1/2}(\mathbf{I}+\mathbf{Q})^{-2}\mathbf{A}^{-1/2}\right]\cdot\mathbf{A}^{1/2}\mathbf{Q}\mathbf{A}^{-1/2}\\
=&\mathbf{A}^{-1/2}\left[\mathbf{Q}(\mathbf{I}+\mathbf{Q})^{-2}\mathbf{Q}\right]\mathbf{A}^{-1/2}\\
=&\mathbf{A}^{-1/2}\left[\mathbf{Q}^2(\mathbf{I}+\mathbf{Q})^{-2}\right]\mathbf{A}^{-1/2},
\end{aligned}
$$

where the last equality holds as $\mathbf{Q}$ and $\mathbf{Q}+\mathbf{I}$ are commutative. Further note that $f(x) = \frac{x^2}{(1+x)^2}$ is increasing, it holds

$$\mathbf{Q}^2(\mathbf{I}+\mathbf{Q})^{-2} = f(\mathbf{Q}) \preceq f(\epsilon)\mathbf{I} = \frac{\epsilon^2}{(1+\epsilon)^2}\mathbf{I}.$$

Hence,

$$\mathbf{A}^{-1}\mathbf{D}(\mathbf{A}+\mathbf{D})^{-1}\mathbf{A}(\mathbf{A}+\mathbf{D})^{-1}\mathbf{D}\mathbf{A}^{-1} \preceq \frac{\epsilon^2}{(1+\epsilon)^2}\mathbf{A}^{-1}.$$

$\square$

**Lemma F.4.** *For $\mathbf{A} \succ 0$, $\mathbf{D} \succeq 0$, if there exists $\epsilon > 0$ such that $\mathbf{D} \preceq \epsilon\mathbf{I}$, then*

$$\mathbf{A}^{-1}\mathbf{D}(\mathbf{A}+\mathbf{D})^{-1}\mathbf{A}(\mathbf{A}+\mathbf{D})^{-1}\mathbf{D}\mathbf{A}^{-1} \preceq \frac{\epsilon^2}{(\epsilon+\mu_{\min}(\mathbf{A}))^2}\mathbf{A}^{-1}.$$

*Proof.* Denote $\mathbf{Q} = \mathbf{A}^{-1/2}\mathbf{D}\mathbf{A}^{-1/2}$. Rewriting

$$
\begin{aligned}
&\mathbf{A}^{-1}\mathbf{D}(\mathbf{A}+\mathbf{D})^{-1}\mathbf{A}(\mathbf{A}+\mathbf{D})^{-1}\mathbf{D}\mathbf{A}^{-1}\\
=&\mathbf{A}^{-1}\left[\mathbf{A}^{1/2}\mathbf{Q}\mathbf{A}^{1/2}\right]\cdot\left[\mathbf{A}^{-1/2}(\mathbf{I}+\mathbf{Q})^{-1}\mathbf{A}^{-1/2}\right]\cdot\mathbf{A}\cdot\left[\mathbf{A}^{-1/2}(\mathbf{I}+\mathbf{Q})^{-1}\mathbf{A}^{-1/2}\right]\cdot\left[\mathbf{A}^{1/2}\mathbf{Q}\mathbf{A}^{1/2}\right]\mathbf{A}^{-1}\\
=&\left(\mathbf{A}^{-1/2}\mathbf{Q}\mathbf{A}^{1/2}\right)\left(\mathbf{A}^{-1/2}(\mathbf{I}+\mathbf{Q})^{-1}\mathbf{A}^{-1/2}\right)\mathbf{A}\left(\mathbf{A}^{-1/2}(\mathbf{I}+\mathbf{Q})^{-1}\mathbf{A}^{-1/2}\right)\left(\mathbf{A}^{1/2}\mathbf{Q}\mathbf{A}^{-1/2}\right)\\
=&\mathbf{A}^{-1/2}\mathbf{Q}\mathbf{A}^{1/2}\cdot\left[\mathbf{A}^{-1/2}(\mathbf{I}+\mathbf{Q})^{-2}\mathbf{A}^{-1/2}\right]\cdot\mathbf{A}^{1/2}\mathbf{Q}\mathbf{A}^{-1/2}\\
=&\mathbf{A}^{-1/2}\left[\mathbf{Q}(\mathbf{I}+\mathbf{Q})^{-2}\mathbf{Q}\right]\mathbf{A}^{-1/2}\\
=&\mathbf{A}^{-1/2}\left[\mathbf{Q}^2(\mathbf{I}+\mathbf{Q})^{-2}\right]\mathbf{A}^{-1/2},
\end{aligned}
$$

where the last equality holds as $\mathbf{Q}$ and $\mathbf{Q}+\mathbf{I}$ are commutative. Given $\mathbf{D} \preceq \epsilon\mathbf{I}$, we substitute $\mathbf{D}$:

$$\mathbf{Q} = \mathbf{A}^{-1/2}\mathbf{D}\mathbf{A}^{-1/2} \preceq \epsilon\mathbf{A}^{-1/2}\mathbf{I}\mathbf{A}^{-1/2} = \epsilon\mathbf{A}^{-1}.$$

Let $\mu_{\min}(\mathbf{A})$ denote the minimum eigenvalue of $\mathbf{A}$. Since $\mathbf{A}$ is positive definite, the maximum eigenvalue of $\mathbf{A}^{-1}$ is $1/\mu_{\min}(\mathbf{A})$. Thus:

$$\mathbf{A}^{-1} \preceq \frac{1}{\mu_{\min}(\mathbf{A})}\mathbf{I}.$$

Combining these, we obtain a scalar upper bound for $\mathbf{Q}$:

$$\mathbf{Q} \preceq \frac{\epsilon}{\mu_{\min}(\mathbf{A})}\mathbf{I}.$$

Further note that $f(x) = \frac{x^2}{(1+x)^2}$ is increasing, it holds

$$\mathbf{Q}^2(\mathbf{I} + \mathbf{Q})^{-2} = f(\mathbf{Q}) \preceq f(\frac{\epsilon}{\mu_{\min}(\mathbf{A})})\mathbf{I} = \frac{\epsilon^2}{(\mu_{\min}(\mathbf{A}) + \epsilon)^2}\mathbf{I}.$$

Hence,

$$\mathbf{A}^{-1}\mathbf{D}(\mathbf{A} + \mathbf{D})^{-1}\mathbf{A}(\mathbf{A} + \mathbf{D})^{-1}\mathbf{D}\mathbf{A}^{-1} \preceq \frac{\epsilon^2}{(\mu_{\min}(\mathbf{A}) + \epsilon)^2}\mathbf{A}^{-1}.$$

□

**Lemma F.5.** *For $\mathbf{A} \succ 0$, $\mathbf{D} \succeq 0$, if there exists $\epsilon > 0$ such that $\mathbf{D} \succeq \epsilon\mathbf{A}$, then*

$$\mathbf{A}^{-1}\mathbf{D}(\mathbf{A} + \mathbf{D})^{-1}\mathbf{A}(\mathbf{A} + \mathbf{D})^{-1}\mathbf{D}\mathbf{A}^{-1} \succeq \frac{\epsilon^2}{(\epsilon + 1)^2}\mathbf{A}^{-1}.$$

*Proof.* Denote $\mathbf{Q} = \mathbf{A}^{-1/2}\mathbf{D}\mathbf{A}^{-1/2}$, we have

$$\mathbf{Q} \succeq \epsilon\mathbf{I}.$$

Rewriting

$$
\begin{aligned}
&\mathbf{A}^{-1}\mathbf{D}(\mathbf{A} + \mathbf{D})^{-1}\mathbf{A}(\mathbf{A} + \mathbf{D})^{-1}\mathbf{D}\mathbf{A}^{-1}\\
=&\mathbf{A}^{-1}\left[\mathbf{A}^{1/2}\mathbf{Q}\mathbf{A}^{1/2}\right] \cdot \left[\mathbf{A}^{-1/2}(\mathbf{I} + \mathbf{Q})^{-1}\mathbf{A}^{-1/2}\right] \cdot \mathbf{A} \cdot \left[\mathbf{A}^{-1/2}(\mathbf{I} + \mathbf{Q})^{-1}\mathbf{A}^{-1/2}\right] \cdot \left[\mathbf{A}^{1/2}\mathbf{Q}\mathbf{A}^{1/2}\right]\mathbf{A}^{-1}\\
=&\left(\mathbf{A}^{-1/2}\mathbf{Q}\mathbf{A}^{1/2}\right)\left(\mathbf{A}^{-1/2}(\mathbf{I} + \mathbf{Q})^{-1}\mathbf{A}^{-1/2}\right)\mathbf{A}\left(\mathbf{A}^{-1/2}(\mathbf{I} + \mathbf{Q})^{-1}\mathbf{A}^{-1/2}\right)\left(\mathbf{A}^{1/2}\mathbf{Q}\mathbf{A}^{-1/2}\right)\\
=&\mathbf{A}^{-1/2}\mathbf{Q}\mathbf{A}^{1/2} \cdot \left[\mathbf{A}^{-1/2}(\mathbf{I} + \mathbf{Q})^{-2}\mathbf{A}^{-1/2}\right] \cdot \mathbf{A}^{1/2}\mathbf{Q}\mathbf{A}^{-1/2}\\
=&\mathbf{A}^{-1/2}\left[\mathbf{Q}(\mathbf{I} + \mathbf{Q})^{-2}\mathbf{Q}\right]\mathbf{A}^{-1/2}\\
=&\mathbf{A}^{-1/2}\left[\mathbf{Q}^2(\mathbf{I} + \mathbf{Q})^{-2}\right]\mathbf{A}^{-1/2},
\end{aligned}
$$

where the last equality holds as $\mathbf{Q}$ and $\mathbf{Q} + \mathbf{I}$ are commutative. Further note that $f(x) = \frac{x^2}{(1+x)^2}$ is increasing, it holds

$$\mathbf{Q}^2(\mathbf{I} + \mathbf{Q})^{-2} = f(\mathbf{Q}) \succeq f(\epsilon)\mathbf{I} = \frac{\epsilon^2}{(1 + \epsilon)^2}\mathbf{I}.$$

Hence,

$$\mathbf{A}^{-1}\mathbf{D}(\mathbf{A} + \mathbf{D})^{-1}\mathbf{A}(\mathbf{A} + \mathbf{D})^{-1}\mathbf{D}\mathbf{A}^{-1} \succeq \frac{\epsilon^2}{(1 + \epsilon)^2}\mathbf{A}^{-1}.$$

□

**Lemma F.6.** *For $\mathbf{A} \succ 0$, $\mathbf{D} \succeq 0$, if there exists $\epsilon > 0$ such that $\mathbf{D} \succeq \epsilon\mathbf{I}$, then*

$$\mathbf{A}^{-1}\mathbf{D}(\mathbf{A} + \mathbf{D})^{-1}\mathbf{A}(\mathbf{A} + \mathbf{D})^{-1}\mathbf{D}\mathbf{A}^{-1} \succeq \frac{\epsilon^2}{(\epsilon + \mu_{\max}(\mathbf{A}))^2}\mathbf{A}^{-1}.$$

*Proof.* Denote $\mathbf{Q} = \mathbf{A}^{-1/2}\mathbf{D}\mathbf{A}^{-1/2}$. Rewriting

$$
\begin{aligned}
&\mathbf{A}^{-1}\mathbf{D}(\mathbf{A} + \mathbf{D})^{-1}\mathbf{A}(\mathbf{A} + \mathbf{D})^{-1}\mathbf{D}\mathbf{A}^{-1}\\
=&\mathbf{A}^{-1}\left[\mathbf{A}^{1/2}\mathbf{Q}\mathbf{A}^{1/2}\right] \cdot \left[\mathbf{A}^{-1/2}(\mathbf{I} + \mathbf{Q})^{-1}\mathbf{A}^{-1/2}\right] \cdot \mathbf{A} \cdot \left[\mathbf{A}^{-1/2}(\mathbf{I} + \mathbf{Q})^{-1}\mathbf{A}^{-1/2}\right] \cdot \left[\mathbf{A}^{1/2}\mathbf{Q}\mathbf{A}^{1/2}\right]\mathbf{A}^{-1}\\
=&\left(\mathbf{A}^{-1/2}\mathbf{Q}\mathbf{A}^{1/2}\right)\left(\mathbf{A}^{-1/2}(\mathbf{I} + \mathbf{Q})^{-1}\mathbf{A}^{-1/2}\right)\mathbf{A}\left(\mathbf{A}^{-1/2}(\mathbf{I} + \mathbf{Q})^{-1}\mathbf{A}^{-1/2}\right)\left(\mathbf{A}^{1/2}\mathbf{Q}\mathbf{A}^{-1/2}\right)\\
=&\mathbf{A}^{-1/2}\mathbf{Q}\mathbf{A}^{1/2} \cdot \left[\mathbf{A}^{-1/2}(\mathbf{I} + \mathbf{Q})^{-2}\mathbf{A}^{-1/2}\right] \cdot \mathbf{A}^{1/2}\mathbf{Q}\mathbf{A}^{-1/2}\\
=&\mathbf{A}^{-1/2}\left[\mathbf{Q}(\mathbf{I} + \mathbf{Q})^{-2}\mathbf{Q}\right]\mathbf{A}^{-1/2}\\
=&\mathbf{A}^{-1/2}\left[\mathbf{Q}^2(\mathbf{I} + \mathbf{Q})^{-2}\right]\mathbf{A}^{-1/2},
\end{aligned}
$$

where the last equality holds as $\mathbf{Q}$ and $\mathbf{Q} + \mathbf{I}$ are commutative. Given $\mathbf{D} \preceq \epsilon\mathbf{I}$, we substitute $\mathbf{D}$:

$$\mathbf{Q} = \mathbf{A}^{-1/2}\mathbf{D}\mathbf{A}^{-1/2} \succeq \epsilon\mathbf{A}^{-1/2}\mathbf{I}\mathbf{A}^{-1/2} = \epsilon\mathbf{A}^{-1}.$$

Let $\mu_{\max}(\mathbf{A})$ denote the maximum eigenvalue of $\mathbf{A}$. Since $\mathbf{A}$ is positive definite, the minimum eigenvalue of $\mathbf{A}^{-1}$ is $1/\mu_{\max}(\mathbf{A})$. Thus:

$$\mathbf{A}^{-1} \succeq \frac{1}{\mu_{\max}(\mathbf{A})}\mathbf{I}.$$

Combining these, we obtain a scalar upper bound for $\mathbf{Q}$:

$$\mathbf{Q} \succeq \frac{\epsilon}{\mu_{\max}(\mathbf{A})}\mathbf{I}.$$

Further note that $f(x) = \frac{x^2}{(1+x)^2}$ is increasing, it holds

$$\mathbf{Q}^2(\mathbf{I} + \mathbf{Q})^{-2} = f(\mathbf{Q}) \succeq f(\frac{\epsilon}{\mu_{\max}(\mathbf{A})})\mathbf{I} = \frac{\epsilon^2}{(\mu_{\max}(\mathbf{A}) + \epsilon)^2}\mathbf{I}.$$

Hence,

$$\mathbf{A}^{-1}\mathbf{D}(\mathbf{A} + \mathbf{D})^{-1}\mathbf{A}(\mathbf{A} + \mathbf{D})^{-1}\mathbf{D}\mathbf{A}^{-1} \succeq \frac{\epsilon^2}{(\mu_{\max}(\mathbf{A}) + \epsilon)^2}\mathbf{A}^{-1}.$$

$\square$

## G. Concentration Lemmas

**Lemma G.1** (Eigenvalues of $\mathbf{SHS}^\top$, Lemma G.4 in Lin et al. (2024)). *Under the power-law spectrum Assumption 3.6, there exists $a$-dependent constants $0 < c_1 < c_2$ such that it holds with probability at least $1 - e^{-\Omega(M)}$ for all $j \in [M]$ that*

$$c_1 j^{-a} \leq \mu_j(\mathbf{SHS}^\top) \leq c_2 j^{-a}.$$

**Lemma G.2** (Eigenvalues of $\mathbf{H}_f^{(q)}$, multiplicative quantization, an upper bound). *If there exist constants $\overline{\epsilon}_s, \overline{\epsilon}_d, \overline{\epsilon}_f$ such that for $i \in \{s, d, f\}$, $\mathcal{Q}_i(\cdot)$ is $\overline{\epsilon}_i$-multiplicative, under the unbiased quantization Assumption 3.1, the power-law spectrum Assumption 3.6, it holds with probability at least $1 - e^{-\Omega(M)}$ for all $j \in [M]$ that*

$$\mu_j(\mathbf{H}_f^{(q)}) \lesssim (1 + \overline{\epsilon}_f)(1 + \overline{\epsilon}_d)(1 + \overline{\epsilon}_s)j^{-a}.$$

*Proof.* Under multiplicative quantization, it holds

$$\begin{aligned} \mathbf{H}_f^{(q)} &\preceq (1 + \overline{\epsilon}_f)(1 + \overline{\epsilon}_d)\mathbb{E}_{\mathcal{Q}_s}\left[\mathcal{Q}_s(\mathbf{S})\mathbf{H}\mathcal{Q}_s(\mathbf{S})^\top\right] \\ &\preceq (1 + \overline{\epsilon}_f)(1 + \overline{\epsilon}_d)\left(\mathbb{E}_{\mathcal{Q}_s}\left[\boldsymbol{\epsilon}^{(s)}\mathbf{H}\boldsymbol{\epsilon}^{(s)^\top}\right] + \mathbf{SHS}^\top\right). \end{aligned}$$

Note that $\mathbb{E}_{\mathcal{Q}_s}\left[\boldsymbol{\epsilon}^{(s)}\mathbf{H}\boldsymbol{\epsilon}^{(s)^\top}\right] \preceq \overline{\epsilon}_s\mathbf{SHS}^\top$, we have,

$$\mathbf{H}_f^{(q)} \preceq (1 + \overline{\epsilon}_f)(1 + \overline{\epsilon}_d)(1 + \overline{\epsilon}_s)\mathbf{SHS}^\top. \tag{93}$$

By Lemma G.1, we have with probability at least $1 - e^{-\Omega(M)}$,

$$\mu_j(\mathbf{H}_f^{(q)}) \precsim (1 + \overline{\epsilon}_f)(1 + \overline{\epsilon}_d)(1 + \overline{\epsilon}_s)j^{-a}. \tag{94}$$

$\square$

**Lemma G.3** (Eigenvalues of $\mathbf{H}_f^{(q)}$, multiplicative quantization, a lower bound). *If there exist constants $\underline{\epsilon}_s, \underline{\epsilon}_d, \underline{\epsilon}_f$ such that for $i \in \{s, d, f\}$, $\mathcal{Q}_i(\cdot)$ is $\underline{\epsilon}_i$-multiplicative, under the unbiased quantization Assumption 3.1, the power-law spectrum Assumption 3.6, it holds with probability at least $1 - e^{-\Omega(M)}$ for all $j \in [M]$ that*

$$\mu_j(\mathbf{H}_f^{(q)}) \gtrsim (1 + \underline{\epsilon}_f)(1 + \underline{\epsilon}_d)(1 + \underline{\epsilon}_s)j^{-a}.$$

*Proof.* Under multiplicative quantization, it holds

$$\mathbf{H}_f^{(q)} \succeq (1 + \underline{\epsilon}_f)(1 + \underline{\epsilon}_d)\mathbb{E}_{\mathcal{Q}_s}\left[\mathcal{Q}_s(\mathbf{S})\mathbf{H}\mathcal{Q}_s(\mathbf{S})^\top\right]$$
$$\succeq (1 + \underline{\epsilon}_f)(1 + \underline{\epsilon}_d)\left(\mathbb{E}_{\mathcal{Q}_s}\left[\boldsymbol{\epsilon}^{(s)}\mathbf{H}\boldsymbol{\epsilon}^{(s)\top}\right] + \mathbf{SHS}^\top\right).$$

Note that $\mathbb{E}_{\mathcal{Q}_s}\left[\boldsymbol{\epsilon}^{(s)}\mathbf{H}\boldsymbol{\epsilon}^{(s)\top}\right] \succeq \underline{\epsilon}_s\mathbf{SHS}^\top$, we have,

$$\mathbf{H}_f^{(q)} \succeq (1 + \underline{\epsilon}_f)(1 + \underline{\epsilon}_d)(1 + \underline{\epsilon}_s)\mathbf{SHS}^\top. \tag{95}$$

By Lemma G.1, we have with probability at least $1 - e^{-\Omega(M)}$,

$$\mu_j(\mathbf{H}_f^{(q)}) \succeq (1 + \underline{\epsilon}_f)(1 + \underline{\epsilon}_d)(1 + \underline{\epsilon}_s)j^{-a}. \tag{96}$$

$\square$

**Lemma G.4** (Eigenvalues of $\mathbf{H}_f^{(q)}$, additive quantization, an upper bound)**.** *If there exist constants $\bar{\epsilon}_s, \bar{\epsilon}_d, \bar{\epsilon}_f$ such that for $i \in \{s, d, f\}$, $\mathcal{Q}_i(\cdot)$ is $\bar{\epsilon}_i$-additive, under the unbiased quantization Assumption 3.1, the power-law spectrum Assumption 3.6, it holds with probability at least $1 - e^{-\Omega(M)}$ for all $j \in [M]$ that*

$$\mu_j(\mathbf{H}_f^{(q)}) \lesssim j^{-a} + \bar{\epsilon}_f + (1 + \bar{\epsilon}_d p)\bar{\epsilon}_s + \bar{\epsilon}_d\frac{p}{M}.$$

*Proof.* Under additive quantization, it holds

$$\mathbf{H}_f^{(q)} \preceq \bar{\epsilon}_f\mathbf{I}_M + \mathbb{E}_{\mathcal{Q}_s}\left[\mathcal{Q}_s(\mathbf{S})\mathbf{H}\mathcal{Q}_s(\mathbf{S})^\top\right] + \bar{\epsilon}_d\mathbb{E}_{\mathcal{Q}_s}\left[\mathcal{Q}_s(\mathbf{S})\mathcal{Q}_s(\mathbf{S})^\top\right]$$
$$= \bar{\epsilon}_f\mathbf{I}_M + \mathbb{E}_{\mathcal{Q}_s}\left[\mathcal{Q}_s(\mathbf{S})\mathbf{H}\mathcal{Q}_s(\mathbf{S})^\top\right] + \bar{\epsilon}_d\mathbf{SS}^\top + \bar{\epsilon}_d\mathbb{E}_{\mathcal{Q}_s}\left[\boldsymbol{\epsilon}^{(s)}\boldsymbol{\epsilon}^{(s)\top}\right]$$
$$= \bar{\epsilon}_f\mathbf{I}_M + \mathbb{E}_{\mathcal{Q}_s}\left[\boldsymbol{\epsilon}^{(s)}\mathbf{H}\boldsymbol{\epsilon}^{(s)\top}\right] + \mathbf{SHS}^\top + \bar{\epsilon}_d\mathbf{SS}^\top + \bar{\epsilon}_d\mathbb{E}_{\mathcal{Q}_s}\left[\boldsymbol{\epsilon}^{(s)}\boldsymbol{\epsilon}^{(s)\top}\right]$$
$$\preceq \bar{\epsilon}_f\mathbf{I}_M + \mathrm{tr}(\mathbf{H})\bar{\epsilon}_s\mathbf{I}_M + \mathbf{SHS}^\top + \bar{\epsilon}_d\mathbf{SS}^\top + \bar{\epsilon}_d\bar{\epsilon}_sp\mathbf{I}_M$$
$$= (\bar{\epsilon}_f + \mathrm{tr}(\mathbf{H})\bar{\epsilon}_s + \bar{\epsilon}_d\bar{\epsilon}_sp)\mathbf{I}_M + \mathbf{SHS}^\top + \bar{\epsilon}_d\mathbf{SS}^\top.$$

We then focus on the eigenvalues of $\mathbf{SS}^\top$. We write

$$\mathbf{S} = (\mathbf{s}_1, ..., \mathbf{s}_p), \quad \mathbf{s}_i \sim \mathcal{N}\left(0, \frac{1}{M}\mathbf{I}_M\right), \quad i \geq 1.$$

For any unit vector $\mathbf{v} \in \mathbb{R}^M$, each $\mathbf{s}_i^\top\mathbf{v}$ is sub-Gaussian. By Bernstein inequality, with probability at least $1 - e^{-\Omega(M)}$, for every unit vector $\mathbf{v} \in \mathbb{R}^M$, [14]

$$\mathbf{v}^\top\mathbf{SS}^\top\mathbf{v} = \sum_{i=1}^p(\mathbf{s}_i^\top\mathbf{v})^2 \asymp \frac{p}{M}.$$

That is, with probability at least $1 - e^{-\Omega(M)}$,

$$\mu_{\min}(\bar{\epsilon}_d\mathbf{SS}^\top) \asymp \bar{\epsilon}_d\frac{p}{M}, \quad \mu_{\max}(\bar{\epsilon}_d\mathbf{SS}^\top) \asymp \bar{\epsilon}_d\frac{p}{M}.$$

Therefore, together with Lemma G.1, with probability at least $1 - e^{-\Omega(M)}$ for all $j \in [M]$ that

$$\mu_j(\mathbf{H}_f^{(q)}) \lesssim j^{-a} + \bar{\epsilon}_f + (1 + \bar{\epsilon}_d p)\bar{\epsilon}_s + \bar{\epsilon}_d\frac{p}{M}. \tag{97}$$

$\square$

---

[14] We consider $p \gg M$.

**Lemma G.5** (Eigenvalues of $\mathbf{H}_f^{(q)}$, additive quantization, a lower bound). *If there exist constants $\underline{\epsilon}_s, \underline{\epsilon}_d, \underline{\epsilon}_f$ such that for $i \in \{s, d, f\}$, $\mathcal{Q}_i(\cdot)$ is $\underline{\epsilon}_i$-additive, under the unbiased quantization Assumption 3.1, the power-law spectrum Assumption 3.6, it holds with probability at least $1 - e^{-\Omega(M)}$ for all $j \in [M]$ that*

$$\mu_j(\mathbf{H}_f^{(q)}) \gtrsim j^{-a} + \underline{\epsilon}_f + \underline{\epsilon}_s(1 + \underline{\epsilon}_d p) + \underline{\epsilon}_d \frac{p}{M}.$$

*Proof.* Under additive quantization, it holds

$$
\begin{aligned}
\mathbf{H}_f^{(q)} &\succeq \underline{\epsilon}_f \mathbf{I}_M + \mathbb{E}_{\mathcal{Q}_s}\left[\mathcal{Q}_s(\mathbf{S})\mathbf{H}\mathcal{Q}_s(\mathbf{S})^\top\right] + \underline{\epsilon}_d \mathbb{E}_{\mathcal{Q}_s}\left[\mathcal{Q}_s(\mathbf{S})\mathcal{Q}_s(\mathbf{S})^\top\right] \\
&= \underline{\epsilon}_f \mathbf{I}_M + \mathbb{E}_{\mathcal{Q}_s}\left[\mathcal{Q}_s(\mathbf{S})\mathbf{H}\mathcal{Q}_s(\mathbf{S})^\top\right] + \underline{\epsilon}_d \mathbf{S}\mathbf{S}^\top + \underline{\epsilon}_d \mathbb{E}_{\mathcal{Q}_s}\left[\boldsymbol{\epsilon}^{(s)}\boldsymbol{\epsilon}^{(s)\top}\right] \\
&= \underline{\epsilon}_f \mathbf{I}_M + \mathbb{E}_{\mathcal{Q}_s}\left[\boldsymbol{\epsilon}^{(s)}\mathbf{H}\boldsymbol{\epsilon}^{(s)\top}\right] + \mathbf{S}\mathbf{H}\mathbf{S}^\top + \underline{\epsilon}_d \mathbf{S}\mathbf{S}^\top + \underline{\epsilon}_d \mathbb{E}_{\mathcal{Q}_s}\left[\boldsymbol{\epsilon}^{(s)}\boldsymbol{\epsilon}^{(s)\top}\right] \\
&\succeq \underline{\epsilon}_f \mathbf{I}_M + \underline{\epsilon}_s \mathrm{tr}(\mathbf{H})\mathbf{I}_M + \mathbf{S}\mathbf{H}\mathbf{S}^\top + \underline{\epsilon}_d \mathbf{S}\mathbf{S}^\top + \underline{\epsilon}_d \underline{\epsilon}_s p \mathbf{I}_M \\
&= (\underline{\epsilon}_f + (\mathrm{tr}(\mathbf{H}) + \underline{\epsilon}_d p)\underline{\epsilon}_s)\mathbf{I}_M + \mathbf{S}\mathbf{H}\mathbf{S}^\top + \underline{\epsilon}_d \mathbf{S}\mathbf{S}^\top.
\end{aligned}
$$

We then focus on the eigenvalues of $\mathbf{S}\mathbf{S}^\top$. We write

$$\mathbf{S} = (\mathbf{s}_1, ..., \mathbf{s}_p), \quad \mathbf{s}_i \sim \mathcal{N}\left(0, \frac{1}{M}\mathbf{I}_M\right), \quad i \geq 1.$$

For any unit vector $\mathbf{v} \in \mathbb{R}^M$, each $\mathbf{s}_i^\top \mathbf{v}$ is sub-Gaussian. By Bernstein inequality, with probability at least $1 - e^{-\Omega(M)}$, for every unit vector $\mathbf{v} \in \mathbb{R}^M$, [15]

$$\mathbf{v}^\top \mathbf{S}\mathbf{S}^\top \mathbf{v} = \sum_{i=1}^p (\mathbf{s}_i^\top \mathbf{v})^2 \asymp \frac{p}{M}.$$

That is, with probability at least $1 - e^{-\Omega(M)}$,

$$\mu_{\min}(\underline{\epsilon}_d \mathbf{S}\mathbf{S}^\top) \asymp \underline{\epsilon}_d \frac{p}{M}, \quad \mu_{\max}(\underline{\epsilon}_d \mathbf{S}\mathbf{S}^\top) \asymp \underline{\epsilon}_d \frac{p}{M}.$$

Therefore, together with Lemma G.1, with probability at least $1 - e^{-\Omega(M)}$ for all $j \in [M]$ that

$$\mu_j(\mathbf{H}_f^{(q)}) \gtrsim j^{-a} + \underline{\epsilon}_f + \underline{\epsilon}_s(1 + \underline{\epsilon}_d p) + \underline{\epsilon}_d \frac{p}{M}. \tag{98}$$

$\square$

**Lemma G.6** (Ratio of eigenvalues of $\mathbf{S}\mathbf{I}_{k:\infty}\mathbf{H}_{k:\infty}\mathbf{I}_{k:\infty}\mathbf{S}^\top$, Lemma G.5 in Lin et al. (2024)). *Under Assumption 3.6, there exists some $a$-dependent constant $c > 0$ such that for all $k \geq 1$, the ratio between the $M/2$-th and $M$-th eigenvalues*

$$\frac{\mu_{M/2}\left(\mathbf{S}\mathbf{I}_{k:\infty}\mathbf{H}_{k:\infty}\mathbf{I}_{k:\infty}\mathbf{S}^\top\right)}{\mu_M\left(\mathbf{S}\mathbf{I}_{k:\infty}\mathbf{H}_{k:\infty}\mathbf{I}_{k:\infty}\mathbf{S}^\top\right)} \leq c$$

*with probability at least $1 - e^{-\Omega(M)}$. Further, for $k \leq M$,*

$$\mu_{M/2}\left(\mathbf{S}\mathbf{I}_{k:\infty}\mathbf{H}_{k:\infty}\mathbf{I}_{k:\infty}\mathbf{S}^\top\right) \lesssim M^{-a}, \quad \mu_M\left(\mathbf{S}\mathbf{I}_{k:\infty}\mathbf{H}_{k:\infty}\mathbf{I}_{k:\infty}\mathbf{S}^\top\right) \gtrsim M^{-a}.$$

## H. Discussions on Assumptions

In this section, we aim to verify Assumption 3.4 and Assumption 3.5 under the fourth order assumption and noise assumption with respect to full-precision data. For simplicity, we verify the upper bounds here.

**Assumption H.1.** There is a constant $\alpha_0 > 0$, such that for every PSD matrix $\mathbf{A}$, we have

$$\mathbb{E}[\mathbf{x}^\top \mathbf{A}\mathbf{x}\mathbf{x}\mathbf{x}^\top] \preceq \alpha_0 \mathrm{tr}(\mathbf{H}\mathbf{A})\mathbf{H}.$$

---

[15]We consider $p \gg M$.

**Assumption H.2.** There exist constants $\bar{\sigma}_0^2, C_y$ such that

$$\mathbb{E}\left[(y - \langle \mathbf{w}^*, \mathbf{x} \rangle)^2 \mathbf{x}\mathbf{x}^\top\right] \preceq \bar{\sigma}_0^2 \mathbf{H}, \ \mathbb{E}[y^2 \mathbf{x}\mathbf{x}^\top] \preceq C_y \mathbf{H}, \ \mathbb{E}\left[(y - \langle \mathbf{w}^*, \mathbf{x} \rangle)^2\right] \le \bar{\sigma}_0^2.$$

We consider specific quantization schemes.

**Example H.1.** *Consider the following element-wise stochastic quantization*

$$\mathcal{Q}(x) = s \cdot \begin{cases} \lfloor \frac{x}{s} \rfloor, \ \text{w.p.} \ \frac{\lceil x/s \rceil - x/s}{\lceil x/s \rceil - \lfloor x/s \rfloor} \\ \lceil \frac{x}{s} \rceil, \ \text{w.p.} \ \frac{x/s - \lfloor x/s \rfloor}{\lceil x/s \rceil - \lfloor x/s \rfloor} \end{cases} \ , \quad s = \begin{cases} 2^{\lfloor \log_2 x \rfloor - m}, & \text{multiplicative} \\ 2^{-b}, & \text{additive} \end{cases}.$$

We first compute the conditional second moment and fourth moment under Example H.1. Regarding the conditional second moment,

$$\begin{aligned} \mathbb{E}\left[(\mathcal{Q}(x) - x)^2 | x\right] &= \left(x - s\lfloor \frac{x}{s} \rfloor\right)^2 \frac{\lceil x/s \rceil - x/s}{\lceil x/s \rceil - \lfloor x/s \rfloor} + \left(s\lceil \frac{x}{s} \rceil - x\right)^2 \frac{x/s - \lfloor x/s \rfloor}{\lceil x/s \rceil - \lfloor x/s \rfloor} \\ &= s^2 \left(\lceil x/s \rceil - x/s\right)\left(x/s - \lfloor x/s \rfloor\right) \\ &\lesssim \begin{cases} x^2 2^{-2m}, & \text{multiplicative} \\ 2^{-2b}, & \text{additive} \end{cases}. \end{aligned} \tag{99}$$

Regarding the fourth moment,

$$\begin{aligned} \mathbb{E}\left[(\mathcal{Q}(x) - x)^4 | x\right] &= \left(x - s\lfloor \frac{x}{s} \rfloor\right)^4 \frac{\lceil x/s \rceil - x/s}{\lceil x/s \rceil - \lfloor x/s \rfloor} + \left(s\lceil \frac{x}{s} \rceil - x\right)^4 \frac{x/s - \lfloor x/s \rfloor}{\lceil x/s \rceil - \lfloor x/s \rfloor} \\ &= s^4 \left(\lceil x/s \rceil - x/s\right)\left(x/s - \lfloor x/s \rfloor\right)\left[\left(x/s - \lfloor \frac{x}{s} \rfloor\right)^3 + \left(\lceil \frac{x}{s} \rceil - x/s\right)^3\right] \\ &\lesssim \begin{cases} x^4 2^{-4m}, & \text{multiplicative} \\ 2^{-4b}, & \text{additive} \end{cases}. \end{aligned} \tag{100}$$

Motivated by (100), we consider the strong multiplicative and additive quantization below for theoretical simplicity. We then verify the upper bounds in Assumption 3.4 and Assumption 3.5 under these quantization schemes.

**Definition H.3** (Strong multiplicative quantization). We call quantization $\mathcal{Q}$ is strong $\epsilon$-multiplicative if

$$\mathbb{E}\left[\boldsymbol{\epsilon}^\top \mathbf{A}\boldsymbol{\epsilon}\boldsymbol{\epsilon}^\top | \mathbf{u}\right] \preceq \epsilon \mathbf{u}^\top \mathbf{A}\mathbf{u}\mathbf{u}^\top, \quad \forall \mathbf{A},$$

where $\boldsymbol{\epsilon} = \mathcal{Q}(\mathbf{u}) - u$. We would like to remark that, for multiplicative quantization to matrix $\mathbf{X}$, we extend the definition to $\mathbb{E}\left[\mathrm{tr}\left(\mathbf{A}\boldsymbol{\Xi}\mathbf{B}\boldsymbol{\Xi}^\top\right)\boldsymbol{\Xi}\mathbf{B}\boldsymbol{\Xi}^\top | \mathbf{U}\right] \preceq \epsilon \mathrm{tr}\left(\mathbf{A}\mathbf{U}\mathbf{B}\mathbf{U}^\top\right)\mathbf{U}\mathbf{B}\mathbf{U}^\top$ for any PSD matrix $\mathbf{B}$, where $\boldsymbol{\Xi} = \mathcal{Q}(\mathbf{U}) - \mathbf{U}$.

**Definition H.4** (Strong additive quantization). We call quantization $\mathcal{Q}$ is strong $\epsilon$-additive if

$$\mathbb{E}\left[\boldsymbol{\epsilon}^\top \mathbf{A}\boldsymbol{\epsilon}\boldsymbol{\epsilon}^\top | \mathbf{u}\right] \preceq \epsilon \mathrm{tr}(\mathbf{A})\mathbf{I}, \quad \forall \mathbf{A},$$

where $\boldsymbol{\epsilon} = \mathcal{Q}(\mathbf{u}) - u$. We would like to remark that, for additive quantization to matrix $\mathbf{X}$, we extend the definition to $\mathbb{E}\left[\mathrm{tr}\left(\mathbf{A}\boldsymbol{\Xi}\mathbf{B}\boldsymbol{\Xi}^\top\right)\boldsymbol{\Xi}\mathbf{B}\boldsymbol{\Xi}^\top | \mathbf{U}\right] \preceq \epsilon \mathrm{tr}(\mathbf{A})\mathrm{tr}(\mathbf{B})^2\mathbf{I}$ for any PSD matrix $\mathbf{B}$, where $\boldsymbol{\Xi} = \mathcal{Q}(\mathbf{U}) - \mathbf{U}$.

### H.1. Fourth-order Assumption

We aim to verify the fourth-order Assumption 3.4 in this subsection. Rewrite $\tilde{\mathbf{x}}^{(q)}$ as $\tilde{\mathbf{x}}^{(q)} = \mathbf{S}^{(q)}\mathbf{x}^{(q)} + \boldsymbol{\epsilon}^{(f)}$, $\mathbf{x}^{(q)} = \mathbf{x} + \boldsymbol{\epsilon}^{(d)}$.

$$\begin{aligned} \mathbb{E}\left[(\tilde{\mathbf{x}}^{(q)})^\top \mathbf{A}\tilde{\mathbf{x}}^{(q)}\tilde{\mathbf{x}}^{(q)}(\tilde{\mathbf{x}}^{(q)})^\top\right] &= \mathbb{E}\left[(\tilde{\mathbf{x}}^{(q)})^\top \mathbf{A}\tilde{\mathbf{x}}^{(q)}(\mathbf{S}^{(q)}\mathbf{x}^{(q)} + \boldsymbol{\epsilon}^{(f)})(\mathbf{S}^{(q)}\mathbf{x}^{(q)} + \boldsymbol{\epsilon}^{(f)})^\top\right] \\ &\preceq 2\mathbb{E}\left[(\tilde{\mathbf{x}}^{(q)})^\top \mathbf{A}\tilde{\mathbf{x}}^{(q)}\mathbf{S}^{(q)}\mathbf{x}^{(q)}\mathbf{x}^{(q)\top}\mathbf{S}^{(q)\top}\right] + 2\mathbb{E}\left[(\tilde{\mathbf{x}}^{(q)})^\top \mathbf{A}\tilde{\mathbf{x}}^{(q)}\boldsymbol{\epsilon}^{(f)}\boldsymbol{\epsilon}^{(f)\top}\right]. \end{aligned}$$

Note that by $\tilde{\mathbf{x}}^{(q)} = \mathbf{S}^{(q)}\mathbf{x}^{(q)} + \boldsymbol{\epsilon}^{(f)}$, we have

$$\begin{aligned} (\tilde{\mathbf{x}}^{(q)})^\top \mathbf{A}\tilde{\mathbf{x}}^{(q)} &= (\mathbf{S}^{(q)}\mathbf{x}^{(q)} + \boldsymbol{\epsilon}^{(f)})^\top \mathbf{A}(\mathbf{S}^{(q)}\mathbf{x}^{(q)} + \boldsymbol{\epsilon}^{(f)}) \\ &\le 2(\mathbf{S}^{(q)}\mathbf{x}^{(q)})^\top \mathbf{A}(\mathbf{S}^{(q)}\mathbf{x}^{(q)}) + 2\boldsymbol{\epsilon}^{(f)\top}\mathbf{A}\boldsymbol{\epsilon}^{(f)}, \end{aligned}$$

it follows that

$$\mathbb{E}\left[(\tilde{\mathbf{x}}^{(q)})^\top \mathbf{A}\tilde{\mathbf{x}}^{(q)}\tilde{\mathbf{x}}^{(q)}(\tilde{\mathbf{x}}^{(q)})^\top\right] \preceq 4\mathbb{E}\left[\mathbf{x}^{(q)^\top}\mathbf{S}^{(q)^\top}\mathbf{A}\mathbf{S}^{(q)}\mathbf{x}^{(q)}\mathbf{S}^{(q)}\mathbf{x}^{(q)}\mathbf{x}^{(q)^\top}\mathbf{S}^{(q)^\top}\right] + 4\mathbb{E}\left[\boldsymbol{\epsilon}^{(f)^\top}\mathbf{A}\boldsymbol{\epsilon}^{(f)}\boldsymbol{\epsilon}^{(f)}\boldsymbol{\epsilon}^{(f)^\top}\right]$$
$$+ 4\mathbb{E}\left[\mathbf{x}^{(q)^\top}\mathbf{S}^{(q)^\top}\mathbf{A}\mathbf{S}^{(q)}\mathbf{x}^{(q)}\boldsymbol{\epsilon}^{(f)}\boldsymbol{\epsilon}^{(f)^\top}\right] + 4\mathbb{E}\left[\boldsymbol{\epsilon}^{(f)^\top}\mathbf{A}\boldsymbol{\epsilon}^{(f)}\mathbf{S}^{(q)}\mathbf{x}^{(q)}\mathbf{x}^{(q)^\top}\mathbf{S}^{(q)^\top}\right].$$

Further note that $\mathbf{x}^{(q)} = \mathbf{x} + \boldsymbol{\epsilon}^{(d)}$, we have

$$\mathbf{x}^{(q)^\top}\mathbf{S}^{(q)^\top}\mathbf{A}\mathbf{S}^{(q)}\mathbf{x}^{(q)} \leq 2\mathbf{x}^\top\mathbf{S}^{(q)^\top}\mathbf{A}\mathbf{S}^{(q)}\mathbf{x} + 2\boldsymbol{\epsilon}^{(d)^\top}\mathbf{S}^{(q)^\top}\mathbf{A}\mathbf{S}^{(q)}\boldsymbol{\epsilon}^{(d)},$$
$$\mathbf{S}^{(q)}\mathbf{x}^{(q)}\mathbf{x}^{(q)^\top}\mathbf{S}^{(q)^\top} \preceq 2\mathbf{S}^{(q)}\mathbf{x}\mathbf{x}^\top\mathbf{S}^{(q)^\top} + 2\mathbf{S}^{(q)}\boldsymbol{\epsilon}^{(d)}\boldsymbol{\epsilon}^{(d)^\top}\mathbf{S}^{(q)^\top},$$

it follows that

$$\mathbb{E}\left[(\tilde{\mathbf{x}}^{(q)})^\top \mathbf{A}\tilde{\mathbf{x}}^{(q)}\tilde{\mathbf{x}}^{(q)}(\tilde{\mathbf{x}}^{(q)})^\top\right] \preceq 16\mathbb{E}\left[\mathbf{x}^\top\mathbf{S}^{(q)^\top}\mathbf{A}\mathbf{S}^{(q)}\mathbf{x}\mathbf{S}^{(q)}\mathbf{x}\mathbf{x}^\top\mathbf{S}^{(q)^\top}\right]$$
$$+ 16\mathbb{E}\left[\mathbf{x}^\top\mathbf{S}^{(q)^\top}\mathbf{A}\mathbf{S}^{(q)}\mathbf{x}\mathbf{S}^{(q)}\boldsymbol{\epsilon}^{(d)}\boldsymbol{\epsilon}^{(d)^\top}\mathbf{S}^{(q)^\top}\right]$$
$$+ 16\mathbb{E}\left[\boldsymbol{\epsilon}^{(d)^\top}\mathbf{S}^{(q)^\top}\mathbf{A}\mathbf{S}^{(q)}\boldsymbol{\epsilon}^{(d)}\mathbf{S}^{(q)}\mathbf{x}\mathbf{x}^\top\mathbf{S}^{(q)^\top}\right]$$
$$+ 16\mathbb{E}\left[\boldsymbol{\epsilon}^{(d)^\top}\mathbf{S}^{(q)^\top}\mathbf{A}\mathbf{S}^{(q)}\boldsymbol{\epsilon}^{(d)}\mathbf{S}^{(q)}\boldsymbol{\epsilon}^{(d)}\boldsymbol{\epsilon}^{(d)^\top}\mathbf{S}^{(q)^\top}\right] \qquad (101)$$
$$+ 4\mathbb{E}\left[\boldsymbol{\epsilon}^{(f)^\top}\mathbf{A}\boldsymbol{\epsilon}^{(f)}\boldsymbol{\epsilon}^{(f)}\boldsymbol{\epsilon}^{(f)^\top}\right]$$
$$+ 8\mathbb{E}\left[\mathbf{x}^\top\mathbf{S}^{(q)^\top}\mathbf{A}\mathbf{S}^{(q)}\mathbf{x}\boldsymbol{\epsilon}^{(f)}\boldsymbol{\epsilon}^{(f)^\top}\right] + 8\mathbb{E}\left[\boldsymbol{\epsilon}^{(d)^\top}\mathbf{S}^{(q)^\top}\mathbf{A}\mathbf{S}^{(q)}\boldsymbol{\epsilon}^{(d)}\boldsymbol{\epsilon}^{(f)}\boldsymbol{\epsilon}^{(f)^\top}\right]$$
$$+ 8\mathbb{E}\left[\boldsymbol{\epsilon}^{(f)^\top}\mathbf{A}\boldsymbol{\epsilon}^{(f)}\mathbf{S}^{(q)}\mathbf{x}\mathbf{x}^\top\mathbf{S}^{(q)^\top}\right] + 8\mathbb{E}\left[\boldsymbol{\epsilon}^{(f)^\top}\mathbf{A}\boldsymbol{\epsilon}^{(f)}\mathbf{S}^{(q)}\boldsymbol{\epsilon}^{(d)}\boldsymbol{\epsilon}^{(d)^\top}\mathbf{S}^{(q)^\top}\right].$$

We then bound $\mathbb{E}\left[(\tilde{\mathbf{x}}^{(q)})^\top \mathbf{A}\tilde{\mathbf{x}}^{(q)}\tilde{\mathbf{x}}^{(q)}(\tilde{\mathbf{x}}^{(q)})^\top\right]$ in (strong) multiplicative and (strong) additive quantization cases respectively, under assumptions on full-precision data.

### H.1.1. MULTIPLICATIVE QUANTIZATION

**Lemma H.5.** *If for each $i = d, f, s$, $\mathcal{Q}_i$ is $(\underline{\epsilon}_i, \bar{\epsilon}_i)$-multiplicative and $\mathcal{Q}_i$ is strong $\bar{\epsilon}'_i$-multiplicative, then under Assumption H.1, we have*

$$\mathbb{E}\left[(\tilde{\mathbf{x}}^{(q)})^\top \mathbf{A}\tilde{\mathbf{x}}^{(q)}\tilde{\mathbf{x}}^{(q)}(\tilde{\mathbf{x}}^{(q)})^\top\right] \leq \alpha_M \text{tr}\left(\mathbf{H}_f^{(q)}\mathbf{A}\right)\mathbf{H}_f^{(q)},$$

*where $\alpha_M \lesssim \frac{\alpha_0(1+\bar{\epsilon}_f+\bar{\epsilon}'_f)(1+\bar{\epsilon}_d+\bar{\epsilon}'_d)(1+\bar{\epsilon}_s+\bar{\epsilon}'_s)}{(1+\underline{\epsilon}_f)^2(1+\underline{\epsilon}_d)^2(1+\underline{\epsilon}_s)^2}$.*

*Proof.* We prove by (101). Under Assumption H.1,

$$\mathbb{E}\left[\mathbf{x}^\top\mathbf{S}^{(q)^\top}\mathbf{A}\mathbf{S}^{(q)}\mathbf{x}\mathbf{S}^{(q)}\mathbf{x}\mathbf{x}^\top\mathbf{S}^{(q)^\top}\right] \preceq \alpha_0\mathbb{E}\left[\mathbf{S}^{(q)}\text{tr}\left(\mathbf{H}\mathbf{S}^{(q)^\top}\mathbf{A}\mathbf{S}^{(q)}\right)\mathbf{H}\mathbf{S}^{(q)^\top}\right]. \qquad (102)$$

As $\mathcal{Q}_d$ is $\bar{\epsilon}_d$-multiplicative, it holds

$$\mathbb{E}\left[\mathbf{x}^\top\mathbf{S}^{(q)^\top}\mathbf{A}\mathbf{S}^{(q)}\mathbf{x}\mathbf{S}^{(q)}\boldsymbol{\epsilon}^{(d)}\boldsymbol{\epsilon}^{(d)^\top}\mathbf{S}^{(q)^\top}\right] \preceq \bar{\epsilon}_d\mathbb{E}\left[\mathbf{x}^\top\mathbf{S}^{(q)^\top}\mathbf{A}\mathbf{S}^{(q)}\mathbf{x}\mathbf{S}^{(q)}\mathbf{x}\mathbf{x}^\top\mathbf{S}^{(q)^\top}\right]$$
$$\preceq \bar{\epsilon}_d\alpha_0\mathbb{E}\left[\mathbf{S}^{(q)}\text{tr}\left(\mathbf{H}\mathbf{S}^{(q)^\top}\mathbf{A}\mathbf{S}^{(q)}\right)\mathbf{H}\mathbf{S}^{(q)^\top}\right], \qquad (103)$$

where the last inequality reuses (102). Similarly,

$$\mathbb{E}\left[\boldsymbol{\epsilon}^{(d)^\top}\mathbf{S}^{(q)^\top}\mathbf{A}\mathbf{S}^{(q)}\boldsymbol{\epsilon}^{(d)}\mathbf{S}^{(q)}\mathbf{x}\mathbf{x}^\top\mathbf{S}^{(q)^\top}\right] \preceq \bar{\epsilon}_d\mathbb{E}\left[\mathbf{x}^\top\mathbf{S}^{(q)^\top}\mathbf{A}\mathbf{S}^{(q)}\mathbf{x}\mathbf{S}^{(q)}\mathbf{x}\mathbf{x}^\top\mathbf{S}^{(q)^\top}\right]$$
$$\preceq \bar{\epsilon}_d\alpha_0\mathbb{E}\left[\mathbf{S}^{(q)}\text{tr}\left(\mathbf{H}\mathbf{S}^{(q)^\top}\mathbf{A}\mathbf{S}^{(q)}\right)\mathbf{H}\mathbf{S}^{(q)^\top}\right]. \qquad (104)$$

Note that $\mathcal{Q}_f$ is $\bar{\epsilon}_f$-multiplicative, it follows

$$
\begin{aligned}
\mathbb{E}\left[\mathbf{x}^\top {\mathbf{S}^{(q)}}^\top \mathbf{A}\mathbf{S}^{(q)}\mathbf{x}\boldsymbol{\epsilon}^{(f)}{\boldsymbol{\epsilon}^{(f)}}^\top\right] \preceq &\ \bar{\epsilon}_f \mathbb{E}\left[\mathbf{x}^\top {\mathbf{S}^{(q)}}^\top \mathbf{A}\mathbf{S}^{(q)}\mathbf{x}\mathbf{S}^{(q)}(\mathbf{x}+\boldsymbol{\epsilon}^{(d)})(\mathbf{x}+\boldsymbol{\epsilon}^{(d)})^\top {\mathbf{S}^{(q)}}^\top\right] \\
\preceq &\ 2\bar{\epsilon}_f \mathbb{E}\left[\mathbf{x}^\top {\mathbf{S}^{(q)}}^\top \mathbf{A}\mathbf{S}^{(q)}\mathbf{x}\mathbf{S}^{(q)}\mathbf{x}\mathbf{x}^\top {\mathbf{S}^{(q)}}^\top\right] \\
&+2\bar{\epsilon}_f \mathbb{E}\left[\mathbf{x}^\top {\mathbf{S}^{(q)}}^\top \mathbf{A}\mathbf{S}^{(q)}\mathbf{x}\mathbf{S}^{(q)}\boldsymbol{\epsilon}^{(d)}{\boldsymbol{\epsilon}^{(d)}}^\top {\mathbf{S}^{(q)}}^\top\right] \\
\preceq &\ 2\bar{\epsilon}_f(1+\bar{\epsilon}_d)\mathbb{E}\left[\mathbf{x}^\top {\mathbf{S}^{(q)}}^\top \mathbf{A}\mathbf{S}^{(q)}\mathbf{x}\mathbf{S}^{(q)}\mathbf{x}\mathbf{x}^\top {\mathbf{S}^{(q)}}^\top\right] \\
\preceq &\ 2\bar{\epsilon}_f(1+\bar{\epsilon}_d)\alpha_0 \mathbb{E}\left[\mathbf{S}^{(q)}\mathrm{tr}\left(\mathbf{H}{\mathbf{S}^{(q)}}^\top \mathbf{A}\mathbf{S}^{(q)}\right)\mathbf{H}{\mathbf{S}^{(q)}}^\top\right],
\end{aligned}
\tag{105}
$$

where the third inequality holds as $\mathcal{Q}_d$ is $\bar{\epsilon}_d$-multiplicative and the last inequality reuses (102). Similarly,

$$
\begin{aligned}
\mathbb{E}\left[{\boldsymbol{\epsilon}^{(d)}}^\top {\mathbf{S}^{(q)}}^\top \mathbf{A}\mathbf{S}^{(q)}\boldsymbol{\epsilon}^{(d)}\boldsymbol{\epsilon}^{(f)}{\boldsymbol{\epsilon}^{(f)}}^\top\right] \preceq &\ \bar{\epsilon}_f \mathbb{E}\left[{\boldsymbol{\epsilon}^{(d)}}^\top {\mathbf{S}^{(q)}}^\top \mathbf{A}\mathbf{S}^{(q)}\boldsymbol{\epsilon}^{(d)}\mathbf{S}^{(q)}(\mathbf{x}+\boldsymbol{\epsilon}^{(d)})(\mathbf{x}+\boldsymbol{\epsilon}^{(d)})^\top {\mathbf{S}^{(q)}}^\top\right] \\
\preceq &\ 2\bar{\epsilon}_f \mathbb{E}\left[{\boldsymbol{\epsilon}^{(d)}}^\top {\mathbf{S}^{(q)}}^\top \mathbf{A}\mathbf{S}^{(q)}\boldsymbol{\epsilon}^{(d)}\mathbf{S}^{(q)}(\mathbf{x}\mathbf{x}^\top+\boldsymbol{\epsilon}^{(d)}{\boldsymbol{\epsilon}^{(d)}}^\top){\mathbf{S}^{(q)}}^\top\right] \\
\preceq &\ 2\bar{\epsilon}_f\bar{\epsilon}_d\alpha_0 \mathbb{E}\left[\mathbf{S}^{(q)}\mathrm{tr}\left(\mathbf{H}{\mathbf{S}^{(q)}}^\top \mathbf{A}\mathbf{S}^{(q)}\right)\mathbf{H}{\mathbf{S}^{(q)}}^\top\right] \\
&+2\bar{\epsilon}_f \mathbb{E}\left[{\boldsymbol{\epsilon}^{(d)}}^\top {\mathbf{S}^{(q)}}^\top \mathbf{A}\mathbf{S}^{(q)}\boldsymbol{\epsilon}^{(d)}\mathbf{S}^{(q)}\boldsymbol{\epsilon}^{(d)}{\boldsymbol{\epsilon}^{(d)}}^\top {\mathbf{S}^{(q)}}^\top\right].
\end{aligned}
\tag{106}
$$

$$
\begin{aligned}
\mathbb{E}\left[{\boldsymbol{\epsilon}^{(f)}}^\top \mathbf{A}\boldsymbol{\epsilon}^{(f)}\mathbf{S}^{(q)}\mathbf{x}\mathbf{x}^\top {\mathbf{S}^{(q)}}^\top\right] \preceq &\ \bar{\epsilon}_f \mathbb{E}\left[(\mathbf{x}+\boldsymbol{\epsilon}^{(d)})^\top {\mathbf{S}^{(q)}}^\top \mathbf{A}\mathbf{S}^{(q)}(\mathbf{x}+\boldsymbol{\epsilon}^{(d)})\mathbf{S}^{(q)}\mathbf{x}\mathbf{x}^\top {\mathbf{S}^{(q)}}^\top\right] \\
\preceq &\ 2\bar{\epsilon}_f \mathbb{E}\left[\mathbf{x}^\top {\mathbf{S}^{(q)}}^\top \mathbf{A}\mathbf{S}^{(q)}\mathbf{x}\mathbf{S}^{(q)}\mathbf{x}\mathbf{x}^\top {\mathbf{S}^{(q)}}^\top\right] \\
&+2\bar{\epsilon}_f \mathbb{E}\left[{\boldsymbol{\epsilon}^{(d)}}^\top {\mathbf{S}^{(q)}}^\top \mathbf{A}\mathbf{S}^{(q)}\boldsymbol{\epsilon}^{(d)}\mathbf{S}^{(q)}\mathbf{x}\mathbf{x}^\top {\mathbf{S}^{(q)}}^\top\right] \\
\preceq &\ 2\bar{\epsilon}_f(1+\bar{\epsilon}_d)\alpha_0 \mathbb{E}\left[\mathbf{S}^{(q)}\mathrm{tr}\left(\mathbf{H}{\mathbf{S}^{(q)}}^\top \mathbf{A}\mathbf{S}^{(q)}\right)\mathbf{H}{\mathbf{S}^{(q)}}^\top\right].
\end{aligned}
\tag{107}
$$

$$
\begin{aligned}
\mathbb{E}\left[{\boldsymbol{\epsilon}^{(f)}}^\top \mathbf{A}\boldsymbol{\epsilon}^{(f)}\mathbf{S}^{(q)}\boldsymbol{\epsilon}^{(d)}{\boldsymbol{\epsilon}^{(d)}}^\top {\mathbf{S}^{(q)}}^\top\right] \preceq &\ \bar{\epsilon}_f \mathbb{E}\left[(\mathbf{x}+\boldsymbol{\epsilon}^{(d)})^\top {\mathbf{S}^{(q)}}^\top \mathbf{A}\mathbf{S}^{(q)}(\mathbf{x}+\boldsymbol{\epsilon}^{(d)})\mathbf{S}^{(q)}\boldsymbol{\epsilon}^{(d)}{\boldsymbol{\epsilon}^{(d)}}^\top {\mathbf{S}^{(q)}}^\top\right] \\
\preceq &\ 2\bar{\epsilon}_f \mathbb{E}\left[\mathbf{x}^\top {\mathbf{S}^{(q)}}^\top \mathbf{A}\mathbf{S}^{(q)}\mathbf{x}\mathbf{S}^{(q)}\boldsymbol{\epsilon}^{(d)}{\boldsymbol{\epsilon}^{(d)}}^\top {\mathbf{S}^{(q)}}^\top\right] \\
&+2\bar{\epsilon}_f \mathbb{E}\left[{\boldsymbol{\epsilon}^{(d)}}^\top {\mathbf{S}^{(q)}}^\top \mathbf{A}\mathbf{S}^{(q)}\boldsymbol{\epsilon}^{(d)}\mathbf{S}^{(q)}\boldsymbol{\epsilon}^{(d)}{\boldsymbol{\epsilon}^{(d)}}^\top {\mathbf{S}^{(q)}}^\top\right] \\
\preceq &\ 2\bar{\epsilon}_f\bar{\epsilon}_d\alpha_0 \mathbb{E}\left[\mathbf{S}^{(q)}\mathrm{tr}\left(\mathbf{H}{\mathbf{S}^{(q)}}^\top \mathbf{A}\mathbf{S}^{(q)}\right)\mathbf{H}{\mathbf{S}^{(q)}}^\top\right] \\
&+2\bar{\epsilon}_f \mathbb{E}\left[{\boldsymbol{\epsilon}^{(d)}}^\top {\mathbf{S}^{(q)}}^\top \mathbf{A}\mathbf{S}^{(q)}\boldsymbol{\epsilon}^{(d)}\mathbf{S}^{(q)}\boldsymbol{\epsilon}^{(d)}{\boldsymbol{\epsilon}^{(d)}}^\top {\mathbf{S}^{(q)}}^\top\right].
\end{aligned}
\tag{108}
$$

Regarding the fourth-order quantization terms, by the definition of strong multiplicative quantization (Definition H.3), it holds

$$
\begin{aligned}
\mathbb{E}\left[{\boldsymbol{\epsilon}^{(d)}}^\top {\mathbf{S}^{(q)}}^\top \mathbf{A}\mathbf{S}^{(q)}\boldsymbol{\epsilon}^{(d)}\mathbf{S}^{(q)}\boldsymbol{\epsilon}^{(d)}{\boldsymbol{\epsilon}^{(d)}}^\top {\mathbf{S}^{(q)}}^\top\right] \preceq &\ \bar{\epsilon}'_d \mathbb{E}\left[\mathbf{x}^\top {\mathbf{S}^{(q)}}^\top \mathbf{A}\mathbf{S}^{(q)}\mathbf{x}\mathbf{S}^{(q)}\mathbf{x}\mathbf{x}^\top {\mathbf{S}^{(q)}}^\top\right] \\
\preceq &\ \bar{\epsilon}'_d\alpha_0 \mathbb{E}\left[\mathbf{S}^{(q)}\mathrm{tr}\left(\mathbf{H}{\mathbf{S}^{(q)}}^\top \mathbf{A}\mathbf{S}^{(q)}\right)\mathbf{H}{\mathbf{S}^{(q)}}^\top\right].
\end{aligned}
\tag{109}
$$

$$\mathbb{E}\left[\boldsymbol{\epsilon}^{(f)\top}\mathbf{A}\boldsymbol{\epsilon}^{(f)}\boldsymbol{\epsilon}^{(f)\top}\right] \preceq \bar{\epsilon}'_f\mathbb{E}\left[(\mathbf{x}+\boldsymbol{\epsilon}^{(d)})^\top\mathbf{S}^{(q)\top}\mathbf{S}^{(q)\top}\mathbf{A}\mathbf{S}^{(q)}(\mathbf{x}+\boldsymbol{\epsilon}^{(d)})\mathbf{S}^{(q)}(\mathbf{x}+\boldsymbol{\epsilon}^{(d)})(\mathbf{x}+\boldsymbol{\epsilon}^{(d)})^\top\mathbf{S}^{(q)\top}\right]$$

$$\preceq 4\bar{\epsilon}'_f\mathbb{E}\left[\mathbf{x}^\top\mathbf{S}^{(q)\top}\mathbf{S}^{(q)\top}\mathbf{A}\mathbf{S}^{(q)}\mathbf{x}\mathbf{S}^{(q)}\mathbf{x}\mathbf{x}^\top\mathbf{S}^{(q)\top}\right]$$

$$+4\bar{\epsilon}'_f\mathbb{E}\left[\mathbf{x}^\top\mathbf{S}^{(q)\top}\mathbf{S}^{(q)\top}\mathbf{A}\mathbf{S}^{(q)}\mathbf{x}\mathbf{S}^{(q)}\boldsymbol{\epsilon}^{(d)}\boldsymbol{\epsilon}^{(d)\top}\mathbf{S}^{(q)\top}\right]$$

$$+4\bar{\epsilon}'_f\mathbb{E}\left[\boldsymbol{\epsilon}^{(d)\top}\mathbf{S}^{(q)\top}\mathbf{S}^{(q)\top}\mathbf{A}\mathbf{S}^{(q)}\boldsymbol{\epsilon}^{(d)}\mathbf{S}^{(q)}\boldsymbol{\epsilon}^{(d)}\boldsymbol{\epsilon}^{(d)\top}\mathbf{S}^{(q)\top}\right] \tag{110}$$

$$+4\bar{\epsilon}'_f\mathbb{E}\left[\boldsymbol{\epsilon}^{(d)\top}\mathbf{S}^{(q)\top}\mathbf{S}^{(q)\top}\mathbf{A}\mathbf{S}^{(q)}\boldsymbol{\epsilon}^{(d)}\mathbf{S}^{(q)}\mathbf{x}\mathbf{x}^\top\mathbf{S}^{(q)\top}\right]$$

$$\preceq 4\bar{\epsilon}'_f\alpha_0\mathbb{E}\left[\mathbf{S}^{(q)}\mathrm{tr}\left(\mathbf{H}\mathbf{S}^{(q)\top}\mathbf{A}\mathbf{S}^{(q)}\right)\mathbf{H}\mathbf{S}^{(q)\top}\right]$$

$$+8\bar{\epsilon}'_f\bar{\epsilon}_d\alpha_0\mathbb{E}\left[\mathbf{S}^{(q)}\mathrm{tr}\left(\mathbf{H}\mathbf{S}^{(q)\top}\mathbf{A}\mathbf{S}^{(q)}\right)\mathbf{H}\mathbf{S}^{(q)\top}\right]$$

$$+4\bar{\epsilon}'_f\bar{\epsilon}'_d\alpha_0\mathbb{E}\left[\mathbf{S}^{(q)}\mathrm{tr}\left(\mathbf{H}\mathbf{S}^{(q)\top}\mathbf{A}\mathbf{S}^{(q)}\right)\mathbf{H}\mathbf{S}^{(q)\top}\right].$$

Therefore, applying (101), it holds

$$\mathbb{E}\left[(\tilde{\mathbf{x}}^{(q)})^\top\mathbf{A}\tilde{\mathbf{x}}^{(q)}\tilde{\mathbf{x}}^{(q)}(\tilde{\mathbf{x}}^{(q)})^\top\right] \preceq C\alpha_0(1+\bar{\epsilon}_f+\bar{\epsilon}'_f)(1+\bar{\epsilon}_d+\bar{\epsilon}'_d)\mathbb{E}\left[\mathrm{tr}\left(\mathbf{S}^{(q)}\mathbf{H}\mathbf{S}^{(q)\top}\mathbf{A}\right)\mathbf{S}^{(q)}\mathbf{H}\mathbf{S}^{(q)\top}\right], \tag{111}$$

where $C > 0$ is a constant. Note that

$$\mathbb{E}\left[\mathrm{tr}\left(\mathbf{S}^{(q)}\mathbf{H}\mathbf{S}^{(q)\top}\mathbf{A}\right)\mathbf{S}^{(q)}\mathbf{H}\mathbf{S}^{(q)\top}\right] = \mathbb{E}\left[\mathrm{tr}\left((\mathbf{S}+\boldsymbol{\epsilon}^{(s)})\mathbf{H}(\mathbf{S}+\boldsymbol{\epsilon}^{(s)})^\top\mathbf{A}\right)\mathbf{S}^{(q)}\mathbf{H}\mathbf{S}^{(q)\top}\right]$$

$$\preceq 2\mathbb{E}\left[\mathrm{tr}\left(\mathbf{S}\mathbf{H}\mathbf{S}^\top\mathbf{A}\right)\mathbf{S}^{(q)}\mathbf{H}\mathbf{S}^{(q)\top}\right]$$

$$+2\mathbb{E}\left[\mathrm{tr}\left(\boldsymbol{\epsilon}^{(s)}\mathbf{H}\boldsymbol{\epsilon}^{(s)\top}\mathbf{A}\right)\mathbf{S}^{(q)}\mathbf{H}\mathbf{S}^{(q)\top}\right]$$

$$\preceq 4\mathbb{E}\left[\mathrm{tr}\left(\mathbf{S}\mathbf{H}\mathbf{S}^\top\mathbf{A}\right)\mathbf{S}\mathbf{H}\mathbf{S}^\top\right]$$

$$+4\mathbb{E}\left[\mathrm{tr}\left(\mathbf{S}\mathbf{H}\mathbf{S}^\top\mathbf{A}\right)\boldsymbol{\epsilon}^{(s)}\mathbf{H}\boldsymbol{\epsilon}^{(s)\top}\right] \tag{112}$$

$$+4\mathbb{E}\left[\mathrm{tr}\left(\boldsymbol{\epsilon}^{(s)}\mathbf{H}\boldsymbol{\epsilon}^{(s)\top}\mathbf{A}\right)\mathbf{S}\mathbf{H}\mathbf{S}^\top\right]$$

$$+4\mathbb{E}\left[\mathrm{tr}\left(\boldsymbol{\epsilon}^{(s)}\mathbf{H}\boldsymbol{\epsilon}^{(s)\top}\mathbf{A}\right)\boldsymbol{\epsilon}^{(s)}\mathbf{H}\boldsymbol{\epsilon}^{(s)\top}\right]$$

$$\preceq 4(1+2\bar{\epsilon}_s+\bar{\epsilon}'_s)\mathbb{E}\left[\mathrm{tr}\left(\mathbf{S}\mathbf{H}\mathbf{S}^\top\mathbf{A}\right)\mathbf{S}\mathbf{H}\mathbf{S}^\top\right],$$

we have

$$\mathbb{E}\left[(\tilde{\mathbf{x}}^{(q)})^\top\mathbf{A}\tilde{\mathbf{x}}^{(q)}\tilde{\mathbf{x}}^{(q)}(\tilde{\mathbf{x}}^{(q)})^\top\right] \preceq C'\alpha_0(1+\bar{\epsilon}_f+\bar{\epsilon}'_f)(1+\bar{\epsilon}_d+\bar{\epsilon}'_d)(1+\bar{\epsilon}_s+\bar{\epsilon}'_s)\mathrm{tr}\left(\mathbf{S}\mathbf{H}\mathbf{S}^\top\mathbf{A}\right)\mathbf{S}\mathbf{H}\mathbf{S}^\top, \tag{113}$$

where $C' > 0$ is a constant. By the definition of

$$\mathbf{H}_f^{(q)} = \mathbb{E}\left[\mathcal{Q}_f(\mathcal{Q}_s(\mathbf{S})\mathcal{Q}_d(\mathbf{x}))\mathcal{Q}_f(\mathcal{Q}_s(\mathbf{S})\mathcal{Q}_d(\mathbf{x}))^\top\right]$$

$$= \mathbb{E}\left[(\mathcal{Q}_s(\mathbf{S})\mathcal{Q}_d(\mathbf{x})+\boldsymbol{\epsilon}^{(f)})(\mathcal{Q}_s(\mathbf{S})\mathcal{Q}_d(\mathbf{x})+\boldsymbol{\epsilon}^{(f)})^\top\right]$$

$$\succeq (1+\underline{\epsilon}_f)\mathbb{E}\left[\mathcal{Q}_s(\mathbf{S})\mathcal{Q}_d(\mathbf{x})(\mathcal{Q}_s(\mathbf{S})\mathcal{Q}_d(\mathbf{x}))^\top\right] \tag{114}$$

$$\succeq (1+\underline{\epsilon}_f)(1+\underline{\epsilon}_d)\mathbb{E}\left[\mathcal{Q}_s(\mathbf{S})\mathbf{x}\mathbf{x}^\top\mathcal{Q}_s(\mathbf{S})^\top\right]$$

$$\succeq (1+\underline{\epsilon}_f)(1+\underline{\epsilon}_d)(1+\underline{\epsilon}_s)\mathbf{S}\mathbf{H}\mathbf{S}^\top,$$

together with (111) we have

$$\mathbb{E}\left[(\tilde{\mathbf{x}}^{(q)})^\top\mathbf{A}\tilde{\mathbf{x}}^{(q)}\tilde{\mathbf{x}}^{(q)}(\tilde{\mathbf{x}}^{(q)})^\top\right] \preceq C'\frac{\alpha_0(1+\bar{\epsilon}_f+\bar{\epsilon}'_f)(1+\bar{\epsilon}_d+\bar{\epsilon}'_d)(1+\bar{\epsilon}_s+\bar{\epsilon}'_s)}{(1+\underline{\epsilon}_f)^2(1+\underline{\epsilon}_d)^2(1+\underline{\epsilon}_s)^2}\mathrm{tr}\left(\mathbf{H}_f^{(q)}\mathbf{A}\right)\mathbf{H}_f^{(q)}. \tag{115}$$

$\square$

## H.1.2. ADDITIVE QUANTIZATION

**Lemma H.6.** *If for each $i = d, f, s$, $\mathcal{Q}_i$ is $(\underline{\epsilon}_i, \bar{\epsilon}_i)$-additive and $\mathcal{Q}_i$ is strong $\epsilon_i'$-additive, then under Assumption H.1, we have*

$$\mathbb{E}\left[(\tilde{\mathbf{x}}^{(q)})^\top \mathbf{A}\tilde{\mathbf{x}}^{(q)}\tilde{\mathbf{x}}^{(q)}(\tilde{\mathbf{x}}^{(q)})^\top\right] \le \alpha_A \mathrm{tr}\left(\mathbf{H}_f^{(q)}\mathbf{A}\right)\mathbf{H}_f^{(q)},$$

*where $\alpha_A \lesssim (1 + \alpha_0)\left(1 + \frac{\bar{\epsilon}_d'}{\bar{\epsilon}_d^2} + \frac{\bar{\epsilon}_f'}{\bar{\epsilon}_f^2}\right)\left(\frac{\bar{\epsilon}_d}{\underline{\epsilon}_d}\left(1 + \frac{\bar{\epsilon}_s + \sqrt{\bar{\epsilon}_s'}}{\underline{\epsilon}_s}\right) + \frac{\bar{\epsilon}_f}{\underline{\epsilon}_f}\right)^2.$*

*Proof.* By Assumption H.1,

$$\mathbb{E}\left[\mathbf{x}^\top \mathbf{S}^{(q)}{}^\top \mathbf{A}\mathbf{S}^{(q)}\mathbf{x}\mathbf{S}^{(q)}\mathbf{x}\mathbf{x}^\top \mathbf{S}^{(q)}{}^\top\right] \preceq \alpha_0 \mathbb{E}\left[\mathbf{S}^{(q)}\mathrm{tr}\left(\mathbf{H}\mathbf{S}^{(q)}{}^\top \mathbf{A}\mathbf{S}^{(q)}\right)\mathbf{H}\mathbf{S}^{(q)}{}^\top\right]. \tag{116}$$

As $\mathcal{Q}_d$ is $\bar{\epsilon}_d$-additive,

$$\begin{aligned}
\mathbb{E}\left[\mathbf{x}^\top \mathbf{S}^{(q)}{}^\top \mathbf{A}\mathbf{S}^{(q)}\mathbf{x}\mathbf{S}^{(q)}\boldsymbol{\epsilon}^{(d)}\boldsymbol{\epsilon}^{(d)}{}^\top \mathbf{S}^{(q)}{}^\top\right] &\preceq \bar{\epsilon}_d \mathbb{E}\left[\mathbf{x}^\top \mathbf{S}^{(q)}{}^\top \mathbf{A}\mathbf{S}^{(q)}\mathbf{x}\mathbf{S}^{(q)}\mathbf{S}^{(q)}{}^\top\right] \\
&= \bar{\epsilon}_d \mathbb{E}\left[\mathbf{S}^{(q)}\mathrm{tr}\left(\mathbf{H}\mathbf{S}^{(q)}{}^\top \mathbf{A}\mathbf{S}^{(q)}\right)\mathbf{S}^{(q)}{}^\top\right].
\end{aligned} \tag{117}$$

Similarly,

$$\mathbb{E}\left[\boldsymbol{\epsilon}^{(d)}{}^\top \mathbf{S}^{(q)}{}^\top \mathbf{A}\mathbf{S}^{(q)}\boldsymbol{\epsilon}^{(d)}\mathbf{S}^{(q)}\mathbf{x}\mathbf{x}^\top \mathbf{S}^{(q)}{}^\top\right] \preceq \bar{\epsilon}_d \mathbb{E}\left[\mathbf{S}^{(q)}\mathrm{tr}\left(\mathbf{S}^{(q)}{}^\top \mathbf{A}\mathbf{S}^{(q)}\right)\mathbf{H}\mathbf{S}^{(q)}{}^\top\right]. \tag{118}$$

As $\mathcal{Q}_f$ is $\bar{\epsilon}_f$-additive,

$$\begin{aligned}
\mathbb{E}\left[\mathbf{x}^\top \mathbf{S}^{(q)}{}^\top \mathbf{A}\mathbf{S}^{(q)}\mathbf{x}\boldsymbol{\epsilon}^{(f)}\boldsymbol{\epsilon}^{(f)}{}^\top\right] &\preceq \bar{\epsilon}_f \mathbb{E}\left[\mathbf{x}^\top \mathbf{S}^{(q)}{}^\top \mathbf{A}\mathbf{S}^{(q)}\mathbf{x}\mathbf{I}\right] \\
&= \bar{\epsilon}_f \mathbb{E}\left[\mathrm{tr}\left(\mathbf{H}\mathbf{S}^{(q)}{}^\top \mathbf{A}\mathbf{S}^{(q)}\right)\mathbf{I}\right].
\end{aligned} \tag{119}$$

By the fact that both $\mathcal{Q}_d$ and $\mathcal{Q}_f$ are additive quantization,

$$\begin{aligned}
\mathbb{E}\left[\boldsymbol{\epsilon}^{(d)}{}^\top \mathbf{S}^{(q)}{}^\top \mathbf{A}\mathbf{S}^{(q)}\boldsymbol{\epsilon}^{(d)}\boldsymbol{\epsilon}^{(f)}\boldsymbol{\epsilon}^{(f)}{}^\top\right] &\preceq \bar{\epsilon}_f \mathbb{E}\left[\boldsymbol{\epsilon}^{(d)}{}^\top \mathbf{S}^{(q)}{}^\top \mathbf{A}\mathbf{S}^{(q)}\boldsymbol{\epsilon}^{(d)}\mathbf{I}\right] \\
&\preceq \bar{\epsilon}_f \bar{\epsilon}_d \mathbb{E}\left[\mathrm{tr}\left(\mathbf{S}^{(q)}{}^\top \mathbf{A}\mathbf{S}^{(q)}\right)\mathbf{I}\right].
\end{aligned} \tag{120}$$

Similarly,

$$\mathbb{E}\left[\boldsymbol{\epsilon}^{(f)}{}^\top \mathbf{A}\boldsymbol{\epsilon}^{(f)}\mathbf{S}^{(q)}\mathbf{x}\mathbf{x}^\top \mathbf{S}^{(q)}{}^\top\right] \preceq \bar{\epsilon}_f \mathbb{E}\left[\mathrm{tr}(\mathbf{A})\mathbf{S}^{(q)}\mathbf{H}\mathbf{S}^{(q)}{}^\top\right]. \tag{121}$$

$$\mathbb{E}\left[\boldsymbol{\epsilon}^{(f)}{}^\top \mathbf{A}\boldsymbol{\epsilon}^{(f)}\mathbf{S}^{(q)}\boldsymbol{\epsilon}^{(d)}\boldsymbol{\epsilon}^{(d)}{}^\top \mathbf{S}^{(q)}{}^\top\right] \preceq \bar{\epsilon}_f \bar{\epsilon}_d \mathbb{E}\left[\mathrm{tr}(\mathbf{A})\mathbf{S}^{(q)}\mathbf{S}^{(q)}{}^\top\right]. \tag{122}$$

Under the strong additive property of $\mathcal{Q}_d$ and $\mathcal{Q}_s$, it follows

$$\mathbb{E}\left[\boldsymbol{\epsilon}^{(d)}{}^\top \mathbf{S}^{(q)}{}^\top \mathbf{A}\mathbf{S}^{(q)}\boldsymbol{\epsilon}^{(d)}\mathbf{S}^{(q)}\boldsymbol{\epsilon}^{(d)}\boldsymbol{\epsilon}^{(d)}{}^\top \mathbf{S}^{(q)}{}^\top\right] \preceq \bar{\epsilon}_d' \mathbb{E}\left[\mathbf{S}^{(q)}\mathrm{tr}\left(\mathbf{S}^{(q)}{}^\top \mathbf{A}\mathbf{S}^{(q)}\right)\mathbf{S}^{(q)}{}^\top\right]. \tag{123}$$

$$\mathbb{E}\left[\boldsymbol{\epsilon}^{(f)}{}^\top \mathbf{A}\boldsymbol{\epsilon}^{(f)}\boldsymbol{\epsilon}^{(f)}\boldsymbol{\epsilon}^{(f)}{}^\top\right] \preceq \bar{\epsilon}_f' \mathrm{tr}(\mathbf{A})\mathbf{I}. \tag{124}$$

Applying (101), it holds

$$\begin{aligned}
&\mathbb{E}\left[(\tilde{\mathbf{x}}^{(q)})^\top \mathbf{A}\tilde{\mathbf{x}}^{(q)}\tilde{\mathbf{x}}^{(q)}(\tilde{\mathbf{x}}^{(q)})^\top\right] \\
&\preceq C(1 + \alpha_0)\left(1 + \frac{\bar{\epsilon}_d'}{\bar{\epsilon}_d^2} + \frac{\bar{\epsilon}_f'}{\bar{\epsilon}_f^2}\right)\mathbb{E}\left[\mathrm{tr}\left((\mathbf{S}^{(q)}(\mathbf{H} + \bar{\epsilon}_d\mathbf{I})\mathbf{S}^{(q)}{}^\top + \bar{\epsilon}_f\mathbf{I})\mathbf{A}\right)(\mathbf{S}^{(q)}(\mathbf{H} + \bar{\epsilon}_d\mathbf{I})\mathbf{S}^{(q)}{}^\top + \bar{\epsilon}_f\mathbf{I})\right].
\end{aligned} \tag{125}$$

where $C > 0$ is constant. Note that

$$
\mathbb{E}\left[\mathrm{tr}\left(\left(\mathbf{S}^{(q)}(\mathbf{H} + \overline{\epsilon}_d\mathbf{I})\mathbf{S}^{(q)\top} + \overline{\epsilon}_f\mathbf{I}\right)\mathbf{A}\right)\left(\mathbf{S}^{(q)}(\mathbf{H} + \overline{\epsilon}_d\mathbf{I})\mathbf{S}^{(q)\top} + \overline{\epsilon}_f\mathbf{I}\right)\right]
$$
$$
\preceq 4\mathbb{E}\left[\mathrm{tr}\left(\left(\mathbf{S}(\mathbf{H} + \overline{\epsilon}_d\mathbf{I})\mathbf{S}^\top + \overline{\epsilon}_f\mathbf{I})\mathbf{A}\right)\left(\mathbf{S}(\mathbf{H} + \overline{\epsilon}_d\mathbf{I})\mathbf{S}^\top + \overline{\epsilon}_f\mathbf{I}\right)\right]
$$
$$
+ 4\mathbb{E}\left[\mathrm{tr}\left(\left(\mathbf{S}(\mathbf{H} + \overline{\epsilon}_d\mathbf{I})\mathbf{S}^\top + \overline{\epsilon}_f\mathbf{I})\mathbf{A}\right)\left(\boldsymbol{\epsilon}^{(s)}(\mathbf{H} + \overline{\epsilon}_d\mathbf{I})\boldsymbol{\epsilon}^{(s)\top} + \overline{\epsilon}_f\mathbf{I}\right)\right]
$$
$$
+ 4\mathbb{E}\left[\mathrm{tr}\left(\left(\boldsymbol{\epsilon}^{(s)}(\mathbf{H} + \overline{\epsilon}_d\mathbf{I})\boldsymbol{\epsilon}^{(s)\top} + \overline{\epsilon}_f\mathbf{I})\mathbf{A}\right)\left(\mathbf{S}(\mathbf{H} + \overline{\epsilon}_d\mathbf{I})\mathbf{S}^\top + \overline{\epsilon}_f\mathbf{I}\right)\right]
$$
$$
+ 4\mathbb{E}\left[\mathrm{tr}\left(\left(\boldsymbol{\epsilon}^{(s)}(\mathbf{H} + \overline{\epsilon}_d\mathbf{I})\boldsymbol{\epsilon}^{(s)\top} + \overline{\epsilon}_f\mathbf{I})\mathbf{A}\right)\left(\boldsymbol{\epsilon}^{(s)}(\mathbf{H} + \overline{\epsilon}_d\mathbf{I})\boldsymbol{\epsilon}^{(s)\top} + \overline{\epsilon}_f\mathbf{I}\right)\right]
$$
$$
\preceq 16\,\mathrm{tr}\left(\left(\mathbf{S}(\mathbf{H} + \overline{\epsilon}_d\mathbf{I})\mathbf{S}^\top + \overline{\epsilon}_f\mathbf{I} + (\overline{\epsilon}_s + \sqrt{\overline{\epsilon}'_s})\mathrm{tr}(\mathbf{H} + \overline{\epsilon}_d\mathbf{I})\mathbf{I}\right)\mathbf{A}\right)\left[\mathbf{S}(\mathbf{H} + \overline{\epsilon}_d\mathbf{I})\mathbf{S}^\top + \overline{\epsilon}_f\mathbf{I} + (\overline{\epsilon}_s + \sqrt{\overline{\epsilon}'_s})\mathrm{tr}(\mathbf{H} + \overline{\epsilon}_d\mathbf{I})\mathbf{I}\right],
$$
$$(126)$$

further by the definition of

$$
\begin{aligned}
\mathbf{H}_f^{(q)} &= \mathbb{E}\left[\mathcal{Q}_f(\mathcal{Q}_s(\mathbf{S})\mathcal{Q}_d(\mathbf{x}))\mathcal{Q}_f(\mathcal{Q}_s(\mathbf{S})\mathcal{Q}_d(\mathbf{x}))^\top\right] \\
&= \mathbb{E}\left[(\mathcal{Q}_s(\mathbf{S})\mathcal{Q}_d(\mathbf{x}) + \boldsymbol{\epsilon}^{(f)})(\mathcal{Q}_s(\mathbf{S})\mathcal{Q}_d(\mathbf{x}) + \boldsymbol{\epsilon}^{(f)})^\top\right] \\
&\succeq \mathbb{E}\left[\mathcal{Q}_s(\mathbf{S})\mathcal{Q}_d(\mathbf{x})(\mathcal{Q}_s(\mathbf{S})\mathcal{Q}_d(\mathbf{x}))^\top\right] + \underline{\epsilon}_f\mathbf{I} \\
&\succeq \mathbb{E}\left[\mathcal{Q}_s(\mathbf{S})(\mathbf{H} + \underline{\epsilon}_d\mathbf{I})\mathcal{Q}_s(\mathbf{S})^\top\right] + \underline{\epsilon}_f\mathbf{I} \\
&\succeq \mathbb{E}\left[\mathcal{Q}_s(\mathbf{S})\mathcal{Q}_d(\mathbf{x})(\mathcal{Q}_s(\mathbf{S})\mathcal{Q}_d(\mathbf{x}))^\top\right] + \underline{\epsilon}_f\mathbf{I} \\
&= \mathbf{S}(\mathbf{H} + \underline{\epsilon}_d\mathbf{I})\mathbf{S}^\top + \mathbb{E}\left[\boldsymbol{\epsilon}^{(s)}(\mathbf{H} + \underline{\epsilon}_d\mathbf{I})\boldsymbol{\epsilon}^{(s)\top}\right] + \underline{\epsilon}_f\mathbf{I} \\
&\succeq \mathbf{S}(\mathbf{H} + \underline{\epsilon}_d\mathbf{I})\mathbf{S}^\top + \underline{\epsilon}_s\,\mathrm{tr}\left(\mathbf{H} + \underline{\epsilon}_d\mathbf{I}\right)\mathbf{I} + \underline{\epsilon}_f\mathbf{I},
\end{aligned}
$$
$$(127)$$

we have

$$
\mathbb{E}\left[\mathrm{tr}\left(\left(\mathbf{S}^{(q)}(\mathbf{H} + \overline{\epsilon}_d\mathbf{I})\mathbf{S}^{(q)\top} + \overline{\epsilon}_f\mathbf{I}\right)\mathbf{A}\right)\left(\mathbf{S}^{(q)}(\mathbf{H} + \overline{\epsilon}_d\mathbf{I})\mathbf{S}^{(q)\top} + \overline{\epsilon}_f\mathbf{I}\right)\right]
$$
$$
\preceq 16\left(\frac{\overline{\epsilon}_d}{\underline{\epsilon}_d}\left(1 + \frac{\overline{\epsilon}_s + \sqrt{\overline{\epsilon}'_s}}{\underline{\epsilon}_s}\right) + \frac{\overline{\epsilon}_f}{\underline{\epsilon}_f}\right)^2 \mathrm{tr}(\mathbf{H}_f^{(q)}\mathbf{A})\mathbf{H}_f^{(q)}.
$$
$$(128)$$

Therefore, together with (125) and (128),

$$
\mathbb{E}\left[(\tilde{\mathbf{x}}^{(q)})^\top\mathbf{A}\tilde{\mathbf{x}}^{(q)}\tilde{\mathbf{x}}^{(q)}(\tilde{\mathbf{x}}^{(q)})^\top\right] \preceq C'(1 + \alpha_0)\left(1 + \frac{\overline{\epsilon}'_d}{\underline{\epsilon}_d^2} + \frac{\overline{\epsilon}'_f}{\underline{\epsilon}_f^2}\right)\left(\frac{\overline{\epsilon}_d}{\underline{\epsilon}_d}\left(1 + \frac{\overline{\epsilon}_s + \sqrt{\overline{\epsilon}'_s}}{\underline{\epsilon}_s}\right) + \frac{\overline{\epsilon}_f}{\underline{\epsilon}_f}\right)^2 \mathrm{tr}(\mathbf{H}_f^{(q)}\mathbf{A})\mathbf{H}_f^{(q)}. \quad (129)
$$

$\square$

## H.2. Second-order Noise Assumption

In this section, we aim to verify the second-order noise assumption. Recall that $\xi := \mathcal{Q}_l(y) - \langle \mathbf{v}^{(q)*}, \tilde{\mathbf{x}}^{(q)} \rangle$ and $\mathbf{v}^{(q)*} = (\mathbf{H}_f^{(q)})^{-1}\mathbf{S}\mathbf{H}\mathbf{w}^*$.

$$
\begin{aligned}
\mathbb{E}\left[\xi^2\tilde{\mathbf{x}}^{(q)}(\tilde{\mathbf{x}}^{(q)})^\top\right] &= \mathbb{E}\left[(\mathcal{Q}_l(y) - \langle \mathbf{v}^{(q)*}, \tilde{\mathbf{x}}^{(q)}\rangle)^2\tilde{\mathbf{x}}^{(q)}(\tilde{\mathbf{x}}^{(q)})^\top\right] \\
&= \mathbb{E}\left[\left(\mathcal{Q}_l(y) - \langle (\mathbf{H}_f^{(q)})^{-1}\mathbf{S}\mathbf{H}\mathbf{w}^*, \tilde{\mathbf{x}}^{(q)}\rangle\right)^2\tilde{\mathbf{x}}^{(q)}(\tilde{\mathbf{x}}^{(q)})^\top\right] \\
&= \mathbb{E}\left[\left(\mathcal{Q}_l(y) - y + y - \langle \mathbf{w}^*, \mathbf{x}\rangle + \langle \mathbf{w}^*, \mathbf{x}\rangle - \left\langle (\mathbf{H}_f^{(q)})^{-1}\mathbf{S}\mathbf{H}\mathbf{w}^*, \tilde{\mathbf{x}}^{(q)}\right\rangle\right)^2\tilde{\mathbf{x}}^{(q)}(\tilde{\mathbf{x}}^{(q)})^\top\right] \\
&\preceq 3\mathbb{E}\left[(\mathcal{Q}_l(y) - y)^2\tilde{\mathbf{x}}^{(q)}(\tilde{\mathbf{x}}^{(q)})^\top\right] + 3\mathbb{E}\left[(y - \langle \mathbf{w}^*, \mathbf{x}\rangle)^2\tilde{\mathbf{x}}^{(q)}(\tilde{\mathbf{x}}^{(q)})^\top\right] \\
&\quad + 3\mathbb{E}\left[\left(\langle \mathbf{w}^*, \mathbf{x}\rangle - \left\langle (\mathbf{H}_f^{(q)})^{-1}\mathbf{S}\mathbf{H}\mathbf{w}^*, \tilde{\mathbf{x}}^{(q)}\right\rangle\right)^2\tilde{\mathbf{x}}^{(q)}(\tilde{\mathbf{x}}^{(q)})^\top\right].
\end{aligned}
$$

Recall that $\tilde{\mathbf{x}}^{(q)} = \mathbf{S}^{(q)}(\mathbf{x} + \boldsymbol{\epsilon}^{(d)}) + \boldsymbol{\epsilon}^{(f)}$, it follows that

$$\left( \langle \mathbf{w}^*, \mathbf{x} \rangle - \left\langle (\mathbf{H}_f^{(q)})^{-1} \mathbf{S} \mathbf{H} \mathbf{w}^*, \tilde{\mathbf{x}}^{(q)} \right\rangle \right)^2 \leq 2 \left\langle \mathbf{w}^* - \mathbf{S}^{(q)\top}(\mathbf{H}_f^{(q)})^{-1} \mathbf{S} \mathbf{H} \mathbf{w}^*, \mathbf{x} \right\rangle^2 + 2 \left\langle (\mathbf{H}_f^{(q)})^{-1} \mathbf{S} \mathbf{H} \mathbf{w}^*, \mathbf{S}^{(q)} \boldsymbol{\epsilon}^{(d)} + \boldsymbol{\epsilon}^{(f)} \right\rangle^2 .$$

Further note that

$$\tilde{\mathbf{x}}^{(q)}(\tilde{\mathbf{x}}^{(q)})^\top \preceq 2\mathbf{S}^{(q)}(\mathbf{x} + \boldsymbol{\epsilon}^{(d)})(\mathbf{x} + \boldsymbol{\epsilon}^{(d)})^\top \mathbf{S}^{(q)\top} + 2\boldsymbol{\epsilon}^{(f)}\boldsymbol{\epsilon}^{(f)\top}$$
$$\preceq 4\mathbf{S}^{(q)}\mathbf{x}\mathbf{x}^\top \mathbf{S}^{(q)\top} + 4\mathbf{S}^{(q)}\boldsymbol{\epsilon}^{(d)}\boldsymbol{\epsilon}^{(d)\top}\mathbf{S}^{(q)\top} + 2\boldsymbol{\epsilon}^{(f)}\boldsymbol{\epsilon}^{(f)\top},$$

we have

$$
\begin{aligned}
\mathbb{E}\left[\xi^2 \tilde{\mathbf{x}}^{(q)}(\tilde{\mathbf{x}}^{(q)})^\top\right] \preceq{} & 12\mathbb{E}\left[(\mathcal{Q}_l(y) - y)^2 \mathbf{S}^{(q)}\mathbf{x}\mathbf{x}^\top \mathbf{S}^{(q)\top}\right] + 12\mathbb{E}\left[(\mathcal{Q}_l(y) - y)^2 \mathbf{S}^{(q)}\boldsymbol{\epsilon}^{(d)}\boldsymbol{\epsilon}^{(d)\top}\mathbf{S}^{(q)\top}\right] \\
& + 6\mathbb{E}\left[(\mathcal{Q}_l(y) - y)^2 \boldsymbol{\epsilon}^{(f)}\boldsymbol{\epsilon}^{(f)\top}\right] + 6\mathbb{E}\left[(y - \langle \mathbf{w}^*, \mathbf{x} \rangle)^2 \boldsymbol{\epsilon}^{(f)}\boldsymbol{\epsilon}^{(f)\top}\right] \\
& + 12\mathbb{E}\left[(y - \langle \mathbf{w}^*, \mathbf{x} \rangle)^2 \mathbf{S}^{(q)}\mathbf{x}\mathbf{x}^\top \mathbf{S}^{(q)\top}\right] + 12\mathbb{E}\left[(y - \langle \mathbf{w}^*, \mathbf{x} \rangle)^2 \mathbf{S}^{(q)}\boldsymbol{\epsilon}^{(d)}\boldsymbol{\epsilon}^{(d)\top}\mathbf{S}^{(q)\top}\right] \\
& + 24\mathbb{E}\left[\left\langle \mathbf{w}^* - \mathbf{S}^{(q)\top}(\mathbf{H}_f^{(q)})^{-1}\mathbf{S}\mathbf{H}\mathbf{w}^*, \mathbf{x} \right\rangle^2 \mathbf{S}^{(q)}\mathbf{x}\mathbf{x}^\top \mathbf{S}^{(q)\top}\right] \\
& + 24\mathbb{E}\left[\left\langle \mathbf{w}^* - \mathbf{S}^{(q)\top}(\mathbf{H}_f^{(q)})^{-1}\mathbf{S}\mathbf{H}\mathbf{w}^*, \mathbf{x} \right\rangle^2 \mathbf{S}^{(q)}\boldsymbol{\epsilon}^{(d)}\boldsymbol{\epsilon}^{(d)\top}\mathbf{S}^{(q)\top}\right] \\
& + 12\mathbb{E}\left[\left\langle \mathbf{w}^* - \mathbf{S}^{(q)\top}(\mathbf{H}_f^{(q)})^{-1}\mathbf{S}\mathbf{H}\mathbf{w}^*, \mathbf{x} \right\rangle^2 \boldsymbol{\epsilon}^{(f)}\boldsymbol{\epsilon}^{(f)\top}\right] \\
& + 24\mathbb{E}\left[\left\langle (\mathbf{H}_f^{(q)})^{-1}\mathbf{S}\mathbf{H}\mathbf{w}^*, \mathbf{S}^{(q)}\boldsymbol{\epsilon}^{(d)} + \boldsymbol{\epsilon}^{(f)} \right\rangle^2 \mathbf{S}^{(q)}\mathbf{x}\mathbf{x}^\top \mathbf{S}^{(q)\top}\right] \\
& + 24\mathbb{E}\left[\left\langle (\mathbf{H}_f^{(q)})^{-1}\mathbf{S}\mathbf{H}\mathbf{w}^*, \mathbf{S}^{(q)}\boldsymbol{\epsilon}^{(d)} + \boldsymbol{\epsilon}^{(f)} \right\rangle^2 \mathbf{S}^{(q)}\boldsymbol{\epsilon}^{(d)}\boldsymbol{\epsilon}^{(d)\top}\mathbf{S}^{(q)\top}\right] \\
& + 12\mathbb{E}\left[\left\langle (\mathbf{H}_f^{(q)})^{-1}\mathbf{S}\mathbf{H}\mathbf{w}^*, \mathbf{S}^{(q)}\boldsymbol{\epsilon}^{(d)} + \boldsymbol{\epsilon}^{(f)} \right\rangle^2 \boldsymbol{\epsilon}^{(f)}\boldsymbol{\epsilon}^{(f)\top}\right].
\end{aligned}
\tag{130}
$$

For simplicity, we merely verify the assumption under multiplicative quantization.

### H.2.1. MULTIPLICATIVE QUANTIZATION

**Lemma H.7.** *Under Assumption H.1, H.2, if for each $i = d, f, s$, $\mathcal{Q}_i$ is $(\underline{\epsilon}_i, \overline{\epsilon}_i)$-multiplicative and $\mathcal{Q}_i$ is strong $\overline{\epsilon}_i'$-multiplicative, if $\mathcal{Q}_l$ is $\overline{\epsilon}_l$-multiplicative, then*

$$\mathbb{E}\left[\xi^2 \tilde{\mathbf{x}}^{(q)}(\tilde{\mathbf{x}}^{(q)})^\top\right] \preceq \overline{\sigma}_M^2 \mathbf{H}_f^{(q)},$$

*where* $\overline{\sigma}_M^2 \lesssim \dfrac{(1 + \alpha_0 \|\mathbf{w}^*\|_{\mathbf{H}}^2)(1 + \overline{\epsilon}_s + \overline{\epsilon}_s')\left[(1 + \overline{\epsilon}_d)(1 + \overline{\epsilon}_f)(\overline{\epsilon}_l C_y + \overline{\sigma}_0^2) + (1 + \overline{\epsilon}_d + \overline{\epsilon}_d')(1 + \overline{\epsilon}_f + \overline{\epsilon}_f')\right]}{(1 + \underline{\epsilon}_f)(1 + \underline{\epsilon}_d)(1 + \underline{\epsilon}_s)}.$

*Proof.* Under Assumption H.2 and $\mathcal{Q}_l$ is $(\underline{\epsilon}_l, \overline{\epsilon}_l)$-multiplicative

$$\mathbb{E}\left[(\mathcal{Q}_l(y) - y)^2 \mathbf{S}^{(q)}\mathbf{x}\mathbf{x}^\top \mathbf{S}^{(q)\top}\right] \preceq \overline{\epsilon}_l C_y \mathbb{E}\left[\mathbf{S}^{(q)}\mathbf{H}\mathbf{S}^{(q)}\right]. \tag{131}$$

Under Assumption H.2 and for $i = l, d$, $\mathcal{Q}_i$ is $(\underline{\epsilon}_i, \overline{\epsilon}_i)$-multiplicative

$$\mathbb{E}\left[(\mathcal{Q}_l(y) - y)^2 \mathbf{S}^{(q)}\boldsymbol{\epsilon}^{(d)}\boldsymbol{\epsilon}^{(d)\top}\mathbf{S}^{(q)\top}\right] \preceq \overline{\epsilon}_l \overline{\epsilon}_d C_y \mathbb{E}\left[\mathbf{S}^{(q)}\mathbf{H}\mathbf{S}^{(q)}\right]. \tag{132}$$

Under Assumption H.2 and for $i = l, f, d$, $\mathcal{Q}_i$ is $(\underline{\epsilon}_i, \overline{\epsilon}_i)$-multiplicative

$$
\begin{aligned}
\mathbb{E}\left[(\mathcal{Q}_l(y) - y)^2 \boldsymbol{\epsilon}^{(f)}\boldsymbol{\epsilon}^{(f)\top}\right] \preceq{} & \overline{\epsilon}_f \overline{\epsilon}_l \mathbb{E}\left[y^2 \mathbf{S}^{(q)}(\mathbf{x} + \boldsymbol{\epsilon}^{(d)})(\mathbf{x} + \boldsymbol{\epsilon}^{(d)})^\top \mathbf{S}^{(q)\top}\right] \\
\preceq{} & 2\overline{\epsilon}_f \overline{\epsilon}_l \mathbb{E}\left[y^2 \mathbf{S}^{(q)}\mathbf{x}\mathbf{x}^\top \mathbf{S}^{(q)\top}\right] + 2\overline{\epsilon}_f \overline{\epsilon}_l \mathbb{E}\left[y^2 \mathbf{S}^{(q)}\boldsymbol{\epsilon}^{(d)}\boldsymbol{\epsilon}^{(d)\top}\mathbf{S}^{(q)\top}\right] \\
\preceq{} & 2\overline{\epsilon}_f \overline{\epsilon}_l (1 + \overline{\epsilon}_d) C_y \mathbb{E}\left[\mathbf{S}^{(q)}\mathbf{H}\mathbf{S}^{(q)}\right].
\end{aligned}
\tag{133}
$$

Under Assumption H.2 and for $i = f, d$, $\mathcal{Q}_i$ is $(\underline{\epsilon}_i, \overline{\epsilon}_i)$-multiplicative

$$
\begin{aligned}
\mathbb{E}\left[(y - \langle \mathbf{w}^*, \mathbf{x} \rangle)^2 \, \boldsymbol{\epsilon}^{(f)} \boldsymbol{\epsilon}^{(f)\top}\right] \preceq & \overline{\epsilon}_f \mathbb{E}\left[(y - \langle \mathbf{w}^*, \mathbf{x} \rangle)^2 \, \mathbf{S}^{(q)}(\mathbf{x} + \boldsymbol{\epsilon}^{(d)})(\mathbf{x} + \boldsymbol{\epsilon}^{(d)})^\top \mathbf{S}^{(q)\top}\right] \\
\preceq & 2\overline{\epsilon}_f \mathbb{E}\left[(y - \langle \mathbf{w}^*, \mathbf{x} \rangle)^2 \, \mathbf{S}^{(q)} \mathbf{x} \mathbf{x}^\top \mathbf{S}^{(q)\top}\right] \\
& + 2\overline{\epsilon}_f \mathbb{E}\left[(y - \langle \mathbf{w}^*, \mathbf{x} \rangle)^2 \, \mathbf{S}^{(q)} \boldsymbol{\epsilon}^{(d)} \boldsymbol{\epsilon}^{(d)\top} \mathbf{S}^{(q)\top}\right] \\
\preceq & 2\overline{\epsilon}_f(1 + \overline{\epsilon}_d) \mathbb{E}\left[(y - \langle \mathbf{w}^*, \mathbf{x} \rangle)^2 \, \mathbf{S}^{(q)} \mathbf{x} \mathbf{x}^\top \mathbf{S}^{(q)\top}\right] \\
\preceq & 2\overline{\epsilon}_f(1 + \overline{\epsilon}_d) \overline{\sigma}_0^2 \mathbb{E}\left[\mathbf{S}^{(q)} \mathbf{H} \mathbf{S}^{(q)\top}\right].
\end{aligned}
\tag{134}
$$

Under Assumption H.2,

$$
\mathbb{E}\left[(y - \langle \mathbf{w}^*, \mathbf{x} \rangle)^2 \, \mathbf{S}^{(q)} \mathbf{x} \mathbf{x}^\top \mathbf{S}^{(q)\top}\right] \preceq \overline{\sigma}_0^2 \mathbb{E}\left[\mathbf{S}^{(q)} \mathbf{H} \mathbf{S}^{(q)\top}\right].
\tag{135}
$$

Similarly,

$$
\mathbb{E}\left[(y - \langle \mathbf{w}^*, \mathbf{x} \rangle)^2 \, \mathbf{S}^{(q)} \boldsymbol{\epsilon}^{(d)} \boldsymbol{\epsilon}^{(d)\top} \mathbf{S}^{(q)\top}\right] \preceq \overline{\epsilon}_d \mathbb{E}\left[(y - \langle \mathbf{w}^*, \mathbf{x} \rangle)^2 \, \mathbf{S}^{(q)} \mathbf{x} \mathbf{x}^\top \mathbf{S}^{(q)\top}\right] \preceq \overline{\epsilon}_d \overline{\sigma}_0^2 \mathbb{E}\left[\mathbf{S}^{(q)} \mathbf{H} \mathbf{S}^{(q)\top}\right].
\tag{136}
$$

Under Assumption H.2,

$$
\begin{aligned}
& \mathbb{E}\left[\left\langle \mathbf{w}^* - \mathbf{S}^{(q)\top}(\mathbf{H}_f^{(q)})^{-1} \mathbf{S} \mathbf{H} \mathbf{w}^*, \mathbf{x} \right\rangle^2 \mathbf{S}^{(q)} \mathbf{x} \mathbf{x}^\top \mathbf{S}^{(q)\top}\right] \\
= & \mathbb{E}\left[\mathbf{x}^\top \left(\mathbf{w}^* - \mathbf{S}^{(q)\top}(\mathbf{H}_f^{(q)})^{-1} \mathbf{S} \mathbf{H} \mathbf{w}^*\right) \left(\mathbf{w}^* - \mathbf{S}^{(q)\top}(\mathbf{H}_f^{(q)})^{-1} \mathbf{S} \mathbf{H} \mathbf{w}^*\right)^\top \mathbf{x} \mathbf{S}^{(q)} \mathbf{x} \mathbf{x}^\top \mathbf{S}^{(q)\top}\right] \\
\preceq & \alpha_0 \mathbb{E}\left[\operatorname{tr}\left(\mathbf{H}\left(\mathbf{I} - \mathbf{S}^{(q)\top}(\mathbf{H}_f^{(q)})^{-1} \mathbf{S} \mathbf{H}\right) \mathbf{w}^* \mathbf{w}^{*\top} \left(\mathbf{I} - \mathbf{S}^{(q)\top}(\mathbf{H}_f^{(q)})^{-1} \mathbf{S} \mathbf{H}\right)^\top\right) \mathbf{S}^{(q)} \mathbf{H} \mathbf{S}^{(q)\top}\right].
\end{aligned}
\tag{137}
$$

Further,

$$
\begin{aligned}
& \mathbb{E}\left[\left\langle \mathbf{w}^* - \mathbf{S}^{(q)\top}(\mathbf{H}_f^{(q)})^{-1} \mathbf{S} \mathbf{H} \mathbf{w}^*, \mathbf{x} \right\rangle^2 \mathbf{S}^{(q)} \boldsymbol{\epsilon}^{(d)} \boldsymbol{\epsilon}^{(d)\top} \mathbf{S}^{(q)\top}\right] \\
\preceq & \overline{\epsilon}_d \mathbb{E}\left[\left\langle \mathbf{w}^* - \mathbf{S}^{(q)\top}(\mathbf{H}_f^{(q)})^{-1} \mathbf{S} \mathbf{H} \mathbf{w}^*, \mathbf{x} \right\rangle^2 \mathbf{S}^{(q)} \mathbf{x} \mathbf{x}^\top \mathbf{S}^{(q)\top}\right] \\
\preceq & \overline{\epsilon}_d \alpha_0 \mathbb{E}\left[\operatorname{tr}\left(\mathbf{H}\left(\mathbf{I} - \mathbf{S}^{(q)\top}(\mathbf{H}_f^{(q)})^{-1} \mathbf{S} \mathbf{H}\right) \mathbf{w}^* \mathbf{w}^{*\top} \left(\mathbf{I} - \mathbf{S}^{(q)\top}(\mathbf{H}_f^{(q)})^{-1} \mathbf{S} \mathbf{H}\right)^\top\right) \mathbf{S}^{(q)} \mathbf{H} \mathbf{S}^{(q)\top}\right].
\end{aligned}
\tag{138}
$$

Similarly,

$$
\begin{aligned}
& \mathbb{E}\left[\left\langle \mathbf{w}^* - \mathbf{S}^{(q)\top}(\mathbf{H}_f^{(q)})^{-1} \mathbf{S} \mathbf{H} \mathbf{w}^*, \mathbf{x} \right\rangle^2 \boldsymbol{\epsilon}^{(f)} \boldsymbol{\epsilon}^{(f)\top}\right] \\
= & \overline{\epsilon}_f \mathbb{E}\left[\left\langle \mathbf{w}^* - \mathbf{S}^{(q)\top}(\mathbf{H}_f^{(q)})^{-1} \mathbf{S} \mathbf{H} \mathbf{w}^*, \mathbf{x} \right\rangle^2 \mathbf{S}^{(q)}(\mathbf{x} + \boldsymbol{\epsilon}^{(d)})(\mathbf{x} + \boldsymbol{\epsilon}^{(d)})^\top \mathbf{S}^{(q)\top}\right] \\
\preceq & 2(1 + \overline{\epsilon}_d) \overline{\epsilon}_f \mathbb{E}\left[\left\langle \mathbf{w}^* - \mathbf{S}^{(q)\top}(\mathbf{H}_f^{(q)})^{-1} \mathbf{S} \mathbf{H} \mathbf{w}^*, \mathbf{x} \right\rangle^2 \mathbf{S}^{(q)} \mathbf{x} \mathbf{x}^\top \mathbf{S}^{(q)\top}\right] \\
\preceq & 2(1 + \overline{\epsilon}_d) \overline{\epsilon}_f \alpha_0 \mathbb{E}\left[\operatorname{tr}\left(\mathbf{H}\left(\mathbf{I} - \mathbf{S}^{(q)\top}(\mathbf{H}_f^{(q)})^{-1} \mathbf{S} \mathbf{H}\right) \mathbf{w}^* \mathbf{w}^{*\top} \left(\mathbf{I} - \mathbf{S}^{(q)\top}(\mathbf{H}_f^{(q)})^{-1} \mathbf{S} \mathbf{H}\right)^\top\right) \mathbf{S}^{(q)} \mathbf{H} \mathbf{S}^{(q)\top}\right].
\end{aligned}
\tag{139}
$$

For the last three terms in (130), by the definition of multiplicative quantization,

$$
\mathbb{E}\left[\left\langle (\mathbf{H}_f^{(q)})^{-1}\mathbf{SHw}^*, \mathbf{S}^{(q)}\boldsymbol{\epsilon}^{(d)} + \boldsymbol{\epsilon}^{(f)} \right\rangle^2 \mathbf{S}^{(q)}\mathbf{xx}^\top \mathbf{S}^{(q)\top}\right]
$$

$$
\preceq 2\mathbb{E}\left[\left\langle (\mathbf{H}_f^{(q)})^{-1}\mathbf{SHw}^*, \mathbf{S}^{(q)}\boldsymbol{\epsilon}^{(d)} \right\rangle^2 \mathbf{S}^{(q)}\mathbf{xx}^\top \mathbf{S}^{(q)\top}\right]
$$

$$
+ 2\bar{\epsilon}_f \mathbb{E}\left[\left\langle (\mathbf{H}_f^{(q)})^{-1}\mathbf{SHw}^*, \mathbf{S}^{(q)}(\mathbf{x} + \boldsymbol{\epsilon}^{(d)}) \right\rangle^2 \mathbf{S}^{(q)}\mathbf{xx}^\top \mathbf{S}^{(q)\top}\right] \tag{140}
$$

$$
\preceq [4(1+\bar{\epsilon}_d)\bar{\epsilon}_f + 2\bar{\epsilon}_d]\mathbb{E}\left[\left\langle (\mathbf{H}_f^{(q)})^{-1}\mathbf{SHw}^*, \mathbf{S}^{(q)}\mathbf{x} \right\rangle^2 \mathbf{S}^{(q)}\mathbf{xx}^\top \mathbf{S}^{(q)\top}\right]
$$

$$
\preceq [4(1+\bar{\epsilon}_d)\bar{\epsilon}_f + 2\bar{\epsilon}_d]\alpha_0 \mathbb{E}\left[\mathrm{tr}\left(\mathbf{H}\left(\mathbf{S}^{(q)\top}(\mathbf{H}_f^{(q)})^{-1}\mathbf{SH}\right)\mathbf{w}^*\mathbf{w}^{*\top}\left(\mathbf{S}^{(q)\top}(\mathbf{H}_f^{(q)})^{-1}\mathbf{SH}\right)^\top\right)\mathbf{S}^{(q)}\mathbf{HS}^{(q)\top}\right].
$$

Similarly, under the definition of multiplicative quantization and strong multiplicative quantization,

$$
\mathbb{E}\left[\left\langle (\mathbf{H}_f^{(q)})^{-1}\mathbf{SHw}^*, \mathbf{S}^{(q)}\boldsymbol{\epsilon}^{(d)} + \boldsymbol{\epsilon}^{(f)} \right\rangle^2 \mathbf{S}^{(q)}\boldsymbol{\epsilon}^{(d)}\boldsymbol{\epsilon}^{(d)\top}\mathbf{S}^{(q)\top}\right]
$$

$$
\preceq 2\mathbb{E}\left[\left\langle (\mathbf{H}_f^{(q)})^{-1}\mathbf{SHw}^*, \mathbf{S}^{(q)}\boldsymbol{\epsilon}^{(d)} \right\rangle^2 \mathbf{S}^{(q)}\boldsymbol{\epsilon}^{(d)}\boldsymbol{\epsilon}^{(d)\top}\mathbf{S}^{(q)\top}\right]
$$

$$
+ 2\mathbb{E}\left[\left\langle (\mathbf{H}_f^{(q)})^{-1}\mathbf{SHw}^*, \boldsymbol{\epsilon}^{(f)} \right\rangle^2 \mathbf{S}^{(q)}\boldsymbol{\epsilon}^{(d)}\boldsymbol{\epsilon}^{(d)\top}\mathbf{S}^{(q)\top}\right]
$$

$$
\preceq 2\bar{\epsilon}'_d \mathbb{E}\left[\left\langle (\mathbf{H}_f^{(q)})^{-1}\mathbf{SHw}^*, \mathbf{S}^{(q)}\mathbf{x} \right\rangle^2 \mathbf{S}^{(q)}\mathbf{xx}^\top \mathbf{S}^{(q)\top}\right]
$$

$$
+ 2\bar{\epsilon}_f \mathbb{E}\left[\left\langle (\mathbf{H}_f^{(q)})^{-1}\mathbf{SHw}^*, \mathbf{S}^{(q)}(\mathbf{x} + \boldsymbol{\epsilon}^{(d)}) \right\rangle^2 \mathbf{S}^{(q)}\boldsymbol{\epsilon}^{(d)}\boldsymbol{\epsilon}^{(d)\top}\mathbf{S}^{(q)\top}\right]
$$

$$
\preceq 2\bar{\epsilon}'_d \mathbb{E}\left[\left\langle (\mathbf{H}_f^{(q)})^{-1}\mathbf{SHw}^*, \mathbf{S}^{(q)}\mathbf{x} \right\rangle^2 \mathbf{S}^{(q)}\mathbf{xx}^\top \mathbf{S}^{(q)\top}\right]
$$

$$
+ 4\bar{\epsilon}_f \mathbb{E}\left[\left\langle (\mathbf{H}_f^{(q)})^{-1}\mathbf{SHw}^*, \mathbf{S}^{(q)}\mathbf{x} \right\rangle^2 \mathbf{S}^{(q)}\boldsymbol{\epsilon}^{(d)}\boldsymbol{\epsilon}^{(d)\top}\mathbf{S}^{(q)\top}\right] \tag{141}
$$

$$
+ 4\bar{\epsilon}_f \mathbb{E}\left[\left\langle (\mathbf{H}_f^{(q)})^{-1}\mathbf{SHw}^*, \mathbf{S}^{(q)}\boldsymbol{\epsilon}^{(d)} \right\rangle^2 \mathbf{S}^{(q)}\boldsymbol{\epsilon}^{(d)}\boldsymbol{\epsilon}^{(d)\top}\mathbf{S}^{(q)\top}\right]
$$

$$
\preceq 2(1 + 2\bar{\epsilon}_f)\bar{\epsilon}'_d \mathbb{E}\left[\left\langle (\mathbf{H}_f^{(q)})^{-1}\mathbf{SHw}^*, \mathbf{S}^{(q)}\mathbf{x} \right\rangle^2 \mathbf{S}^{(q)}\mathbf{xx}^\top \mathbf{S}^{(q)\top}\right]
$$

$$
+ 4\bar{\epsilon}_f \bar{\epsilon}_d \mathbb{E}\left[\left\langle (\mathbf{H}_f^{(q)})^{-1}\mathbf{SHw}^*, \mathbf{S}^{(q)}\mathbf{x} \right\rangle^2 \mathbf{S}^{(q)}\mathbf{xx}^\top \mathbf{S}^{(q)\top}\right]
$$

$$
= [2(1 + 2\bar{\epsilon}_f)\bar{\epsilon}'_d + 4\bar{\epsilon}_f \bar{\epsilon}_d]\mathbb{E}\left[\left\langle (\mathbf{H}_f^{(q)})^{-1}\mathbf{SHw}^*, \mathbf{S}^{(q)}\mathbf{x} \right\rangle^2 \mathbf{S}^{(q)}\mathbf{xx}^\top \mathbf{S}^{(q)\top}\right]
$$

$$
\preceq [2(1 + 2\bar{\epsilon}_f)\bar{\epsilon}'_d + 4\bar{\epsilon}_f \bar{\epsilon}_d]\alpha_0 \mathbb{E}\left[\mathrm{tr}\left(\mathbf{H}\left(\mathbf{S}^{(q)\top}(\mathbf{H}_f^{(q)})^{-1}\mathbf{SH}\right)\mathbf{w}^*\mathbf{w}^{*\top}\left(\mathbf{S}^{(q)\top}(\mathbf{H}_f^{(q)})^{-1}\mathbf{SH}\right)^\top\right)\mathbf{S}^{(q)}\mathbf{HS}^{(q)\top}\right].
$$

Further,

$$\mathbb{E}\left[\left\langle(\mathbf{H}_f^{(q)})^{-1}\mathbf{SHw}^*,\mathbf{S}^{(q)}\boldsymbol{\epsilon}^{(d)}+\boldsymbol{\epsilon}^{(f)}\right\rangle^2\boldsymbol{\epsilon}^{(f)}\boldsymbol{\epsilon}^{(f)\top}\right]$$

$$\preceq 2\mathbb{E}\left[\left\langle(\mathbf{H}_f^{(q)})^{-1}\mathbf{SHw}^*,\mathbf{S}^{(q)}\boldsymbol{\epsilon}^{(d)}\right\rangle^2\boldsymbol{\epsilon}^{(f)}\boldsymbol{\epsilon}^{(f)\top}\right]$$

$$+2\mathbb{E}\left[\left\langle(\mathbf{H}_f^{(q)})^{-1}\mathbf{SHw}^*,\boldsymbol{\epsilon}^{(f)}\right\rangle^2\boldsymbol{\epsilon}^{(f)}\boldsymbol{\epsilon}^{(f)\top}\right]$$

$$\preceq 2\bar{\epsilon}_f\mathbb{E}\left[\left\langle(\mathbf{H}_f^{(q)})^{-1}\mathbf{SHw}^*,\mathbf{S}^{(q)}\boldsymbol{\epsilon}^{(d)}\right\rangle^2\mathbf{S}^{(q)}(\mathbf{x}+\boldsymbol{\epsilon}^{(d)})(\mathbf{x}+\boldsymbol{\epsilon}^{(d)})^\top\mathbf{S}^{(q)\top}\right]$$

$$+2\bar{\epsilon}_f'\mathbb{E}\left[\left\langle(\mathbf{H}_f^{(q)})^{-1}\mathbf{SHw}^*,\mathbf{S}^{(q)}(\mathbf{x}+\boldsymbol{\epsilon}^{(d)})\right\rangle^2\mathbf{S}^{(q)}(\mathbf{x}+\boldsymbol{\epsilon}^{(d)})(\mathbf{x}+\boldsymbol{\epsilon}^{(d)})^\top\mathbf{S}^{(q)\top}\right]$$

$$\preceq 2\bar{\epsilon}_f\mathbb{E}\left[\left\langle(\mathbf{H}_f^{(q)})^{-1}\mathbf{SHw}^*,\mathbf{S}^{(q)}\boldsymbol{\epsilon}^{(d)}\right\rangle^2\mathbf{S}^{(q)}(\mathbf{x}+\boldsymbol{\epsilon}^{(d)})(\mathbf{x}+\boldsymbol{\epsilon}^{(d)})^\top\mathbf{S}^{(q)\top}\right]$$

$$+4\bar{\epsilon}_f'\mathbb{E}\left[\left\langle(\mathbf{H}_f^{(q)})^{-1}\mathbf{SHw}^*,\mathbf{S}^{(q)}\mathbf{x}\right\rangle^2\mathbf{S}^{(q)}(\mathbf{x}+\boldsymbol{\epsilon}^{(d)})(\mathbf{x}+\boldsymbol{\epsilon}^{(d)})^\top\mathbf{S}^{(q)\top}\right]$$

$$+4\bar{\epsilon}_f'\mathbb{E}\left[\left\langle(\mathbf{H}_f^{(q)})^{-1}\mathbf{SHw}^*,\mathbf{S}^{(q)}\boldsymbol{\epsilon}^{(d)}\right\rangle^2\mathbf{S}^{(q)}(\mathbf{x}+\boldsymbol{\epsilon}^{(d)})(\mathbf{x}+\boldsymbol{\epsilon}^{(d)})^\top\mathbf{S}^{(q)\top}\right]$$

$$\preceq[4\bar{\epsilon}_f+8\bar{\epsilon}_f']\mathbb{E}\left[\left\langle(\mathbf{H}_f^{(q)})^{-1}\mathbf{SHw}^*,\mathbf{S}^{(q)}\boldsymbol{\epsilon}^{(d)}\right\rangle^2\mathbf{S}^{(q)}\mathbf{x}\mathbf{x}^\top\mathbf{S}^{(q)\top}\right]$$

$$+[4\bar{\epsilon}_f+8\bar{\epsilon}_f']\mathbb{E}\left[\left\langle(\mathbf{H}_f^{(q)})^{-1}\mathbf{SHw}^*,\mathbf{S}^{(q)}\boldsymbol{\epsilon}^{(d)}\right\rangle^2\mathbf{S}^{(q)}\boldsymbol{\epsilon}^{(d)}\boldsymbol{\epsilon}^{(d)\top}\mathbf{S}^{(q)\top}\right]$$

$$+8\bar{\epsilon}_f'\mathbb{E}\left[\left\langle(\mathbf{H}_f^{(q)})^{-1}\mathbf{SHw}^*,\mathbf{S}^{(q)}\mathbf{x}\right\rangle^2\mathbf{S}^{(q)}\mathbf{x}\mathbf{x}^\top\mathbf{S}^{(q)\top}\right]$$

$$+8\bar{\epsilon}_f'\mathbb{E}\left[\left\langle(\mathbf{H}_f^{(q)})^{-1}\mathbf{SHw}^*,\mathbf{S}^{(q)}\mathbf{x}\right\rangle^2\mathbf{S}^{(q)}\boldsymbol{\epsilon}^{(d)}\boldsymbol{\epsilon}^{(d)\top}\mathbf{S}^{(q)\top}\right]$$

$$\preceq[8\bar{\epsilon}_f'(1+\bar{\epsilon}_d)+(\bar{\epsilon}_d+\bar{\epsilon}_d')(4\bar{\epsilon}_f+8\bar{\epsilon}_f')]\mathbb{E}\left[\left\langle(\mathbf{H}_f^{(q)})^{-1}\mathbf{SHw}^*,\mathbf{S}^{(q)}\mathbf{x}\right\rangle^2\mathbf{S}^{(q)}\mathbf{x}\mathbf{x}^\top\mathbf{S}^{(q)\top}\right]$$

$$\preceq\alpha_0[8\bar{\epsilon}_f'(1+\bar{\epsilon}_d)+(\bar{\epsilon}_d+\bar{\epsilon}_d')(4\bar{\epsilon}_f+8\bar{\epsilon}_f')]\mathbb{E}\left[\mathrm{tr}\left(\mathbf{S}^{(q)\top}(\mathbf{H}_f^{(q)})^{-1}\mathbf{SHw}^*\mathbf{w}^{*\top}\mathbf{HS}^\top(\mathbf{H}_f^{(q)})^{-1}\mathbf{S}^{(q)}\mathbf{H}\right)\mathbf{S}^{(q)}\mathbf{HS}^{(q)\top}\right].$$

(142)

Specifically,

$$\mathbb{E}\left[\mathrm{tr}\left(\mathbf{H}\left(\mathbf{I}-\mathbf{S}^{(q)\top}(\mathbf{H}_f^{(q)})^{-1}\mathbf{SH}\right)\mathbf{w}^*\mathbf{w}^{*\top}\left(\mathbf{I}-\mathbf{S}^{(q)\top}(\mathbf{H}_f^{(q)})^{-1}\mathbf{SH}\right)^\top\right)\mathbf{S}^{(q)}\mathbf{HS}^{(q)\top}\right]$$

$$\preceq 2\mathbb{E}\left[\mathrm{tr}\left(\mathbf{H}\left(\mathbf{I}-\mathbf{S}^\top(\mathbf{H}_f^{(q)})^{-1}\mathbf{SH}\right)\mathbf{w}^*\mathbf{w}^{*\top}\left(\mathbf{I}-\mathbf{S}^\top(\mathbf{H}_f^{(q)})^{-1}\mathbf{SH}\right)^\top\right)\mathbf{S}^{(q)}\mathbf{HS}^{(q)\top}\right]$$

$$+2\mathbb{E}\left[\mathrm{tr}\left(\mathbf{H}\left(\boldsymbol{\epsilon}^{(s)\top}(\mathbf{H}_f^{(q)})^{-1}\mathbf{SH}\right)\mathbf{w}^*\mathbf{w}^{*\top}\left(\boldsymbol{\epsilon}^{(s)\top}(\mathbf{H}_f^{(q)})^{-1}\mathbf{SH}\right)^\top\right)\mathbf{S}^{(q)}\mathbf{HS}^{(q)\top}\right]$$

$$\preceq 8\|\mathbf{w}^*\|_\mathbf{H}^2\mathbb{E}\left[\mathbf{S}^{(q)}\mathbf{HS}^{(q)\top}\right]+4\mathbb{E}\left[\mathrm{tr}\left(\mathbf{H}\left(\boldsymbol{\epsilon}^{(s)\top}(\mathbf{H}_f^{(q)})^{-1}\mathbf{SH}\right)\mathbf{w}^*\mathbf{w}^{*\top}\left(\boldsymbol{\epsilon}^{(s)\top}(\mathbf{H}_f^{(q)})^{-1}\mathbf{SH}\right)^\top\right)\mathbf{SHS}^\top\right]$$ (143)

$$+4\mathbb{E}\left[\mathrm{tr}\left(\mathbf{H}\left(\boldsymbol{\epsilon}^{(s)\top}(\mathbf{H}_f^{(q)})^{-1}\mathbf{SH}\right)\mathbf{w}^*\mathbf{w}^{*\top}\left(\boldsymbol{\epsilon}^{(s)\top}(\mathbf{H}_f^{(q)})^{-1}\mathbf{SH}\right)^\top\right)\boldsymbol{\epsilon}^{(s)}\mathbf{H}\boldsymbol{\epsilon}^{(s)\top}\right]$$

$$\preceq 8\|\mathbf{w}^*\|_\mathbf{H}^2\mathbb{E}\left[\mathbf{S}^{(q)}\mathbf{HS}^{(q)\top}\right]+4(\bar{\epsilon}_s+\bar{\epsilon}_s')\mathbb{E}\left[\mathrm{tr}\left(\mathbf{H}\left(\mathbf{S}^\top(\mathbf{H}_f^{(q)})^{-1}\mathbf{SH}\right)\mathbf{w}^*\mathbf{w}^{*\top}\left(\mathbf{S}^\top(\mathbf{H}_f^{(q)})^{-1}\mathbf{SH}\right)^\top\right)\mathbf{SHS}^\top\right]$$

$$\preceq 8\|\mathbf{w}^*\|_\mathbf{H}^2\mathbb{E}\left[\mathbf{S}^{(q)}\mathbf{HS}^{(q)\top}\right]+4(\bar{\epsilon}_s+\bar{\epsilon}_s')\|\mathbf{w}^*\|_\mathbf{H}^2\mathbf{SHS}^\top,$$

and

$$\mathbb{E}\left[\mathrm{tr}\left(\mathbf{H}\left(\mathbf{S}^{(q)\top}(\mathbf{H}_f^{(q)})^{-1}\mathbf{S}\mathbf{H}\right)\mathbf{w}^*\mathbf{w}^{*\top}\left(\mathbf{S}^{(q)\top}(\mathbf{H}_f^{(q)})^{-1}\mathbf{S}\mathbf{H}\right)^\top\right)\mathbf{S}^{(q)}\mathbf{H}\mathbf{S}^{(q)\top}\right]$$

$$\preceq 2\mathbb{E}\left[\mathrm{tr}\left(\mathbf{H}\left(\mathbf{S}^{\top}(\mathbf{H}_f^{(q)})^{-1}\mathbf{S}\mathbf{H}\right)\mathbf{w}^*\mathbf{w}^{*\top}\left(\mathbf{S}^{\top}(\mathbf{H}_f^{(q)})^{-1}\mathbf{S}\mathbf{H}\right)^\top\right)\mathbf{S}^{(q)}\mathbf{H}\mathbf{S}^{(q)\top}\right]$$

$$+2\mathbb{E}\left[\mathrm{tr}\left(\mathbf{H}\left(\boldsymbol{\epsilon}^{(s)\top}(\mathbf{H}_f^{(q)})^{-1}\mathbf{S}\mathbf{H}\right)\mathbf{w}^*\mathbf{w}^{*\top}\left(\boldsymbol{\epsilon}^{(s)\top}(\mathbf{H}_f^{(q)})^{-1}\mathbf{S}\mathbf{H}\right)^\top\right)\mathbf{S}^{(q)}\mathbf{H}\mathbf{S}^{(q)\top}\right] \tag{144}$$

$$\preceq 2\|\mathbf{w}^*\|_{\mathbf{H}}^2\mathbb{E}\left[\mathbf{S}^{(q)}\mathbf{H}\mathbf{S}^{(q)\top}\right]+4\mathbb{E}\left[\mathrm{tr}\left(\mathbf{H}\left(\boldsymbol{\epsilon}^{(s)\top}(\mathbf{H}_f^{(q)})^{-1}\mathbf{S}\mathbf{H}\right)\mathbf{w}^*\mathbf{w}^{*\top}\left(\boldsymbol{\epsilon}^{(s)\top}(\mathbf{H}_f^{(q)})^{-1}\mathbf{S}\mathbf{H}\right)^\top\right)\mathbf{S}\mathbf{H}\mathbf{S}^\top\right]$$

$$+4\mathbb{E}\left[\mathrm{tr}\left(\mathbf{H}\left(\boldsymbol{\epsilon}^{(s)\top}(\mathbf{H}_f^{(q)})^{-1}\mathbf{S}\mathbf{H}\right)\mathbf{w}^*\mathbf{w}^{*\top}\left(\boldsymbol{\epsilon}^{(s)\top}(\mathbf{H}_f^{(q)})^{-1}\mathbf{S}\mathbf{H}\right)^\top\right)\boldsymbol{\epsilon}^{(s)}\mathbf{H}\boldsymbol{\epsilon}^{(s)\top}\right]$$

$$\preceq 2\|\mathbf{w}^*\|_{\mathbf{H}}^2\mathbb{E}\left[\mathbf{S}^{(q)}\mathbf{H}\mathbf{S}^{(q)\top}\right]+4(\bar{\epsilon}_s+\bar{\epsilon}_s')\|\mathbf{w}^*\|_{\mathbf{H}}^2\mathbf{S}\mathbf{H}\mathbf{S}^\top.$$

Here we use two key inequalities: firstly,

$$\mathrm{tr}\left(\mathbf{H}\left(\mathbf{S}^{\top}(\mathbf{H}_f^{(q)})^{-1}\mathbf{S}\mathbf{H}\right)\mathbf{w}^*\mathbf{w}^{*\top}\left(\mathbf{S}^{\top}(\mathbf{H}_f^{(q)})^{-1}\mathbf{S}\mathbf{H}\right)^\top\right)$$

$$=\mathbf{w}^{*\top}\mathbf{H}\mathbf{S}^{\top}(\mathbf{H}_f^{(q)})^{-1}\mathbf{S}\mathbf{H}\mathbf{S}^{\top}(\mathbf{H}_f^{(q)})^{-1}\mathbf{S}\mathbf{H}\mathbf{w}^*$$

$$\leq\mathbf{w}^{*\top}\mathbf{H}\mathbf{S}^{\top}(\mathbf{H}_f^{(q)})^{-1}\mathbf{S}\mathbf{H}\mathbf{w}^*$$

$$\leq\|\mathbf{w}^*\|_{\mathbf{H}}^2\|\mathbf{H}^{1/2}\mathbf{S}^{\top}(\mathbf{H}_f^{(q)})^{-1}\mathbf{S}\mathbf{H}^{1/2}\|$$

$$\leq\|\mathbf{w}^*\|_{\mathbf{H}}^2,$$

and secondly,

$$\mathrm{tr}\left(\mathbf{H}\left(\mathbf{I}-\mathbf{S}^{\top}(\mathbf{H}_f^{(q)})^{-1}\mathbf{S}\mathbf{H}\right)\mathbf{w}^*\mathbf{w}^{*\top}\left(\mathbf{I}-\mathbf{S}^{\top}(\mathbf{H}_f^{(q)})^{-1}\mathbf{S}\mathbf{H}\right)^\top\right)$$

$$\leq 2\|\mathbf{w}^*\|_{\mathbf{H}}^2+2\mathrm{tr}\left(\mathbf{H}\left(\mathbf{S}^{\top}(\mathbf{H}_f^{(q)})^{-1}\mathbf{S}\mathbf{H}\right)\mathbf{w}^*\mathbf{w}^{*\top}\left(\mathbf{S}^{\top}(\mathbf{H}_f^{(q)})^{-1}\mathbf{S}\mathbf{H}\right)^\top\right)$$

$$\leq 4\|\mathbf{w}^*\|_{\mathbf{H}}^2.$$

Overall, together with (130)-(144) and $\mathbb{E}\left[\mathbf{S}^{(q)}\mathbf{H}\mathbf{S}^{(q)}\right]\preceq(1+\bar{\epsilon}_s)\mathbf{S}\mathbf{H}\mathbf{S}^\top$, it holds

$$\mathbb{E}\left[\xi^2\tilde{\mathbf{x}}^{(q)}(\tilde{\mathbf{x}}^{(q)})^\top\right]$$
$$\precsim(1+\alpha_0\|\mathbf{w}^*\|_{\mathbf{H}}^2)(1+\bar{\epsilon}_s+\bar{\epsilon}_s')\left[(1+\bar{\epsilon}_d)(1+\bar{\epsilon}_f)(\bar{\epsilon}_lC_y+\overline{\sigma}_0^2)+(1+\bar{\epsilon}_d+\bar{\epsilon}_d')(1+\bar{\epsilon}_f+\bar{\epsilon}_f')\right]\mathbf{S}\mathbf{H}\mathbf{S}^\top, \tag{145}$$

together with $\mathbf{H}_f^{(q)}\succeq(1+\underline{\epsilon}_f)(1+\underline{\epsilon}_d)(1+\underline{\epsilon}_s)\mathbf{S}\mathbf{H}\mathbf{S}^\top$, we have

$$\mathbb{E}\left[\xi^2\tilde{\mathbf{x}}^{(q)}(\tilde{\mathbf{x}}^{(q)})^\top\right]\preceq\overline{\sigma}_M^2\mathbf{H}_f^{(q)}, \tag{146}$$

where $\overline{\sigma}_M^2\precsim\frac{(1+\alpha_0\|\mathbf{w}^*\|_{\mathbf{H}}^2)(1+\bar{\epsilon}_s+\bar{\epsilon}_s')\left[(1+\bar{\epsilon}_d)(1+\bar{\epsilon}_f)(\bar{\epsilon}_lC_y+\overline{\sigma}_0^2)+(1+\bar{\epsilon}_d+\bar{\epsilon}_d')(1+\bar{\epsilon}_f+\bar{\epsilon}_f')\right]}{(1+\underline{\epsilon}_f)(1+\underline{\epsilon}_d)(1+\underline{\epsilon}_s)}$. $\qquad\square$

