# OpenReview forum: "Scaling Laws for Precision in High-Dimensional Linear Regression"
_ICML.cc/2026/Conference — ICML 2026 regular_

### Official Review · Reviewer_dD5U · 2026-02-23

**Soundness:** 3
**Presentation:** 2
**Significance:** 3
**Originality:** 3
**Overall Recommendation:** 4
**Confidence:** 3

**Summary:**

This paper develops a theoretical framework for some scaling laws for low precision optimization that where empirically established in prior literature. The authors focus on the high dimensional linear regression setting with a power-law spectrum, which has been shown to exhibit several complex behaviors and phenomenology of large neural networks.

The paper establishes a set of upper and lower bounds for the out-of-sample risk of one-pass SGD with either additive or multiplicative quantization. The theory predicts a qualitative distinction: multiplicative quantization preserves the effective model size, whereas additive quantization injects a signal-independent error floor that can shrink effective model capacity and introduce an implicit penalty term.

The authors present a set of synthetic experiments with power-law spectrum, for various model sizes, and report parameters that closely match the theoretical predictions, supporting the claimed scaling behavior, as well as prior empirical results that wrote the test loss as a power law in model size and data size.

**Compliance With Llm Reviewing Policy:**

Affirmed.

**Final Justification:**

The rebuttal phase has clarified all of my questions, my original view and comments on the paper are unchanged and I lean towards acceptance.

**Key Questions For Authors:**

1. Can you further comment on the commutativity condition ? In particular, does it hold in some practical settings ? If it typically fails, can you comment on the tightness of your results ?

2. How do you expect the scaling laws to change under multi-epoch training and common optimization choices (learning-rate schedules, momentum/Adam)? Can you provide either theory or empirical evidence that the effective-size picture remains stable?

3. Do you have evidence that the predicted “effective shrinkage + additive floor” decomposition appears in practical deep learning training (e.g., vision or language models), where the data representation and spectrum evolve during training?

4. How do the proposed multiplicative vs. additive quantization models map to real training pipelines? In particular, which common low-precision mechanisms (e.g., stochastic rounding, per-channel scaling...) correspond to your $\varepsilon$-multiplicative vs $\varepsilon$-additive assumptions, and where might those assumptions fail?

**Limitations:**

yes

**Strengths And Weaknesses:**

# Strengths

### Direct connection to recent empirical observations

The paper positions its results as a theoretical basis for reported empirical behavior; model shrinkage under integer quantization + an additive penalty due to truncated/flattened tail directions becoming un-learnable. The proposed decomposition also yields an immediately actionable lens for practitioners by suggesting how to allocate compute depending on which regime dominates.

### Theoretical framework appears sound

The theoretical framework is unified, conceptually clean and clearly presented. The authors reduce quantization effects to a small set of interpretable terms, and back their decomposition by mathematics that look well-supported and sound.

### Empirical results tightly follow the theory

The results of the experiments are extremely well aligned with the paper's theoretical claims (all fits achieving close to $1$ $R^2$, across a range of parameters and quantization schemes).

# Weaknesses

### Bounds are not fully tight

The theory provides both upper and lower bounds, but they do not fully match in all regimes. This weakens how sharp the proposed scaling picture is (e.g., the conclusion explicitly flags the lack of matching upper/lower bounds as an open limitation), and the paper also notes specific mismatches such as the multiplicative-quantization lower bound on $N_{eff}$ not strictly matching its upper bound.

### Results seem to rely on an artificial commutativity condition

Several key statements assume a commutativity condition between the quantized feature covariance and the sketched covariance. While this helps isolate the main mechanisms cleanly, it is a fairly strong/algebraic assumption that may not hold generically; the paper itself indicates that removing it can introduce additional penalties (e.g., involving conditioning) and treats the more general case separately. While there should not be much questioning about the soundness of the generalized bounds, prior works do not highlight any commutator term, suggesting that these general bounds might not be tight.

### Limited validation beyond the stylized synthetic regime

While the experiments are tightly aligned with the theory, they are confined to a controlled synthetic setup. This leaves some uncertainty about how robust the proposed theoretical picture is. In particular, whether the derived power law generalizes to any dataset with potentially un-clean covariates, to more complex (non-linear) models, or to more realistic optimization schemes (multi-epoch optimization).

---

> ### Author Rebuttal · Authors · 2026-03-31
>
> **Q1:** Commutativity assumption.
>
> **A1:**  Thanks for the comment. We **verify** commutativity in the common case where the quantization error variance is exactly proportional either to the raw data or to the identity, i.e., $\overline{ϵ}=\underline{ϵ}$, and then **approximate commutativity** in general case. Without commutativity, our framework **remains valid** and the **scaling** of "effective shrinkage+additive floor" persists, though upper-bound shrinkage is enhanced and lower-bound floor is suppressed. Overall, we make this assumption for concise scaling laws. For mathematical details, please refer to our response to Q2 of Reviewer X1yg.
>
> **Q2:** Tightness.
>
> **Q2.1:** Tightness regarding commutator term.
>
> **A2.1:** We note that prior work [5] only studies the case where **commutativity holds** and our framework recovers them as a special case. The mismatch induced by the commutator term reflects a **quantization mechanism effect** rather than a technical limitation. Specifically, the term arises from $\overline{ε}≠\underline{ε}$, making $H_f^{(q)}$ no longer proportional to $SHS^T$. This misalignment inherently widens the worst-case bounds.
>
> **Q2.2:** Tightness with commutativity.
>
> **A2.2:** Beyond the intrinsic non-commutativity, **general risk bounds** are tight (Theorem C.17 / C.18 vs. Theorem D.14 / D.15) in commutative case. The apparent mismatch arises from **aggregating** risks across (M,N) regimes to derive **unifying concise scaling laws**. Though widening the gap, we remark that this is a deliberate design for interpretability rather than a technical limitation.
>
> **Q3:** Optimization algorithm.
>
> **A3:** Thanks for the comment. Our **technique** can extend to **practical algorithms**, such as multi-epoch SGD, decaying learning rate, SGD with momentum and preconditioned SGD and the proposed "effective shrinkage + additive floor" **scaling** still holds. We note the fact that Adam acts as the combination of momentum and preconditioning implies the **applicability to Adam**.
> - **Multi-pass SGD**
>
>   [1] decomposes the risk of full-precision multi-pass SGD as the risk of GD and the fluctuation from stochastic sampling. The decomposition **carries to low-precision setting** by replacing $H$ with $H_f^{(q)}$. As this parallels the standard-to-quantized one-pass SGD transition, our **technique** and **scaling law structure** still hold.
> - **SGD with geometrically decaying learning rate**
>
>   The key technique is a **multi-phase strategy** [2] where the trajectory is partitioned into phases of constant stepsize. As this is independent of quantization, our **technique** and the **scaling structure** still hold.
> - **SGD with momentum**
>
>   [3] introduces an augmented iterate taking the same form of one-pass SGD to model the momentum. This model **carrires to low-precision setting** by replacing $x_t$ with quantized $\tilde{x}_t^{(q)}$. As this parallels the standard-to-quantized one-pass SGD transition, our **technique** and **scaling structure** still hold.
> - **Preconditioned SGD**
>
>   Preconditioning [4] transforms the covariance by matrix G. Our **technique** is applicable by transforming the quantized covariance using G in the same way and the **scaling structure** still holds.
>
> **Adam** combines momentum and adaptive preconditioning. As our framework supports both mechanisms independently, we believe their composition for Adam inherits the scaling structure.
>
> **Q4:** Quantization scheme.
>
> **A4:** Thanks for the comment. Definitions of multiplicative and additive quantization capture practical FP and INT formats when data lies in the feasible range (**no clipping**). Otherwise it fails. For simplicity, we verify the stochastic rounding (Example H.1 in the manuscript) and consider the upper bounds and scalar case of our Definition 3.2. For unclipped x,
>   - FP: $D ≤ x^2 2^{-2m}/4$, aligning with multiplicative quantization
>   - INT: $D ≤ 2^{-2b}/4$, aligning with additive quantization
>
> **Q5:** Experiments.
>
> **A5:** We conduct additional experiments for linear regression with
> - SGD on **real-world dataset** Communities and Crimes
> - **Adam** on synthetic dataset
>
> Experiment results in https://anonymous.4open.science/r/Experiment-Results-ICML_2026/ validate our theory of scaling laws. Works [6, 7] show our scaling structure holds for complex non-linear models in practice. Due to rebuttal computational constraints, we defer large-scale language/vision experiments to future work.
>
> **References:**
>
> [1] Risk bounds of multi-pass SGD for least squares in the interpolation regime.
>
> [2] Last iterate risk bounds of SGD with decaying stepsize for overparameterized linear regression.
>
> [3] Risk bounds of accelerated SGD for overparameterized linear regression.
>
> [4] Improving implicit regularization of SGD with preconditioning for least square problems.
>
> [5] Learning under quantization for high-dimensional linear regression.
>
> [6] Scaling laws for precision.
>
> [7] Scaling laws for floating point quantization training.

---

> > ### Author Rebuttal · Reviewer_dD5U · 2026-04-01
> >
> > Thank you for the detailed rebuttal. It addresses my questions constructively, and overall it reinforces my positive assessment of the paper.
> >
> > On the commutativity assumption, your clarification is helpful. I understand better now.
> >
> > On optimization, I appreciate the effort to connect the analysis to multi-pass SGD, decaying step sizes, momentum, preconditioning, and Adam. That said, this part of the rebuttal remains somewhat speculative from my perspective: the transfer principle sounds plausible, but in the absence of either explicit theorem statements in the paper or more substantial empirical evidence, I would still describe this as an important expected extension. Since I had framed this as a question rather than a major weakness, this does not change my recommendation.
> >
> > Finally, the added experiments are a welcome addition. They strengthen the case that the theory is not confined only to the synthetic one-pass setting, even if the validation beyond stylized linear models remains limited for now. I agree that large-scale vision/language evidence is beyond the scope of a rebuttal.
> >
> > Overall, the rebuttal successfully addresses most of my concerns at the level I expected for this paper. My final view remains that this is a technically solid and interesting contribution with a clear theoretical message, I'm maintaining my score.

---

> > > ### Author Response · Authors · 2026-04-04
> > >
> > > Dear Reviewer dD5U,
> > >
> > > We extend our thanks for your time, and we sincerely appreciate your positive feedback on our work. Thank you again for your effort in reviewing our work.
> > >
> > > Best,
> > >
> > > Authors

---

### Official Review · Reviewer_X1yg · 2026-03-11

**Soundness:** 3
**Presentation:** 3
**Significance:** 4
**Originality:** 4
**Overall Recommendation:** 5
**Confidence:** 3

**Summary:**

This paper investigates how low-precision training impacts empirical scaling laws, using a sketched linear regression framework in the high-dimensional regime. The authors analyze the learning dynamics under additive and multiplicative quantization and prove that while both schemes reduce effective data size and introduce an unavoidable additive error floor, multiplicative quantization preserves the model's full capacity, whereas additive quantization actively shrinks it. The work provides a unified mathematical explanation for empirical observations regarding quantized model scaling, by establishing rigorous upper and lower risk bounds.

**Compliance With Llm Reviewing Policy:**

Affirmed.

**Final Justification:**

The authors effectively addressed my concerns. Their explanations and commitment to revisions mitigate the initially identified weaknesses. I am increasing my score.

**Key Questions For Authors:**

Regarding the commutativity assumption between $H_f^{(q)}$ and $SHS^\top$ used in the main theorems:
1. How justifiable is this assumption in practical scenarios? Are there specific data distributions or quantization schemes where this condition can be shown to be valid?
2. How difficult is it to extract interpretable insights from the bounds in the non-commutative setting? Specifically, do you expect significant alterations of the qualitative phenomenology and scaling behavior?

**Limitations:**

Yes

**Strengths And Weaknesses:**

### Strengths
The paper provides a formal mathematical explanation for phenomena previously observed only empirically in low-precision training. By studying an analytically tractable problem, the authors offer interpretable insights into how quantization affects scaling behavior. Specifically, the derivation of rigorous upper and lower bounds and the formalization of the dichotomy between additive and multiplicative quantization, make the theoretical contribution particularly solid and original.

### Weaknesses
I find some core definitions (e.g. the quantized SGD) particularly hard to parse, and the overall clarity is compromised by the introduction of heavily nested notation without sufficient pedagogical build-up. I would suggest to expand the heaviest definitions in order improve readability and make the paper and its insights more accessible to a broader audience.

---

> ### Author Rebuttal · Authors · 2026-03-31
>
> **Q1:** Quantized SGD illustration.
>
> **A1:** Thanks for the comment. We will detail the **step-by-step construction** of the quantized SGD for clarity. Standard SGD updates by $\mathbf{v}\_t=\mathbf{v}\_{t-1}+γ(y_t-(\mathbf{S}\mathbf{x}\_t)^T \mathbf{v}_{t-1})\mathbf{S}\mathbf{x}_t.$ As low-precision training requires **every multiplication must be executed using low-precision formats**, we introduce quantization operations to convert those components from full precision into low precision. Specifically, we define
> - the **quantized feature** $\mathbf{f}\_t^{(q)} =\mathcal{Q}_{f}(\mathcal{Q}_s(\mathbf{S})\mathcal{Q}_d(\mathbf{x}_t))$ regarding the multiplication of $\mathbf{S}$ and $\mathbf{x}_t$
> - the **quantized activation** $a\_t^{(q)}=\mathcal{Q}\_p(\mathbf{v}\_{t-1})^T \mathbf{f}_t^{(q)}$ regarding the multiplication of $\mathbf{S}\mathbf{x}\_t$ and $\mathbf{v}\_{t-1}$
> - the **quantized output gradient** $g_t^{(q)} =\mathcal{Q}\_o(\mathcal{Q}\_l(y_t){-}\mathcal{Q}\_a(a_t^{(q)}))$ for the multiplication of $y_t-(\mathbf{S}\mathbf{x}\_t)^T \mathbf{v}_{t-1}$ and $\mathbf{S}\mathbf{x}_t$
>
> Finally, mirroring the empirical practice in [1] that the master weight $\mathbf{v}_t$ is maintained and updated in full precision, we obtain the quantized SGD $\mathbf{v}\_t=\mathbf{v}\_{t-1}+ γ g\_t^{(q)}\mathbf{f}\_t^{(q)}.$
>
> **Q2:** Commutativity assumption.
>
> **A2:** Thanks for the comment. We first **verify** commutativity in the common case where the quantization error variance is exactly proportional either to the raw data (under multiplicative quantization) or to the identity (under additive error), i.e., $\overline{ϵ}=\underline{ϵ}$, and then **approximate commutativity** in general case. Without commutativity, our framework **remains valid** and the **scaling** of "effective shrinkage+additive floor" persists, though upper-bound shrinkage is enhanced and lower-bound floor is suppressed. Overall, we make this assumption for concise scaling laws. For mathematical details, we denote $\overline{ε}\_M=\prod\_{i=d,f,s}(1+\overline{ε}_i)-1, \ \underline{ε}\_M=\prod\_{i=d,f,s}(1+\underline{ε}_i)-1,\ \overline{ε}_A=\overline{ε}_f+\overline{ε}_s(1+p\overline{ε}_d)+\overline{ε}_d\frac{p}{M},\ \underline{ε}_A=\underline{ε}_f+\underline{ε}_s(1+p\underline{ε}_d)+\underline{ε}_d\frac{p}{M}.$
>
> **I. Exact commutativity in case $\overline{ε}_i = \underline{ε}_i = ε_i$.**
>
> In this case, $\mathbf{H}_f^{(q)} = (1+ε_M) \mathbf{SHS}^T$ under multiplicative quantization and $\mathbf{H}_f^{(q)}\eqsim \mathbf{SHS}^⊤+ ε_A \mathbf{I}$ with high probability under additive quantization. Since any matrix commutes with any scalar multiple of itself or identity, the commutativity holds.
>
> **II. Approximate commutativity in general.**
>
> Denote $I=\\|\mathbf{H}_f^{(q)}\mathbf{SHS}^T-\mathbf{SHS}^T\mathbf{H}_f^{(q)}\\|$.​ With high probability, $I \lesssim \overline{ε}_M-\underline{ε}_M$ under multiplicative quantization and $I \lesssim \overline{ε}_A-\underline{ε}_A$ under additive quantization. The commutativity approximately holds as ε is small.
>
> **III. Theory without commutativity.**
>
> We emphasize that the commutativity assumption is used primarily for **cleaner presentation** of the scaling laws. Our framework **generalizes** to the non-commutative setting. We re-derive the following bounds in the non-commutative setting.
> * **Additive error lower bounds** (Lemmas D.16, D.17)
>
>   Denote $J=\mathrm{tr}\left( (\mathbf{H}_f^{(q)})^{-1}\frac{1}{Nγ}(\mathbf{I}-(\mathbf{I}-γ\mathbf{H}_f^{(q)})^N)(\mathbf{H}_f^{(q)})^{-1}(\mathbf{H}_f^{(q)} - \mathbf{SHS}^T)(\mathbf{H}_f^{(q)})^{-1}\mathbf{SH}^2\mathbf{S}^T\right).$ With high probability, $J \gtrsim \frac{1}{N(1+\overline{ε}_M)}\left[\frac{\underline{ε}_M}{1+\underline{ε}_M}-(\frac{\overline{ε}_M}{1+\overline{ε}_M} - \frac{\underline{ε}_M}{1+\underline{ε}_M})\sqrt{M^{3a}\frac{(1+\overline{ε}_M)}{(1+\underline{ε}_M)}}\right]$ under multiplicative quantization and $J \gtrsim \frac{1}{N}\frac{M^{-a}}{M^{-a}+\overline{ε}_A}\left[\frac{\underline{ε}_A}{1+\underline{ε}_A}-(\frac{\overline{ε}_A}{M^{-a}+\overline{ε}_A}-\frac{\underline{ε}_A}{1+\underline{ε}_A})\sqrt{M^{3a}\frac{(1+\overline{ε}_A)}{(1+\underline{ε}_A)}}\right]$ under additive quantization. The proof is mainly based on the approximate commutativity discussed above, which we omit for brevity.
> * **Bias upper bounds** (Lemmas C.25, C.27)
>
>   Without commutativity, the bias upper bounds acquire additional terms:
>   * Multiplicative: $\frac{\overline{ε}_M - \underline{ε}_M}{1+\overline{ε}_M}M^{a/2}$
>   * Additive: $M^{a/2}\frac{M^a\overline{ε}_A - \underline{ε}_A}{1+M^a\overline{ε}_A}$
>
> Overall, without commutativity, the qualitative scaling behavior of "**effective shrinkage + additive floor**" remains intact. The quantitative changes are:
> 1. Larger bias upper bounds, **boosting the effective shrinkage in data size**.
> 2. Smaller additive error lower bounds, **suppressing the additive floor**.
>
> **Reference:**
>
> [1] DeepSeek-V3 technical report.

---

> > ### Author Rebuttal · Reviewer_X1yg · 2026-04-01
> >
> > Thanks for your detailed rebuttal and clarification. I have now a better understanding of the algorithm and the commutativity assumption.

---

> > > ### Author Response · Authors · 2026-04-04
> > >
> > > Dear Reviewer X1yg,
> > >
> > > We extend our thanks for your time, and we sincerely appreciate your positive feedback on our work. Thank you again for your effort in reviewing our work.
> > >
> > > Best,
> > >
> > > Authors

---

### Official Review · Reviewer_xGyK · 2026-03-12

**Soundness:** 2
**Presentation:** 2
**Significance:** 3
**Originality:** 3
**Overall Recommendation:** 3
**Confidence:** 2

**Summary:**

Within a high-dimensional sketched linear regression framework, this paper provides the first rigorous theoretical exploration of the scaling laws concerning model size, data size, and quantization precision in low-precision training. By comparatively analyzing multiplicative quantization (signal-dependent, akin to floating-point quantization) and additive quantization (signal-independent, akin to integer quantization), the study reveals a critical dichotomy in how quantization affects scaling behaviors: although both schemes inevitably introduce an additive error and reduce the effective data size, multiplicative quantization fully preserves the effective capacity of the full-precision model, whereas additive quantization strictly compresses the effective model size. This core finding not only perfectly explains recent empirical observations in the industry but also provides a solid theoretical foundation and principled guidance for jointly optimizing model scale, dataset size, and training protocols under practical hardware constraints.

**Compliance With Llm Reviewing Policy:**

Affirmed.

**Final Justification:**

Although I acknowledge that the authors have provided substantial work under the current problem setup, I still do not feel that the two issues I raised have been adequately addressed, particularly in terms of providing further guidance for practical real-world applications. Therefore, I maintain my score.

**Key Questions For Authors:**

see weakness

**Limitations:**

yes

**Strengths And Weaknesses:**

Strengths

First, the proof approach of the paper appears to be correct. I did not check it line by line, but judging from the proof structure, it should be correct.

Second, the proof workload is huge.

Third, the paper focuses on a very important point, namely the performance issues of quantized optimization algorithms.

Weaknesses

First, I noticed that the proofs in the paper rely on many properties of quantized SGD. However, the SGD algorithm itself is not widely used in the training of large models (in fact, scaling laws are primarily needed only for large models). It would have broader significance if the conclusions could be generalized to general first-order algorithms, including Adam.

Second, the mathematical assumptions required in this paper are relatively strong. For instance, it assumes that the quantized feature covariance matrix and the sketched covariance matrix are commutative.

Third, the model adopted in this paper is very simple. Although the workload of the proofs is substantial, the actual difficulty is not very high due to the simplicity of the model combined with broad assumptions.

Fourth, for linear models, there is a massive amount of binary classification data available in the community, such as the LIBSVM datasets. I believe it is inappropriate to use simulated (synthetic) data here.

---

> ### Author Rebuttal · Authors · 2026-03-31
>
> **Q1:** Quantized SGD algorithm.
>
> **A1:** Thanks for the comment. SGD is the **standard setup** in theoretical studies of scaling laws [1-6]. We follow to ensure **interpretable and comparable** results. Going further, our **technique** can extend to more **practical algorithms**, including multi-pass SGD, SGD with decaying stepsize, SGD with momentum and preconditioned SGD, and the proposed **scaling law structure** is preserved. We remark the fact that Adam can be viewed as the combination of momentum and preconditioning implies the **applicability of our results to Adam**. For mathematical details, please refer to our response to Q3 of Reviewer dD5U.
>
> **Q2:** Commutative assumption.
>
> **A2:** Thanks for the comment. We first **verify** commutativity in the common case [7] where the quantization error variance is exactly proportional either to the raw data (under multiplicative quantization) or to the identity (under additive quantization), i.e., $\overline{ϵ}=\underline{ϵ}$, and then **approximate commutativity** in general case. Without commutativity, our framework **remains valid** and the **scaling** of "effective shrinkage+additive floor" persists, though upper-bound shrinkage is enhanced and lower-bound floor is suppressed. Overall, we make this assumption for concise scaling laws. For mathematical details, please refer to our response to Q2 of Reviewer X1yg.
>
> **Q3:** Technique and model.
>
> **A3:** Thanks for the comment. We first restate the **technical contribution** which has been emphasized in Section 5 to highlight that our technique is non-trivial, then illustrate the benefit of using **linear model**, and lastly show the technical **extensibility to non-linear models**.
>
> As detailed in Section 5 of the manuscript, neither the techniques for full-precision scaling laws [1] nor the techniques for low-precision excess risk upper bounds [7] are sufficient for our analysis. In particular, our work **resolves two challenges**:
>
> - overcome the potential **indefiniteness** of the cross-term between quantization error and the optimal parameter by first deriving crude lower bounds and then bootstraping into refined bounds;
>
> - overcome the **destruction of polynomial spectrum** structure of $\mathbf{H}_f^{(q)}$ by concentration and spectrum decomposition.
>
> We remark that the linear model setup is **well-established and standard** in the scaling law literature [1-6]. This line of research has repeatedly showed that linear models serve as a powerful and well-established testbed for **capturing key phenomenological aspects** of deep learning, including learning-rate effects [5,6], batch-size effects [5], multi-epoch effects [2,4] and even quantization effects [7]. Using linear model setup ensures our results are **interpretable**.
>
> Going further, we highlight that our framework can be extended to non-linear models.
>
> - **Single-neuron models**
>
>   For $f(\mathbf{x}) = σ(\mathbf{w}^T \mathbf{x})$ with a known activation σ, the SGD dynamics can be analyzed via the same error covariance recursion $\mathbb{E}[\boldsymbol{η}_t \otimes \boldsymbol{η}_t]$, with the Hessian of the population risk replacing the data covariance. Combining the technique for full-precision single-neuron (ReLU) models in [8], our technique can extend to single-neuron models.
>
> - **Multiple-neuron models in NTK regime**
>
>   In the NTK regime, our framework applies by replacing $\mathbf{H}_f^{(q)}$ with the quantized NTK Gram matrix $\widetilde{\mathbf{H}}^{(q)}:=\mathbb{E}[\mathcal{Q}_g(\phi(\mathcal{Q}_d(\mathbf{x}_t)))\mathcal{Q}_g(\phi(\mathcal{Q}_d(\mathbf{x}_t)))^T].$ The spectral structure of the $\widetilde{\mathbf{H}}^{(q)}$ fits directly into current structure of $\mathbf{H}_f^{(q)}$ in sketched linear regression setting, showing that the "effective shrinkage + additive floor" scaling is expected to persist.
>
> **Q4:** Experiments in more practical settings.
>
> **A4:** Thanks for the comment. We provide empirical evidence for scaling laws in more practical settings, including
>
> * linear regression with SGD on **real-world dataset** Communities and Crimes
>
> * linear regression with **Adam** on synthetic dataset
>
> Experiment results are shown in https://anonymous.4open.science/r/Experiment-Results-ICML_2026/ and validate our theory.
>
>
>
> **References:**
>
> [1] Scaling laws in linear regression: compute, parameters, and data.
>
> [2] Improved scaling laws in linear regression via data reuse.
>
> [3] Scaling law for stochastic gradient descent in quadratically parameterized linear regression.
>
> [4] Larger datasets can be repeated more: A theoretical analysis of multi-epoch scaling in linear regression.
>
> [5] Functional scaling laws in kernel regression: Loss dynamics and learning rate schedules.
>
> [6] Optimal learning-rate schedules under functional scaling laws: power decay and warmup-stable-decay.
>
> [7] Learning under quantization for high-dimensional linear regression.
>
> [8] Finite-sample analysis of learning high-dimensional single ReLU neuron.

---

> > ### Author Rebuttal · Reviewer_xGyK · 2026-04-02
> >
> > First, Adam and SGD exhibit vastly different mathematical properties in many aspects; for instance, Adam may even fail to converge under certain conditions[1]. Therefore, forcibly imposing operational similarities is highly inappropriate. Furthermore, I don't believe linear models can substitute for deep learning models in experiments, especially when discussing effectiveness. If authors believe the conclusions can be extended, please explicitly state the extended conclusion rather than merely citing relevant literature.
> >
> > [1] Reddi, S. J., Kale, S., & Kumar, S. (2018). On the convergence of Adam and beyond. In International Conference on Learning Representations (ICLR). https://openreview.net/forum?id=ryQu7f-RZ

---

> > > ### Author Response · Authors · 2026-04-04
> > >
> > > **Q1:** SGD vs. Adam.
> > >
> > > **A1:** Thanks for the comment. We clarify Adam's convergence and show the extensibility to Adam via a theoretical heuristic.
> > >
> > > **(i) Convergence.** To identify special cases where Adam diverges, [9] **first picks the hyperparameters** of Adam, **then picks the problem**. However, [10] points out that practical applications often **fix the problem first and then tune the hyperparameters**. Under this workflow, [10] proves convergence where $β_2$ is large and $β_1<\sqrt{β_2}<1$. Thus the special case in [9] does not preclude extensibility as we can focus on this convergence region.
> > >
> > > **(ii) Extensibility.** Beyond operations composition, we show the extensibility to a heuristic design (take $β_2\to 1$) for Adam, which is widely used as an effective proxy to approximate Adam in theory [11,12].
> > > - **Results in full precision** (Section J in [11])
> > >
> > >   Adam updates$$θ_{k+1}=θ_k-γ\hat{m}_k\odot(ε\mathbf{1}+\hat{v}_k)^{-1/2}.$$
> > >
> > >   Consider a heuristic by taking $β_2\to 1$ and setting $β_1=0$ (momentum preserves scaling structure [11]),$$θ_{k+1}-θ_k \eqsim -γ\frac{(\langle Sx_k,θ_k\rangle-y_k)\,Sx_k}{\sqrt{diag(K)\cdot L(θ_k)}},\ K=SHS^T. \tag{2}$$Decompose excess risk $R(N)=\sum_{i=1}^M r_i(N)$ with$$r_i(N)=(θ_N-θ^ *)^T(Ku_i\otimes w_i)(θ_N-θ^ *),$$where $u_i,w_i$ are right/left eigenvectors of $\bar{K}=diag(K)^{-1/2}K,$ use (2) and Taylor expansion,$$\mathbb{E}[r_i(k+1)-r_i(k)|\mathcal{F}\_k]=-(\frac{2γ}{\sqrt{L(θ_k)}}-\frac{2γ^2}{L(θ_k)})λ_i(\bar{K})r_i(k)+γ^2(w_i^\top K_τKu_i), \tag{3}$$
> > >
> > >   where $K_τ=\bar{K} diag(K)^{-1/2}.$ Approximate (3) using continuous-time ODE via continuous time t=kγ and take integral, we obtain$$R(M,N)\eqsim M^{-2α+\max(0,1-2β)}+(M^{\min(α,0.5)}N)^{-\frac{2(2α+2β-1)}{2α+1-2β}}+(M^{\frac{6α-1}{4α-2}}N)^{-\frac{2(2α-1)}{2α+1}}+M^{2-2\min(α,0.5)}. \tag{*}$$
> > > - **Extensibility to low precision**
> > >
> > >   In low precision, (2) changes to
> > >   $$θ_{k+1} - θ_k \eqsim -γ \frac{(\langle \tilde{x}_k^{(q)}, Q_p(θ_k)\rangle - Q_l(y_k)) \tilde{x}_k^{(q)}}{\sqrt{diag(H_f^{(q)}) \cdot L^{(q)}(θ_k)}}+quantization\ error\ terms.$$Using our technique, excess risk is decomposed into the **risk in the quantized data space** and the **quantization-induced additive error**. With Taylor expansion and unbiased quantization assumption, the update of risk in quantized space satisfies
> > >
> > >   $$\mathbb{E}[r_i^{(q)}(k+1) - r_i^{(q)}(k) | \mathcal{F}\_k] ≈ \underbrace{-(\frac{2γ}{\sqrt{L^{(q)}}} - \frac{2γ^2}{L^{(q)}}) λ_i(\bar{H}\_f^{(q)}) r_i^{(q)}(k)}\_{drift} + \underbrace{γ^2 (w_i^\top H_{f,τ}^{(q)} H_f^{(q)} u_i) + σ_q^2}_{noise}, \tag{4}$$
> > >
> > >   Comparing (3) and (4), though drift is transformed into quantized space and noise is amplified with quantization noise, their structures preserve. Accompanied with quantization additive error, the scaling laws structure preserves with effective shrinkage and additive floor.$$R^{(q)}(M,N)\eqsim R(M_{eff},N_{eff})+additive\ floor,$$where $R(\cdot,\cdot)$ is defined in (*), additive floor, effective model size $M_{eff}$ and effective data size $N_{eff}$ depend on quantization strength ε. The detailed dependency can be derived based on similar techniques in our paper.
> > >
> > > We emphasize that deriving scaling laws for Adam remains highly complex even in full precision. But based on the discussion above, we believe our technique is applicable once the risk bounds are established in full-precision setting.
> > >
> > > **Q2:** Linear setup.
> > >
> > > **A2:** Thanks for the comment. We address this regarding both theoretical precedent and practical relevance.
> > >
> > > **(i) Principle in theory.** The linear setup is a **standard methodology**, tractable enough to yield **explicit joint scaling laws** while capturing core optimization mechanisms (multi-epoch [2], learning-rate [5], and batch-size schedules [5]). Conversely, the entanglement of quantization with non-linearities and attention mechanisms in complex models renders such analytical clarity **unattainable**.  Therefore, we follow the established linear setup to cleanly **isolate how precision affects scaling**.
> > >
> > > **(ii) Connection to practice.** Though linear model is not a substitute for LLM training, this style of theory has repeatedly provided useful **qualitative explanation and guidance for practice**. For example, functional scaling laws [5] derived in linear regression are verified in 0.1B-1B LLM pretraining. Similarly, our low-precision laws align with empirical large-model pretraining results that quantization induces effective shrinkage and additive error [13], implying that **linear model remains a powerful, valid testbed** for low-precision training.
> > >
> > > **References:** (follow-up of references in rebuttal)
> > >
> > > [9] On the convergence of Adam and beyond.
> > >
> > > [10] Adam can converge without any modification on update rules.
> > >
> > > [11] Scaling laws of SignSGD in linear regression: when does it outperform SGD?
> > >
> > > [12] Exact risk curves of SignSGD in high-dimensions: quantifying preconditioning and noise-compression effects.
> > >
> > > [13] Scaling laws for precision.

---

### Decision · Program_Chairs · 2026-04-30

**Decision:**

Accept (regular)

**Comment:**

Paper presents theoretical study of effects of quantization in the random features model with linear activation, which was previously studied by [Lin et al'24]. This paper extends theory to quantized case. Reviewers raised several important points (absence of Adam, commutativity assumption) and less important ones (relevance of linear model). The recommendation is split exactly 50/50 and I had to cast a tie-breaker. I do not find linear model problematic, since this is a theory paper and Adam version was sketched during the rebuttal. My main issue is the constant overclaiming: "additive quant reduces model size" -- this is not proved as the lower bound is missing! Normally, I would reject paper but I will recommend acceptance hoping that the authors will remove these claims or withdraw the paper if they disagree with this conditional acceptance.

On a stylistic note I think that assuming uncheckable and unrealistic commutativity is not great. Authors should just set \underbar{\eps}=\overbar{\eps} for readability. It's fine to analyze "hypothetical" purely-additive or purely-multiplicative version, especially given the backing of the numerical experiments.